# Optimal Best Arm Identification under Differential Privacy

**Marc Jourdan**\*
EPFL, Lausanne, Switzerland
marc.jourdan@epfl.ch

**Achraf Azize**\*
FairPlay Joint Team, CREST, ENSAE Paris
achraf.azize@ensae.fr

## Abstract

Best Arm Identification (BAI) algorithms are deployed in data-sensitive applications, such as adaptive clinical trials or user studies. Driven by the privacy concerns of these applications, we study the problem of fixed-confidence BAI under global Differential Privacy (DP) for Bernoulli distributions. While numerous asymptotically optimal BAI algorithms exist in the non-private setting, a significant gap remains between the best lower and upper bounds in the global DP setting. This work reduces this gap to a small multiplicative constant, for any privacy budget $\epsilon$. First, we provide a tighter lower bound on the expected sample complexity of any $\delta$-correct and $\epsilon$-global DP strategy. Our lower bound replaces the Kullback–Leibler (KL) divergence in the transportation cost used by the non-private characteristic time with a new information-theoretic quantity that optimally trades off between the KL divergence and the Total Variation distance scaled by $\epsilon$. Second, we introduce a stopping rule based on these transportation costs and a private estimator of the means computed using an arm-dependent geometric batching. En route to proving the correctness of our stopping rule, we derive concentration results of independent interest for the Laplace distribution and for the sum of Bernoulli and Laplace distributions. Third, we propose a Top Two sampling rule based on these transportation costs. For any budget $\epsilon$, we show an asymptotic upper bound on its expected sample complexity that matches our lower bound to a multiplicative constant smaller than $8$. Our algorithm outperforms existing $\delta$-correct and $\epsilon$-global DP BAI algorithms for different values of $\epsilon$.

## 1 Introduction

The stochastic Multi-Armed Bandit (MAB) is an interactive sequential decision-making model [Bubeck et al., 2012; Lattimore and Szepesvári, 2020], introduced by William R. Thompson [Thompson, 1933]. Thompson's motivation for studying MABs is to design clinical trials that adapt treatment allocations on the fly as the medicines appear more or less effective. Specifically, in MABs, a learner interacts with $K \in \mathbb{N}$ unknown probability distributions, referred to as *arms*. In clinical trials, the arms are the candidate medicines, while the observations are patient reactions, $1$ if the patient is cured and $0$ otherwise. The learner aims to identify the arm with the highest average efficiency, i.e., the medicine that cures most patients in expectation. Given a fixed error $\delta \in (0, 1)$, Best Arm Identification (BAI) [Audibert and Bubeck, 2010b; Jamieson and Nowak, 2014] algorithms in the fixed confidence setting [Even-Dar et al., 2006; Gabillon et al., 2012; Garivier and Kaufmann, 2016] suggest a candidate answer that coincides with the optimal arm with probability more than $1 - \delta$, while using as few samples as possible.

BAI algorithms have been increasingly deployed in data-sensitive applications, such as adaptive clinical trials [Thompson, 1933; Robbins, 1952; Aziz et al., 2021], pandemic mitigation [Libin et al., 2019],

---

\*Equal Contribution

39th Conference on Neural Information Processing Systems (NeurIPS 2025).

user studies [Losada et al., 2022], crowdsourcing [Zhou et al., 2014], online advertisement [Chen et al., 2014], hyperparameter tuning [Li et al., 2017], and communication networks [Lindståhl et al., 2022], to name a few. Due to the adaptive nature of these procedures, critical data privacy concerns are raised [Tucker et al., 2016], as exemplified by the adaptive dose finding trial. For each new patient $n$, a physician chooses a dose level $a_n \in [K] := \{1, \cdots, K\}$ based on previous observations, and collects a binary observation measuring the effect of the selected dose on the patient. Crucially, the patients' reactions might reveal information regarding their health. Subsequently, these outcomes will guide the physician's decision for future patients. Eventually, the physician adaptively decides to stop the trial and recommends a dose $\hat{a}_{\tau_\delta}$ after collecting $\tau_\delta$ samples, referred to as *sample complexity*. Even if those outcomes are kept secret, the experimental findings and protocol are detailed thoroughly to the health authorities. This report contains the sequence of chosen dose levels $(a_n)_{n \leq \tau_\delta}$ and the recommended dose level $\hat{a}_{\tau_\delta}$, both indirectly leaking information regarding the patients involved in the trial. This example underscores the need for privacy-preserving fixed-confidence BAI algorithms.

We adopt the Differential Privacy (DP) framework [Dwork and Roth, 2014], which bounds the influence of any single data point. Given a privacy budget $\epsilon$, we consider the $\epsilon$-global DP constraint that assumes the existence of a trusted curator (e.g., the physician running the clinical trial), who observes the outcomes and ensures privacy when publishing these findings. While $\epsilon$-global DP is well-studied in regret minimization [Mishra and Thakurta, 2015; Azize and Basu, 2022; Azize et al., 2025], its impact on fixed-confidence BAI is less understood [Sajed and Sheffet, 2019; Kalogerias et al., 2021]. A significant gap remains between the existing lower and upper bounds [Azize et al., 2023, 2024]. This paper reduces this gap to a small constant for any privacy budget $\epsilon$. Appendix C.1 contains a detailed literature review.

**Contributions.** Our contributions for fixed-confidence BAI under $\epsilon$-global DP are threefold.

**1. Lower bound under global DP.** We derive a novel information-theoretic lower bound on the expected sample complexity of any $\delta$-correct and $\epsilon$-global DP BAI algorithms (Theorem 2). Our lower bound replaces the Kullback-Leibler (KL) divergence in the transportation cost of the non-private characteristic time with an information-theoretic quantity $d_\epsilon$ (Eq. (1)) that smoothly interpolates between the KL divergence and the Total Variation (TV) distance scaled by $\epsilon$.

**2. Private estimator and Generalized Likelihood Ratio (GLR) stopping rule.** We introduce a private estimator using arm-dependent geometric batching without forgetting and a GLR stopping rule based on the $d_\epsilon$ refined transportation costs. Its correctness (Theorem 6) required novel tails concentration results for Laplace distributions and the sum of Bernoulli and Laplace distributions, which could be of independent interest.

**3. Asymptotically optimal algorithm.** We propose a new Top Two sampling rule (DP-TT, Algorithm 1) based on the $d_\epsilon$-transportation costs suggested by our lower bound. We show that the asymptotic expected sample complexity of DP-TT matches our lower bound for any privacy budget $\epsilon$ up to a constant smaller than $8$ (Theorem 7). DP-TT outperforms all the other $\delta$-correct $\epsilon$-global DP BAI algorithms on all tested instances and all $\epsilon$.

## 2 Background: Best Arm Identification under Differential Privacy

In this section, we present the Best Arm Identification (BAI) under fixed confidence problem [Garivier and Kaufmann, 2016], introduce the Differential Privacy (DP) [Dwork et al., 2006] constraint, and finally extend DP to BAI algorithms.

**BAI under Fixed Confidence.** Let $\mathcal{F} := \{\text{Ber}(p) \mid p \in (0, 1)\}$ be the set of Bernoulli distributions. A bandit *instance* $\nu := (\nu_a)_{a \in [K]} \in \mathcal{F}^K$ is characterized by its means $\mu := (\mu_a)_{a \in [K]} \in (0, 1)^K$. The *best* (optimal) arm $a^\star$ is assumed to be unique, i.e., $a^\star(\nu) = a^\star(\mu) := \arg\max_{a \in [K]} \mu_a = \{a^\star\}$. Let $\delta \in (0, 1)$ be the risk parameter. A fixed confidence BAI algorithm $\pi$ specifies three rules that rely on previously observed samples and some exogenous randomness. The *sampling rule* determines the next arm to pull $a_n \in [K]$ for which $X_{n,a_n} \sim \nu_{a_n}$ is observed. The *recommendation rule* recommends a *candidate* arm $\tilde{a} \in [K]$. The *stopping rule* decides when to stop collecting additional samples and output the current candidate arm. The stopping time $\tau_{\epsilon,\delta}$ is the *sample complexity*. Let $\mathbb{P}_{\nu\pi}$ and $\mathbb{E}_{\nu\pi}$ denote the probability and expectation taken over the randomness of the observations from $\nu$ and the algorithm $\pi$ (e.g., due to its privacy mechanism). A fixed-confidence BAI algorithm $\pi$ is $\delta$-correct when $\mathbb{P}_{\nu\pi}(\tau_{\epsilon,\delta} < +\infty, \ \tilde{a} \notin a^\star(\nu)) \leq \delta$ for all $\nu \in \mathcal{F}^K$.

**Differential Privacy (DP).** An algorithm satisfies the Differential Privacy constraint if the algorithm's outputs are "essentially" equally likely to occur, for any two input datasets that only differ in one individual's data. An adversary only observing the mechanism's output cannot distinguish whether any individual's data was included. A privacy budget $\epsilon$ captures the closeness of the output distributions. Smaller $\epsilon$ means stronger privacy.

**Definition 1** ($\epsilon$-DP [Dwork et al., 2006]). *A mechanism $\mathcal{M}$ satisfies $\epsilon$-DP for a given $\epsilon \geq 0$, if, for all neighboring datasets $D \sim D'$, where $D \sim D'$ if and only if $d_{Ham}(D, D') := \sum_{t=1}^{T} \mathbb{1}\{D_t \neq D'_t\} \leq 1$, i.e., $D$ and $D'$ differ by at most one record, and for all sets of output $\mathcal{O} \subseteq \mathrm{Range}(\mathcal{M})$, $\Pr[\mathcal{M}(\mathcal{D}) \in \mathcal{O}] \leq e^\epsilon \Pr[\mathcal{M}(\mathcal{D}') \in \mathcal{O}]$ where the probability space is over the coin flips of the mechanism $\mathcal{M}$.*

To ensure $\epsilon$-DP, the Laplace mechanism [Dwork et al., 2010; Dwork and Roth, 2014] adds calibrated Laplacian noise to the algorithm's output. Let $\mathrm{Lap}(b)$ be the Laplace distribution with mean/variance $(0, 2b^2)$.

**Theorem 1** (Laplace mechanism, Theorem 3.6 [Dwork and Roth, 2014]). *Let $f : \mathcal{X} \to \mathbb{R}^d$ be an algorithm with sensitivity $s(f) \triangleq \max_{\mathcal{D}, \mathcal{D}' \ s.t \ d_{Ham}(\mathcal{D}, \mathcal{D}')=1} \|f(\mathcal{D}) - f(\mathcal{D}')\|_1$, where $\|\cdot\|_1$ is the $\ell_1$ norm. Let $(Z_i)_{i \in [d]}$ be i.i.d. from $\mathrm{Lap}(s(f)/\epsilon)$, then the noisy output $f(\mathcal{D}) + (Z_i)_{i \in [d]}$ satisfies $\epsilon$-DP.*

To be consistent with the literature on private bandits, we use global DP to denote the central DP model with a trusted central decision maker. While the notation $\delta$ is standard in the $(\epsilon, \delta)$-DP relaxation of the $(\epsilon, 0)$-DP constraint considered in this paper, we use $\delta$ for the confidence parameter to be consistent with the literature on pure exploration problems.

**DP for BAI.** In BAI algorithms, the private input is the observation dataset and the output is the recommended candidate arm $\tilde{a}$ and the sequence of sampled actions $(a_n)_{n < \tau_{\epsilon,\delta}}$ until stopping at $\tau_{\epsilon,\delta}$. Let $R = \{r_1, \dots\}$ be a sequence of private observations. Given a fixed sequence of observations $R$, we denote by $\Pr[\pi(R) = (T + 1, \tilde{a}, (a_1, \dots, a_T))]$ the probability that the BAI algorithm $\pi$ stops at step $T + 1$, recommending action $\tilde{a}$ and sampling actions $(a_1, \dots, a_T)$ when interacting with $R$. The randomisation in this probability comes only from the BAI algorithm's sampling, recommendation and stopping rules, whereas the observations are fixed. Then, a BAI algorithm $\pi$ is said to be $\epsilon$-*global DP* if, for every two neighboring sequences of observations $R$ and $R'$, and for every possible stopping time, recommendation and sampled actions $(T + 1, \tilde{a}, (a_1, \dots, a_T))$, we have that

$$\Pr[\pi(R) = (T + 1, \tilde{a}, (a_1, \dots, a_T))] \leq e^\epsilon \Pr[\pi(R') = (T + 1, \tilde{a}, (a_1, \dots, a_T))] .$$

**Main Goal:** Design $\epsilon$-global DP $\delta$-correct BAI algorithms, with the smallest sample complexity $\tau_{\epsilon,\delta}$.

**Notation.** Let $[x]_0^1 := \max\{0, \min\{1, x\}\}$ be the clipping operator to $[0, 1]$. Let $\mathbb{1}(\cdot)$ be the indicator function. For two probability distributions $\mathbb{P}$ and $\mathbb{Q}$ on the measurable space $(\Omega, \mathcal{G})$, the Total Variation (TV) distance is $\mathrm{TV}(\mathbb{P} \| \mathbb{Q}) := \sup_{A \in \mathcal{G}}\{\mathbb{P}(A) - \mathbb{Q}(A)\}$ and the Kullback-Leibler (KL) divergence is $\mathrm{KL}(\mathbb{P} \| \mathbb{Q}) := \int \log\left(\frac{d\mathbb{P}}{d\mathbb{Q}}(\omega)\right) d\mathbb{P}(\omega)$, when $\mathbb{P} \ll \mathbb{Q}$, and $+\infty$ otherwise. The KL divergence and TV distance between two Bernoulli distributions with means $(p, q) \in (0, 1)^2$ are the relative entropy denoted by kl, i.e., $\mathrm{KL}(\mathrm{Ber}(p) \| \mathrm{Ber}(q)) = \mathrm{kl}(p, q) := p \log(p/q) + (1 - p) \log((1 - p)/(1 - q))$, and the absolute mean difference, i.e., $\mathrm{TV}(\mathrm{Ber}(p) \| \mathrm{Ber}(q)) = |p - q|$. Let $\triangle_K := \{w \in \mathbb{R}^K \mid w \geq 0, \sum_{a \in [K]} w_a = 1\}$ be the probability simplex of dimension $K - 1$. For all $a \in [K]$, let $N_{n,a} := \sum_{t \in [n-1]} \mathbb{1}(a_t = a)$ be the *global* pulling count of arm $a$ before time $n$.

## 3 Lower Bound on the Expected Sample Complexity

In order to be $\delta$-correct, an algorithm has to be able to distinguish $\boldsymbol{\nu}$ from *alternative* instances with different best arms, i.e., an instance $\boldsymbol{\kappa} \in \mathrm{Alt}(\boldsymbol{\nu}) := \{\boldsymbol{\kappa} \in \mathcal{F}^K \mid a^\star(\boldsymbol{\kappa}) \neq a^\star(\boldsymbol{\nu})\}$. On the other hand, being $\epsilon$-global DP forces an algorithm to have similar behaviour on similar instances. The interplay between the stochasticity of the bandit instance, controlled with the KL divergence, and the stochasticity of the privacy mechanism, controlled with the TV distance, is smoothly captured by

$$d_\epsilon(\nu, \kappa) := \inf_{\varphi \in \mathcal{F}} \{\epsilon \cdot \mathrm{TV}(\nu \| \varphi) + \mathrm{KL}(\varphi \| \kappa)\} , \tag{1}$$

recently introduced by Azize et al. [2025] for $\epsilon$-global DP regret minimization. The $d_\epsilon$ divergence measures the shortest two-parts path between the two distributions $\nu$ and $\kappa$, by finding the best

intermediate distribution $\varphi \in \mathcal{F}$. The cost of moving from $\nu$ to $\varphi$ is measured with the TV distance rescaled by $\epsilon$, while it is measured with the KL divergence when moving from $\varphi$ to $\kappa$.

The tension between the $\delta$-correct and $\epsilon$-global DP constraints yields the following problem-dependent non-asymptotic lower bound on the expected sample complexity $\mathbb{E}_{\nu\pi}[\tau_{\epsilon,\delta}]$ for any algorithm $\pi$ on any instance $\nu$, holding for any values of $\epsilon$ and $\delta$.

**Theorem 2.** *Let $(\epsilon, \delta) \in \mathbb{R}_+^\star \times (0,1)$. For any algorithm $\pi$ that is $\delta$-correct and $\epsilon$-global DP on $\mathcal{F}^K$,*

$$\mathbb{E}_{\nu\pi}[\tau_{\epsilon,\delta}] \geq T_\epsilon^\star(\nu)\log(1/(3\delta))$$

*for all $\nu \in \mathcal{F}^K$ with unique best arm. The inverse of the* characteristic time $T_\epsilon^\star(\nu)$ *is defined as*

$$T_\epsilon^\star(\nu)^{-1} := \sup_{w \in \triangle_K} \inf_{\kappa \in \mathrm{Alt}(\nu)} \sum_{a=1}^K w_a \mathrm{d}_\epsilon(\nu_a, \kappa_a) \,. \tag{2}$$

**Comments.** (a) The characteristic time in the lower bound is the value of a two-player zero-sum game between a MIN player, who plays instances $\kappa$ close of $\nu$ is order to confuse the MAX player, who in order plays an arm allocation $w \in \triangle_K$ to distinguish between $\nu$ and $\kappa$.

(b) The crucial part in characteristic times similar to Eq. (2) is finding the "right" measure capturing the "distinguishability" between instances. In the non-private lower bounds, this is captured by the KL divergence for parametric distributions [Garivier and Kaufmann, 2016] and by the Kinf (i.e., $\inf$ KL under mean constraint) for non-parametric distributions [Agrawal et al., 2020]. In the DP lower bounds of Azize et al. [2023], it is captured by $\min\{\mathrm{KL}, \epsilon\mathrm{TV}\}$. In Theorem 2, it is captured by $\mathrm{d}_\epsilon$ (as in Eq. (1)) that smoothly interpolates between KL and TV. Azize et al. [2025] recently introduced $\mathrm{d}_\epsilon$ for $\epsilon$-global DP regret minimization. Our results show that $\mathrm{d}_\epsilon$ also tightly captures the hardness of fixed-confidence BAI under $\epsilon$-global DP. Namely, our DP-TT algorithm achieves a matching upper bound when $\delta \to 0$ (up to a constant smaller than 8), for all instances with distinct means and all values of $\epsilon$.

(c) Azize et al. [2023, Theorem 2] provides a lower bound on the sample complexity of any $\epsilon$-global $\delta$-correct algorithm, where the inverse characteristic time is $\sup_{w \in \triangle_K} \inf_{\kappa \in \mathrm{Alt}(\nu)} \min\{\sum_{a=1}^K w_a \mathrm{KL}(\nu_a \| \kappa_a), 6\epsilon \sum_{a=1}^K w_a \mathrm{TV}(\nu_a \| \kappa_a)\}$. The lower bound of Theorem 2 is strictly tighter than that of Theorem 2 in [Azize et al., 2023], for all instances $\nu$ and values of $\epsilon$. The reason is that $\mathrm{d}_\epsilon(\mathbb{P}, \mathbb{Q}) \leq \min\{\mathrm{KL}(\mathbb{P}, \mathbb{Q}), \epsilon\mathrm{TV}(\mathbb{P}, \mathbb{Q})\}$ for any two distributions.

(d) The lower bound of Theorem 2 suggests the existence of two privacy regimes, depending on the value of $\epsilon$ and the instance $\nu$. Specifically, when $\epsilon$ is big, $\mathrm{d}_\epsilon$ reduces to the KL, and we retrieve the classic non-private lower bound. On the other hand, as $\epsilon \to 0$, $\mathrm{d}_\epsilon$ reduces to $\epsilon$ TV, and the characteristic time reduces to $\frac{1}{\epsilon}T_{\mathrm{TV}}^\star(\nu) := \frac{1}{\epsilon}\left(\sup_{w \in \triangle_K} \inf_{\kappa \in \mathrm{Alt}(\nu)} \sum_{a=1}^K w_a \mathrm{TV}(\nu_a \| \kappa_a)\}\right)^{-1} = \frac{1}{\epsilon}\sum_{a=1}^k \frac{1}{\Delta_a}$, where $\Delta_a = \mu^\star - \mu_a$ for $a \neq a^\star$ and $\Delta_{a^\star} = \min_{a \neq a^\star} \Delta_a$ . This improves the high privacy regime lower bound of prior work by a factor 6. Also, the value of $\epsilon$ at which the privacy regimes change can be tightly specified, which we quantify for Bernoulli instances in the following.

**Proof Sketch and Techniques.** The proof uses the standard reduction to hypothesis testing [Garivier and Kaufmann, 2016], using the data-processing inequality. The asymptotic techniques used by Azize et al. [2025] for regret cannot be adapted for our non-asymptotic lower bound. Thus, new techniques are needed. The main technical novelty of the proof is a tighter quantification of the "similar" behaviour of a DP mechanism when applied to stochastic datasets. Specifically, let $\mathcal{M}$ be an $\epsilon$-DP mechanism. Given two data-generating distributions $\mathbb{P}$ and $\mathbb{Q}$, letting $\mathbb{M}_{\mathbb{P},\mathcal{M}}$ (resp. $\mathbb{M}_{\mathbb{Q},\mathcal{M}}$) be the marginal over outputs of the mechanism when the input dataset is generated through $\mathbb{P}$ (resp. $\mathbb{Q}$), then we show that

$$\mathrm{KL}\left(\mathbb{M}_{\mathbb{P},\mathcal{M}} \| \mathbb{M}_{\mathbb{Q},\mathcal{M}}\right) \leq \inf_{\mathbb{L}} \left\{ \epsilon \inf_{\mathbb{C}_{\mathbb{P},\mathbb{L}}} \left\{ \mathbb{E}_{D,D'\sim\mathbb{C}_{\mathbb{P},\mathbb{L}}}\left[d_{\mathrm{Ham}}(D, D')\right]\right\} + \mathrm{KL}\left(\mathbb{L} \| \mathbb{Q}\right)\right\} \,,$$

where the first infimum is over all distributions $\mathbb{L}$ on the input space, and the second infimum is an optimal transport problem over all couplings between $\mathbb{P}$ and $\mathbb{L}$, where the cost is the Hamming distance (introduced in Definition 1). This bound of general interest could be applied to get tighter lower bounds in any DP application using stochastic inputs. For product and bandit distributions, we solve the optimal transport using maximal couplings, where the Total Variation naturally appears,

while keeping the first infimum unchanged, giving rise to the $d_\epsilon$ quantity. Finally, plugging the new upper bound on the KL in the hypothesis reduction concludes the sample complexity lower bound proof. A detailed proof and discussion of all these claims is given in Appendix D.

**Transportation Costs Based on Signed Divergences.** In non-parametric BAI, the ordering between the mean parameters is captured by a signed Kinf [Jourdan et al., 2022]. We adopt this convention by introducing a signed divergences: $d_\epsilon^+$ and $d_\epsilon^-$ defined in Lemma 3 on means rather than probability distributions, where $d_\epsilon^\pm$ refers to both of them. Given $(\kappa, \nu) \in \mathcal{F}^2$ with means $(\lambda, \mu) \in (0, 1)^2$, they satisfy $\mathrm{d}_\epsilon(\kappa, \nu) = d_\epsilon^+(\lambda, \mu)$ when $\mu > \lambda$, and $\mathrm{d}_\epsilon(\kappa, \nu) = d_\epsilon^-(\lambda, \mu)$ otherwise (Lemma 22).

**Lemma 3.** *For all $x \in [0, 1]$, let us define $g_\epsilon^-(x) := \frac{x e^\epsilon}{x(e^\epsilon - 1) + 1}$ and $g_\epsilon^+(x) := 1 - g_\epsilon^-(1 - x) = (g_\epsilon^-)^{-1}(x)$. For all $(\lambda, \mu) \in \mathbb{R} \times [0, 1]$, the signed divergences are defined as*

$$d_\epsilon^+(\lambda, \mu) := \mathbb{1}\left(\mu > [\lambda]_0^1\right) \inf_{z \in [[\lambda]_0^1, \mu]} \left\{ \mathrm{kl}(z, \mu) + \epsilon(z - [\lambda]_0^1) \right\}$$

$$= \begin{cases} 0 & \text{if } \mu \in [0, [\lambda]_0^1] \\ -\log\left(1 - \mu(1 - e^{-\epsilon})\right) - \epsilon[\lambda]_0^1 & \text{if } \mu \in (g_\epsilon^-([\lambda]_0^1), 1] \\ \mathrm{kl}(\lambda, \mu) & \text{if } \lambda \in (0, 1) \text{ and } \mu \in ([\lambda]_0^1, g_\epsilon^-([\lambda]_0^1)] \end{cases},$$

$$d_\epsilon^-(\lambda, \mu) := \mathbb{1}\left(\mu < [\lambda]_0^1\right) \inf_{z \in [\mu, [\lambda]_0^1]} \left\{ \mathrm{kl}(z, \mu) + \epsilon([\lambda]_0^1 - z) \right\} = d_\epsilon^+(1 - \lambda, 1 - \mu). \quad (3)$$

When two distributions are close enough compared to the privacy $\epsilon$, the signed divergences reduce to the KL divergence: the indistinguishability due to the stochasticity of the instance is stronger than the indistinguishability due to DP. The function $g_\epsilon^-$ (resp. $g_\epsilon^+$) represents the maximal (resp. minimal) mean for which this property hold for $d_\epsilon^+$ (resp. $d_\epsilon^-$).

For $(\mu, w) \in \mathbb{R}^K \times \mathbb{R}_+^K$, the *transportation cost* of the pair of arms $(a, b) \in [K]^2$ is defined as

$$W_{\epsilon, a, b}(\mu, w) := \mathbb{1}\left([\mu_a]_0^1 > [\mu_b]_0^1\right) \inf_{u \in [0, 1]} \left\{ w_a d_\epsilon^-(\mu_a, u) + w_b d_\epsilon^+(\mu_b, u) \right\}. \quad (4)$$

The signed divergences $d_\epsilon^\pm$ and the transportation costs $(W_{\epsilon, a, b})_{(a, b) \in [K]^2}$ satisfy all the desired properties required to study BAI algorithms based on the empirical version of $W_{\epsilon, a, b}$ (see Lemmas 23, 24, 25 and 26, as well as Lemmas 35, 36, 37 and 38), e.g., symmetry, explicit formula, monotonicity, strict convexity, etc.

**Properties of the Characteristic Time and Optimal Allocation.** The set $w_\epsilon^\star(\nu)$ of *optimal allocations* is the maximizer of the outer supremum on $\triangle_K$ that defines $T_\epsilon^\star(\nu)^{-1}$ in Eq. (2). Theorem 4 gathers key properties satisfied by $T_\epsilon^\star(\nu)$ and $w_\epsilon^\star(\nu)$, for Bernoulli distributions. See lemmas proven in Appendix G, i.e., Lemmas 36, 43 and 47.

**Theorem 4.** *Let $\nu \in \mathcal{F}^K$ having means $\mu \in (0, 1)^K$ with unique best arm $a^\star$. Then, we have*

$$T_\epsilon^\star(\nu)^{-1} = \max_{w \in \triangle_K} \min_{a \neq a^\star} W_{\epsilon, a^\star, a}(\mu, w) \quad \text{and} \quad T_\epsilon^\star(\nu) \geq \sum_{a \in [K]} \Delta_{\epsilon, a}^{-1}. \quad (5)$$

*where $\Delta_{\epsilon, a^\star} := \min_{a \neq a^\star} d_\epsilon^-(\mu_{a^\star}, \mu_a)$ and $\Delta_{\epsilon, a} := d_\epsilon^+(\mu_a, \mu_{a^\star})$ for all $a \neq a^\star$. The optimal allocation is unique, has dense support and ensures the equality of the transportation costs with $T_\epsilon^\star(\nu)^{-1}$ (i.e., information balance equation), namely $w_\epsilon^\star(\nu) = \{w_\epsilon^\star\}$, $\min_{a \in [K]} w_{\epsilon, a}^\star > 0$ and $W_{\epsilon, a^\star, a}(\mu, w_\epsilon^\star) = T_\epsilon^\star(\nu)^{-1}$ for all $a \neq a^\star$.*

In Garivier and Kaufmann [2016], the characteristic time and its optimal allocation can be computed with a simpler optimisation problem. A simpler optimization problem can also be solved to compute $T_\epsilon^\star(\nu)$ and $w_\epsilon^\star(\nu)$ explicitly (Lemma 47).

**Allocation Dependent Low Privacy Regime.** Let $(\mu, w, a, b) \in (0, 1)^K \times \mathbb{R}_+^K \times [K]^2$ such that $\mu_a > \mu_b$ and $\min\{w_a, w_b\} > 0$. The non-private Bernoulli transportation costs [Garivier and Kaufmann, 2016] are defined as

$$W_{a, b}(\mu, w) := w_a \mathrm{kl}(\mu_a, \mu_{a, b}^w) + w_b \mathrm{kl}(\mu_b, \mu_{a, b}^w) \quad \text{with} \quad \mu_{a, b}^w := \frac{w_a \mu_a + w_b \mu_b}{w_a + w_b}.$$

We provide an *allocation-dependent* low-privacy condition that depends on $(\epsilon, \mu, w)$ (Lemma 45), i.e., $W_{\epsilon, a, b}(\mu, w) = W_{a, b}(\mu, w)$ is implied by

$$\mu_a - \mu_b \leq (1 - e^{-\epsilon}) \min\left\{ (1 + w_a/w_b) \mu_a g_\epsilon^-(1 - \mu_a), (1 + w_b/w_a)(1 - \mu_b) g_\epsilon^-(\mu_b) \right\}. \quad (6)$$

Plugging $w_\epsilon^\star$ from Theorem 4 in Eq. (6) would give an implicit condition on $(\epsilon, \mu)$ under which the non-private characteristic time $T^\star(\nu)$ for Bernoulli distributions is recovered, i.e., $T_\epsilon^\star(\nu) = T^\star(\nu)$. From a privacy-utility tradeoff perspective, the choice of $\epsilon$ that provides the highest privacy protection while maintaining a low sample complexity is exactly this change-of-regime $\epsilon$, depending on the unknown $\mu$. A weaker (yet explicit) allocation-independent sufficient condition for $T_\epsilon^\star(\nu) = T^\star(\nu)$ is $\epsilon \geq \max_{a \neq a^\star} \epsilon_{a^\star,a}$ where $\epsilon_{a,b} := \log\left(\frac{\mu_a(1-\mu_b)}{\mu_b(1-\mu_a)}\right)$.

# 4   Generalized Likelihood Ratio Stopping Rule

Designing appropriate recommendation and stopping rules for the BAI problem can be framed as a sequential hypothesis testing task with multiple hypotheses $\{\mu_a = \max_{b \in [K]} \mu_b\}$. One of the earliest approaches to active hypothesis testing—where data collection is also optimized—was introduced by Chernoff [1959], who advocated for the use of Generalized Likelihood Ratio (GLR) tests for stopping decisions. This methodology is also popular in the context of BAI [Garivier and Kaufmann, 2016]. Despite its relevance, fewer works attempted to extend it for private sequential hypothesis testing, see, e.g., Zhang et al. [2022] under Rényi DP and Azize et al. [2024] under $\epsilon$-local and $\epsilon$-global DP.

**Mean Estimator.** Three rules need to be specified to define a BAI algorithm: recommendation, sampling, and stopping rules. An important remark in designing BAI algorithms is that the dependence of these rules on the private input observation dataset comes solely through the sequence of mean estimators. Thus, designing a sequence of mean estimators that satisfy DP is crucial when defining a $\epsilon$-global DP BAI algorithm. To estimate the sequence of means, defined in Lines 5-8 of Algorithm 1, we rely on two ingredients: adaptive arm-dependent episodes with a geometric update grid and the Laplace mechanism. We call this mechanism estimating the sequence of means the Geometric Private Estimator, i.e., $\mathrm{GPE}_\eta(\epsilon)$. Most notably, we eliminate "observation forgetting" from $\mathrm{GPE}_\eta(\epsilon)$, an important design choice made in all past BAI algorithms Sajed and Sheffet [2019]; Azize et al. [2023, 2024]. Specifically, for some $\eta > 0$ called the geometric grid parameter, $\mathrm{GPE}_\eta(\epsilon)$ estimates the noisy means in arm-dependent phases: a phase changes when the counts of an arm has increased multiplicatively by $1 + \eta$ (Line 5). Then, $\mathrm{GPE}_\eta(\epsilon)$ only updates the mean of the arm that changed phases, by accumulating the observations collected from its last phase and adding Laplace noise (Line 7). Due to this accumulation step, we *do not forget* the observations from past phases. Thus, each estimated noisy mean $\tilde{\mu}_{n,a}$ in Line 7 contains $\tilde{N}_{n,a}$ i.i.d. observations from $\nu_a$ and $k_{n,a} \approx \log_{1+\eta} \tilde{N}_{n,a}$ i.i.d. observations from $\mathrm{Lap}(1/\epsilon)$. In contrast, using forgetting produces a noisy mean that contains fewer i.i.d. observations from $\nu_a$ (e.g. $\tilde{N}_{n,a}/2$ samples for forgetting with $\eta = 1$), but only *one* Laplace noise. While removing forgetting allows us to keep more signal, i.e., more i.i.d samples from $\nu_a$, we need more noise, i.e., the cumulative sum of $\mathrm{Lap}(1/\epsilon)$, which is logarithmic in the number of samples from $\nu_a$. Tighter concentration inequalities allow controlling the cumulative sum of Laplace noise. See below for a detailed discussion about our novel concentration results. As long as the number of samples from the $\mathrm{Lap}(1/\epsilon)$ is logarithmic in the number of samples from $\nu_a$, the effect of noise on the sample complexity is similar to having only *one* additional Laplace noise.

**Privacy Analysis.** By adaptive post-processing, the following lemma is proved naturally for any $\epsilon$.

**Lemma 5.** *Any BAI algorithm using only $\mathrm{GPE}_\eta(\epsilon)$ to access observations is $\epsilon$-global DP on $[0, 1]$.*

**Proof Sketch.** The proof combines two steps. First, we show that the sequence of mean estimators produced by $\mathrm{GPE}_\eta(\epsilon)$ is $\epsilon$-DP. The crucial observation is that a change in one observation *only* affects the partial sum collected in just *one* arm-phase. By the Laplace mechanism, adding one $\mathrm{Lap}(1/\epsilon)$ to the partial sum is enough to make it $\epsilon$-DP. Then, by post-processing, the sequence of accumulated partial sums $(\tilde{S}_{k_{n,a},a})$ and noisy means $(\mu_{n,a})$ (Line 7) are also $\epsilon$-DP. The second step shows how to use the sequential nature of the process and adaptive post-processing to conclude that BAI algorithms using only $\mathrm{GPE}_\eta(\epsilon)$ are $\epsilon$-global DP. The detailed proof is in Appendix E.

**Recommendation Rule.** The recommendation rule $\tilde{a}_n$ is defined as the arm with the highest clipped noisy empirical mean, i.e., $\tilde{a}_n \in \arg\max_{a \in [K]} [\tilde{\mu}_{n,a}]_0^1$ where ties are broken uniformly at random.

**GLR Stopping Rule.** The GLR stopping rule runs $K$ sequential GLR tests in parallel, and stops as soon as one of these tests can reject the null hypothesis. When comparing the recommendation $\tilde{a}_n$ with an alternative arm $a$, the GLR statistic is defined as the transportation cost $W_{\epsilon,\tilde{a}_n,a}$ evaluated

empirically at $(\tilde{\mu}_n, \tilde{N}_n)$ (see Eq. (4)). Intuitively, $W_{\epsilon, \tilde{a}_n, a}(\tilde{\mu}_n, \tilde{N}_n)$ represents the amount of empirical evidence to reject the hypothesis that arm $a$ has a higher mean than $\tilde{a}_n$. One can stop and recommend $\tilde{a}_n$ when all these statistics exceed a given stopping threshold. Given a privacy budget and risk $(\epsilon, \delta) \in \mathbb{R}_+^\star \times (0, 1)$ and a stopping threshold $c : \mathbb{N} \times \mathbb{R}_+^\star \times (0, 1) \to \mathbb{R}_+$, we define

$$\tau_{\epsilon, \delta} = \inf\{\, n \mid \forall a \neq \tilde{a}_n, \ W_{\epsilon, \tilde{a}_n, a}(\tilde{\mu}_n, \tilde{N}_n) > c(\tilde{N}_{n, \tilde{a}_n}, \epsilon, \delta) + c(\tilde{N}_{n, a}, \epsilon, \delta) \,\} \,. \tag{7}$$

Given its proximity to the characteristic time $T_\epsilon^\star(\boldsymbol{\nu})$, see Eq. (5) (Theorem 4), the GLR stopping rule is a good candidate to match the lower bound, i.e., if one could sample arms according to $w_\epsilon^\star(\boldsymbol{\nu})$ and use the stopping threshold $\log(1/\delta)$. Unfortunately, this threshold is too good to be $\delta$-correct and $w_\epsilon^\star(\boldsymbol{\nu})$ should be estimated as it is unknown (Section 5).

**Calibration of the Stopping Threshold.** Regardless of the sampling rule, the stopping threshold should ensure $\delta$-correctness of the GLR stopping rule for any pair $(\epsilon, \delta)$, see Theorem 6.

**Theorem 6.** *Let $(\epsilon, \delta, \eta) \in \mathbb{R}_+^\star \times (0, 1) \times \mathbb{R}_+^\star$. Let $s > 1$, $\zeta$ be the Riemann $\zeta$ function and $\overline{W}_{-1}(x) = -W_{-1}(-e^{-x})$ for all $x \geq 1$, where $W_{-1}$ is the negative branch of the Lambert $W$ function, satisfying $\overline{W}_{-1}(x) \approx x + \log x$ (Lemma 52). Given any sampling rule using the $GPE_\eta(\epsilon)$, using the GLR stopping rule as in Eq. (7) with the $GPE_\eta(\epsilon)$ and the stopping threshold $c(n, \epsilon, \delta) := c_1(n, \delta) + c_2(n, \epsilon)$ where*

$$c_1(n, \delta) = \overline{W}_{-1}\left(\log\left(K\zeta(s)/\delta\right) + s\log(k_\eta(n)) + 3 - \log 2\right) - 3 + \log 2\,, \tag{8}$$
$$c_2(n, \epsilon) = k_\eta(n)\left(\log\left(1 + 2\epsilon n/k_\eta(n)\right) + 1\right) \quad \text{with} \quad k_\eta(x) := 1 + \log_{1+\eta} x\,,$$

*yields a $\delta$-correct and $\epsilon$-global DP algorithm for all Bernoulli instances with a unique best arm.*

The proof of Theorem 6 builds on novel concentration results of independent interest (Appendix F.2). Our explicit instance-independent upper bounds are pivotal to derive the stopping threshold in Eq. (8), which avoids the large instance-dependent constants used in the regret minimisation literature [Azize et al., 2025].

**Concentration Results.** First, we give tail bounds for the cumulative sum of i.i.d. Laplace observations (Lemma 16). We use Chernoff's method with the convex conjugate of the moment generating function of $\mathrm{Lap}(1/\epsilon)$, hence improving on Azize et al. [2025, Lemma 18] that approximates it. Second, we derive tail bounds for the sum between independent cumulative sums of $t$ i.i.d. Bernoulli and $n_t$ i.i.d. Laplace observations (Lemmas 18 and 19). They involve the *modified* signed divergences $\widetilde{d}_\epsilon^\pm$ that better capture the non-asymptotic tails behaviour, and are equivalent to $d_\epsilon^\pm$ to an additive term $\Theta(\log(1 + 2\epsilon r_t)/r_t)$ where $r_t := t/n_t$ (Lemma 30). Whenever $r_t \to +\infty$, we recover the same noise effect as adding only one $\mathrm{Lap}(1/\epsilon)$ observation. For $x > 0$, the exponential decrease of the probability of exceeding $\mu + x$ (resp. being lower than $\mu - x$) scales as $t\widetilde{d}_\epsilon^-(\mu + x, \mu, r_t)$ (resp. $t\widetilde{d}_\epsilon^+(\mu - x, \mu, r_t)$). The proof builds on fine-grained tail bounds of the sum of two independent random variables, i.e., we bound those probabilities by the maximal product between their respective survival functions (Lemma 10). While Azize et al. [2025, Lemma 19] directly integrates their tail bounds, Lemma 12 can be used with any tail bounds. Third, we obtain time-uniform upper tail bounds for $\tilde{N}_{n,a}\widetilde{d}_\epsilon^\pm(\tilde{\mu}_{n,a}, \mu_a, \tilde{N}_{n,a}/k_{n,a})$ by exploiting the geometric-grid update of $(\tilde{\mu}_n, \tilde{N}_n, k_n)$.

**Threshold Scaling.** The threshold $c_1$ in Eq. (8) ensures $\delta$-correctness of the *modified* GLR stopping rule, defined in Appendix F.1 with the *modified* transportation costs $\widetilde{W}_{\epsilon,a,b}$ and divergences $\widetilde{d}_\epsilon^\pm$ (Appendix F.1). Independent of $\epsilon$, it scales as $\log(1/\delta) + \Theta(\log\log(1/\delta))$ when $\delta \to 0$ and $\Theta(\log\log(n))$ when $n \to +\infty$. The threshold $c_2$ in Eq. (8) is an upper bound on $\tilde{N}_{n,a}(d_\epsilon^\pm(\tilde{\mu}_{n,a}, \mu_a) - \widetilde{d}_\epsilon^\pm(\tilde{\mu}_{n,a}, \mu_a, \tilde{N}_{n,a}/k_{n,a}))$ (Lemma 30) that scales as $\Theta(\epsilon n)$ when $\epsilon \to 0$ and as $\Theta((\log n)^2)$ when $n \to +\infty$. Both $c_1$ and $c_2$ scales as $\Theta(1/\log(1 + \eta))$ when $\eta \to 0$.

**Limitation.** As the threshold in Eq. (7) is the sum of per-arm thresholds, it scales as $2\log(1/\delta)$ when $\delta \to 0$, hence incurs a suboptimal factor 2 asymptotically. Obtaining a threshold in $\log(1/\delta)$ is left for future work. It requires controlling the re-weighted sum of modified divergences $\widetilde{d}_\epsilon^\pm$. Azize et al. [2023, Theorem 4] has a suboptimal factor 2 for the same reason and incurs an additive factor $\frac{1}{n\epsilon^2}\log(1/\delta)^2$ due to the separate control of the Laplace and the Bernoulli observations (based on sub-Gaussian concentration results). Azize et al. [2024, Lemma 18] alleviates this factor 2 in their low privacy regime, yet it also pays $\frac{1}{n\epsilon^2}\log(1/\delta)^2$.

---

**Algorithm 1** Differentially Private Top Two (DP-TT) Algorithm.

---

1: **Input:** setting parameters $(\epsilon, \delta) \in \mathbb{R}_+^\star \times (0,1)$, algorithmic hyperparameters $(\eta, \beta) \in \mathbb{R}_+^\star \times (0,1)$ and threshold $c$, e.g., $(\eta, \beta) = (1, 1/2)$ and $c$ as in Eq. (8). $(W_{\epsilon,a,b})_{(a,b) \in [K]}$ as in Eq. (4).

2: **Output:** Stopping time $\tau_{\epsilon,\delta}$, recommendation $\tilde{a}_{\tau_{\epsilon,\delta}}$ and pulling history $(a_n)_{n < \tau_{\epsilon,\delta}}$.

3: **Initialization:** For all $a \in [K]$, pull arm $a$, observe $X_{a,a} \sim \nu_a$ and draw $Y_{1,a} \sim \text{Lap}(1/\epsilon)$. Set $n = K + 1$. For all $a \in [K]$, set $\tilde{S}_{n,a} = X_{a,a} + Y_{1,a}$, $k_{n,a} = 1$, $T_1(a) = n$, $N_{n,a} = \tilde{N}_{n,a} = 1$, $\tilde{\mu}_{n,a} = \tilde{S}_{n,a}/\tilde{N}_{n,a}$, $L_{n,a} = 0$ and $N_{n,a}^a = 0$.

4: **for** $n \geq K + 1$ **do**

5:      **if** there exists $a \in [K]$ such that $N_{n,a} \geq (1 + \eta)^{k_{n,a}}$ **then**     ▷ Per-arm geometric update grid

6:          For this arm $a$, change phase $k_{n,a} \hookleftarrow k_{n,a} + 1$, and $(T_{k_{n,a}}(a), \tilde{N}_{n,a}) = (n, N_{T_{k_{n,a}}(a),a})$;

7:          Set $\tilde{S}_{k_{n,a},a} = \sum_{t=T_{k_{n,a}-1}(a)}^{T_{k_{n,a}}(a)-1} X_{t,a} \mathbb{1}(a_t = a) + Y_{k_{n,a},a} + \tilde{S}_{k_{n,a}-1,a}$ with $Y_{k_{n,a},a} \sim \text{Lap}(1/\epsilon)$, and update the mean $\tilde{\mu}_{n,a} = \tilde{S}_{k_{n,a},a}/\tilde{N}_{n,a}$;

8:      **end if**

9:      Set $\tilde{a}_n \in \arg\max_{a \in [K]} [\tilde{\mu}_{n,a}]_0^1$;                         ▷ Recommendation rule

10:      **if** $W_{\epsilon,\tilde{a}_n,a}(\tilde{\mu}_n, \tilde{N}_n) > \sum_{b \in \{\tilde{a}_n,a\}} c(\tilde{N}_{n,b}, \epsilon, \delta)$ for all $a \neq \hat{a}_n$ **then**     ▷ GLR stopping rule

11:          **return** $(n, \tilde{a}_n, (a_t)_{t<n})$.

12:      **end if**

13:      Set $B_n = \tilde{a}_n$ and $C_n \in \arg\min_{a \neq B_n} \{W_{\epsilon,B_n,a}(\tilde{\mu}_n, N_n) + \log N_{n,a}\}$;        ▷ EB-TCI

14:      Set $a_n = B_n$ if $N_{n,B_n}^{B_n} \leq \beta L_{n+1,B_n}$, and $a_n = C_n$ otherwise;        ▷ $\beta$-tracking

15:      Pull $a_n$, observe and store $X_{n,a_n} \sim \nu_{a_n}$;

16:      Update $(N_{n+1,a_n}, L_{n+1,B_n}, N_{n+1,B_n}^{B_n}) = (N_{n,a_n}, L_{n,B_n}, N_{n,B_n}^{B_n}) + (1, 1, \mathbb{1}(B_n = a_n))$;

17: **end for**

---

## 5 Top Two Sampling Rule

Equipped with a recommendation and stopping rules, we define a sampling rule using the $\text{GPE}_\eta(\epsilon)$. Within the fixed-confidence BAI literature, we adopt the Top Two approach [Russo, 2016; Qin et al., 2017; Shang et al., 2020; Jourdan et al., 2022] that recently received increased scrutiny due to its good theoretical guarantees [Jourdan and Degenne, 2024; You et al., 2023; Jourdan et al., 2024; Bandyopadhyay et al., 2024], competitive empirical performance, and low computational cost. The Differentially Private Top Two (DP-TT) algorithm (Algorithm 1) uses the EB-TCI-$\beta$ sampling rule [Jourdan et al., 2022]. In Appendix I, we introduce the Track-and-Stop [Garivier and Kaufmann, 2016] and LUCB [Kalyanakrishnan et al., 2012] sampling rules for fixed-confidence BAI under $\epsilon$-global DP.

After initialization, a Top Two sampling rule specifies four choices [Jourdan, 2024]: a leader arm $B_n \in [K]$, a challenger arm $C_n \in [K] \setminus \{B_n\}$, a target allocation $\beta_n(B_n, C_n) \in [0,1]$ and a mechanism to choose the next arm to sample from, i.e., $a_n \in \{B_n, C_n\}$ by using $\beta_n(B_n, C_n)$. The leader should select a good estimator of the best arm $a^\star$. We use the empirical best (EB) leader that coincides with our recommendation rule, i.e., $B_n := \tilde{a}_n$. The challenger should be a confusing alternative arm, for which the empirical evidence that the leader has a better mean is low. We use the TCI challenger [Jourdan et al., 2022] that penalizes oversampled challenger to foster implicit exploration, i.e., $C_n \in \arg\min_{a \neq B_n} \{W_{\epsilon,B_n,a}(\tilde{\mu}_n, N_n) + \log N_{n,a}\}$ where ties are broken uniformly at random. Crucially, we leverage our novel transportation costs $(W_{\epsilon,B_n,a})_{a \neq B_n}$ featuring the signed divergences $d_\epsilon^\pm$ that are evaluated empirically at $(\tilde{\mu}_n, N_n)$, see Eq. (3) and (4). The target should be chosen to balance the allocation between the leader and the challenger arms. Let $\beta \in (0,1)$, e.g., $\beta = 1/2$. We use a fixed $\beta$-design $\beta_n(B_n, C_n) := \beta$. The mechanism to choose the next arm to sample should enforce that this target is reached on average. We use $K$ independent $\beta$-tracking procedures (one per leader), i.e., $a_n = B_n$ if $N_{n,B_n}^{B_n} \leq \beta L_{n+1,B_n}$ and $a_n = B_n$ otherwise, where $N_{n,a}^a = \sum_{t \in [n-1]} \mathbb{1}((B_t, a_t) = (a,a))$ and $L_{n,a} = \sum_{t \in [n-1]} \mathbb{1}(B_t = a)$. Using Degenne et al. [2020b, Theorem 6] for each tracking procedure yields $-1/2 \leq N_{n,a}^a - \beta L_{n,a} \leq 1$ for all $a \in [K]$.

**Computational and Memory Cost.** The $\text{GPE}_\eta(\epsilon)$ sums the observations, and the recommendation and GLR stopping rules are updated when an arm is updated. Using the closed-form formula for $W_{\epsilon,a,b}$ (Lemma 38), the per-iteration computational and global memory costs of DP-TT are $\mathcal{O}(K)$.

**Asymptotic Upper Bound on the Expected Sample Complexity.** Given a fixed target $\beta$, the empirical allocation of $a^\star$ converges towards $\beta$, that differs from $w^\star_{\epsilon,a^\star}$. At best, we can estimate the $\beta$-optimal allocation $w^\star_{\epsilon,\beta}(\boldsymbol{\nu})$, i.e., maximizer of the inverse $\beta$-characteristic time $T^\star_{\epsilon,\beta}(\boldsymbol{\nu})^{-1}$ defined as in Eq. (5) with the constraint $w_{a^\star} = \beta$. While being only nearly asymptotic optimal, i.e., $T^\star_\epsilon(\boldsymbol{\nu}) = \min_{\beta \in (0,1)} T^\star_{\epsilon,\beta}(\boldsymbol{\nu})$, it satisfies $T^\star_{\epsilon,1/2}(\boldsymbol{\nu}) \leq 2T^\star_\epsilon(\boldsymbol{\nu})$ (Lemma 44).

DP-TT is $\epsilon$-global DP, $\delta$-correct and matches $T^\star_\epsilon(\boldsymbol{\nu})$ to a small constant, for any privacy budget $\epsilon$. While Theorem 7 is an asymptotic upper bound for $\delta \to 0$, it holds for any $\epsilon$.

**Theorem 7.** *Let $(\epsilon, \delta, \eta, \beta) \in \mathbb{R}^\star_+ \times (0,1) \times \mathbb{R}^\star_+ \times (0,1)$ and $c$ as in Eq. (8). The DP-TT algorithm is $\epsilon$-global DP, $\delta$-correct and satisfies that, for any Bernoulli instance $\boldsymbol{\nu}$ with distinct means $\mu \in (0,1)^K$,*

$$\limsup_{\delta \to 0} \frac{\mathbb{E}_{\boldsymbol{\nu}\pi}[\tau_{\epsilon,\delta}]}{\log(1/\delta)} \leq 2(1+\eta)T^\star_{\epsilon,\beta}(\boldsymbol{\nu}) \, .$$

*Proof.* The proof (Appendix H) builds on the unified analysis of Jourdan et al. [2022] and relies heavily on the derived regularity properties for $d^\pm_\epsilon$, $(W_{\epsilon,a,b})_{(a,b)}$, $T^\star_{\epsilon,\beta}(\boldsymbol{\nu})$ and $w^\star_{\epsilon,\beta}(\boldsymbol{\nu})$ (Appendix G). □

For $(\eta, \beta) = (1, 1/2)$, the asymptotic upper bound is $4T^\star_{\epsilon,1/2}(\boldsymbol{\nu}) \leq 8T^\star_\epsilon(\boldsymbol{\nu})$. For any privacy budget $\epsilon$, we reduced the gap between known lower and upper bounds for fixed-confidence BAI under $\epsilon$-global DP to a constant lower than 8, hence closing the open problem in Azize et al. [2024]. A discussion on how to improve this constant is deferred to Appendix C.2. Since DP-TT enjoys good empirical performance for moderate value of $\delta$ (Figure 1), adapting the non-asymptotic upper bound in Jourdan et al. [2024] to the private setting is an interesting direction for future work.

**Comparison with Azize et al. [2023, 2024].** AdaP-TT and AdaP-TT$^\star$ use the DAF($\epsilon$) estimator, GLR-inspired recommendation/stopping rules and the TTUCB [Jourdan and Degenne, 2024] sampling rule (i.e., UCB-TC-$\beta$ [Jourdan, 2024]), all based on arm-dependent doubling, forgetting and unclipped estimators. While AdaP-TT relies on the non-private *Gaussian* transportation costs, AdaP-TT$^\star$ accounts for a high privacy regime by clipping the mean gap, i.e., $(\mu_a - \mu_b)_+ \min\{3\epsilon, (\mu_a - \mu_b)_+\}$ instead of $(\mu_a - \mu_b)^2$. The AdaP-TT and AdaP-TT$^\star$ algorithms are $\epsilon$-global DP and $\delta$-correct. The sample complexity of AdaP-TT only matches the *high privacy* lower bound for instances where the means are of similar order. AdaP-TT$^\star$ improves on AdaP-TT by matching the *high privacy* lower bound for all instances with distinct means. However, both AdaP-TT and AdaP-TT$^\star$ fail to match the lower bound beyond the high-privacy regime, due to the use of non-adapted transportation costs. In contrast, DP-TT uses the $d_\epsilon$-inspired transportation costs, matching the lower bound up to a small constant for all values of $\epsilon$.

**Comparison with Sajed and Sheffet [2019].** DP-SE [Sajed and Sheffet, 2019] is an $\epsilon$-global DP version of the Successive Elimination algorithm introduced for the regret minimisation setting, modified by Azize et al. [2023] into a $\epsilon$-global and $\delta$-correct BAI algorithm. Compared to DP-TT, DP-SE is less adaptive and not anytime, since it relies on uniform sampling within each phase and the phase length depends explicitly on the risk $\delta$. The high probability upper bound on the sample complexity scales as $\mathcal{O}(\sum_{a \neq a^\star} (\Delta_a \min\{\epsilon, \Delta_a\})^{-1})$ with $\Delta_a = \mu_{a^\star} - \mu_a$. This matches the lower bound when $\epsilon \to 0$ (to a constant), but fails to recover the sample-complexity lower bound beyond this regime.

## 6 Experiments

The empirical performance of DP-TT with $(\eta, \beta) = (1, 1/2)$ is compared to AdaP-TT [Azize et al., 2023], AdaP-TT$^\star$ [Azize et al., 2024], and DP-SE [Sajed and Sheffet, 2019] on different Bernoulli instances for varying privacy budget, ranging from 0.001 to 125 (see Appendix J.1). The first instance has means $\mu_1 = (0.95, 0.9, 0.9, 0.9, 0.5)$ and the second instance has means $\mu_2 = (0.75, 0.7, 0.7, 0.7, 0.7)$. As a benchmark, we also compare to the non-private EB-TCI-$\beta$

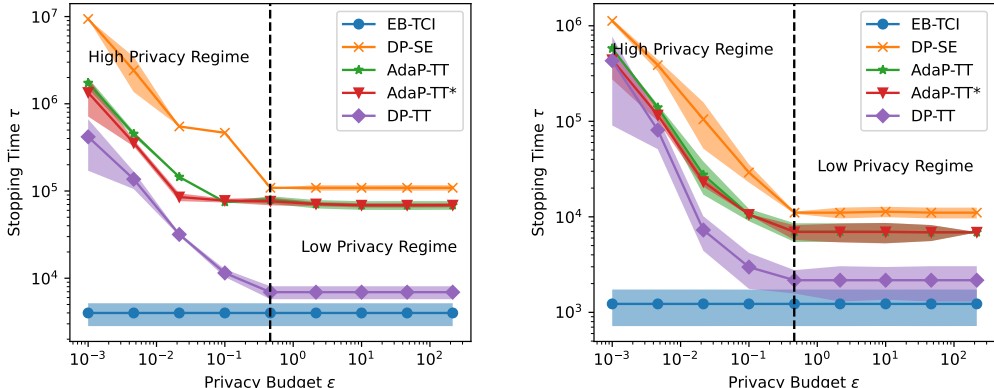

Figure 1: Empirical stopping time $\tau_{\epsilon,\delta}$ (mean $\pm 2$ std) for $\delta = 10^{-2}$ with respect to the privacy budget $\epsilon$ on Bernoulli instances (a) $\mu_1$ and (b) $\mu_2$. The vertical line separates the two privacy regimes.

algorithm with $\beta = 1/2$. For $\delta = 10^{-2}$, we run each algorithm $1000$ times, and plot the averaged empirical stopping times in Figure 1. Additional experiments are in Appendix J.

Figure 1 shows that DP-TT outperforms all the other $\delta$-correct and $\epsilon$-global DP BAI algorithms, for different values of $\epsilon$ and in all the instances tested. The empirical performance of DP-TT demonstrates two regimes. A high-privacy regime, where the stopping time depends on the privacy budget $\epsilon$, and a low privacy regime, where the performance of DP-TT is independent of $\epsilon$, and requires twice the number of samples used by the non-private EB-TCI-$\beta$. In Figure 1, the vertical lines are both at $\epsilon = 0.45$. We compare with our allocation-independent condition, i.e., $\max_{a \neq a^\star} \epsilon_{a^\star,a}$ where $\epsilon_{a,b} = \log\left(\frac{\mu_{a^\star}(1-\mu_a)}{\mu_a(1-\mu_{a^\star})}\right)$. For $\mu_2$, we have $\epsilon_{1,a} = 0.25$ for all $a \neq 1$ that is close to the empirical separation. For $\mu_1$, we have $\epsilon_{1,5} = 2.94$ and $\epsilon_{1,a} = 0.75$ for all $a \in \{2,3,4\}$. When an arm has a significantly lower mean, an allocation-dependent condition, such as Eq. (6), is required to reflect the empirical separation. Intuitively, an arm with a significantly lower mean will also be sampled significantly less, hence there is a compounding effect.

## 7 Conclusion

Motivated by the privacy requirements of sensitive applications of BAI, we address the problem of fixed-confidence BAI under $\epsilon$-global DP. We narrow the gap between the lower and upper bounds on the expected sample complexity to a multiplicative constant smaller than $8$, for all $\epsilon$ values. Our novel lower bound incorporates $d_\epsilon$ an information-theoretic quantity smoothly balancing KL divergence and TV distance, scaled by $\epsilon$. We design a private, arm-dependent geometric grid estimator *without forgetting* and a GLR stopping rule based on the $d_\epsilon$-transportation costs, whose correctness requires novel concentration results for Laplace and mixed distributions. Finally, we proposed a Top Two sampling rule that achieves an asymptotic upper bound matching our lower bound to a small constant.

We detailed research directions to further reduce the constant gap between the lower and upper bounds, by improving both the calibration of the stopping threshold and the analysis of the sampling rule. The most exciting direction for future work is to extend our results to other classes of distributions (e.g., Gaussian or bounded distributions), structured settings (e.g., linear or unimodal), or other identification problems (e.g., approximate BAI or Good Arm Identification). Another interesting research direction is to extend the proposed technique to other variants of pure DP (e.g., $(\epsilon, \delta)$-DP or Rényi DP [Mironov, 2017]) or other trust models (e.g., shuffle DP [Cheu, 2021; Girgis et al., 2021]).

## Acknowledgments and Disclosure of Funding

We thank Junya Honda for the interesting discussions. Marc Jourdan was supported by the Swiss National Science Foundation (grant number 212111) and by an unrestricted gift from Google. Achraf Azize thanks the support of the FairPlay Joint Team.

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

# A   Outline

The appendices are organized as follows:

- Notation are summarized in Appendix B.

- A detailed related work and an extended discussion is given in Appendix C.

- The lower bound on the expected sample complexity under $\epsilon$-global DP (Theorem 2) is proven in Appendix D.

- The proof of Lemma 5 is given in Appendix E.

- The proof of our concentration results are detailed in Appendix F. In particular, this includes the proof of Theorem 6.

- Appendix G gathers key properties on the (resp. modified) divergence $d_\epsilon^\pm$ (resp. $\widetilde{d}_\epsilon^\pm$), the (resp. modified) transportation costs $W_{\epsilon,a,b}$ (resp. $\widetilde{W}_{\epsilon,a,b}$) and (resp. $\beta$-)characteristic times $T_\epsilon^\star(\boldsymbol{\nu})$ (resp. $T_{\epsilon,\beta}^\star(\boldsymbol{\nu})$) and their (resp. $\beta$-)optimal allocation $w_\epsilon^\star(\boldsymbol{\nu})$ (resp. $w_{\epsilon,\beta}^\star(\boldsymbol{\nu})$). In particular, this includes the proof of Theorem 4 based on Lemmas 43 and 47.

- The proof of the upper bound on the asymptotic expected sample complexity of DP-TT (Theorem 7) is given in Appendix H.

- In Appendix I, we propose variants of algorithms to tackle $\epsilon$-global DP BAI. We aim at providing several choices for the interested practitioners.

- Implementation details and additional experiments are presented in Appendix J.

Table 1: Notation for the setting.

| Notation | Type | Description |
|---|---|---|
| $K$ | $\mathbb{N}$ | Number of arms |
| $\mathcal{F}$ | $\subseteq \mathcal{P}([0,1])$ | Class of Bernoulli distributions |
| $\nu_a$ | $\mathcal{F}$ | Bernoulli distribution of arm $a \in [K]$ |
| $\boldsymbol{\nu}$ | $\mathcal{F}^K$ | Vector of Bernoulli distributions, $\nu := (\nu_a)_{a \in [K]}$ |
| $\mu_a$ | $(0,1)$ | Mean of arm $a \in [K]$ |
| $\mu$ | $(0,1)^K$ | Vector of means, $\mu := (\mu_a)_{a \in [K]}$ |
| $a^\star(\mu), a^\star(\boldsymbol{\nu})$ | $\subseteq [K]$ | Set of best arms, $a^\star(\boldsymbol{\nu}) = a^\star(\mu) := \arg\max_{a \in [K]} \mu_a$ |
| $a^\star$ | $[K]$ | Unique best arm, i.e., $a^\star(\mu) = \{a^\star\}$ |
| $\epsilon$ | $\mathbb{R}_+^\star$ | Privacy budget for $\epsilon$-global DP |
| $\delta$ | $(0,1)$ | Risk for $\delta$-correctness |
| $\mathrm{Alt}(\boldsymbol{\nu})$ | $\subseteq \mathcal{F}^K$ | Alternative instances with different best arms |

# B   Notation

We recall some commonly used notation: the set of integers $[n] := \{1, \cdots, n\}$, the complement $X^{\complement}$ and interior $\mathring{X}$ of a set $X$, the indicator function $\mathbb{1}(X)$ of an event, the probability $\mathbb{P}_{\boldsymbol{\nu}\pi}$ and the expectation $\mathbb{E}_{\boldsymbol{\nu}\pi}$ taken over the randomness of the observations from $\boldsymbol{\nu}$ and the algorithm $\pi$, Landau's notation $o$, $\mathcal{O}$, $\Omega$ and $\Theta$, the $(K-1)$-dimensional probability simplex $\triangle_K := \left\{ w \in \mathbb{R}_+^K \mid w \geq 0, \sum_{i \in [K]} w_i = 1 \right\}$. The functions $[x]_0^1 := \max\{0, \min\{1, x\}\}$, $k_\eta(x) := 1 + \log_{1+\eta} x$, $\overline{W}_{-1}$ in Lemma 52, $h$ in Eq. (31), $r$ in Eq. (33), $\zeta$ is the Riemann $\zeta$ function. Moreover, we recall the definitions: $d_\epsilon^\pm$ in Eq. (3), $\mathrm{d}_\epsilon$ in Eq. (1), $\widetilde{d}_\epsilon^\pm$ in Eq. (32), $W_{\epsilon,a,b}^\pm$ in Eq. (4), $\widetilde{W}_{\epsilon,a,b}^\pm$ in Eq. (34), $(T_\epsilon^\star(\boldsymbol{\nu}), T_{\epsilon,\beta}^\star(\boldsymbol{\nu}), w_\epsilon^\star(\boldsymbol{\nu}), w_{\epsilon,\beta}^\star(\boldsymbol{\nu}))$ in Eq.35. While Table 1 gathers problem-specific notation, Table 2 groups notation for the algorithms.

Table 2: Notation for the algorithm.

| Notation | Type | Description |
|---|---|---|
| $B_n$ | $[K]$ | (EB) Leader at time $n$ |
| $C_n$ | $[K]$ | (TC) Challenger at time $n$ |
| $a_n$ | $[K]$ | Arm sampled at time $n$ |
| $X_{n,a_n}$ | $\{0,1\}$ | Sample observed at the end of time $n$, i.e. $X_{n,a_n} \sim \nu_{a_n}$ |
| $Y_{k_{n,a},a}$ | $\mathbb{R}$ | Noisy perturbation drawn at the beginning of phase $k_{n,a}$ for arm $a$, i.e. $Y_{k_{n,a},a} \sim \mathrm{Lap}(1/\epsilon)$ |
| $\mathcal{F}_n$ | | History before time $n$ |
| $\tilde{a}_n$ | $[K]$ | Arm recommended before time $n$ |
| $\tau_{\epsilon,\delta}$ | $\mathbb{N}$ | Sample complexity (stopping time) |
| $c(n,\epsilon,\delta)$ | $\mathbb{N} \times \mathbb{R}_+^\star \times (0,1) \to \mathbb{R}_+^\star$ | Stopping threshold function |
| $c_1(n,\delta)$ | $\mathbb{N} \times (0,1) \to \mathbb{R}_+^\star$ | Stopping threshold function |
| $c_2(n,\epsilon)$ | $\mathbb{N} \times \mathbb{R}_+^\star \to \mathbb{R}_+^\star$ | Approximation threshold function |
| $N_{n,a}$ | $\mathbb{N}$ | Number of pulls of arm $a$ before time $n$ |
| $k_{n,a}$ | $\mathbb{N}$ | Current phase of arm $a$ at time $n$ |
| $T_k(a)$ | $\mathbb{N}$ | Time $n$ where the arm $a$ changes to phase $k$ |
| $\tilde{S}_{k,a}$ | $\mathbb{R}$ | Private sum of observations for arm $a$ at phase $k$ |
| $\tilde{N}_{n,a}$ | $\mathbb{N}$ | Number of pulls of arm $a$ at the beginning of phase $k_{n,a}$ |
| $\tilde{\mu}_{n,a}$ | $\mathbb{R}$ | Private estimator of the empirical mean of arm $a$ at the beginning of phase $k_{n,a}$ |
| $L_{n,a}$ | $\mathbb{N}$ | Counts of $B_t = a$ before time $n$ |
| $N_{n,a}^a$ | $\mathbb{N}$ | Counts of $(B_t, a_t) = (a, a)$ before time $n$ |
| $\beta$ | $(0,1)$ | Fixed proportion |

## C Related Work and Extended Discussion

We provide a more detailed literature review in Appendix C.1, and an extended discussion in Appendix C.2.

### C.1 Related Work

**Structured Bandits.** While we consider unstructured bandits [Auer et al., 2002], numerous structural assumptions have been studied: linear bandits [Soare et al., 2014], generalized linear bandits [Filippi et al., 2010] such as logistic bandits [Jun et al., 2021], combinatorial bandits [Chen et al., 2013], sparse bandits [Jamieson et al., 2015], spectral bandits [Kocák and Garivier, 2021], unimodal bandits [Combes and Proutière, 2014], Lipschitz [Magureanu et al., 2014], partial monitoring [Audibert and Bubeck, 2010a], etc. Coping for the structural assumption while preserving $\epsilon$-global DP is an interesting direction for future works.

**Pure Exploration Problems.** While we consider only BAI [Even-Dar et al., 2002], other pure exploration problems have been studied in the literature: $\epsilon$-BAI [Mannor and Tsitsiklis, 2004], thresholding bandits [Carpentier and Locatelli, 2016], Top-$k$ identification [Katz-Samuels and Scott, 2019], Pareto set identification [Auer et al., 2016], best partition identification [Chen et al., 2017], etc. Extending our $\epsilon$-global DP results to answer these identification problems is an interesting research direction.

**Performance Metrics.** In pure exploration problems, the two major theoretical frameworks are the *fixed-confidence* setting [Even-Dar et al., 2006; Jamieson and Nowak, 2014; Garivier and Kaufmann, 2016], which is the focus of this paper, and the *fixed-budget* setting [Audibert et al., 2010; Gabillon et al., 2012]. In the fixed-budget setting, the objective is to minimize the probability of misidentifying a correct answer with a fixed number of samples $T$. Recent works have also considered the anytime setting, in which the agent aims at achieving a low probability of error at any deterministic time [Zhao et al., 2023; Jourdan et al., 2024]. Extending our findings to support $\epsilon$-global DP in the fixed-budget or the anytime setting is an interesting direction for future works, see e.g., Chen et al. [2024].

**DP in Bandits.** DP has been studied for multi-armed bandits under different bandit settings: finite-armed stochastic [Mishra and Thakurta, 2015; Sajed and Sheffet, 2019; Zheng et al., 2020; Hu et al., 2021; Azize and Basu, 2022; Hu and Hegde, 2022; Azize and Basu, 2024; Wang and Zhu, 2024; Hu et al., 2025], adversarial [Thakurta and Smith, 2013; Agarwal and Singh, 2017; Tossou and Dimitrakakis, 2017], linear [Hanna et al., 2024; Li et al., 2022; Azize and Basu, 2024], contextual linear [Shariff and Sheffet, 2018; Neel and Roth, 2018; Zheng et al., 2020; Azize and Basu, 2024], and kernel bandits [Pavlovic et al., 2025], among others. Most of these works were for regret minimisation, but the problem has also been explored for best-arm identification, with fixed confidence [Azize et al., 2023, 2024] and fixed budget [Chen et al., 2024]. The problem has also been studied under three different DP trust models: (a) global DP where the users trust the centralised decision maker [Mishra and Thakurta, 2015; Shariff and Sheffet, 2018; Sajed and Sheffet, 2019; Azize and Basu, 2022; Hu and Hegde, 2022], (b) local DP where each user deploys a local perturbation mechanism to send a "noisy" version of the rewards to the policy [Basu et al., 2019; Zheng et al., 2020; Han et al., 2021], and (c) shuffle DP where users still feed their data to a local perturbation, but now they trust an intermediary to apply a uniformly random permutation on all users' data before sending to the central servers [Tenenbaum et al., 2021; Garcelon et al., 2022; Chowdhury and Zhou, 2022].

In the first papers on DP for bandits, the tree-based mechanism [Dwork et al., 2010; Chan et al., 2011] was used to compute the sum of rewards privately. However, this mechanism was proven to be sub-optimal, matching the lower bounds up to logarithmic factors. Then, forgetting was first proposed by Sajed and Sheffet [2019] to get rid of the tree-based mechanism, then adapted to UCB in [Hu et al., 2021; Azize and Basu, 2022]. Finally, if the KL is the divergence that controls the complexity of bandits without privacy [Lai and Robbins, 1985; Garivier and Kaufmann, 2016], then Azize and Basu [2022] were the first to show that the TV controls the complexity of private bandits, in the high privacy regime.

In this paper, *we focus on $\epsilon$-pure DP, under a global trust model, in stochastic finite-armed bandits, for best arm identification under fixed confidence*.

**Gap in the Literature.** This problem setting is first studied by Azize et al. [2023], who proposed the first problem-dependent sample complexity lower bound, and introduced AdaP-TT, an $\epsilon$-global DP version of the Top Algorithm. However, the sample complexity upper bound of AdaP-TT only matches the lower bound in *the high privacy regime $\epsilon \to 0$*, and for instances where the means have similar order (see Condition 1 in [Azize et al., 2023] in the discussion after Theorem 5 in [Azize et al., 2023]).

Azize et al. [2024] proposes AdaP-TT$^\star$, an improved version of AdaP-TT. The improvement is achieved by using a transport inspired by the sample complexity lower bound from [Azize et al., 2023]. Using the new transport, AdaP-TT$^\star$ gets rid of Condition 1 needed by AdaP-TT, and achieves the high privacy lower bound for all instances up to a multiplicative factor 48.

However, both AdaP-TT and AdaP-TT$^\star$ do not match the lower bound, beyond the high privacy regime, i.e. for both the low privacy regime and transitional regimes.

## C.2 Extended Discussions

**Limitations of Theorem 7** Using adaptive targets $\beta_n(B_n, C_n)$ in DP-TT could replace $T^\star_{\epsilon,\beta}(\boldsymbol{\nu})$ by $T^\star_\epsilon(\boldsymbol{\nu})$, which is the optimal scaling asymptotically (Theorem 2). While we propose two adaptive choices of target based on IDS [You et al., 2023] or BOLD [Bandyopadhyay et al., 2024] (Appendix I), we leave their analysis for future work. Based on finer concentration results, the stopping threshold could scale asymptotically as $\log(1/\delta)$ instead of $2\log(1/\delta)$, hence improving by a factor 2. The assumption that the means are distinct is used to prove sufficient exploration; it can be removed by using forced exploration or a fine-grained analysis [Jourdan and Degenne, 2024; Jourdan et al., 2024]. Provided these improvements are proven, the multiplicative factor would be reduced to $1 + \eta$, where $\eta$ is the parameter that controls the geometrically increasing phases. While it improves the asymptotic upper bound, choosing $\eta$ too close to $0$ negatively impacts the performance of DP-TT, due to the dependency in $\mathcal{O}(1/\log(1+\eta))$ of the stopping threshold.

**Beyond Bernoulli Distributions** The non-asymptotic lower bound on the expected sample complexity (Theorem 2) holds for any class of distributions $\mathcal{F}$, and thus is already true beyond Bernoullis (e.g. exponential families, sub-Gaussians, etc). However, by going beyond Bernoullis, $d_\epsilon$ might not

admit a closed-form solution, e.g., the TV between Gaussian distributions has no simple formula. We conjecture that most regularity properties used to derive Theorem 4, and needed for the upper bound proofs, also hold for other classes of distributions. From a theoretical perspective, this might render the proof of the sufficient regularity properties more challenging. From a practical perspective, this increases the computational cost of the stopping rule and the challenger arm, as they require computing an empirical transportation cost based on the $d_\epsilon$ divergence. Finally, since the objective function inside the $d_\epsilon$ is continuous and convex over a compact interval, using off-the-shelf convex optimisation solvers to compute $d_\epsilon$ is still straightforward.

Our privacy analysis (Lemma 8) holds for any distribution with support in $[0, 1]$, and thus is already true beyond Bernoullis. It is straightforward to extend this to any distribution with a support in $[a, b]$, by multiplying the noise terms by the range $b - a$. Also, all prior works in bandits with DP are either for Bernoulli or bounded distributions, for the simple reason that this assumption makes estimating the empirical mean privately using the Laplace mechanism straightforward, as the sensitivity is controlled. This helps focus on the more interesting tradeoffs between DP, exploration and exploitation, without any additional technical overheads that may be introduced by estimating privately the mean of unbounded distributions.

The concentration results (Theorem 6) used to derive a stopping threshold ensuring $\delta$-correctness are highly specific to Bernoulli distributions. However, the proof builds on fine-grained tail bounds of the sum of two independent random variables, i.e., we bound those probabilities by the maximal product between their respective survival functions (Lemma 10). Therefore, the technical tools used for this intermediate result can be also used to study other one-parameter exponential families, e.g., Gaussian observation, with other noise mechanisms, e.g., Gaussian noise.

The proof of Theorem 7 (Appendix H) builds on the unified analysis of the Top Two algorithm Jourdan et al. [2022], coping with many classes of distributions, e.g., one-parameter exponential families and non-parametric bounded distributions. It relies heavily on the derived regularity properties (Appendix G) for the signed divergences, transportation costs, characteristic times, and optimal allocations. Based on the non-private BAI literature, we conjecture that most regularity properties used to derive Theorem 7 also hold for other classes of distributions. Due to the lack of explicit formulae, the proofs are challenging. The DP-TT algorithm becomes computationally costly due to the intertwined optimisation procedure when computing the transportation costs numerically.

**Beyond BAI** The technical arguments used in the proof of Theorem 2 allow obtaining a similar lower bound for (i) one-parameter exponential families, e.g., Gaussian, (ii) pure exploration problems with a unique correct answer, e.g., top-$m$ best arm identification, and (iii) structured instances without arm correlation, e.g., unimodal bandits. This is straightforwardly done by adapting the definition of $\text{Alt}(\boldsymbol{\nu})$, $a^\star(\boldsymbol{\nu})$, and $\mathcal{F}$ in Eq. (2). For settings that do not fall into the above three categories, the characteristic time requires more subtle modifications; yet our intermediate technical results can still be used. Based on the non-private fixed-confidence literature, we conjecture the changes of characteristic times described below.

- For bounded distribution (non-parametric) or Gaussian with unknown variance (two-parameters exponential family), the KL should be replaced by an infimum over KL Agrawal et al. [2020]; Jourdan et al. [2023]. The plurality of distributions having the same mean implies the existence of a distribution that is the most confusing given a specific constraint on the mean, captured by the Kinf.

- For $\gamma$-Best Arm Identification (multiple correct answers), an outer maximization over all the correct answers should be added Degenne and Koolen [2019]. The plurality of correct answers implies the existence of a correct answer that is the easiest to verify.

- For linear bandits (correlated means), the $\inf$ in the definition of $d_\epsilon$ per-arm should be replaced by a joint infimum over all the arms and moved outside the sum over arms Degenne et al. [2020a].

**Beyond $\epsilon$-DP** Our lower bound technique (Theorem 2) can be seen as a generalisation of the packing argument, and thus depends on the group privacy property. Specifically, in Appendix D, just before Eq. (11), we used the group privacy of $\epsilon$-DP to bound $\text{KL}(\mathcal{M}_D, \mathcal{M}_{D'}) \leq \epsilon\text{dham}(D, D')$.

- For $\rho$-zCDP, there is a similar group privacy property that can directly be plugged here, stating that $\mathrm{KL}(\mathcal{M}_D, \mathcal{M}_{D'}) \leq \rho \mathrm{dham}(D, D')^2$. This means that for $\rho$-zCDP, $\epsilon \times \mathrm{dham}$ is replaced by $\rho \times \mathrm{dham}^2$ in the fundamental optimal transport inequality of Eq. (11). We leave solving tightly this new optimal transport for future work.

- For $(\epsilon, \delta)$-DP, the group privacy property is not tight. This is a classic issue in $(\epsilon, \delta)$-DP lower bounds, where other techniques are used (e.g., fingerprinting). Adapting these techniques for bandits is an interesting open problem (even for bandits with regret minimisation).

It is straightforward to make DP-TT achieve either $\rho$-zCDP or $(\epsilon, \delta)$-DP, by replacing the Laplace mechanism with the Gaussian mechanism. To design a correct stopping rule with the Gaussian mechanism, it is possible to use the same fine-grained tail bounds of the sum of two independent random variables (Lemma 10) by only replacing the concentration of Laplace random variables with the concentration of Gaussian random variables. In the Top Two family of algorithms, an important design choice is the transportation cost, used for stopping and sampling the challenger. DP-TT uses a $d_\epsilon$ based transport, inspired by the lower bound of Theorem 2. Thus, to go beyond $\epsilon$-DP for algorithm design, it is important to derive a tight lower bound that suggests the use of a new transportation cost.

**Behavior with Near-optimal Arms**  The unique best arm assumption is standard in the fixed-confidence BAI setting. Even in the non-private setting, this assumption is necessary to obtain a $\delta$-correct algorithm with finite expected stopping time, as the characteristic time becomes infinite otherwise. In the private setting, this phenomenon persists as can be seen by the lower bound in Eq. (5), i.e., $T_\epsilon^\star(\nu) \to +\infty$ when $\Delta_{\epsilon, a^\star} \to 0$. In near-tie scenarios, DP-TT will perform as badly as any other $\delta$-correct and $\epsilon$-global DP algorithms due to the fundamental lower bound. Empirically, this will be reflected by empirical transportation costs that are too small for the stopping condition to be met. In applications where the near-tie scenario is common, practitioners consider $\gamma$-Best Arm Identification, i.e., identify an arm that is $\gamma$-close to the best arm. For the non-private fixed-confidence $\gamma$-BAI setting, we refer to Jourdan et al. [2024] for references and guarantees satisfied by Top Two algorithms. To tackle $\gamma$-BAI, DP-TT should be modified by using the appropriate transportation costs in both the stopping rule and the definition of the challenger, i.e., $W_{\epsilon, \gamma, a, b}$ instead of $W_{\epsilon, a, b}$.

**Implications of Not Forgetting**  DP-TT has the same memory complexity as the forgetting algorithms. The reason is that DP-TT only needs to store one accumulated noisy sum of rewards across phases, while the forgetting ones store the noisy sum of rewards of only the last phase. For privacy considerations, under our threat model where the adversary only observes the output of the algorithm (sequence of actions, final recommendation and stopping time), both forgetting and non-forgetting algorithms provide the same DP guarantees thanks to post-processing.

One can imagine other threat models where forgetting the rewards provides better privacy guarantees. One possible example of this is the pan-private threat model Dwork et al. [2010], where the adversary can also intrude into the internal states of the algorithm during its execution. In this threat model, if an adversary observes the internal states of the execution of DP-TT at some phase $\ell > 1$, they can see the full sum of rewards and maybe infer the membership of, say, the first user (up to tradeoffs guaranteed by the $\epsilon$-DP constraint, since the sum is noisy). However, for the forgetting algorithms, if the adversary observes the internal states of the execution of forgetting algorithms at some phase $\ell > 1$, the adversary can infer nothing about the reward of the first patient, since its reward has been deleted completely.

## D   Lower Bound

Let $\mathcal{M} : \mathcal{X}^n \to \mathcal{O}$ be an $\epsilon$-DP mechanism. For $D \in \mathcal{X}^n$ an input dataset, we denote by $\mathcal{M}_D$ the distribution over outputs, when the input is $D$, and $\mathcal{M}_D(E)$ the probability of observing output $E$ when the input is $D$.

Let $\mathbb{P}$ and $\mathbb{Q}$ be two data-generating distributions over $\mathcal{X}^n$. We denote by $\mathbb{M}_{\mathbb{P}, \mathcal{M}}$ the marginal over outputs of the mechanism $\mathcal{M}$ when the input dataset is generated through $\mathbb{P}$, i.e.

$$\mathbb{M}_{\mathbb{P}, \mathcal{M}}(A) := \int_{D \in \mathcal{X}^n} \mathcal{M}_D(A) \, \mathrm{d}\mathbb{P}(D) \,, \tag{9}$$

for any event $A$ in the output space. We define similarly $\mathbb{M}_{\mathbb{Q}, \mathcal{M}}$ the marginal over outputs of the mechanism $\mathcal{M}$ when the input dataset is generated through $\mathbb{Q}$.

The main question is to control the divergence $\text{KL}\left(\mathbb{M}_{\mathbb{P},\mathcal{M}} \parallel \mathbb{M}_{\mathbb{Q},\mathcal{M}}\right)$ when the mechanism $\mathcal{M}$ satisfies DP. In general, for any mechanism $\mathcal{M}$, the data-processing inequality provides the following bound

$$\text{KL}\left(\mathbb{M}_{\mathbb{P},\mathcal{M}} \parallel \mathbb{M}_{\mathbb{Q},\mathcal{M}}\right) \leq \text{KL}\left(\mathbb{P} \parallel \mathbb{Q}\right) . \tag{10}$$

Now, for $\epsilon$-DP mechanisms, we want to translate the DP constraint to a tight bound on the divergence $\text{KL}\left(\mathbb{M}_{\mathbb{P},\mathcal{M}} \parallel \mathbb{M}_{\mathbb{Q},\mathcal{M}}\right)$. To do so, let $\mathbb{L}$ be any other distribution on $\mathcal{X}^n$. Let $\mathbb{C}_{\mathbb{P},\mathbb{L}}$ be a coupling of $(\mathbb{P}, \mathbb{L})$, i.e., the marginals of $\mathbb{C}_{\mathbb{P},\mathbb{L}}$ are $\mathbb{P}$ and $\mathbb{L}$. We can now rewrite our the marginals using the definition of couplings. For $\mathbb{M}_{\mathbb{P},\mathcal{M}}$, we have

$$\mathbb{M}_{\mathbb{P},\mathcal{M}}(A) := \int_{D \in \mathcal{X}^n} \mathcal{M}_D(A) \, d\mathbb{P}(D) = \int_{D,D' \in \mathcal{X}^n} \mathcal{M}_D(A) \, d\mathbb{C}_{\mathbb{P},\mathbb{L}}(D, D') \, ,$$

and for $\mathbb{Q}$ we get

$$\mathbb{M}_{\mathbb{Q},\mathcal{M}}(A) := \int_{D' \in \mathcal{X}^n} \mathcal{M}_{D'}(A) \, d\mathbb{Q}(D') = \int_{D' \in \mathcal{X}^n} \mathcal{M}_{D'}(A) \frac{d\mathbb{Q}(D')}{d\mathbb{L}(D')} \, d\mathbb{L}(D')$$

$$= \int_{D,D' \in \mathcal{X}^n} \mathcal{M}_{D'}(A) \frac{d\mathbb{Q}(D')}{d\mathbb{L}(D')} \, d\mathbb{C}_{\mathbb{P},\mathbb{L}}(D, D') \, .$$

Using the data-processing inequality, we get

$$\text{KL}\left(\mathbb{M}_{\mathbb{P},\mathcal{M}} \parallel \mathbb{M}_{\mathbb{Q},\mathcal{M}}\right) \leq \int_{D,D' \in \mathcal{X}^n} \int_{o \in \mathcal{O}} \log\left(\frac{\mathcal{M}_D(o)}{\mathcal{M}_{D'}(o)\frac{d\mathbb{Q}(D')}{d\mathbb{L}(D')}}\right) \mathcal{M}_D(o) \, do \, d\mathbb{C}_{\mathbb{P},\mathbb{L}}(D, D')$$

$$= \int_{D,D' \in \mathcal{X}^n} \left(\text{KL}\left(\mathcal{M}_D \parallel \mathcal{M}_{D'}\right) + \log\left(\frac{d\mathbb{L}(D')}{d\mathbb{Q}(D')}\right)\right) d\mathbb{C}_{\mathbb{P},\mathbb{L}}(D, D')$$

$$= \mathbb{E}_{D,D' \sim \mathbb{C}_{\mathbb{P},\mathbb{L}}}\left[\text{KL}\left(\mathcal{M}_D \parallel \mathcal{M}_{D'}\right)\right] + \text{KL}\left(\mathbb{L} \parallel \mathbb{Q}\right) \, .$$

Since this is true for any coupling $\mathbb{C}_{\mathbb{P},\mathbb{L}}$ and any distribution $\mathbb{L}$, we get the final bound

$$\text{KL}\left(\mathbb{M}_{\mathbb{P},\mathcal{M}} \parallel \mathbb{M}_{\mathbb{Q},\mathcal{M}}\right) \leq \inf_{\mathbb{L} \in \mathcal{P}(\mathcal{X}^n)} \left\{\inf_{\mathbb{C}_{\mathbb{P},\mathbb{L}} \in \mathcal{C}(\mathbb{P},\mathbb{L})} \left\{\mathbb{E}_{D,D' \sim \mathbb{C}_{\mathbb{P},\mathbb{L}}}\left[\text{KL}\left(\mathcal{M}_D \parallel \mathcal{M}_{D'}\right)\right]\right\} + \text{KL}\left(\mathbb{L} \parallel \mathbb{Q}\right)\right\}$$

where $\mathcal{P}(\mathcal{X}^n)$ is the set of all distributions over $\mathcal{X}^n$ and $\mathcal{C}(\mathbb{P},\mathbb{L})$ is the set of all couplings between $\mathbb{P}$ and $\mathbb{L}$. Using that the $\mathcal{M}$ is $\epsilon$-DP, we can use the simple bound $\text{KL}\left(\mathcal{M}_D \parallel \mathcal{M}_{D'}\right) \leq \epsilon d_{\text{Ham}}(D, D')$ which gives

$$\text{KL}\left(\mathbb{M}_{\mathbb{P},\mathcal{M}} \parallel \mathbb{M}_{\mathbb{Q},\mathcal{M}}\right) \leq \inf_{\mathbb{L} \in \mathcal{P}(\mathcal{X}^n)} \left\{\epsilon \inf_{\mathbb{C}_{\mathbb{P},\mathbb{L}} \in \mathcal{C}(\mathbb{P},\mathbb{L})} \left\{\mathbb{E}_{D,D' \sim \mathbb{C}_{\mathbb{P},\mathbb{L}}}\left[d_{\text{Ham}}(D, D')\right]\right\} + \text{KL}\left(\mathbb{L} \parallel \mathbb{Q}\right)\right\} \, . \tag{11}$$

### D.1   Product Distributions

Suppose that $\mathbb{P} := \bigotimes_{i=1}^n \mathbb{P}_i$ and $\mathbb{Q} := \bigotimes_{i=1}^n \mathbb{Q}_i$ are product distributions. Consider the subset of product distributions $\mathbb{L} := \bigotimes_{i=1}^n \mathbb{L}_i$, and the maximal coupling $\mathbb{C}_\infty(\mathbb{P},\mathbb{L}) := \prod_{i=1}^n \mathbb{C}_\infty(\mathbb{P}_i,\mathbb{L}_i)$. Plugging these in Equation (11), we get

$$\text{KL}\left(\mathbb{M}_{\mathbb{P},\mathcal{M}} \parallel \mathbb{M}_{\mathbb{Q},\mathcal{M}}\right) \leq \inf_{\mathbb{L}_1,\dots,\mathbb{L}_n} \left\{\epsilon \sum_{i=1}^n \mathbb{E}_{D_i,D_i' \sim \mathbb{C}_\infty(\mathbb{P}_i,\mathbb{L}_i)}\left[\mathbb{1}\left\{D_i \neq D_i'\right\}\right] + \sum_{i=1}^n \text{KL}\left(\mathbb{L}_i \parallel \mathbb{Q}_i\right)\right\}$$

$$= \inf_{\mathbb{L}_1,\dots,\mathbb{L}_n} \left\{\sum_{i=1}^n \left(\epsilon \, \text{TV}\left(\mathbb{P}_i \parallel \mathbb{L}_i\right) + \text{KL}\left(\mathbb{L}_i \parallel \mathbb{Q}_i\right)\right)\right\}$$

$$= \sum_{i=1}^n \inf_{\mathbb{L}_i \in \mathcal{P}(\mathcal{X})} \left\{\epsilon \, \text{TV}\left(\mathbb{P}_i \parallel \mathbb{L}_i\right) + \text{KL}\left(\mathbb{L}_i \parallel \mathbb{Q}_i\right)\right\} = \sum_{i=1}^n d_\epsilon(\mathbb{P}_i, \mathbb{Q}_i) \, ,$$

where

$$d_\epsilon(\mathbb{P}, \mathbb{Q}) := \inf_{\mathbb{L} \in \mathcal{P}(\mathcal{X})} \left\{\epsilon \, \text{TV}\left(\mathbb{P} \parallel \mathbb{L}\right) + \text{KL}\left(\mathbb{L} \parallel \mathbb{Q}\right)\right\} \, . \tag{12}$$

---

**Algorithm 2** Bandit interaction between a policy and an environment

---

1: **Input:** A policy $\pi$ and an environment $\nu \triangleq (P_a : a \in [K])$
2: **for** $t = 1, \ldots$ **do**
3:      The policy samples an action $a_t \sim \pi_t(. \mid a_1, r_1, \ldots, a_{t-1}, r_{t-1})$
4:      The policy observes a reward $r_t \sim P_{a_t}$
5: **end for**
6: **if** Regret minimisation **then**
7:      The interaction ends after $T$ steps
8: **else** FC-BAI
9:      The policy decides to stop the interaction at step $\tau_{\epsilon,\delta}$ and recomends the final guess $\hat{a}$
10: **end if**

---

### D.2 Sequential KL decomposition for bandits under DP

In this section, we adapt the techniques from product distributions to bandit marginals.

First, we introduce the bandit canonical model.

**The bandit canonical model.** A stochastic bandit (or environment) is a collection of distributions $\nu \triangleq (P_a : a \in [K])$, where $[K]$ is the set of available $K$ actions. The learner and the environment interact sequentially over $T$ rounds. In each round $t \in 1, \ldots, T$, the learner chooses an action $a_t \in [K]$, which is fed to the environment. The environment then samples a reward $r_t \in \mathbb{R}$ from distribution $P_{a_t}$ and reveals $r_t$ to the learner. The interaction between the learner (or policy) and environment induces a probability measure on the sequence of outcomes $H_T \triangleq (a_1, r_1, a_2, r_2, \ldots, a_T, r_T)$. In the following, we construct the probability space that carries these random variables.

Let $T \in \mathbb{N}^\star$ be the horizon. Let $\nu = (P_a : a \in [K])$ a bandit instance with $K \in \mathbb{N}^\star$ finite arms, and each $P_a$ is a probability measure on $(\mathbb{R}, \mathcal{B}(\mathbb{R}))$ with $\mathfrak{B}$ being the Borel set. For each $t \in [T]$, let $\Omega_t = ([K] \times \mathbb{R})^t \subset \mathbb{R}^{2t}$ and $\mathcal{F}_t = \mathfrak{B}(\Omega_t)$. We first formalise the definition of a policy.

**Definition 2** (The policy)**.** *A policy $\pi$ is a sequence $(\pi_t)_{t=1}^T$, where $\pi_t$ is a probability kernel from $(\Omega_t, \mathcal{F}_t)$ to $([K], 2^{[K]})$. Since $[K]$ is discrete, we adopt the convention that for $a \in [K]$,*

$$\pi_t(a \mid a_1, r_1, \ldots, a_{t-1}, r_{t-1}) = \pi_t(\{a\} \mid a_1, r_1, \ldots, a_{t-1}, r_{t-1})$$

We want to define a probability measure on $(\Omega_T, \mathcal{F}_T)$ that respects our understanding of the sequential nature of the interaction between the learner and a stationary stochastic bandit. Specifically, the sequence of outcomes should satisfy the following two assumptions:

(a) The conditional distribution of action $a_t$ given $a_1, r_1, \ldots, a_{t-1}, r_{t-1}$ is $\pi(a_t \mid H_{t-1})$ almost surely.

(b) The conditional distribution of reward $r_t$ given $a_1, r_1, \ldots, a_{t-1}, r_{t-1}, a_t$ is $P_{a_t}$ almost surely.

The probability measure on $(\Omega_T, \mathcal{F}_T)$ depends on both the environment $\nu$ and the policy $\pi$. To construct this probability, let $\lambda$ be a $\sigma$-finite measure on $(\mathbb{R}, \mathcal{B}(\mathbb{R}))$ for which $P_a$ is absolutely continuous with respect to $\lambda$ for all $a \in [K]$. Let $p_a = dP_a/d\lambda$ be the Radon–Nikodym derivative of $P_a$ with respect to $\lambda$. Letting $\rho$ be the counting measure with $\rho(B) = |B|$, the density $p_{\nu\pi} : \Omega_T \to \mathbb{R}$ can now be defined with respect to the product measure $(\rho \times \lambda)^T$ by

$$p_{\nu\pi}(a_1, r_1, \ldots, a_T, r_T) \triangleq \prod_{t=1}^T \pi_t(a_t \mid a_1, r_1, \ldots, a_{t-1}, r_{t-1}) p_{a_t}(r_t)$$

and $\mathcal{P}_{\nu\pi}$ is defined as

$$\mathbb{P}_{\nu\pi}(B) \triangleq \int_B p_{\nu\pi}(\omega)(\rho \times \lambda)^T(d\omega) \quad \text{for all } B \in \mathcal{F}_T$$

Hence $(\Omega_T, \mathcal{F}_T, \mathbb{P}_{\nu\pi})$ is a probability space over histories induced by the interaction between $\pi$ and $\nu$. We define also a marginal distribution over the sequence of actions by

$$m_{\nu\pi}(a_1, \ldots, a_T) \triangleq \int_{r_1,\ldots,r_T} p_{\nu\pi}(a_1, r_1, \ldots, a_T, r_T)\, dr_1 \ldots dr_T,$$

and forall $C \in \mathcal{P}([K]^T)$,

$$\mathbb{M}_{\nu\pi}(C) \triangleq \sum_{(a_1,\ldots,a_T)\in C} m_{\nu\pi}(a_1, a_2, \ldots, a_T).$$

Finally, $([K]^T, \mathcal{P}([K]^T), \mathbb{M}_{\nu\pi})$ is a probability space over sequence of actions produced when $\pi$ interacts with $\nu$ for $T$ time-steps.

**The KL upper bound.** Now, we adapt the techniques for the bandit marginals. Let $\nu = \{P_a, a \in [K]\}$ and $\nu' = \{P'_a, a \in [K]\}$ be two bandit instances in $\mathcal{F}^K$. We recall that, when the policy $\pi$ interacts with the bandit instance $\nu$, it induces a marginal distribution $\mathbb{M}_{\nu\pi}$ over the sequence of actions. We define $\mathbb{M}_{\nu'\pi}$ similarly.

The goal is to upper bound the quantity $\mathrm{KL}\left(\mathbb{M}_{\nu\pi} \parallel \mathbb{M}_{\nu'\pi}\right)$. The marginals $\mathbb{M}_{\nu\pi}$ and $\mathbb{M}_{\nu\pi}$ in the sequential setting "look like" marginals generated by "product distributions". However, the hardness of the sequential setting lies in the fact that the data-generating distributions depend on the stochastic sequential actions chosen. Thus, the results of the previous section cannot be directly applied. To adapt the proof ideas of the previous section to the bandit case, we introduce the idea of a coupled bandit instance.

Let $\nu'' = \{P''_a : a \in [K]\}$ be any "intermediary" bandit instance from $\mathcal{F}^K$. Define $c_a$ as the maximal coupling between $P_a$ and $P''_a$, i.e., $c_a := \mathbb{C}_\infty(P_a, P''_a)$. Fix a policy $\pi = \{\pi_t\}_{t=1}^T$.

Here, we build a coupled environment $\gamma$ of $\nu$ and $\nu''$. The policy $\pi$ interacts with the coupled environment $\gamma$ up to a given time horizon $T$ to produce an augmented history $\{(a_t, r_t, r''_t)\}_{t=1}^T$. The iterative steps of this interaction process are:

1. The probability of choosing an action $a_t = a$ at time $t$ is dictated only by the policy $\pi_t$ and $a_1, r_1, a_2, r_2, \ldots, a_{t-1}, r_{t-1}$, *i.e.* the policy ignores $\{r''_s\}_{s=1}^{t-1}$.
2. The distribution of rewards $(r_t, r''_t)$ is $c_{a_t}$ and is conditionally independent of the previous observed history $\{(a_s, r_s, r''_s)\}_{t=1}^{t-1}$.

This interaction is similar to the interaction process of policy $\pi$ with the first bandit instance $\nu$, with the addition of sampling an extra $r''_t$ from the coupling of $P_{a_t}$ and $P''_{a_t}$.

The distribution of the augmented history induced by the interaction of $\pi$ and the coupled environment can be defined as

$$p_{\gamma\pi}(a_1, r_1, r''_1 \ldots, a_T, r_T, r''_T) := \prod_{t=1}^T \pi_t(a_t \mid a_1, r_1, \ldots, a_{t-1}, r_{t-1}) c_{a_t}(r_t, r''_t).$$

To simplify the notation, let $\mathbf{a} := (a_1, \ldots, a_T)$, $\mathbf{r} := (r_1, \ldots, r_T)$, $\mathbf{r'} := (r'_1, \ldots, r'_T)$ and $\mathbf{r''} := (r'_1, \ldots, r'_T)$. Also, let $c_{\mathbf{a}}(\mathbf{r}, \mathbf{r''}) := \prod_{t=1}^T c_{a_t}(r_t, r''_t)$ and $\pi(\mathbf{a} \mid \mathbf{r}) := \prod_{t=1}^T \pi_t(a_t \mid a_1, r_1, \ldots, a_{t-1}, r_{t-1})$. We put $\mathbf{h} := (\mathbf{a}, \mathbf{r}, \mathbf{r''})$. With the new notation

$$p_{\gamma\pi}(\mathbf{a}, \mathbf{r}, \mathbf{r''}) := \pi(\mathbf{a} \mid \mathbf{r}) c_{\mathbf{a}}(\mathbf{r}, \mathbf{r''}).$$

By the definition of the couplings, we have that $m_{\nu\pi}$ is the marginal of $p_{\gamma\pi}$ when integrated over $(\mathbf{r}, \mathbf{r''})$, i.e.,

$$m_{\nu\pi}(\mathbf{a}) = \int_{\mathbf{r},\mathbf{r''}} p_{\gamma\pi}(\mathbf{a}, \mathbf{r}, \mathbf{r''}) \, \mathrm{d}\mathbf{r} \, \mathrm{d}\mathbf{r''}.$$

Now, we define a new joint distribution $q_{\gamma\pi}$, inspired by the techniques used for product distributions:

$$q_{\gamma\pi}(\mathbf{a}, \mathbf{r}, \mathbf{r''}) := \pi(\mathbf{a} \mid \mathbf{r''}) \frac{p'_{\mathbf{a}}(\mathbf{r''})}{p''_{\mathbf{a}}(\mathbf{r''})} c_{\mathbf{a}}(\mathbf{r}, \mathbf{r''}),$$

where $p'_{\mathbf{a}}(\mathbf{r''}) := \prod_{t=1}^T p'_{a_t}(r''_t)$, and similarly, $p''_{\mathbf{a}}(\mathbf{r''}) := \prod_{t=1}^T p''_{a_t}(r''_t)$.

First, observe that it is indeed a valid joint distribution, i.e.

$$\sum_{\mathbf{a}} \int_{\mathbf{r},\mathbf{r''}} q_{\gamma\pi}(\mathbf{a}, \mathbf{r}, \mathbf{r''}) \, \mathrm{d}\mathbf{r} \, \mathrm{d}\mathbf{r''} = \sum_{\mathbf{a}} \int_{\mathbf{r},\mathbf{r''}} \pi(\mathbf{a} \mid \mathbf{r''}) \frac{p'_{\mathbf{a}}(\mathbf{r''})}{p''_{\mathbf{a}}(\mathbf{r''})} c_{\mathbf{a}}(\mathbf{r}, \mathbf{r''}) \, \mathrm{d}\mathbf{r} \, \mathrm{d}\mathbf{r''}$$

$$= \sum_{\mathbf{a}} \int_{\mathbf{r}"} \pi(\mathbf{a} \mid \mathbf{r}") p'_{\mathbf{a}}(\mathbf{r}") \, d\mathbf{r}" = \int_{\mathbf{r}"} p'_{\mathbf{a}}(\mathbf{r}") \, d\mathbf{r}" = 1 \,,$$

and that $m_{\nu'\pi}$ is the marginal of $q_{\gamma\pi}$ when integrated over $(\mathbf{r}, \mathbf{r}")$, i.e.,

$$\int_{\mathbf{r},\mathbf{r}"} q_{\gamma\pi}(\mathbf{a}, \mathbf{r}, \mathbf{r}") \, d\mathbf{r} \, d\mathbf{r}" = \int_{\mathbf{r},\mathbf{r}"} \pi(\mathbf{a} \mid \mathbf{r}") \frac{p'_{\mathbf{a}}(\mathbf{r}")}{p''_{\mathbf{a}}(\mathbf{r}")} c_{\mathbf{a}}(\mathbf{r}, \mathbf{r}") \, d\mathbf{r} \, d\mathbf{r}"$$

$$= \int_{\mathbf{r}"} \pi(\mathbf{a} \mid \mathbf{r}") p'_{\mathbf{a}}(\mathbf{r}") \, d\mathbf{r}" = m_{\nu'\pi}(\mathbf{a}) \,.$$

Using the data-processing inequality, we get

$$\mathrm{KL} \left( \mathbb{M}_{\nu\pi} \mid\mid \mathbb{M}_{\nu'\pi} \right) \le \mathrm{KL} \left( p_{\gamma\pi} \mid\mid q_{\gamma\pi} \right) \,. \tag{13}$$

Now, we compute

$$\mathrm{KL} \left( p_{\gamma\pi} \mid\mid q_{\gamma\pi} \right) \overset{(a)}{=} \mathbb{E}_{\mathbf{h}:=(\mathbf{a},\mathbf{r},\mathbf{r}")\sim p_{\gamma\pi}} \left[ \log \left( \frac{\pi(\mathbf{a} \mid \mathbf{r}) c_{\mathbf{a}}(\mathbf{r}, \mathbf{r}")}{\pi(\mathbf{a} \mid \mathbf{r}") \frac{p'_{\mathbf{a}}(\mathbf{r}")}{p''_{\mathbf{a}}(\mathbf{r}")} c_{\mathbf{a}}(\mathbf{r}, \mathbf{r}")} \right) \right]$$

$$\overset{(b)}{\le} \mathbb{E}_{\mathbf{h}:=(\mathbf{a},\mathbf{r},\mathbf{r}")\sim p_{\gamma\pi}} \left[ \epsilon d_{\mathrm{Ham}}(\mathbf{r}, \mathbf{r}") + \log \left( \frac{p''_{\mathbf{a}}(\mathbf{r}")}{p'_{\mathbf{a}}(\mathbf{r}")} \right) \right]$$

$$\overset{(c)}{=} \sum_{t=1}^{T} \mathbb{E}_{\mathbf{h}\sim p_{\gamma\pi}} \left[ \epsilon \mathbb{1} \left\{ r_t \ne r''_t \right\} + \log \left( \frac{p''_{a_t}(r''_t)}{p'_{a_t}(r''_t)} \right) \right]$$

$$\overset{(d)}{=} \sum_{t=1}^{T} \mathbb{E}_{\mathbf{h}\sim p_{\gamma\pi}} \left[ \mathbb{E}_{\mathbf{h}\sim p_{\gamma\pi}} [ \epsilon \mathbb{1} \left\{ r_t \ne r''_t \right\} + \log \left( \frac{p''_{a_t}(r''_t)}{p'_{a_t}(r''_t)} \right) \mid a_t ] \right]$$

$$\overset{(e)}{=} \sum_{t=1}^{T} \mathbb{E}_{\mathbf{h}\sim p_{\gamma\pi}} \left[ \epsilon \mathrm{TV} \left( p_{a_t} \mid\mid p''_{a_t} \right) + \mathrm{KL} \left( p''_{a_t} \mid\mid p'_{a_t} \right) \right]$$

$$\overset{(f)}{=} \mathbb{E}_{\nu\pi} \left[ \sum_{t=1}^{T} \epsilon \mathrm{TV} \left( p_{a_t} \mid\mid p''_{a_t} \right) + \mathrm{KL} \left( p''_{a_t} \mid\mid p'_{a_t} \right) \right] \,.$$

where:

(a) by the definition of the KL

(b) the group privacy property, applied to the $\epsilon$-global DP policy, we have

$$\pi(\mathbf{a} \mid \mathbf{r}) \le e^{\epsilon d_{\mathrm{Ham}}(\mathbf{r},\mathbf{r}")} \pi(\mathbf{a} \mid \mathbf{r}")$$

(c) by the definition of dham

(d) by the towering property of conditional expectations

(e) given $a_t$, we have $r_t \sim p_{a_t}$, $r'_t \sim p'_{a_t}$ and $r" \sim p''_{a_t}$

(f) by linearity of the expectation, and the fact that the expression inside the expectation only depends on the actions $a_t$

Since this is true for any "intermediary" bandit instance $\nu'' \in \mathcal{F}^K$, we take $\nu''_\star$ to be the environment where the infimum of the $d_\epsilon(P_a, P'_a)$ is attained for each arm $a \in [K]$. Specifically, let $\nu''_\star = (p^\star_a, a \in [K])$ where

$$p^\star_a = \arg\min_{\mathbb{L} \in \mathcal{F}} \left\{ \epsilon \, \mathrm{TV} \left( p_a \mid\mid \mathbb{L} \right) + \mathrm{KL} \left( \mathbb{L} \mid\mid p'_a \right) \right\}$$

Plugging $\nu''_\star$ gives

$$\mathrm{KL} \left( \mathbb{M}_{\nu\pi} \mid\mid \mathbb{M}_{\nu'\pi} \right) \le \mathbb{E}_{\nu\pi} \left[ \sum_{t=1}^{T} d_\epsilon(p_{a_t}, p'_{a_t}) \right] \tag{14}$$

Let $N_{t,a} = \sum_{s<t} \mathbb{1}\{a_s = a\}$ be the counts of arm $a$ before step $t$. Then, we can rewrite the bound as

$$\mathrm{KL}\left(\mathbb{M}_{\nu\pi} \,\|\, \mathbb{M}_{\nu'\pi}\right) \leq \sum_{a=1}^{K} \mathbb{E}_{\nu\pi}[N_{T+1,a}]\mathrm{d}_\epsilon(p_a, p'_a)\,, \tag{15}$$

**Stopping time version of the KL decomposition for BAI under DP.** Let $\pi$ be an $\epsilon$-DP BAI strategy. Let $\nu$ and $\lambda$ be two bandit instances. Denote by $\mathbb{M}_{\nu\pi}$ the marginal distribution of the output of the BAI strategy when $\pi$ interacts with $\nu$. By using Wald's lemma in the proof technique seen before for the canonical bandit setting under FC-BAI, we get that

$$\mathrm{KL}\left(\mathbb{M}_{\nu\pi} \,\|\, \mathbb{M}_{\lambda\pi}\right) \leq \mathbb{E}_{\nu\pi}\left(\sum_{t=1}^{\tau_{\epsilon,\delta}} \mathrm{d}_\epsilon(\nu_{a_t}, \lambda_{a_t})\right) = \sum_{a=1}^{K} \mathbb{E}_{\nu\pi}[N_{\tau_{\epsilon,\delta}+1,a}]\mathrm{d}_\epsilon(\nu_a, \lambda_a)\,, \tag{16}$$

where $\tau$ is the stopping time.

### D.3 Sample Complexity Lower Bound Proof

**Theorem 2** (Sample complexity lower bound for BAI under $\epsilon$-DP). *Let $(\epsilon, \delta) \in \mathbb{R}^\star_+ \times (0,1)$. For any algorithm $\pi$ that is $\delta$-correct and $\epsilon$-global DP on $\mathcal{F}^K$,*

$$\mathbb{E}_{\boldsymbol{\nu}\pi}[\tau_{\epsilon,\delta}] \geq T^\star_\epsilon(\boldsymbol{\nu}) \log(1/(3\delta))$$

*for all $\boldsymbol{\nu} \in \mathcal{F}^K$ with unique best arm. The inverse of the* characteristic time $T^\star_\epsilon(\boldsymbol{\nu})$ *is defined as*

$$T^\star_\epsilon(\boldsymbol{\nu})^{-1} := \sup_{w \in \triangle_K} \inf_{\boldsymbol{\kappa} \in \mathrm{Alt}(\boldsymbol{\nu})} \sum_{a=1}^{K} w_a \mathrm{d}_\epsilon(\nu_a, \kappa_a)\,, \tag{17}$$

$$\mathrm{d}_\epsilon(\nu_a, \kappa_a) := \inf_{\varphi_a \in \mathcal{F}} \{\mathrm{KL}(\varphi_a \,\|\, \kappa_a) + \epsilon \cdot \mathrm{TV}(\nu_a \,\|\, \varphi_a)\}\,. \tag{18}$$

*Proof.* Let $\pi$ be an $\epsilon$-global DP $\delta$-correct BAI strategy. Let $\nu$ be a bandit instance and $\lambda \in \mathrm{Alt}(\nu)$.

Let $\mathbb{M}_{\nu\pi}$ denote the probability distribution of the output when the BAI strategy $\pi$ interacts with $\nu$. For any alternative instance $\lambda \in \mathrm{Alt}(\nu)$, the data-processing inequality gives that

$$\mathrm{KL}\left(\mathbb{M}_{\nu\pi} \,\|\, \mathbb{M}_{\lambda,\pi}\right) \geq \mathrm{kl}\left(\mathbb{M}_{\nu\pi}\left(\tilde{a} = a^\star(\boldsymbol{\nu})\right), \mathbb{M}_{\lambda,\pi}\left(\tilde{a} = a^\star(\boldsymbol{\nu})\right)\right) \geq \mathrm{kl}(1 - \delta, \delta)\,, \tag{19}$$

where the second inequality is because $\pi$ is $\delta$-correct, i.e., $\mathbb{M}_{\nu\pi}(\tilde{a} = a^\star(\boldsymbol{\nu})) \geq 1 - \delta$ and $\mathbb{M}_{\lambda,\pi}(\tilde{a} = a^\star(\boldsymbol{\nu})) \leq \delta$, and the monotonicity of the kl. Now, using the stopping time version of the KL decomposition for FC-BAI, we get that

$$\mathrm{kl}(1 - \delta, \delta) \leq \mathrm{KL}\left(\mathbb{M}_{\nu,\pi} \,\|\, \mathbb{M}_{\lambda,\pi}\right) \leq \sum_{a=1}^{K} \mathbb{E}_{\nu\pi}[N_{\tau_{\epsilon,\delta}+1,a}]\mathrm{d}_\epsilon(\nu_a, \lambda_a)\,.$$

Since this is true for all $\lambda \in \mathrm{Alt}(\nu)$, we get

$$\mathrm{kl}(1 - \delta, \delta) \leq \inf_{\lambda \in \mathrm{Alt}(\nu)} \sum_{a=1}^{K} \mathbb{E}_{\nu\pi}[N_{\tau_{\epsilon,\delta}+1,a}]\mathrm{d}_\epsilon(\nu_a, \lambda_a)$$

$$\overset{(a)}{=} \mathbb{E}[\tau_{\epsilon,\delta}] \inf_{\lambda \in \mathrm{Alt}(\nu)} \sum_{a=1}^{K} \frac{\mathbb{E}\left[N_{\tau_{\epsilon,\delta}+1,a}\right]}{\mathbb{E}[\tau_{\epsilon,\delta}]}\mathrm{d}_\epsilon(\nu_a, \lambda_a)$$

$$\overset{(b)}{\leq} \mathbb{E}[\tau_{\epsilon,\delta}]\left(\sup_{\omega \in \triangle_K} \inf_{\lambda \in \mathrm{Alt}(\nu)} \sum_{a=1}^{K} \omega_a \mathrm{d}_\epsilon(\nu_a, \lambda_a)\right)\,.$$

(a) is due to the fact that $\mathbb{E}[\tau_{\epsilon,\delta}]$ does not depend on $\lambda$. (b) is obtained by noting that the vector $(\omega_a)_{a \in [K]} \triangleq \left(\frac{\mathbb{E}_{\nu,\pi}\left[N_{\tau_{\epsilon,\delta}+1,a}\right]}{\mathbb{E}_{\nu,\pi}[\tau_{\epsilon,\delta}]}\right)_{a \in [K]}$ belongs to the simplex $\triangle_K$. The theorem follows by noting that for $\delta \in (0,1)$, $\mathrm{kl}(1 - \delta, \delta) \geq \log(1/3\delta)$. $\qquad\square$

# E   Privacy Analysis

In this section, we prove Lemma 5. First, we justify using a geometric grid for updating the means (Lemma 8). Second, we obtain Lemma 5 as a combination of Lemma 8 and the post-processing property of DP (Proposition 1).

## E.1   Releasing partial sums privately

First, the following lemma justifies the use of geometric grids, and provides that the price of getting rid of forgetting is summing the Laplace noise from previous phases.

**Lemma 8** (Privacy of our grid-based mean estimator). *Let $T \in \{1, \dots\}$, $\ell < T$ and $t_1, \dots t_\ell, t_{\ell+1}$ be in $[1, T]$ such that $1 = t_1 < \cdots < t_\ell < t_{\ell+1} - 1 = T$.*

*Let $\mathcal{M}$ be the following mechanism:*

$$\begin{pmatrix} x_1 \\ x_2 \\ \vdots \\ x_T \end{pmatrix} \overset{\mathcal{M}}{\to} \begin{pmatrix} (x_1 + \cdots + x_{t_2-1}) + (Y_1) \\ (x_1 + \cdots + x_{t_3-1}) + (Y_1 + Y_2) \\ \vdots \\ (x_1 + \cdots + x_T) + (Y_1 + Y_2 + \cdots + Y_{\ell-1}) \end{pmatrix}$$

*where $(Y_1, \dots, Y_\ell) \sim^{iid} \mathrm{Lap}(1/\epsilon)$.*

*Then, for any $\{x_1, \dots, x_T\} \in [0,1]^T$, $\mathcal{M}$ is $\epsilon$-DP.*

*Proof.* First, consider the following mechanism, that only computes the partial sums:

$$\begin{pmatrix} x_1 \\ x_2 \\ \vdots \\ x_T \end{pmatrix} \to \begin{pmatrix} x_1 + \cdots + x_{t_2-1} \\ x_{t_2} + \cdots + x_{t_3-1} \\ \vdots \\ x_{t_{\ell-1}} + \cdots + x_T \end{pmatrix} .$$

Because $x_t \in [0,1]$, the sensitivity of each partial sum is 1. Since each partial sum is computed over non-overlapping sequences, combining the Laplace mechanism (Theorem 1) with the parallel composition property of DP (Lemma 3) gives that the following mechanism:

$$\begin{pmatrix} x_1 \\ x_2 \\ \vdots \\ x_T \end{pmatrix} \overset{\mathcal{P}}{\to} \begin{pmatrix} x_1 + \cdots + x_{t_2-1} + Y_1 \\ x_{t_2} + \cdots + x_{t_3-1} + Y_2 \\ \vdots \\ x_{t_{\ell-1}} + \cdots + x_T + Y_{\ell-1} \end{pmatrix}$$

is $\epsilon$-DP, where $(Y_1, \dots, Y_{\ell-1}) \sim^{iid} \mathrm{Lap}(1/\epsilon)$.

Consider the post-processing function $f : (x_1, \dots x_{\ell-1}) \to (x_1, x_1 + x_2, \dots, x_1 + x_2 + \cdots + x_{\ell-1})$. Then, we have that that $\mathcal{M} = f \circ \mathcal{P}$. So, by the post-processing property of DP, $\mathcal{M}$ is $\epsilon$-DP. □

**Remark 1.** *Mechanism $\mathcal{P}$, defined in the proof of Lemma 8, is the fundamental mechanism used by all previous bandit algorithms [Sajed and Sheffet, 2019; Azize and Basu, 2022; Hu and Hegde, 2022; Azize et al., 2024] to justify the use of forgetting. Our mechanism $\mathcal{M}$ is just summing over the partial sums computed on each phase, and thus the price of having sums of $x_i$ that start from the beginning (i.e. do not forget) is that we have to sum now the noise from all previous phases too.*

## E.2   Proof of Lemma 5

We are now ready to prove Lemma 5, i.e. that any BAI algorithm based solely on using $\mathrm{GPE}_\eta(\epsilon)$ to access observations is $\epsilon$-global DP on $[0, 1]$.

*Proof.* Let $\pi$ be a BAI algorithm using only $\mathrm{GPE}_\eta(\epsilon)$ to access observations. Let $R = \{x_1, \dots\}$ and $R' = \{x'_1, \dots\}$ be two neighbouring sequences of private observations, i.e. there exists a $t^\star \in \{1, \dots\}$ such that $x_t = x'_t$ for all $t \neq t^\star$, i.e. that $R$ and $R'$ only differ at $t^\star$.

Fix a stopping time, recommendation and sampled actions $(T+1, \tilde{a}, (a_1, \ldots, a_T))$, we want to show that

$$\Pr[\pi(R) = (T+1, \tilde{a}, (a_1, \ldots, a_T))] \leq e^\epsilon \Pr[\pi(R') = (T+1, \tilde{a}, (a_1, \ldots, a_T))].$$

Step 1: Probability decompositions: First, let us denote by $\tau$, $\tilde{A}$ and $A_1, \ldots, A_\tau$ the random variables of stopping, recommendation and sampled actions, when $\pi$ interacts with $R$. Similarly, let $\tau'$, $\tilde{A}'$ and $A'_1, \ldots, A'_\tau$ the random variables of stopping, recommendation and sampled actions, when $\pi$ interacts with $R'$.

We have

$$\Pr[\pi(R) = (T+1, \tilde{a}, (a_1, \ldots, a_T))] = \Pr[\tau = T+1, \tilde{A} = \tilde{a}, A_1 = a_1, \ldots, A_T = a_T]$$
$$\Pr[\pi(R') = (T+1, \tilde{a}, (a_1, \ldots, a_T))] = \Pr[\tau' = T+1, \tilde{A}' = \tilde{a}, A'_1 = a_1, \ldots, A'_T = a_T]$$

Since for all $t < t^\star$, $x_t = x'_t$, the policy samples the same actions, up to step $t^\star$, i.e.

$$\Pr[A_1 = a_1, \ldots, A_{t^\star} = a_{t^\star}] = \Pr[A'_1 = a_1, \ldots, A'_{t^\star} = a_{t^\star}]$$

And thus

$$\frac{\Pr[\pi(R) = (T+1, \tilde{a}, (a_1, \ldots, a_T))]}{\Pr[\pi(R') = (T+1, \tilde{a}, (a_1, \ldots, a_T))]}$$
$$= \frac{\Pr[\tau = T+1, \tilde{A} = \tilde{a}, A_{t^\star+1} = a_{t^\star+1}, \ldots, A_T = a_T \mid A_1 = a_1, \ldots, A_{t^\star} = a_{t^\star}]}{\Pr[\tau' = T+1, \tilde{A}' = \tilde{a}, A'_{t^\star+1} = a_{t^\star+1}, \ldots, A'_T = a_T \mid A'_1 = a_1, \ldots, A'_{t^\star} = a_{t^\star}]}$$

Let us denote by $t_1, \ldots, t_\ell$ the time step corresponding to the beginning of the phases when $\pi$ interacts with $R$, and $t'_1, \ldots, t'_{\ell'}$ the the time step corresponding to the beginning of the phases $\pi$ interacts with **r'**.

Also, let $t_{k^\star}$ be the beginning of the phase for which $t^\star$ belongs in $R$ phases. Similarly, let $t'_{k'_\star}$ be the beginning of the phase for which $t^\star$ belongs in $R'$ phases.

Since the actions $a_1, \ldots, a_T$ are fixed, and $r_t = r'_t$ for $t < t^\star$, $t^\star$ falls in the same phase under both $R$ and $R'$. Thus, $t_{k_\star} = t'_{k'_\star}$ and $k_\star = k'_\star$.

Step 2: Using the structure of $\mathrm{GPE}_\eta(\epsilon)$

Let $\tilde{S}^p_{k^\star} = \sum_{s=t_{k^\star}}^{t_{k^\star+1}-1} x_s + Y_{k_\star}$ be the noisy partial sum of observations collected at phase $k^\star$ for **r**, where $Y_{k^\star} \sim \mathrm{Lap}(1/\epsilon)$. Similarly, let $\tilde{S}'^p_{k^\star} = \sum_{s=t_{k^\star}}^{t_{k^\star+1}-1} x'_s + Y'_{k_\star}$ be the noisy partial sum of observations collected at phase $k^\star$ for **r'**, where $Y'_{k^\star} \sim \mathrm{Lap}(1/\epsilon)$. Using the structure of $\mathrm{GPE}_\eta(\epsilon)$, we have that:

(a) If the value of the noisy partial sums at phase $k^\star$ is exactly the same between the neighbouring $R$ and $R'$, then the BAI algorithm $\pi$ will sample the same sequence of actions from step $t^\star$ onward, recommend the same final guess and stop at the same time, with the same probability under $R$ and $R'$. Thus, for any $s \in \mathbb{R}$:

$$\Pr[\tau = T+1, \tilde{A} = \tilde{a}, A_{t^\star+1} = a_{t^\star+1}, \ldots, A_T = a_T \mid A_1 = a_1, \ldots, A_{t^\star} = a_{t^\star}, \tilde{S}^p_{k^\star} = s]$$
$$= \Pr[\tau' = T+1, \tilde{A}' = \tilde{a}, A'_{t^\star+1} = a_{t^\star+1}, \ldots, A'_T = a_T \mid A'_1 = a_1, \ldots, A'_{t^\star} = a_{t^\star}, \tilde{S}'^p_{k^\star} = s]$$
(20)

This is due to the fact that, in $\mathrm{GPE}_\eta(\epsilon)$, the reward at step $t^\star$ only affects the statistic $\tilde{S}^p_{k^\star}$, and nothing else.

(b) Since rewards are $[0,1]$, using the Laplace mechanism, we have that

$$\Pr[\tilde{S}^p_{k^\star} = s \mid A_1 = a_1, \ldots, A_{t^\star} = a_{t^\star}] \leq e^\epsilon \Pr(\tilde{S}'^p_{k^\star} = s \mid A_1 = a_1, \ldots, A'_{t^\star} = a_{t^\star}). \quad (21)$$

Step 3: Combining Eq. 20 and Eq. 21, aka post-processing:

We have

$$\Pr[\tau = T + 1, \tilde{A} = \tilde{a}, A_{t^\star+1} = a_{t^\star+1}, \ldots, A_T = a_T \mid A_1 = a_1, \ldots, A_{t^\star} = a_{t^\star}]$$

$$= \int_{s \in \mathbb{R}} \Pr[\tau = T + 1, \tilde{A} = \tilde{a}, A_{t^\star+1} = a_{t^\star+1}, \ldots, A_T = a_T \mid A_1 = a_1, \ldots, A_{t^\star} = a_{t^\star}, \tilde{S}^p_{k^\star} = s]$$

$$\Pr[\tilde{S}^p_{k^\star} = s \mid A_1 = a_1, \ldots, A_{t^\star} = a_{t^\star}]$$

$$\leq \int_{s \in \mathbb{R}} e^\epsilon \Pr[\tau' = T + 1, \tilde{A}' = \tilde{a}, A'_{t^\star+1} = a_{t^\star+1}, \ldots, A'_T = a_T$$

$$\mid A_1 = a_1, \ldots, A'_{t^\star} = a_{t^\star}, \tilde{S}'^p_{k^\star} = s]$$

$$\Pr(\tilde{S}'^p_{k^\star} = s \mid A_1 = a_1, \ldots, A'_{t^\star} = a_{t^\star})$$

$$= e^\epsilon \Pr[\tau' = T + 1, \tilde{A}' = \tilde{a}, A'_{t^\star+1} = a_{t^\star+1}, \ldots, A'_T = a_T \mid A'_1 = a_1, \ldots, A'_{t^\star} = a_{t^\star}] .$$

This concludes the proof:

$$\frac{\Pr[\pi(R) = (T + 1, \tilde{a}, (a_1, \ldots, a_T))]}{\Pr[\pi(R') = (T + 1, \tilde{a}, (a_1, \ldots, a_T))]} \leq e^\epsilon .$$

$\square$

### E.3 Recalling the post-processing and composition properties of DP

**Proposition 1** (Post-processing [Dwork and Roth, 2014])**.** *Let $\mathcal{M}$ be a mechanism and $f$ be an arbitrary randomised function defined on $\mathcal{M}$'s output. If $\mathcal{M}$ is $\epsilon$-DP, then $f \circ \mathcal{M}$ is $\epsilon$-DP.*

The post-processing property ensures that any quantity constructed only from a private output is still private, with the same privacy budget. This is a consequence of the data processing inequality.

**Proposition 2** (Simple Composition)**.** *Let $\mathcal{M}^1, \ldots, \mathcal{M}^k$ be $k$ mechanisms. We define the mechanism*

$$\mathcal{G} : D \to \bigotimes_{i=1}^{k} \mathcal{M}^i_D$$

*as the $k$ composition of the mechanisms $\mathcal{M}^1, \ldots, \mathcal{M}^k$.*

*If each $\mathcal{M}^i$ is $\epsilon_i$-DP, then $\mathcal{G}$ is $\sum_{i=1}^{k} \epsilon_i$-DP.*

**Proposition 3** (Parallel Composition)**.** *Let $\mathcal{M}^1, \ldots, \mathcal{M}^k$ be $k$ mechanisms, such that $k < n$, where $n$ is the size of the input dataset. Let $t_1, \ldots t_k, t_{k+1}$ be indexes in $[1, n]$ such that $1 = t_1 < \cdots < t_k < t_{k+1} - 1 = n$.*
*Let's define the following mechanism*

$$\mathcal{G} : \{x_1, \ldots, x_n\} \to \bigotimes_{i=1}^{k} \mathcal{M}^i_{\{x_{t_i}, \ldots, x_{t_{i+1}-1}\}}$$

*$\mathcal{G}$ is the mechanism that we get by applying each $\mathcal{M}^i$ to the $i$-th partition of the input dataset $\{x_1, \ldots, x_n\}$ according to the indexes $t_1 < \cdots < t_k < t_{k+1}$.*

*If each $\mathcal{M}^i$ is $\epsilon$-DP, then $\mathcal{G}$ is $\epsilon$-DP.*

In parallel composition, the $k$ mechanisms are applied to different "non-overlapping" parts of the input dataset. If each mechanism is DP, then the parallel composition of the $k$ mechanisms is DP, *with the same privacy budget*. This property will be the basis for designing private bandit algorithms.

## F Concentration Results

In Appendix F, we detail the proof of all our concentration results. In Appendix F.1, we start by introducing a variant of GLR-based stopping rule using the modified transportation costs $\widetilde{W}_{\epsilon,a,b}$

(see Appendix G.2.1 for details) which are defined based on the modified divergences $\widetilde{d}_\epsilon^\pm$ (see Appendix G.1.1 for details). The proof of Theorem 6 is given in Appendix F.2. In Appendix F.3, we show tail bounds for a sum between independent Bernoulli and Laplace observations that feature the product of the tail bounds of each process. We prove time-uniform and fixed-time tails concentration for Laplace distribution in Appendix F.4, and recall existing results for Bernoulli in Appendix F.5. In Appendix F.6, we provide tail bounds for a sum between independent Bernoulli and Laplace observations that feature the modified divergence $\widetilde{d}_\epsilon$ defined in Eq. (32). In Appendix F.7, we give geometric grid time uniform tails concentration for the reweighted modified divergence.

## F.1 Modified GLR Stopping Rule

The modified GLR stopping rule is defined as

$$\tau_{\epsilon,\delta}^{\text{MGLR}} = \inf \left\{ n \mid \forall a \neq \tilde{a}_n, \ \widetilde{W}_{\epsilon,\tilde{a}_n,a}(\tilde{\mu}_n, \tilde{N}_n) > \sum_{b \in \{\tilde{a}_n,a\}} \widetilde{c}(k_{n,b}, \delta) \right\} \text{ with } \tilde{a}_n \in \arg\max_{a \in [K]}[\tilde{\mu}_{n,a}]_0^1 ,$$

(22)

where $(\tilde{\mu}_n, \tilde{N}_n)$ are the outputs of $\text{GPE}_\eta(\epsilon)$. The modified transportation costs $(\widetilde{W}_{\epsilon,a,b})_{(a,b) \in [K]^2}$ are defined in Eq. (34), i.e., for all $(\mu, w) \in \mathbb{R}^K \times \mathbb{R}_+^K$ and all $(a, b) \in [K]^2$ such that $a \neq b$,

$$\widetilde{W}_{\epsilon,a,b}(\mu, w) := \mathbb{1}\left([\mu_a]_0^1 > [\mu_b]_0^1\right) \inf_{u \in (0,1)} \left\{ w_a \widetilde{d}_\epsilon^-(\mu_a, u, r(w_a)) + w_b \widetilde{d}_\epsilon^+(\mu_b, u, r(w_b)) \right\} ,$$

where $r(x) := \frac{x}{1 + \log_{1+\eta} x}$ is defined in Eq. (33) for all $x \geq 1$. The modified divergence $\widetilde{d}_\epsilon^\pm$ are defined in Eq. (32), i.e., for all $(\lambda, \mu, r) \in \mathbb{R} \times (0,1) \times \mathbb{R}_+^\star$,

$$\widetilde{d}_\epsilon^-(\lambda, \mu, r) := \mathbb{1}\left(\mu < [\lambda]_0^1\right) \inf_{z \in (\mu, [\lambda]_0^1)} \left\{ \text{kl}(z, \mu) + \frac{1}{r} h(r\epsilon(\lambda - z)) \right\} ,$$

$$\widetilde{d}_\epsilon^+(\lambda, \mu, r) := \mathbb{1}\left(\mu > [\lambda]_0^1\right) \inf_{z \in ([\lambda]_0^1, \mu)} \left\{ \text{kl}(z, \mu) + \frac{1}{r} h(r\epsilon(z - \lambda)) \right\} ,$$

where $h(x) := \sqrt{1 + x^2} - 1 + \log\left(\frac{2}{x^2}\left(\sqrt{1 + x^2} - 1\right)\right)$ is defined in Eq. (31) for all $x > 0$.

Lemma 9 gives a stopping threshold under which the modified GLR stopping rule is $\delta$-correct.

**Lemma 9.** *Let $\delta \in (0,1)$ and $\epsilon > 0$. Let $\eta > 0$. Let $s > 1$ and $\zeta$ be the Riemann $\zeta$ function. Let $\overline{W}_{-1}(x) = -W_{-1}(-e^{-x})$ for all $x \geq 1$, where $W_{-1}$ is the negative branch of the Lambert $W$ function. It satisfies $\overline{W}_{-1}(x) \approx x + \log x$, see Lemma 52. Given any sampling rule using the $\text{GPE}_\eta(\epsilon)$, combining $\text{GPE}_\eta(\epsilon)$ with the modified GLR stopping rule as in Eq. (22) with the stopping threshold*

$$\widetilde{c}(k, \delta) = \overline{W}_{-1}\left(\log\left(\frac{K\zeta(s)}{\delta}\right) + s\log(k) + 3 - \log 2\right) - 3 + \log 2 .$$

(23)

*yields a $\delta$-correct and $\epsilon$-global DP algorithm for Bernoulli instances with unique best arm.*

*Proof.* Lemma 5 yields the $\epsilon$-global DP. Let $\mathcal{E}_\delta = \mathcal{E}_{\delta,a^\star,+} \cap \bigcap_{a \neq a^\star} \mathcal{E}_{\delta,a,-}$ with

$$\mathcal{E}_{\delta,a^\star,+} = \left\{ \forall n \in \mathbb{N}, \ \tilde{N}_{n,a^\star} \widetilde{d}_\epsilon^+(\tilde{\mu}_{n,a^\star}, \mu_{a^\star}, \tilde{N}_{n,a^\star}/k_{n,a^\star}) \leq \widetilde{c}(k_{n,a^\star}, \delta) \right\} ,$$

$$\forall a \neq a^\star, \quad \mathcal{E}_{\delta,a,-} = \left\{ \forall n \in \mathbb{N}, \ \tilde{N}_{n,a} \widetilde{d}_\epsilon^-(\tilde{\mu}_{n,a}, \mu_a, \tilde{N}_{n,a}/k_{n,a}) \leq \widetilde{c}(k_{n,a}, \delta) \right\} ,$$

where $(\tilde{\mu}_n, \tilde{N}_n, k_n)$ are given by $\text{GPE}_\eta(\epsilon)$, $\widetilde{c}$ as in Eq. (23) and $\widetilde{d}_\epsilon^\pm$ as in Eq. (32).

Using Lemmas 20 and 21, we have $\mathbb{P}_{\boldsymbol{\nu}\pi}(\mathcal{E}_{\delta,a,-}^{\complement}) \leq \delta/K$ for all $a \neq a^\star$, and $\mathbb{P}_{\boldsymbol{\nu}\pi}(\mathcal{E}_{\delta,a^\star,+}^{\complement}) \leq \delta/K$. By union bound over $a \in [K]$, we obtain $\mathbb{P}_{\boldsymbol{\nu}\pi}(\mathcal{E}_\delta^{\complement}) \leq \delta$.

Let $\tau_{\epsilon,\delta}^{\text{MGLR}}$ as in Eq. (22) and $\tilde{a}_n \in \arg\max_{a \in [K]}[\tilde{\mu}_{n,a}]_0^1$. Then, we directly have that

$$\mathbb{P}_{\boldsymbol{\nu}\pi}\left(\tau_{\epsilon,\delta}^{\text{MGLR}} < +\infty, \ \tilde{a}_{\tau_{\epsilon,\delta}^{\text{MGLR}}} \neq a^\star\right) \leq \mathbb{P}_{\boldsymbol{\nu}\pi}\left(\mathcal{E}_\delta^{\complement}\right) + \mathbb{P}_{\boldsymbol{\nu}\pi}\left(\mathcal{E}_\delta \cap \{\tau_{\epsilon,\delta}^{\text{MGLR}} < +\infty, \ \tilde{a}_{\tau_{\epsilon,\delta}^{\text{MGLR}}} \neq a^\star\}\right)$$

$$\leq \delta + \mathbb{P}_{\boldsymbol{\nu}\pi}\left(\mathcal{E}_\delta \cap \{\tau_{\epsilon,\delta}^{\mathrm{MGLR}} < +\infty,\ \tilde{a}_{\tau_{\epsilon,\delta}^{\mathrm{MGLR}}} \neq a^\star\}\right).$$

Under $\mathcal{E}_\delta \cap \{\tau_{\epsilon,\delta}^{\mathrm{MGLR}} < +\infty,\ \tilde{a}_{\tau_{\epsilon,\delta}^{\mathrm{MGLR}}} \neq a^\star\}$, by definition of the stopping rule as in Eq. (7) and the stopping threshold in Eq. (8), we obtain that there exists $a \neq a^\star$ and $n \in \mathbb{N}$ such that $[\tilde{\mu}_{n,a}]_0^1 > [\tilde{\mu}_{n,a^\star}]_0^1$ and

$$\sum_{b \in \{a,a^\star\}} \widetilde{c}(k_{n,b}, \delta) < \widetilde{W}_{\epsilon,a,a^\star}(\tilde{\mu}_n, \tilde{N}_n)$$

$$= \inf_{u \in (0,1)} \left\{ \tilde{N}_{n,a}\widetilde{d}_\epsilon^-(\tilde{\mu}_{n,a}, u, r(\tilde{N}_{n,a})) + \tilde{N}_{n,a^\star}\widetilde{d}_\epsilon^+(\tilde{\mu}_{n,a^\star}, u, r(\tilde{N}_{n,a^\star})) \right\}$$

$$= \inf_{(u_a,u_{a^\star}) \in (0,1)^2,\ u_a \leq u_{a^\star}} \{\tilde{N}_{n,a}\widetilde{d}_\epsilon^-(\tilde{\mu}_{n,a}, u_a, r(\tilde{N}_{n,a})) + \tilde{N}_{n,a^\star}\widetilde{d}_\epsilon^+(\tilde{\mu}_{n,a^\star}, u_{a^\star}, r(\tilde{N}_{n,a^\star}))\}$$

$$\leq \tilde{N}_{n,a}\widetilde{d}_\epsilon^-(\tilde{\mu}_{n,a}, \mu_a, r(\tilde{N}_{n,a})) + \tilde{N}_{n,a^\star}\widetilde{d}_\epsilon^+(\tilde{\mu}_{n,a^\star}, \mu_{a^\star}, r(\tilde{N}_{n,a^\star}))$$

$$\leq \tilde{N}_{n,a}\widetilde{d}_\epsilon^-(\tilde{\mu}_{n,a}, \mu_a, \tilde{N}_{n,a}/k_{n,a}) + \tilde{N}_{n,a^\star}\widetilde{d}_\epsilon^+(\tilde{\mu}_{n,a^\star}, \mu_{a^\star}, \tilde{N}_{n,a^\star}/k_{n,a^\star}) \leq \sum_{b \in \{a,a^\star\}} \widetilde{c}(k_{n,b}, \delta),$$

where we used the definition of $\widetilde{W}_{\epsilon,a,a^\star}$ in Eq. (34) and Lemma 40 in the two equalities and $\mu_{a^\star} > \mu_a$ in the following inequality. The second to last inequality uses that $r(\tilde{N}_{n,a}) \leq \tilde{N}_{n,a}/k_{n,a}$ for all $a \in [K]$ by definition of $(k_n, \tilde{N}_n)$, i.e., $k_{n,a} \leq 1 + \log_{1+\eta} \tilde{N}_{n,a} \leq k_{n,a} + 1$, and that $r \mapsto \widetilde{d}_\epsilon^\pm(\lambda, u, r)$ is non-decreasing, see Lemmas 31 and 32. The last inequality is obtained by the concentration event $\mathcal{E}_\delta$. Since this yields a contradiction, we obtain $\mathcal{E}_\delta \cap \{\tau_{\epsilon,\delta}^{\mathrm{MGLR}} < +\infty,\ \tilde{a}_{\tau_{\epsilon,\delta}^{\mathrm{MGLR}}} \neq a^\star\} = \emptyset$. This concludes the proof, i.e., $\mathbb{P}_{\boldsymbol{\nu}\pi}\left(\tau_{\epsilon,\delta}^{\mathrm{MGLR}} < +\infty,\ \tilde{a}_{\tau_{\epsilon,\delta}^{\mathrm{MGLR}}} \neq a^\star\right) \leq \delta$. $\qquad\square$

### F.2 Proof of Theorem 6

Lemma 5 yields the $\epsilon$-global DP. The proof of $\delta$-correctness is the same as the one of Lemma 9 detailed above. In particular, we use the same concentration event $\mathcal{E}_\delta = \mathcal{E}_{\delta,a^\star,+} \cap \bigcap_{a \neq a^\star} \mathcal{E}_{\delta,a,-}$ that satisfies $\mathbb{P}_{\boldsymbol{\nu}\pi}(\mathcal{E}_\delta^\complement) \leq \delta$.

Under $\mathcal{E}_\delta \cap \{\tau_{\epsilon,\delta} < +\infty,\ \tilde{a}_{\tau_{\epsilon,\delta}} \neq a^\star\}$, by definition of the GLR stopping rule as in Eq. (7) and the stopping threshold in Eq. (8), we obtain that there exists $a \neq a^\star$ and $n \in \mathbb{N}$ such that $[\tilde{\mu}_{n,a}]_0^1 > [\tilde{\mu}_{n,a^\star}]_0^1$,

$$\sum_{b \in \{a,a^\star\}} \left(c_1(\tilde{N}_{n,b}, \delta) + c_2(\tilde{N}_{n,b}, \epsilon)\right) = \sum_{b \in \{a,a^\star\}} c(k_{n,b}, \epsilon, \delta) < W_{\epsilon,a,a^\star}(\tilde{\mu}_n, \tilde{N}_n).$$

Then, we obtain

$$W_{\epsilon,a,a^\star}(\tilde{\mu}_n, \tilde{N}_n) = \inf_{u \in [0,1]} \left\{ \tilde{N}_{n,a}d_\epsilon^-(\tilde{\mu}_{n,a}, u) + \tilde{N}_{n,a^\star}d_\epsilon^+(\tilde{\mu}_{n,a^\star}, u) \right\}$$

$$= \inf_{(u_a,u_{a^\star}) \in [0,1]^2,\ u_a \leq u_{a^\star}} \{\tilde{N}_{n,a}d_\epsilon^-(\tilde{\mu}_{n,a}, u_a) + \tilde{N}_{n,a^\star}d_\epsilon^+(\tilde{\mu}_{n,a^\star}, u_{a^\star})\}$$

$$\leq \tilde{N}_{n,a}d_\epsilon^-(\tilde{\mu}_{n,a}, \mu_a) + \tilde{N}_{n,a^\star}d_\epsilon^+(\tilde{\mu}_{n,a^\star}, \mu_{a^\star}),$$

where we used the definition of $W_{\epsilon,a,a^\star}$ in Eq. (4) and Lemma 35 in the two equalities, and $(u_{a^\star}, u_a) = (\mu_{a^\star}, \mu_a) \in [0,1]^2$ that satisfies $u_{a^\star} > u_a$ in the following inequality.

Using Lemma 39 and initialization yields $\min\{r(\tilde{N}_{n,a^\star}), r(\tilde{N}_{n,a})\} > 0$ by . When $[\tilde{\mu}_{n,a}]_0^1 > \mu_a$ and $\mu_{a^\star} > [\tilde{\mu}_{n,a^\star}]_0^1$, Lemma 30 yields

$$\tilde{N}_{n,a^\star}(d_\epsilon^+(\tilde{\mu}_{n,a^\star}, \mu_{a^\star}) - \widetilde{d}_\epsilon^+(\tilde{\mu}_{n,a^\star}, \mu_{a^\star}, r(\tilde{N}_{n,a^\star}))) \leq k_\eta(\tilde{N}_{n,a^\star})(\log(1 + 2\epsilon\frac{\tilde{N}_{n,a^\star}}{k_\eta(\tilde{N}_{n,a^\star})}) + 1)$$

$$\tilde{N}_{n,a}(d_\epsilon^-(\tilde{\mu}_{n,a}, \mu_a) - \widetilde{d}_\epsilon^-(\tilde{\mu}_{n,a}, \mu_a, r(\tilde{N}_{n,a}))) \leq k_\eta(\tilde{N}_{n,a})\left(\log\left(1 + 2\epsilon\frac{\tilde{N}_{n,a}}{k_\eta(\tilde{N}_{n,a})}\right) + 1\right),$$

where we used that $(\mu_a, \mu_{a^\star}) \in (0,1)^2$ and $x/r(x) = 1 + \log_{1+\eta} x = k_\eta(x)$. When $[\tilde{\mu}_{n,a}]_0^1 \leq \mu_a$, we have $d_\epsilon^-(\tilde{\mu}_{n,a}, \mu_a) = 0 = \tilde{d}_\epsilon^-(\tilde{\mu}_{n,a}, \mu_a, r(\tilde{N}_{n,a}))$. When $\mu_{a^\star} \leq [\tilde{\mu}_{n,a^\star}]_0^1$, we have $d_\epsilon^+(\tilde{\mu}_{n,a^\star}, \mu_{a^\star}) = 0 = \tilde{d}_\epsilon^+(\tilde{\mu}_{n,a^\star}, \mu_{a^\star}, r(\tilde{N}_{n,a^\star}))$. In either case, the above inequalities are still valid since the left hand side is null and the right hand side is positive. Therefore, we have

$$\tilde{N}_{n,a} d_\epsilon^-(\tilde{\mu}_{n,a}, \mu_a) + \tilde{N}_{n,a^\star} d_\epsilon^+(\tilde{\mu}_{n,a^\star}, \mu_{a^\star})$$
$$\leq \tilde{N}_{n,a} \tilde{d}_\epsilon^-(\tilde{\mu}_{n,a}, \mu_a, r(\tilde{N}_{n,a})) + \tilde{N}_{n,a^\star} \tilde{d}_\epsilon^+(\tilde{\mu}_{n,a^\star}, \mu_{a^\star}, r(\tilde{N}_{n,a^\star})) + \sum_{b \in \{a, a^\star\}} c_2(\tilde{N}_{n,b}, \epsilon)$$
$$\leq \sum_{b \in \{a, a^\star\}} \tilde{c}(k_{n,b}, \delta) + \sum_{b \in \{a, a^\star\}} c_2(\tilde{N}_{n,b}, \epsilon) \leq \sum_{b \in \{a, a^\star\}} \left( c_1(\tilde{N}_{n,b}, \delta) + c_2(\tilde{N}_{n,b}, \epsilon) \right),$$

where the second inequality uses the proof of Lemma 9, and third leverages that

$$\tilde{c}(k_{n,a}, \delta) \leq \overline{W}_{-1} \left( \log\left( \frac{K\zeta(s)}{\delta} \right) + s \log(k_\eta(\tilde{N}_{n,a})) + 3 - \log 2 \right) - 3 + \log 2,$$

by using that $\overline{W}_{-1}$ is increasing (Lemma 52) and $k_{n,a} \leq 1 + \log_{1+\eta} \tilde{N}_{n,a} = k_\eta(\tilde{N}_{n,a})$ for all $a \in [K]$, as well as $r(x) = x/k_\eta(x)$. Combining all the above inequalities, we have shown that

$$\sum_{b \in \{a, a^\star\}} (c_1(\tilde{N}_{n,b}, \delta) + c_2(\tilde{N}_{n,b}, \epsilon)) < W_{\epsilon, a, a^\star}(\tilde{\mu}_n, \tilde{N}_n) \leq \sum_{b \in \{a, a^\star\}} (c_1(\tilde{N}_{n,b}, \delta) + c_2(\tilde{N}_{n,b}, \epsilon)).$$

This yields a contradiction, hence we have $\mathcal{E}_\delta \cap \{\tau_{\epsilon,\delta} < +\infty, \tilde{a}_{\tau_{\epsilon,\delta}} \neq a^\star\} = \emptyset$. This concludes the proof, i.e., $\mathbb{P}_{\nu\pi}\left( \tau_{\epsilon,\delta} < +\infty, \tilde{a}_{\tau_{\epsilon,\delta}} \neq a^\star \right) \leq \delta$.

### F.3 Fixed Time Tails Bounds for a Convolution of Probability Distributions

We derive general upper and lower bounds on the upper and lower tails of the convolution (i.e., sum) between two independent random variables (Lemma 10). We provide upper (Lemma 12) and lower (Lemma 12) tail bounds for a sum (i.e., convolution) between independent Bernoulli and Laplace i.i.d. observations for a fixed time. The bounds are expressed as a function of the infimum over a bounded interval of a $-\frac{1}{t} \log(\cdot)$ transform of the product between the (upper or lower) tail bounds of each process. Therefore, in Lemmas 12 and 12, we can plug any bounds on the (upper or lower) tail concentration of each process. While those bounds are standard for Bernoulli distribution (Lemma 17 in Appendix F.5), we propose new bounds for Laplace distribution (Lemma 16 in Appendix F.4).

**Sketch of Proof of Lemma 10** The main difficulty when studying the sum of two random variables lies in the fact that it involves the integral of the convolution of their probability measures. In all generality, it is difficult to upper bound such a quantity. The main idea behind our proof technique is to split the event of interest into a partition of carefully chosen events. Then, on each of those smaller events, we derive a "tight" upper bound on the integral of the convolution of their probability measures. It is reasonable to wonder how one could choose those events such that the upper bound is easier to obtain. When the event is defined as the intersection of two independent events, then we obtain a straightforward upper bound by the product of their respective probablities. When the event truly mixes the distributions, we need to use a smarter approach to control the integrated function. First, we upper bound a sub-component of this function by a maximum of the product of their respective probablities (on a small interval that is defined by the smaller event). Second, after this upper bound, the integrated function coincides with the hazard function, whose integral is the cumulative hazard function. To conclude the proof, it only remains to merge together the different upper bounds.

To the best of our knownledge, the proof technique closest to ours is the one used to prove Lemma 64 in Jourdan et al. [2022]. They control the probability that two random variables have an unexpected empirical ranking as a function of the boundary crossing probabilities of each random variable. While tackling a distinct problem, they adopt the same proof structure. They decompose the event into carefully chosen events on which they can upper bound the integral of the convolution of their probability distributions. The upper bounds are obtained similarly as ours, with fewer events to consider.

Lemma 10 gives upper and lower bounds on the upper and lower tails of the sum of two independent random variables. This result is of independent interest.

**Lemma 10.** *Let $\theta$ and $\lambda$ be two independent real random variables such that (i) $\theta$ has bounded support included in $[\alpha, \beta]$ and mean $\mu \in (\alpha, \beta)$ and (ii) $\lambda$ has zero mean. Let*

$$\forall u \in [0,1], \ \forall v \in (0,1], \quad p(u,v) := u(1 - \log(u) + \log(v)) .$$

*Then, for all $x > 0$, we have*

$$\mathbb{P}(\theta + \lambda \geq \mu + x) \leq \mathbb{P}(\lambda \geq x)\mathbb{P}(\theta \in [\alpha, \mu]) + \mathbb{P}(\lambda \leq 0)\mathbb{P}(\theta \in [\min\{\beta, \mu + x\}, \beta])$$

$$+ \, p\left(\sup_{z \in (\mu, \min\{\beta, \mu + x\})} \{\mathbb{P}(\theta \in [z, \beta])\mathbb{P}(\lambda \geq \mu + x - z)\}, \ \mathbb{P}(\theta \in [\mu, \beta])\mathbb{P}(\lambda \geq 0)\right) ,$$

$$\mathbb{P}(\theta + \lambda \geq \mu + x) \geq \sup_{z \in (\mu, \min\{\beta, \mu + x\})} \{\mathbb{P}(\theta \in [z, \beta])\mathbb{P}(\lambda \geq \mu + x - z)\} ,$$

$$\mathbb{P}(\theta + \lambda \leq \mu - x) \leq \mathbb{P}(\lambda \leq -x)\mathbb{P}(\theta \in [\mu, \beta]) + \mathbb{P}(\lambda \geq 0)\mathbb{P}(\theta \in [\alpha, \max\{\alpha, \mu - x\}])$$

$$p\left(\sup_{z \in (\max\{\alpha, \mu - x\}, \mu)} \{\mathbb{P}(\theta \in [\alpha, z])\mathbb{P}(\lambda \leq \mu - x - z)\}, \ \mathbb{P}(\theta \in [\alpha, \mu])\mathbb{P}(\lambda \leq 0)\right) ,$$

$$\mathbb{P}(\theta + \lambda \leq \mu - x) \geq \sup_{z \in (\max\{\alpha, \mu - x\}, \mu)} \{\mathbb{P}(\theta \in [\alpha, z])\mathbb{P}(\lambda \leq \mu - x - z)\} .$$

*Proof.* **I. Upper Bound on Upper Tail.** We start by studying $\mathbb{P}(\theta + \lambda \geq \mu + x)$ where $x > 0$. We can suppose that there exists $y_1 \in (\max\{x + \mu - \beta, 0\}, x)$ such that $\mathbb{P}(\theta \geq x + \mu - y_1)\mathbb{P}(\lambda \geq y_1) > 0$. Otherwise, the probability of $\{\theta + \lambda \geq \mu + x\}$ is 0, and both bounds are 0 as well. Let $y_1$ be such a value, and

$$y_3 \in [x, x + \mu) \quad \text{and} \quad y_2 \in (\min\{x + \mu - \beta, 0\}, 0] .$$

First, we note that $-\log \mathbb{P}(\theta \geq x + \mu - y_1)$ and $-\log \mathbb{P}(\lambda \geq y_1)$ are finite, since $\mathbb{P}(\theta \geq x + \mu - y_1)\mathbb{P}(\lambda \geq y_1) > 0$ implies that $\min\{\mathbb{P}(\theta \geq x + \mu - y_1), \mathbb{P}(\lambda \geq y_1)\} > 0$. Second, we note that $y_2$ only exists when $x + \mu < \beta$, i.e., $(\min\{x + \mu - \beta, 0\}, 0] \neq \emptyset$. In order to study the cases $x + \mu < \beta$ and $x + \mu \geq \beta$ simultaneously, we adopt the convention that the maximum of a positive quantity on an empty set is defined as zero. Note that the situation $x + \mu < \beta$ has more subcases.

We partition of the event $\{\theta + \lambda \geq \mu + x\}$ into eight sets, namely

$$\{\theta + \lambda \geq \mu + x, \ \theta \in [\alpha, \beta], \ \lambda \in \mathbb{R}\} = \{\lambda \in (\max\{x + \mu - \beta, 0\}, y_1), \ \theta \in [x + \mu - \lambda, \beta]\}$$
$$\cup \{\theta \in [x + \mu - y_1, \beta], \ \lambda \geq y_1\}$$
$$\cup \{\theta \in (\mu, x + \mu - y_1), \ \lambda \geq x + \mu - \theta\}$$
$$\cup \{\theta \in [x + \mu - y_3, \mu], \ \lambda \geq x + \mu - \theta\}$$
$$\cup \{\theta \in [\alpha, x + \mu - y_3), \lambda \geq x + \mu\}$$
$$\cup \{\lambda \in [y_3, x + \mu), \ \theta \in [x + \mu - \lambda, x + \mu - y_3)\}$$
$$\cup \{\lambda \in [y_2, 0], \ \theta \in [x + \mu - \lambda, \beta]\}$$
$$\cup \{\lambda \in [x + \mu - \theta, y_2), \ \theta \in [x + \mu - y_2, \beta]\} .$$

First, it is direct to see that

$$\{\lambda \in [y_2, 0], \ \theta \in [x + \mu - \lambda, \beta]\} \cup \{\lambda \in [x + \mu - \theta, y_2), \ \theta \in [x + \mu - y_2, \beta]\}$$
$$\subseteq \{\lambda \leq 0, \theta \in [\min\{\beta, \mu + x\}, \beta]\} ,$$
$$\{\theta \in [x + \mu - y_3, \mu], \ \lambda \geq x + \mu - \theta\} \cup \{\theta \in [\alpha, x + \mu - y_3), \lambda \geq x + \mu\}$$
$$\cup \{\lambda \in [y_3, x + \mu), \ \theta \in [x + \mu - \lambda, x + \mu - y_3)\} \subseteq \{\lambda \geq x, \ \theta \in [\alpha, \mu]\} .$$

By union bound, the probability of the union of those five events is upper bounded by the sum of the probability of those two events, i.e., $\mathbb{P}(\lambda \geq x, \ \theta \in [\alpha, \mu]) + \mathbb{P}(\lambda \leq 0, \theta \in [\min\{\beta, \mu + x\}, \beta])$.

**A. Separate Conditions.** Those two events and one of the three remaining do not require to control $(\theta, \lambda)$ simultaneously, as they separate the conditions on $(\theta, \lambda)$. Thanks to the independence of $(\theta, \lambda)$, the probability of those events can be simply upper bounded by the product of the respective probability of those conditions. Therefore, we obtain

$$\mathbb{P}(\lambda \geq x, \ \theta \in [\alpha, \mu]) + \mathbb{P}(\lambda \leq 0, \theta \in [\min\{\beta, \mu + x\}, \beta]) + \mathbb{P}(\theta \in [x + \mu - y_1, \beta], \ \lambda \geq y_1)$$
$$= \mathbb{P}(\lambda \geq x)\mathbb{P}(\theta \in [\alpha, \mu]) + \mathbb{P}(\lambda \leq 0)\mathbb{P}(\theta \in [\min\{\beta, \mu + x\}, \beta])$$

$$+ \mathbb{P}(\theta \in [x + \mu - y_1, \beta]) \mathbb{P}(\lambda \geq y_1) \,.$$

**B. Mixed Conditions.** The two remaining events truly require to control $(\theta, \lambda)$ simultaneously, i.e., consider their convolution. The proof idea is the following: (1) we integrate one integral to obtain one survival function, (2) we make appear the other survival function artificially, (3) we upper bound the product of their survival functions on the whole set and (4) we integrate the remaining hazard function, whose integral is the cumulative hazard function. Let $\mathrm{d}G$ and $\mathrm{d}F$ be the probability measures of $\theta$ and $\lambda$ on $\mathbb{R}$.

For all $s \in (\max\{x + \mu - \beta, 0\}, y_1)$, we have $\mathbb{P}(\lambda \geq s) \geq \mathbb{P}(\lambda \geq y_1) > 0$. Then, we obtain

$$\mathbb{P}\left(\lambda \in (\max\{x + \mu - \beta, 0\}, y_1), \, \theta \in [x + \mu - \lambda, \beta]\right)$$

$$= \int_{s \in (\max\{x + \mu - \beta, 0\}, y_1)} \mathbb{P}(\theta \in [x + \mu - s, \beta]) \mathrm{d}F(s)$$

$$= \int_{s \in (\max\{x + \mu - \beta, 0\}, y_1)} \mathbb{P}(\lambda \geq s) \mathbb{P}(\theta \in [x + \mu - s, \beta]) \frac{1}{\mathbb{P}(\lambda \geq s)} \mathrm{d}F(s)$$

$$\leq \sup_{s \in (\max\{x + \mu - \beta, 0\}, y_1)} \{\mathbb{P}(\lambda \geq s) \mathbb{P}(\theta \in [x + \mu - s, \beta])\} \int_{s \in (\max\{x + \mu - \beta, 0\}, y_1)} \frac{1}{\mathbb{P}(\lambda \geq s)} \mathrm{d}F(s)$$

$$\leq \sup_{s \in (\max\{x + \mu - \beta, 0\}, y_1)} \{\mathbb{P}(\lambda \geq s) \mathbb{P}(\theta \in [x + \mu - s, \beta])\} \left(-\log(\mathbb{P}(\lambda \geq y_1)) + \log(\mathbb{P}(\lambda \geq 0))\right) \,,$$

where we used that $\mathbb{P}(\lambda \geq \max\{x + \mu - \beta, 0\}) \leq \mathbb{P}(\lambda \geq 0)$.

For all $z \in (\mu, x + \mu - y_1)$, we have $\mathbb{P}(\theta \in [z, \beta]) \geq \mathbb{P}(\theta \in [x + \mu - y_1, \beta]) > 0$. Then, we obtain

$$\mathbb{P}\left(\theta \in (\mu, x + \mu - y_1), \, \lambda \geq x + \mu - \theta\right)$$

$$= \int_{z \in (\mu, x + \mu - y_1)} \mathbb{P}(\lambda \geq x + \mu - z) \mathrm{d}G(z)$$

$$= \int_{z \in (\mu, x + \mu - y_1)} \mathbb{P}(\lambda \geq x + \mu - z) \mathbb{P}(\theta \in [z, \beta]) \frac{1}{\mathbb{P}(\theta \in [z, \beta])} \mathrm{d}G(z)$$

$$\leq \sup_{z \in (\mu, x + \mu - y_1)} \{\mathbb{P}(\lambda \geq x + \mu - z) \mathbb{P}(\theta \in [z, \beta])\} \int_{z \in (\mu, x + \mu - y_1)} \frac{1}{\mathbb{P}(\theta \in [z, \beta])} \mathrm{d}G(z)$$

$$\leq \sup_{s \in (y_1, x)} \{\mathbb{P}(\lambda \geq s) \mathbb{P}(\theta \in [x + \mu - s, \beta])\}$$
$$\cdot \left(-\log(\mathbb{P}(\theta \in [x + \mu - y_1, \beta])) + \log(\mathbb{P}(\theta \in [\mu, \beta]))\right) \,.$$

**C. Combining Results.** Putting everything together, we have, for all $y_1 \in (\max\{x + \mu - \beta, 0\}, x)$,

$$\mathbb{P}\left(\theta + \lambda \geq \mu + x\right) \leq \mathbb{P}(\lambda \geq x) \mathbb{P}(\theta \in [\alpha, \mu]) + \mathbb{P}(\lambda \leq 0) \mathbb{P}(\theta \in [\min\{\beta, \mu + x\}, \beta])$$
$$+ \mathbb{P}(\theta \in [x + \mu - y_1, \beta]) \mathbb{P}(\lambda \geq y_1)$$
$$+ \sup_{s \in (\max\{x + \mu - \beta, 0\}, y_1)} \{\mathbb{P}(\lambda \geq s) \mathbb{P}(\theta \in [x + \mu - s, \beta])\} \left(-\log(\mathbb{P}(\lambda \geq y_1)) + \log(\mathbb{P}(\lambda \geq 0))\right)$$
$$+ \sup_{s \in (y_1, x)} \{\mathbb{P}(\lambda \geq s) \mathbb{P}(\theta \in [x + \mu - s, \beta])\} \left(-\log(\mathbb{P}(\theta \in [x + \mu - y_1, \beta])) + \log(\mathbb{P}(\theta \in [\mu, \beta]))\right)$$
$$\leq \mathbb{P}(\lambda \geq x) \mathbb{P}(\theta \in [\alpha, \mu]) + \mathbb{P}(\lambda \leq 0) \mathbb{P}(\theta \in [\min\{\beta, \mu + x\}, \beta])$$
$$+ \mathbb{P}(\theta \in [x + \mu - y_1, \beta]) \mathbb{P}(\lambda \geq y_1)$$
$$+ \sup_{s \in (\max\{x + \mu - \beta, 0\}, x)} \{\mathbb{P}(\lambda \geq s) \mathbb{P}(\theta \in [x + \mu - s, \beta])\}$$
$$\cdot \left(-\log(\mathbb{P}(\lambda \geq y_1) \mathbb{P}(\theta \in [x + \mu - y_1, \beta])) + \log(\mathbb{P}(\theta \in [\mu, \beta]) \mathbb{P}(\lambda \geq 0))\right) \,,$$

where the second inequality is obtained by extending the two suprema to $(\max\{x + \mu - \beta, 0\}, x)$, which is possible since multiplied by a positive value, and factorizing them together. Taking

$$y_1^\star \in \underset{s \in (\max\{x + \mu - \beta, 0\}, x)}{\arg\max} \{\mathbb{P}(\lambda \geq s) \mathbb{P}(\theta \in [x + \mu - s, \beta])\}\} \,,$$

and the change of variable $z = x + \mu - s$, i.e.,

$$\sup_{s \in (\max\{x + \mu - \beta, 0\}, x)} \{\mathbb{P}(\lambda \geq s) \mathbb{P}(\theta \in [x + \mu - s, \beta])\}$$

$$= \sup_{z \in (\mu, \min\{\beta, x+\mu\})} \{\mathbb{P}(\theta \in [z, \beta]) \mathbb{P}(\lambda \geq x + \mu - z)\} \,,$$

concludes the proof of the upper bound on the upper tail.

**II. Lower Bound on Upper Tail.** Let $z \in (\mu, \min\{\beta, \mu + x\})$. Then, we have directly that

$$\{\theta \in [z, \beta], \ \lambda \geq \mu + x - z\} \subseteq \{\theta + \lambda \geq \mu + x\} \,.$$

Using independence, we obtain

$$\mathbb{P}(\theta \in [z, \beta]) \mathbb{P}(\lambda \geq \mu + x - z) = \mathbb{P}(\theta \in [z, \beta], \lambda \geq \mu + x - z) \leq \mathbb{P}(\theta + \lambda \geq \mu + x) \,.$$

Taking the supremum over $z \in (\mu, \min\{\beta, \mu + x\})$ on the left hand side concludes the proof of the lower bound on the upper tail.

**III. Upper/Lower Bound on Lower Tail.** The third and forth inequalities are a direct consequence of the first and second inequalities applied to the two independent real random variables $-\theta$ and $-\lambda$ since (1) $-\theta$ has bounded support included in $[-\beta, -\alpha]$ and mean $-\mu \in (-\beta, -\alpha)$, and (2) $-\lambda$ has zero mean. Namely,

$$\mathbb{P}(\theta + \lambda \leq \mu - x) = \mathbb{P}(-\theta - \lambda \geq -\mu + x) \,,$$
$$\mathbb{P}(\lambda \leq -x)\mathbb{P}(\theta \in [\mu, \beta]) = \mathbb{P}(-\lambda \geq x)\mathbb{P}(-\theta \in [-\beta, -\mu]) \,,$$
$$\mathbb{P}(\lambda \geq 0)\mathbb{P}(\theta \in [\alpha, \max\{\alpha, \mu - x\}]) = \mathbb{P}(-\lambda \leq 0)\mathbb{P}(-\theta \in [\min\{-\alpha, x - \mu\}, -\alpha]) \,,$$
$$\mathbb{P}(\theta \in [\alpha, \mu])\mathbb{P}(\lambda \leq 0) = \mathbb{P}(-\theta \in [-\mu, -\alpha])\mathbb{P}(-\lambda \geq 0) \,,$$
$$\sup_{z \in (\max\{\alpha, \mu - x\}, \mu)} \{\mathbb{P}(\theta \in [\alpha, z])\mathbb{P}(\lambda \leq \mu - x - z)\}$$
$$= \sup_{\tilde{z} \in (-\mu, \min\{-\alpha, -\mu + x\})} \{\mathbb{P}(-\theta \in [\tilde{z}, -\alpha])\mathbb{P}(-\lambda \geq -\mu + x - \tilde{z})\} \,,$$

where we used the change of variable $\tilde{z} = -z$. $\qquad\square$

**Properties on the Rate Function**  Lemma 11 gathers properties on the rate function $f$ in Lemmas 12 and 12.

**Lemma 11.** *Let us define*

$$\forall x \geq 0, \quad f(x) := (x + 3 - \log 2) \exp(-x) \,. \tag{24}$$

*On $\mathbb{R}_+$, the function $f$ is twice continuously differentiable, positive, decreasing and strictly convex. It satisfies $f(0) > 1$, $\lim_{x \to +\infty} f(x) = 0$ and*

$$f(x) \leq \delta \quad \Longleftrightarrow \quad x \geq \overline{W}_{-1}(\log(1/\delta) + 3 - \log 2) - 3 + \log 2 \,,$$

*where $\overline{W}_{-1}$ is defined in Lemma 52.*

*Proof.* Direct manipulation yields $f(0) = 3 - \log 2 > 1$, $\lim_{x \to +\infty} f(x) = 0$,

$$\forall x \geq 0, \quad f'(x) = -(x + 2 - \log 2) \exp(-x) < 0 \quad \text{and} \quad f''(x) = (x + 1 - \log 2) \exp(-x) > 0 \,.$$

Using that $f(x) = e^{3 - \log 2} \exp(-h(x + 3 - \log 2))$ where $h(x) = x - \log(x)$, Lemma 52 yields

$$f(x) \leq \delta \quad \Longleftrightarrow \quad h(x + 3 - \log 2) \geq \log\left(e^{3 - \log 2}/\delta\right)$$
$$\Longleftrightarrow \quad \overline{W}_{-1}(\log(1/\delta) + 3 - \log 2) - 3 + \log 2 \leq x \,.$$

$\qquad\square$

**Fixed Time Upper and Lower Tails Concentration**  Lemma 12 gives an upper and lower tails bound for a sum between independent Bernoulli and Laplace i.i.d. observations for a fixed time.

**Lemma 12.** *Let $\mu \in (0, 1)$ and $\epsilon > 0$. Let $Z_t = \sum_{s \in [t]} X_s$ where $X_s \sim Ber(\mu)$ are i.i.d. observations. Let $S_t = \sum_{s \in [n_t]} Y_s$ where $Y_s \sim Lap(1/\epsilon)$ are i.i.d. observations where $(n_t)_{t \in \mathbb{N}}$ be a piece-wise constant increasing function from $\mathbb{N}$ to $\mathbb{N}$. Let $f$ as in Eq. (24). Then, for all $t \in \mathbb{N}$ and all $x > 0$,*

$$\mathbb{P}(Z_t + S_t \geq t(x + \mu)) \leq f\left(t \inf_{z \in (\mu, \min\{1, x+\mu\})} \left\{-\frac{1}{t} \log\left(\mathbb{P}(Z_t \geq tz)\mathbb{P}(S_t \geq t(x + \mu - z))\right)\right\}\right)$$

$$\mathbb{P}(Z_t + S_t \leq t(\mu - x)) \leq f\left(t \inf_{z \in (\max\{0, \mu-x\}, \mu)} \left\{-\frac{1}{t} \log\left(\mathbb{P}(Z_t \leq tz)\mathbb{P}(S_t \leq t(\mu - x - z))\right)\right\}\right)$$

*Proof.* Let $t \in \mathbb{N}$ and $x > 0$. Then, $Z_t$ and $S_t$ are two independent real random variables such that (1) $Z_t$ has bounded support included in $[0, t]$ and mean $t\mu \in (0, t)$ and (ii) $S_t$ has zero mean. By symmetry of $\text{Lap}(1/\epsilon)$ around 0, the cumulative sum of $n_t$ observations (i.e., $S_t$) is also symmetric around 0. However, $Z_t$ follows $\text{Bin}(t, \mu)$ which can be skewed. Therefore, we have

$$\mathbb{P}(S_t \geq 0) = 1/2 = \mathbb{P}(S_t \leq 0) \quad \text{and} \quad \max\{\mathbb{P}(Z_t \in [t\mu, t]), \mathbb{P}(Z_t \in [0, t\mu])\} \leq 1 \,,$$
$$\forall z \in [0, 1], \quad \mathbb{P}(Z_t \in [tz, t]) = \mathbb{P}(Z_t \geq tz) \quad \text{and} \quad \mathbb{P}(Z_t \in [0, tz]) = \mathbb{P}(Z_t \leq tz) \,.$$

Using that $z \mapsto \mathbb{P}(Z_t \geq tz)$ is decreasing on $(\mu, \min\{1, x + \mu\})$ and $z \mapsto \mathbb{P}(S_t \geq t(\mu - x - z)$ is increasing on $(\mu, \min\{1, x + \mu\})$, we obtain

$$\max\{\mathbb{P}(Z_t \geq t\min\{1, x + \mu\}))\mathbb{P}(S_t \leq 0), \mathbb{P}(S_t \geq tx)\mathbb{P}(Z_t \leq t\mu)\}$$
$$\leq \sup_{z \in (\mu, \min\{1, x+\mu\})} \{\mathbb{P}(Z_t \geq tz)\mathbb{P}(S_t \geq t(x + \mu - z))\} \,.$$

Let us define $g(x) := x(3 - \log(2) - \log(x))$. Using Lemma 10 for $(Z_t, S_t)$ and considering $tx > 0$ and $z \in (\mu, \min\{\beta, \mu + x\})$ (i.e., $tz \in (t\mu, t\min\{\beta, \mu + x\}))$, we obtain

$$\mathbb{P}(Z_t + S_t \geq t(x + \mu)) \leq g\left( \sup_{z \in (\mu, \min\{1, x+\mu\})} \{\mathbb{P}(Z_t \geq tz)\mathbb{P}(S_t \geq t(x + \mu - z))\} \right) \,.$$

Let $f$ as in Eq. (24) of Lemma 11. Then, we have $f(x) = g(\exp(-x))$. This concludes the proof of the upper bound on the upper tail. The second result is obtained similarly based on Lemma 10 and the above results. □

### F.4 Tails Concentration of Cumulative Laplace Distributions

We derive time-uniform (Lemma 15) and fixed-time (Lemma 16) tails concentration for the cumulative sum of i.i.d. Laplace observations. Our proof technique is based on the Chernoff method and Ville's inequality as in Eq. (26). Therefore, we need to derive the convex conjugate of the moment generating function of a Laplace distribution (Lemma 13). While the time-uniform result requires using the peeling method, the proof of the fixed-time concentration is simpler. To use the peeling method, we need to control the deviation of the process on slices of time (Lemma 14).

**Convex Conjugate of the Moment Generating Function of Laplace Distribution** Let $\epsilon > 0$. The moment generating function of the Laplace distribution $\text{Lap}(1/\epsilon)$ is defined as

$$\forall \lambda \in (0, \epsilon), \quad \psi_{\text{Lap}, \epsilon}(\lambda) = \log \mathbb{E}_{X \sim \text{Lap}(1/\epsilon)} [\exp(\lambda X)] = -\log(1 - \lambda^2/\epsilon^2) \,. \tag{25}$$

Lemma 13 explicits the convex conjugate of $\psi_{\text{Lap}, \epsilon}$ and its associated maximizer.

**Lemma 13.** *Let $\psi_{Lap, \epsilon}$ as in Eq.* (25). *Let us define*

$$\forall x > 0, \quad \psi_{Lap, \epsilon}^{\star}(x) := \max_{\lambda \in (0, \epsilon)} \{\lambda x - \psi_{Lap, \epsilon}(\lambda)\} \quad and \quad \lambda(x) := \arg\max_{\lambda \in (0, \epsilon)} \{\lambda x - \psi_{Lap, \epsilon}(\lambda)\} \,.$$

*Then, for all $x > 0$, we have*

$$\lambda(x) = \frac{1}{x} \left( \sqrt{1 + (x\epsilon)^2} - 1 \right) \in (0, \epsilon) \quad and \quad \psi_{Lap, \epsilon}^{\star}(x) = h(\epsilon x) > 0 \,.$$

*where $h$ is defined in Eq.* (31).

*Proof.* Let $f(\lambda) = \lambda x - \psi_{\text{Lap}, \epsilon}(\lambda)$ for all $\lambda \in (0, \epsilon)$. Direct manipulation yields that

$$\forall \lambda \in (0, \epsilon), \quad f'(\lambda) = x - \frac{2\lambda}{\epsilon^2 - \lambda^2} \quad \text{and} \quad f''(\lambda) = -2\frac{\epsilon^2 + \lambda^2}{(\epsilon^2 - \lambda^2)^2} < 0 \,.$$

Moreover, for all $\lambda \in (0, \epsilon)$, we have

$$f'(\lambda) = 0 \quad \Longleftrightarrow \quad \lambda^2 + 2\lambda/x - \epsilon^2 = 0 \quad \Longleftrightarrow \quad \lambda = \frac{1}{x} \left( \sqrt{1 + (x\epsilon)^2} - 1 \right) \,.$$

We used that the second solution to the second order polynomial equation is negative, hence not in $(0, \epsilon)$. Moreover, it is direct to see that $\frac{1}{x} \left( \sqrt{1 + (x\epsilon)^2} - 1 \right) \in (0, \epsilon)$ since $\sqrt{1 + x^2} - 1 \leq x$, as it

is equivalent to $1 + x^2 \le (x+1)^2$ which is true when $x > 0$. Since $f$ is strictly concave, the above computation gives its unique maximizer on $(0, \epsilon)$, namely we have $\lambda(x) = \frac{1}{x} \left( \sqrt{1 + (x\epsilon)^2} - 1 \right)$. Moreover, the convex conjugate of $\psi_{\text{Lap},\epsilon}$ is

$$\psi_{\text{Lap},\epsilon}^\star(x) = f(\lambda(x)) = \sqrt{1 + (x\epsilon)^2} - 1 + \log \left( 1 - \frac{1}{(x\epsilon)^2} \left( \sqrt{1 + (x\epsilon)^2} - 1 \right)^2 \right)$$

$$= \sqrt{1 + (x\epsilon)^2} - 1 + \log \left( \frac{2}{(x\epsilon)^2} \left( \sqrt{1 + (x\epsilon)^2} - 1 \right) \right) .$$

This concludes the proof. $\qquad \square$

**Test Martingale for Cumulative Laplace Observations** Let $\epsilon > 0$ and $S_t = \sum_{s \in [t]} Y_s$ where $Y_s \sim \text{Lap}(1/\epsilon)$ are i.i.d. observations. Let us define

$$\forall \lambda \in (0, \epsilon), \quad M_t(\lambda) := \exp(\lambda S_t - t\psi_{\text{Lap},\epsilon}(\lambda)) .$$

It is direct to see that $M_0(\lambda) = 0$ almost surely and

$$\mathbb{E}[M_t(\lambda) \mid \mathcal{F}_{-1}] = M_{t-1}(\lambda)\mathbb{E}_{X \sim \text{Lap}(1/\epsilon)}[\exp(\lambda X - \psi_{\text{Lap},\epsilon}(\lambda))] = M_{t-1}(\lambda) .$$

Therefore, $M_t(\lambda)$ is a test martingale. Using Ville's inequality [Ville, 1939] yields that

$$\forall \delta \in (0, 1), \ \forall \lambda \in (0, \epsilon), \quad \mathbb{P}\left( \exists t \in \mathbb{N}, \ \lambda S_t - t\psi_{\text{Lap},\epsilon}(\lambda) \ge \log(1/\delta) \right) \le \delta . \qquad (26)$$

**Time Uniform Tails Concentration** Lemma 14 controls the deviation of the process on slices of time.

**Lemma 14.** *Let $\epsilon > 0$ and $S_t = \sum_{s \in [t]} Y_s$ where $Y_s \sim Lap(1/\epsilon)$ are i.i.d. observations. Let $N > 0$. For all $x > 0$, there exists $\lambda(x)$ such that for all $t \ge N$,*

$$\{S_t \ge tx\} \subseteq \{\lambda(x)S_t - t\psi_{Lap,\epsilon}(\lambda(x)) \ge Nh(\epsilon x)\} ,$$

*where $\lambda(x)$ as in Lemma 13 and $h$ as in Eq. (31).*

*Proof.* Using Lemma 13, we obtain $\lambda(x) \in (0, \epsilon)$ and $\psi_{\text{Lap},\epsilon}^\star(x) = h(\epsilon x) > 0$, hence $t\psi_{\text{Lap},\epsilon}^\star(x) \ge N\psi_{\text{Lap},\epsilon}^\star(x)$ for $t \ge N$. Then, direct computations yield

$$S_t \ge tx \implies \lambda(x)S_t - t\psi_{\text{Lap},\epsilon}(\lambda(x)) \ge t \left( x\lambda(x) - \psi_{\text{Lap},\epsilon}(\lambda(x)) \right) = t\psi_{\text{Lap},\epsilon}^\star(x)$$

$$\implies \lambda S_t - t\psi_{\text{Lap},\epsilon}(\lambda) \ge N\psi_{\text{Lap},\epsilon}^\star(x) = Nh(\epsilon x) .$$

This concludes the proof. $\qquad \square$

Lemma 15 gives time-uniform tails concentration for the cumulative sum of i.i.d. Laplace observations. It is obtained by applying Lemma 14 on slices of time with geometric growth rate.

**Lemma 15.** *Let $\delta \in (0, 1)$. Let $\gamma > 0$, $s > 1$ and $\zeta$ be the Riemann $\zeta$ function. Let $h^{-1}$ be the inverse of $h$ defined as in Eq. (31), which is well-defined by Lemma 28. Let $\epsilon > 0$ and $S_t = \sum_{s \in [t]} Y_s$ where $Y_s \sim Lap(1/\epsilon)$ are i.i.d. observations. Then,*

$$\mathbb{P}\left( \exists t \in \mathbb{N}, \ S_t \ge \frac{t}{\epsilon} h^{-1} \left( \frac{1+\gamma}{t} \left( \log \left( \frac{\zeta(s)}{\delta} \right) + s \log \left( 1 + \log_{1+\gamma} t \right) \right) \right) \right) \le \delta ,$$

$$\mathbb{P}\left( \exists t \in \mathbb{N}, \ S_t \le -\frac{t}{\epsilon} h^{-1} \left( \frac{1+\gamma}{t} \left( \log \left( \frac{\zeta(s)}{\delta} \right) + s \log \left( 1 + \log_{1+\gamma} t \right) \right) \right) \right) \le \delta .$$

*Proof.* Let us define the geometric grid $N_i = (1 + \gamma)^{i-1}$, hence we have $\mathbb{N} = \bigcup_{i \in \mathbb{N}} [N_i, N_{i+1})$. For all $i \in \mathbb{N}$, let $x_i(\delta) > 0$ to be defined later, and $\lambda(x_i(\delta))$ as in Lemma 14. For all $t \in \mathbb{N}$, let $g(t, \delta)$ to be defined later such that $g(t, \delta) \ge x_i(\delta)$ for $t \in [N_i, N_{i+1})$. Using Lemma 14 with $x_i(\delta) > 0$ and $g(t, \delta) \ge x_i(\delta)$ for $t \in [N_i, N_{i+1})$, a union bound yields that

$$\mathbb{P}\left( \exists t \in \mathbb{N}, \ S_t \ge tg(t, \delta) \right)$$

$$\leq \sum_{i \in \mathbb{N}} \mathbb{P}\left(\exists t \in [N_i, N_{i+1}) : S_t \geq t x_i(\delta)\right)$$

$$\leq \sum_{i \in \mathbb{N}} \mathbb{P}\left(\exists t \in [N_i, N_{i+1}) : \lambda(x_i(\delta)) S_t - t \psi_{\text{Lap}, \epsilon}(\lambda(x_i(\delta))) \geq N_i h(\epsilon x_i(\delta))\right)$$

$$\leq \sum_{i \in \mathbb{N}} e^{-N_i h(\epsilon x_i(\delta))},$$

where the last inequality uses Ville's inequality as in Eq. (26) for all $i \in \mathbb{N}$. Let us define

$$g(t, \delta) = \frac{1}{\epsilon} h^{-1}\left(\frac{1+\gamma}{t}\left(\log\left(\frac{\zeta(s)}{\delta}\right) + s \log\left(1 + \log_{1+\gamma}(t)\right)\right)\right),$$

$$x_i(\delta) = \frac{1}{\epsilon} h^{-1}\left(\frac{1}{N_i} \log\left(\frac{i^s \zeta(s)}{\delta}\right)\right).$$

Using Lemma 28, we obtain that $x_i(\delta) > 0$ and that $h^{-1}$ is increasing on $\mathbb{R}_+^\star$. Using $t \in [N_i, N_{i+1})$ and $i = 1 + \log_{1+\gamma} N_i$, we obtain

$$g(t, \delta) \geq \frac{1}{\epsilon} h^{-1}\left(\frac{1}{N_i}\left(\log\left(\frac{\zeta(s)}{\delta}\right) + s \log\left(1 + \log_{1+\gamma}(t)\right)\right)\right)$$

$$\geq \frac{1}{\epsilon} h^{-1}\left(\frac{1}{N_i} \log\left(\frac{i^s \zeta(s)}{\delta}\right)\right) = x_i(\delta).$$

Therefore, we have

$$\mathbb{P}\left(\exists t \in \mathbb{N}, \ S_t \geq t g(t, \delta)\right) \leq \sum_{i \in \mathbb{N}} e^{-N_i h(\epsilon x_i(\delta))} \leq \frac{\delta}{\zeta(s)} \sum_{i \in \mathbb{N}} \frac{1}{i^s} = \delta.$$

This concludes the proof of the first result.

By symmetry of the $\text{Lap}(1/\epsilon)$ around zero, the cumulative sum of i.i.d. observations is symmetric around zero. Combining the first result with the symmetry around zero yields the second result. $\quad\square$

**Fixed Time Tails Concentration**  When the time is fixed and not random, there is no need to consider slices of time and we can directly control the deviation of the process.

**Lemma 16.** *Let $\epsilon > 0$ and $S_t = \sum_{s \in [t]} Y_s$ where $Y_s \sim Lap(1/\epsilon)$ are i.i.d. observations. Let $h$ as in Eq.* (31). *Then,*

$$\forall t \in \mathbb{N}, \forall x > 0, \quad \mathbb{P}(S_t \geq tx) \leq \exp(-th(\epsilon x)),$$

$$\forall t \in \mathbb{N}, \forall x > 0, \quad \mathbb{P}(S_t \leq -tx) \leq \exp(-th(\epsilon x)).$$

*Proof.* The first result can be obtained with the same manipulation as in the proof of Lemma 15, i.e., combining Ville's inequality in Eq. (26) with Lemma 14 at $N = t$.

By symmetry of the $\text{Lap}(1/\epsilon)$ around zero, the cumulative sum of i.i.d. observations is symmetric around zero. Combining the first result with the symmetry around zero yields the second result. $\quad\square$

### F.5  Fixed Time Tails Concentration of Cumulative Bernoulli Distributions

The fixed time upper and lower tail concentration of cumulative Bernoulli distributions are well-studied. Using the Chernoff method yields Lemma 17, whose proof is omitted since it is a classic result.

**Lemma 17** (Chernoff Tail Bound for Bernoulli Distributions [Boucheron et al., 2013]). *Let $\mu \in (0, 1)$ and $Z_t = \sum_{s \in [t]} X_s$ where $X_s \sim Ber(\mu)$ are i.i.d. observations. Then,*

$$\forall t \in \mathbb{N}, \forall x \in (\mu, 1), \quad \mathbb{P}(Z_t \geq tx) \leq \exp(-t\text{kl}(x, \mu)),$$

$$\forall t \in \mathbb{N}, \forall x \in (0, \mu), \quad \mathbb{P}(Z_t \leq tx) \leq \exp(-t\text{kl}(x, \mu)).$$

### F.6 Fixed Time Tails Concentration for a Convolution between Bernoulli and Laplace Distributions

We provide upper (Lemma 18) and lower (Lemma 19) tail concentrations for a sum (i.e., convolution) between independent Bernoulli and Laplace i.i.d. observations for a fixed time.

**Fixed Time Upper Tail Concentration**  Lemma 18 shows an upper tail concentration on the sum (i.e., convolution) between independent Bernoulli and Laplace i.i.d. observations.

**Lemma 18.** *Let $\mu \in (0,1)$ and $Z_t = \sum_{s \in [t]} X_s$ where $X_s \sim Ber(\mu)$ are i.i.d. observations. Let $(n_t)_{t \in \mathbb{N}}$ be a piece-wise constant increasing function from $\mathbb{N}$ to $\mathbb{N}$. Let $\epsilon > 0$ and $S_t = \sum_{s \in [n_t]} Y_s$ where $Y_s \sim Lap(1/\epsilon)$ are i.i.d. observations. Then,*

$$\forall t \in \mathbb{N}, \ \forall x > 0, \quad \mathbb{P}(Z_t + S_t \geq t(\mu + x)) \leq f\left(t\widetilde{d}_\epsilon^-(\mu + x, \mu, t/n_t)\right) ,$$

*where $f$ is defined in Eq. (24) and $\widetilde{d}_\epsilon^-$ is defined in Eq. (32).*

*Proof.* Let $t \in \mathbb{N}$ and $x > 0$. Combining Lemmas 16 and 17, we obtain, for all $x > 0$ and all $z \in (\mu, \min\{1, x + \mu\})$,

$$-\frac{1}{t} \log\left(\bar{G}_t(tz)\bar{F}_{n_t}(t(x + \mu - z))\right) \geq \mathrm{kl}(z, \mu) + \frac{n_t}{t} h\left(\frac{t}{n_t}\epsilon(x + \mu - z)\right) ,$$

where we used that $x + \mu - z > 0$. Taking the infimum on $(\mu, \min\{1, x + \mu\})$ on both sides and using that $[x + \mu]_0^1 = \min\{1, x + \mu\} > \mu$, we obtain

$$\inf_{z \in (\mu, \min\{1, x+\mu\})} \left\{ -\frac{1}{t} \log\left(\bar{G}_t(tz)\bar{F}_{n_t}(t(x + \mu - z))\right) \right\} \geq \widetilde{d}_\epsilon^-(\mu + x, \mu, t/n_t) ,$$

where $\widetilde{d}_\epsilon^-$ is defined in Eq. (32). Since $f$ is decreasing on $\mathbb{R}_+$ (Lemma 11), using Lemma 12 yields

$$\mathbb{P}(Z_t + S_t \geq t(x + \mu)) \leq f\left(t\widetilde{d}_\epsilon^-(\mu + x, \mu, t/n_t)\right) .$$

which concludes the proof. $\qquad\square$

**Fixed Time Lower Tail Concentration**  Lemma 19 shows a lower tail concentration on the sum (i.e., convolution) between independent Bernoulli and Laplace i.i.d. observations.

**Lemma 19.** *Let $\mu \in (0,1)$ and $Z_t = \sum_{s \in [t]} X_s$ where $X_s \sim Ber(\mu)$ are i.i.d. observations. Let $(n_t)_{t \in \mathbb{N}}$ be a piece-wise constant increasing function from $\mathbb{N}$ to $\mathbb{N}$. Let $\epsilon > 0$ and $S_t = \sum_{s \in [n_t]} Y_s$ where $Y_s \sim Lap(1/\epsilon)$ are i.i.d. observations. Then,*

$$\forall t \in \mathbb{N}, \ \forall x > 0, \quad \mathbb{P}(Z_t + S_t \leq t(\mu - x)) \leq f\left(t\widetilde{d}_\epsilon^+(\mu - x, \mu, t/n_t)\right) ,$$

*where $f$ is defined in Eq. (24) and $\widetilde{d}_\epsilon^+$ is defined in Eq. (32).*

*Proof.* Let $t \in \mathbb{N}$ and $x > 0$. Combining Lemmas 16 and 17, we obtain, for all $x > 0$ and all $z \in (\max\{0, \mu - x\}, \mu)$,

$$-\frac{1}{t} \log\left(G_t(tz)F_{n_t}(t(\mu - x - z))\right) \geq \mathrm{kl}(z, \mu) + \frac{n_t}{t} h\left(\frac{t}{n_t}\epsilon(z + x - \mu)\right) ,$$

where we used that $\mu - x - z < 0$. Taking the infimum on $z \in (\max\{0, \mu - x\}, \mu)$ on both sides and using that $[\mu - x]_0^1 = \max\{0, \mu - x\} < \mu$, we obtain

$$\inf_{z \in (\max\{0, \mu-x\}, \mu)} \left\{ -\frac{1}{t} \log\left(G_t(tz)F_{n_t}(t(\mu - x - z))\right) \right\} \geq \widetilde{d}_\epsilon^+(\mu - x, \mu, t/n_t) ,$$

where $\widetilde{d}_\epsilon^+$ is defined in Eq. (32). Since $f$ is decreasing on $\mathbb{R}_+$ (Lemma 11), using Lemma 12 yields

$$\mathbb{P}(Z_t + S_t \geq t(\mu - x)) \leq f\left(t\widetilde{d}_\epsilon^+(\mu - x, \mu, t/n_t)\right) ,$$

which concludes the proof. $\qquad\square$

### F.7 Geometric Grid Time Uniform Tails Concentration for a Convolution between Bernoulli and Laplace Distributions

We provide upper (Lemma 20) and lower (Lemma 21) tail concentrations for a sum (i.e., convolution) between independent Bernoulli and Laplace i.i.d. observations that holds time uniformly on a geometric grid.

**Geometric Grid Time Uniform Upper Tail Concentration**   Lemma 20 gives a threshold ensuring that a geometric grid time uniform upper tail concentration holds with probability at least $1 - \delta$.

**Lemma 20.** *Let $\delta \in (0,1)$. Let $(\tilde{\mu}_n, \tilde{N}_n, k_n)$ are given by $GPE_\eta(\epsilon)$. Let $s > 1$ and $\zeta$ be the Riemann $\zeta$ function. Let $\overline{W}_{-1}(x) = -W_{-1}(-e^{-x})$ for all $x \geq 1$, where $W_{-1}$ is the negative branch of the Lambert W function. It satisfies $\overline{W}_{-1}(x) \approx x + \log x$, see Lemma 52. Let us define*

$$c(k, \delta) = \overline{W}_{-1}\left(\log\left(1/\delta\right) + s\log(k) + \log(\zeta(s)) + 3 - 2\log 2\right) - 3 + 2\log 2 . \qquad (27)$$

*For all $a \in [K]$, let us define*

$$\mathcal{E}_{\delta, a, -} = \left\{\forall n \in \mathbb{N}, \ \tilde{N}_{n,a}\widetilde{d}_\epsilon^-\left(\tilde{\mu}_{n,a}, \mu_a, \tilde{N}_{n,a}/k_{n,a}\right) \leq c(k_{n,a}, \delta)\right\} , \qquad (28)$$

*where $\widetilde{d}_\epsilon^-$ is defined in Eq. (32). Then, we have $\mathbb{P}_{\boldsymbol{\nu}\pi}(\mathcal{E}_{\delta, a, -}^{\complement}) \leq \delta$ for all $a \in [K]$.*

*Proof.* Let us define the geometric grid $N_i = (1 + \eta)^{i-1}$, hence we have $\mathbb{N} = \bigcup_{i \in \mathbb{N}}[N_i, N_{i+1})$. Let $a \in [K]$. If $\tilde{N}_{n,a} \in [N_i, N_{i+1})$, then we have $\tilde{N}_{n,a} = \lceil N_i \rceil$ and $k_{n,a} = i$. By union bound, we obtain

$$\mathbb{P}_{\boldsymbol{\nu}\pi}(\mathcal{E}_{\delta, a, -}^{\complement}) = \mathbb{P}_{\boldsymbol{\nu}\pi}\left(\exists n \in \mathbb{N}, \ \tilde{N}_{n,a}\widetilde{d}_\epsilon^-\left(\tilde{\mu}_{n,a}, \mu_a, \tilde{N}_{n,a}/k_{n,a}\right) \geq c(k_{n,a}, \delta)\right)$$

$$\leq \sum_{i \in \mathbb{N}} \mathbb{P}_{\boldsymbol{\nu}\pi}\left(\exists i \in \mathbb{N}, \ (\tilde{N}_{n,a}, k_{n,a}) = (\lceil N_i \rceil, i) \ \wedge \ \tilde{N}_{n,a}\widetilde{d}_\epsilon^-\left(\tilde{\mu}_{n,a}, \mu_a, \tilde{N}_{n,a}/k_{n,a}\right) \geq c(k_{n,a}, \delta)\right)$$

$$= \sum_{i \in \mathbb{N}} \mathbb{P}\left(\lceil N_i \rceil \widetilde{d}_\epsilon^-\left((Z_{\lceil N_i \rceil} + S_i)/\lceil N_i \rceil, \mu_a, \lceil N_i \rceil/i\right) \geq c(i, \delta)\right) ,$$

where $Z_{\lceil N_i \rceil}$ is the cumulative sum of $\lceil N_i \rceil$ i.i.d. observations from $\mathrm{Ber}(\mu_a)$ and $S_i$ is the cumulative sum of $i$ i.i.d. observations from $\mathrm{Lap}(1/\epsilon)$.

For all $i \in \mathbb{N}$, let $x_i > 0$ be the unique solution of $\lceil N_i \rceil \widetilde{d}_\epsilon^-(x + \mu_a, \mu_a, \lceil N_i \rceil/i) = c(i, \delta)$, which exists by Lemma 33. Then, we obtain

$$\mathbb{P}\left(\lceil N_i \rceil \widetilde{d}_\epsilon^-\left((Z_{\lceil N_i \rceil} + S_i)/\lceil N_i \rceil, \mu_a, \lceil N_i \rceil/i\right) \geq c(i, \delta)\right)$$

$$= \mathbb{P}\left(\widetilde{d}_\epsilon^-\left((Z_{\lceil N_i \rceil} + S_i)/\lceil N_i \rceil, \mu_a, \lceil N_i \rceil/i\right) \geq \widetilde{d}_\epsilon^-\left(x_i + \mu_a, \mu_a, \lceil N_i \rceil/i\right)\right)$$

$$\leq \mathbb{P}(Z_{\lceil N_i \rceil} + S_i \geq \lceil N_i \rceil(x_i + \mu_a)) \leq f\left(\lceil N_i \rceil \widetilde{d}_\epsilon^-\left(x_i + \mu_a, \mu_a, \lceil N_i \rceil/i\right)\right) = f\left(c(i, \delta)\right) ,$$

where $f(x) := (x + 3 - \log 2)\exp(-x)$ for all $x \geq 0$. The first and the last equalities are otained by definition of $x_i$, i.e., $\lceil N_i \rceil \widetilde{d}_\epsilon^-(x + \mu_a, \mu_a, \lceil N_i \rceil/i) = c(i, \delta)$. The first inequality is obtained by using Lemma 34, and the second inequality is obtained by using Lemma 18. Using Lemma 11 yields

$$f(x) \leq \delta \quad \iff \quad \overline{W}_{-1}(\log\left(1/\delta\right) + 3 - \log 2) - 3 + \log 2 \leq x .$$

Taking

$$c(i, \delta) = \overline{W}_{-1}(\log\left(i^s\zeta(s)/\delta\right) + 3 - \log 2) - 3 + \log 2 ,$$

we can conclude the proof since $\mathbb{P}_{\boldsymbol{\nu}\pi}(\mathcal{E}_{\delta, a, -}^{\complement}) \leq \sum_{i \in \mathbb{N}} f\left(c(i, \delta)\right) \leq \sum_{i \in \mathbb{N}} \frac{\delta}{\zeta(s)i^s} \leq \delta.$ $\qquad\square$

**Geometric Grid Time Uniform Lower Tail Concentration**   Lemma 21 gives a threshold ensuring that a geometric grid time uniform lower tail concentration holds with probability at least $1 - \delta$.

**Lemma 21.** *Let $\delta \in (0,1)$. Let $(\tilde{\mu}_n, \tilde{N}_n, k_n)$ are given by $GPE_\eta(\epsilon)$. Let $c$ as in Eq. (27). For all $a \in [K]$, let us define*

$$\mathcal{E}_{\delta,a,+} = \left\{ \forall n \in \mathbb{N}, \ \tilde{N}_{n,a} \widetilde{d}_\epsilon^+ (\tilde{\mu}_{n,a}, \mu_a, \tilde{N}_{n,a}/k_{n,a}) \le c(k_{n,a}, \delta) \right\}, \quad (29)$$

*where $\widetilde{d}_\epsilon^+$ is defined in Eq. (32). Then, we have $\mathbb{P}_{\boldsymbol{\nu}\pi}(\mathcal{E}_{\delta,a,+}^\complement) \le \delta$ for all $a \in [K]$.*

*Proof.* Let us define the geometric grid $N_i = (1+\eta)^{i-1}$, hence we have $\mathbb{N} = \bigcup_{i\in\mathbb{N}}[N_i, N_{i+1})$. Let $a \in [K]$. If $\tilde{N}_{n,a} \in [N_i, N_{i+1})$, then we have $\tilde{N}_{n,a} = \lceil N_i \rceil$ and $k_{n,a} = i$. By union bound, we obtain

$$\mathbb{P}_{\boldsymbol{\nu}\pi}(\mathcal{E}_{\delta,a,+}^\complement) = \mathbb{P}_{\boldsymbol{\nu}\pi}\left( \exists n \in \mathbb{N}, \ \tilde{N}_{n,a} \widetilde{d}_\epsilon^+ (\tilde{\mu}_{n,a}, \mu_a, \tilde{N}_{n,a}/k_{n,a}) \ge c(k_{n,a}, \delta) \right)$$

$$\le \sum_{i\in\mathbb{N}} \mathbb{P}_{\boldsymbol{\nu}\pi}\left( \exists i \in \mathbb{N}, \ (\tilde{N}_{n,a}, k_{n,a}) = (\lceil N_i \rceil, i) \wedge \tilde{N}_{n,a} \widetilde{d}_\epsilon^+ (\tilde{\mu}_{n,a}, \mu_a, \tilde{N}_{n,a}/k_{n,a}) \ge c(k_{n,a}, \delta) \right)$$

$$= \sum_{i\in\mathbb{N}} \mathbb{P}\left( \lceil N_i \rceil \widetilde{d}_\epsilon^+ ((Z_{\lceil N_i \rceil} + S_i)/\lceil N_i \rceil, \mu_a, \lceil N_i \rceil/i) \ge c(i, \delta) \right),$$

where $Z_{\lceil N_i \rceil}$ is the cumulative sum of $\lceil N_i \rceil$ i.i.d. observations from $\mathrm{Ber}(\mu_a)$ and $S_i$ is the cumulative sum of $i$ i.i.d. observations from $\mathrm{Lap}(1/\epsilon)$.

For all $i \in \mathbb{N}$, let $x_i > 0$ be the unique solution of $\lceil N_i \rceil \widetilde{d}_\epsilon^+ (\mu_a - x, \mu_a, \lceil N_i \rceil/i) = c(i, \delta)$, which exists by Lemma 33. Then, we obtain

$$\mathbb{P}\left( \lceil N_i \rceil \widetilde{d}_\epsilon^+ ((Z_{\lceil N_i \rceil} + S_i)/\lceil N_i \rceil, \mu_a, \lceil N_i \rceil/i) \ge c(i, \delta) \right)$$

$$= \mathbb{P}\left( \widetilde{d}_\epsilon^+ ((Z_{\lceil N_i \rceil} + S_i)/\lceil N_i \rceil, \mu_a, \lceil N_i \rceil/i) \ge \widetilde{d}_\epsilon^+ (\mu_a - x_i, \mu_a, \lceil N_i \rceil/i) \right)$$

$$\le \mathbb{P}(Z_{\lceil N_i \rceil} + S_i \le \lceil N_i \rceil (\mu_a - x_i)) \le f\left( \lceil N_i \rceil \widetilde{d}_\epsilon^+ (\mu_a - x_i, \mu_a, \lceil N_i \rceil/i) \right) = f(c(i,\delta)) \le \frac{\delta}{\zeta(s)i^s}$$

where $f(x) := (x + 3 - \log 2)\exp(-x)$ for all $x \ge 0$. The first and the last equalities are otained by definition of $x_i$, i.e., $\lceil N_i \rceil \widetilde{d}_\epsilon^+ (\mu_a - x, \mu_a, \lceil N_i \rceil/i) = c(i, \delta)$. The first inequality is obtained by using Lemma 34, and the second inequality is obtained by using Lemma 19. The last inequality uses the same derivations based on Lemma 11 as in the proof of Lemma 20 by taking

$$c(i,\delta) = \overline{W}_{-1}(\log(i^s \zeta(s)/\delta) + 3 - \log 2) - 3 + \log 2.$$

This concludes the proof since $\mathbb{P}_{\boldsymbol{\nu}\pi}(\mathcal{E}_{\delta,a,-}^\complement) \le \sum_{i\in\mathbb{N}} \frac{\delta}{\zeta(s)i^s} \le \delta$. $\qquad\square$

# G Divergence, Transportation Cost and Characteristic Time

Appendix G is organized as follow. First, we derive regularity properties for the signed (modified) divergences $d_\epsilon^\pm$ (Appendix G.1) and $\widetilde{d}_\epsilon^\pm$ (Appendix G.1.1). Second, we derive regularity properties the (modified) transportation costs $W_{\epsilon,a,b}$ (Appendix G.2) and $\widetilde{W}_{\epsilon,a,b}$ (Appendix G.2.1) for a pair of arms $(a, b)$. Third, we study the characteristic time for $\epsilon$-global DP BAI (Appendix G.3).

## G.1 Signed Divergence

Recall $[x]_0^1 := \max\{0, \min\{1, \lambda\}\}$ and

$$\forall (\lambda, \mu) \in (0,1)^2, \quad \mathrm{kl}(\lambda, \mu) := \lambda \log\left(\frac{\lambda}{\mu}\right) + (1 - \lambda) \log\left(\frac{1-\lambda}{1-\mu}\right)$$

where kl is infinity when $\{\mu, \lambda\} \cap \{0, 1\} \ne \emptyset$. The signed divergences $d_\epsilon^\pm$ are defined in Eq. (3), i.e.,

$$\forall (\lambda, \mu) \in \mathbb{R} \times [0,1], \quad d_\epsilon^-(\lambda, \mu) := \mathbb{1}\left(\mu < [\lambda]_0^1\right) \inf_{z \in [\mu, [\lambda]_0^1]} \left\{ \mathrm{kl}(z, \mu) + \epsilon([\lambda]_0^1 - z) \right\},$$

$$d_\epsilon^+(\lambda, \mu) := \mathbb{1}\left(\mu > [\lambda]_0^1\right) \inf_{z \in [[\lambda]_0^1, \mu]} \left\{ \mathrm{kl}(z, \mu) + \epsilon(z - [\lambda]_0^1) \right\}.$$

Lemma 22 relates $\mathrm{d}_\epsilon$ and $d_\epsilon^\pm$.

**Lemma 22.** *Let $d_\epsilon^\pm$ and $\mathrm{d}_\epsilon$ as in Eq. (3) and (1). Let $(\kappa, \nu) \in \mathcal{F}^2$ with means $(\lambda, \mu) \in (0,1)^2$. Then,*

$$\mathrm{d}_\epsilon(\kappa, \nu) = \begin{cases} 0 & \text{if } \lambda = \mu \\ d_\epsilon^-(\lambda, \mu) & \text{if } \mu < \lambda \\ d_\epsilon^+(\lambda, \mu) & \text{if } \mu > \lambda \end{cases}.$$

*Proof.* When $\lambda = \mu$, we have $\mathrm{d}_\epsilon(\kappa, \nu) = 0$ by taking $\varphi = \nu$ and using the non-negativity of $\mathrm{d}_\epsilon$.

Let $\varphi \in \mathcal{F}$ with mean $z \in (0,1)$. When $\mu < \lambda$, we have

$$\mathrm{d}_\epsilon(\kappa, \nu) = \min\{ \inf_{z \in (0,\mu)} \{\mathrm{kl}(z, \mu) + \epsilon(\lambda - z)\}, \inf_{z \in [\mu, \lambda]} \{\mathrm{kl}(z, \mu) + \epsilon(\lambda - z)\},$$

$$\inf_{z \in (\lambda, 1)} \{\mathrm{kl}(z, \mu) + \epsilon(z - \lambda)\}\}$$

$$= \inf_{z \in [\mu, \lambda]} \{\mathrm{kl}(z, \mu) + \epsilon(\lambda - z)\} = d_\epsilon^-(\lambda, \mu),$$

where we partitioned $(0,1)$ and used that (1) $z \mapsto \mathrm{kl}(z, \mu) + \epsilon(z - \lambda)$ is increasing on $(\lambda, 1)$, hence the infimum on this interval is achieved at $\lambda$, and (2) $z \mapsto \mathrm{kl}(z, \mu) + \epsilon(\lambda - z)$, is decreasing on $(0, \mu)$, hence the infimum on this interval is achieved at $\mu$.

When $\mu > \lambda$, we have

$$\mathrm{d}_\epsilon(\kappa, \nu) = \min\{ \inf_{z \in (0,\lambda)} \{\mathrm{kl}(z, \mu) + \epsilon(\lambda - z)\}, \inf_{z \in [\lambda, \mu]} \{\mathrm{kl}(z, \mu) + \epsilon(z - \lambda)\},$$

$$\inf_{z \in (\mu, 1)} \{\mathrm{kl}(z, \mu) + \epsilon(z - \lambda)\}\}$$

$$= \inf_{z \in [\lambda, \mu]} \{\mathrm{kl}(z, \mu) + \epsilon(z - \lambda)\} = d_\epsilon^+(\lambda, \mu),$$

where we partitioned $(0,1)$ and used that (1) $z \mapsto \mathrm{kl}(z, \mu) + \epsilon(z - \lambda)$ is increasing on $(\mu, 1)$, hence the infimum on this interval is achieved at $\mu$, and (2) $z \mapsto \mathrm{kl}(z, \mu) + \epsilon(\lambda - z)$, is decreasing on $(0, \lambda)$, hence the infimum on this interval is achieved at $\lambda$. $\qquad \square$

Lemma 23 shows a strong link between $d_\epsilon^\pm$. This symmetry property can be used to carry regularity properties from $d_\epsilon^+$ to $d_\epsilon^-$, and vice versa.

**Lemma 23.** *Let $d_\epsilon^\pm$ as in Eq. 3. For all $\mu \in [0,1]$ and all $\lambda \in \mathbb{R}$,*

$$d_\epsilon^+(1 - \lambda, 1 - \mu) = d_\epsilon^-(\lambda, \mu) \quad \text{and} \quad d_\epsilon^-(1 - \lambda, 1 - \mu) = d_\epsilon^+(\lambda, \mu).$$

*Proof.* By definitions and change of variable $\tilde{z} = 1 - z$ and $\mathrm{kl}(1 - \tilde{z}, 1 - \mu) = \mathrm{kl}(\tilde{z}, \mu)$, we obtain

$$d_\epsilon^+(1 - \lambda, 1 - \mu) = \mathbb{1}\left(\mu < [\lambda]_0^1\right) \inf_{z \in [1 - [\lambda]_0^1, 1 - \mu]} \{\mathrm{kl}(z, 1 - \mu) + \epsilon(\max\{0, \min\{1, \lambda\} - (1 - z)\})\}$$

$$= \mathbb{1}\left(\mu < [\lambda]_0^1\right) \inf_{\tilde{z} \in [\mu, [\lambda]_0^1]} \{\mathrm{kl}(1 - \tilde{z}, 1 - \mu) + \epsilon(\max\{0, \min\{1, \lambda\} - \tilde{z}\})\}$$

$$= \mathbb{1}\left(\mu < [\lambda]_0^1\right) \inf_{\tilde{z} \in [\mu, [\lambda]_0^1]} \{\mathrm{kl}(\tilde{z}, \mu) + \epsilon(\max\{0, \min\{1, \lambda\} - \tilde{z}\})\} = d_\epsilon^-(\lambda, \mu).$$

The second equality is a consequence of the first. $\qquad \square$

Lemma 24 gathers regularity properties on the functions $g_\epsilon^\pm$ that appear in the explicit solutions of $d_\epsilon^\pm$, as shown below. Intuitively, those functionals govern locally the separation between the low privacy regime where $d_\epsilon^\pm$ is equals to the $\mathrm{kl}$ and the high privacy regime where the divergence has to be modified to account for the privacy budget $\epsilon$.

**Lemma 24.** *Let $\epsilon > 0$. Let $g_\epsilon^\pm$ defined as*

$$\forall x \in [0,1], \quad g_\epsilon^+(x) := \frac{x}{x(1 - e^\epsilon) + e^\epsilon} \quad \text{and} \quad g_\epsilon^-(x) := \frac{x e^\epsilon}{x(e^\epsilon - 1) + 1}. \tag{30}$$

*On $[0,1]$, the function $g_\epsilon^+$ is twice continuously differentiable, increasing and strictly convex. It satisfies $g_\epsilon^+(0) = 0$, $g_\epsilon^+(1) = 1$ and $g_\epsilon^+(x) < x$ on $(0,1)$. On $[0,1]$, the function $g_\epsilon^-$ is twice continuously differentiable, increasing and strictly concave. It satisfies $g_\epsilon^-(0) = 0$, $g_\epsilon^-(1) = 1$ and $g_\epsilon^-(x) > x$ on $(0,1)$. For all $x \in [0,1]$, we have $g_\epsilon^+(g_\epsilon^-(x)) = x$ and $g_\epsilon^-(1 - x) + g_\epsilon^+(x) = 1$. For all $x \in [0,1]$, we have $\lim_{\epsilon \to 0} g_\epsilon^+(x) = \lim_{\epsilon \to 0} g_\epsilon^-(x) = x$; it satisfies $\lim_{\epsilon \to +\infty} g_\epsilon^-(x) = 1$ if $x \neq 0$ and $\lim_{\epsilon \to +\infty} g_\epsilon^+(x) = 0$ if $x \neq 1$.*

*Proof.* Using that $e^\epsilon > 1$, direct computations yield that, for all $x \in [0,1]$,

$$(g_\epsilon^+)'(x) = \frac{e^\epsilon}{(x(1-e^\epsilon)+e^\epsilon)^2} > 0 \quad \text{and} \quad (g_\epsilon^+)''(x) = -2\frac{e^\epsilon(1-e^\epsilon)}{(x(1-e^\epsilon)+e^\epsilon)^2} > 0\,,$$

$$(g_\epsilon^-)'(x) = \frac{e^\epsilon}{(x(e^\epsilon-1)+1)^2} > 0 \quad \text{and} \quad (g_\epsilon^-)''(x) = -2\frac{e^\epsilon(e^\epsilon-1)}{(x(e^\epsilon-1)+1)^3} < 0\,.$$

Therefore, $g_\epsilon^+$ is twice continuously differentiable, increasing and strictly convex on $[0,1]$ and $g_\epsilon^-$ is twice continuously differentiable, increasing and strictly concave on $[0,1]$. It is direct to see that $g_\epsilon^+(0) = g_\epsilon^-(0) = 0$ and $g_\epsilon^+(1) = g_\epsilon^-(1) = 1$. Since they are strictly convex and strictly concave, we obtain $g_\epsilon^+(x) < x$ and $g_\epsilon^-(x) > x$ for all $x \in (0,1)$. It is direct to see that, for all $x \in [0,1]$, we have $g_\epsilon^+(g_\epsilon^-(x)) = x$ and $1 - g_\epsilon^+(x) = g_\epsilon^-(1-x)$. It is direct to see that, $\lim_{\epsilon\to 0} g_\epsilon^+(x) = \lim_{\epsilon\to 0} g_\epsilon^-(x) = x$ for all $x \in [0,1]$, and $\lim_{\epsilon\to+\infty} g_\epsilon^+(x) = 0$ if $x \neq 1$ and $\lim_{\epsilon\to+\infty} g_\epsilon^-(x) = 1$ if $x \neq 0$. □

Lemma 25 gathers regularity properties of $d_\epsilon^+$. In particular, it gives a closed-form solution, which is a key property used in our implementation to reduce the computational cost.

**Lemma 25.** *Let $d_\epsilon^+$ as in Eq. (3), and $g_\epsilon^\pm$ as in Eq. (30). For all $\mu \in [0,1]$ and $\lambda \in \mathbb{R}$, we have*

$$d_\epsilon^+(\lambda,\mu) = \begin{cases} 0 & \text{if } \mu \in [0,[\lambda]_0^1] \\ -\log\left(1-\mu(1-e^{-\epsilon})\right) - \epsilon[\lambda]_0^1 & \text{if } \mu \in (g_\epsilon^-([\lambda]_0^1),1] \\ \mathrm{kl}\,(\lambda,\mu) & \text{if } \lambda \in (0,1) \,\wedge\, \mu \in ([\lambda]_0^1, g_\epsilon^-([\lambda]_0^1)] \end{cases}.$$

*The function $(\lambda,\mu) \mapsto d_\epsilon^+(\lambda,\mu)$ is jointly continuous on $\mathbb{R} \times [0,1]$. For all $\mu \in [0,1]$, the function $\lambda \mapsto d_\epsilon^+(\lambda,\mu)$ is constant on $(-\infty,0]$ and on $[1,+\infty)$. Then,*

$$\forall \lambda \in (0,1), \forall \mu \in [0,1], \quad d_\epsilon^+(\lambda,\mu) = \begin{cases} 0 & \text{if } \mu \in [0,\lambda] \\ \mathrm{kl}\,(\lambda,\mu) & \mu \in (\lambda, g_\epsilon^-(\lambda)] \\ -\log\left(1-\mu(1-e^{-\epsilon})\right) - \epsilon\lambda & \text{if } \mu \in (g_\epsilon^-(\lambda),1] \end{cases}.$$

*For all $\mu \in [0,1]$, the function $\lambda \mapsto d_\epsilon^+(\lambda,\mu)$ is continuously differentiable, positive, decreasing and convex on $(0,\mu)$; it is affine with negative slope $-\epsilon$ on $(0,g_\epsilon^+(\mu))$ and twice continuously differentiable and strictly convex on $(g_\epsilon^+(\mu),\mu)$.*

*For all $\lambda \in (0,1)$, the function $\mu \mapsto d_\epsilon^+(\lambda,\mu)$ is positive, three times differentiable with continuous first derivative, increasing and strictly convex on $(\lambda,1]$; its second derivative is discontinuous at $g_\epsilon^-(\lambda)$ with gap $\frac{\partial^2 d_\epsilon^+}{\partial\mu^2}(\lambda,g_\epsilon^-(\lambda)) - \lim_{\mu\to g_\epsilon^-(\lambda)^+} \frac{\partial^2 d_\epsilon^+}{\partial\mu^2}(\lambda,\mu) > 0$. Moreover, we have*

$$\forall \mu \in (\lambda,1], \quad \frac{\partial d_\epsilon^+}{\partial\mu}(\lambda,\mu) = \begin{cases} \frac{1-e^{-\epsilon}}{1-\mu(1-e^{-\epsilon})} & \text{if } \mu \in (g_\epsilon^-(\lambda),1] \\ \frac{\mu-\lambda}{\mu(1-\mu)} & \text{if } \mu \in (\lambda, g_\epsilon^-(\lambda)] \end{cases}.$$

*The function $d_\epsilon^+$ is jointly convex on $(0,1) \times [0,1]$.*

*Proof.* Recall that $d_\epsilon^+(\lambda,\mu) = \mathbb{1}\left(\mu > [\lambda]_0^1\right) \inf_{z \in [[\lambda]_0^1,\mu]} f_\epsilon^+([\lambda]_0^1,\mu,z)$ where $f_\epsilon^+(\lambda,\mu,z) = \mathrm{kl}(z,\mu) + \epsilon(z-\lambda)$. Direct computations yield that, for all $z \in ([\lambda]_0^1,\mu)$,

$$\frac{\partial f_\epsilon^+}{\partial z}(\lambda,\mu,z) = \log\left(\frac{z(1-\mu)}{(1-z)\mu}\right) + \epsilon \quad \text{and} \quad \frac{\partial f_\epsilon^+}{\partial z}(\lambda,\mu,z) = 0 \iff z = g_\epsilon^+(\mu)\,,$$

$$\frac{\partial^2 f_\epsilon^+}{\partial z^2}(\lambda,\mu,z) = \frac{1}{z(1-z)} > 0\,.$$

Therefore, $z \to f_\epsilon^+(\lambda,\mu,z)$ is twice continuously differentiable, positive and strictly convex on $([\lambda]_0^1,\mu)$. Moreover, $z \to f_\epsilon^+(\lambda,\mu,z)$ is decreasing on $([\lambda]_0^1, \max\{g_\epsilon^+(\mu),\lambda\})$ and increasing on $(\max\{g_\epsilon^+(\mu),\lambda\},\mu)$. Using Lemma 24, we obtain

$$f_\epsilon^+(\lambda,\mu,\lambda) = \mathrm{kl}\,(\lambda,\mu)\,,$$

$$\mathrm{kl}(g_\epsilon^+(\mu),\mu) = -(g_\epsilon^+(\mu)+g_\epsilon^-(1-\mu))\log\left(\mu(1-e^\epsilon)+e^\epsilon\right) + \epsilon g_\epsilon^-(1-\mu)$$

$$= -\log\left(1 - \mu(1 - e^{-\epsilon})\right) - \epsilon g_\epsilon^+(\mu)\,,$$
$$f_\epsilon^+(\lambda, \mu, g_\epsilon^+(\mu)) = -\log\left(1 - \mu(1 - e^{-\epsilon})\right) - \epsilon\lambda\,.$$

By definition of the indicator function, we have $d_\epsilon^+(\lambda, \mu) = 0$ if $\mu \in [0, [\lambda]_0^1]$. When $\lambda \leq 0$, for all $\mu \in (0, 1)$, we have

$$\forall \mu \in (0, 1), \quad d_\epsilon^+(\lambda, \mu) = f_\epsilon^+([\lambda]_0^1, \mu, g_\epsilon^+(\mu)) = -\log\left(1 - \mu(1 - e^{-\epsilon})\right) - \epsilon[\lambda]_0^1\,,$$

by using the properties of $z \to f_\epsilon^+(\lambda, \mu, z)$ on $(0, 1) = (g_\epsilon^-([\lambda]_0^1), 1)$ by Lemma 24. This function can be extended by continuity to $\mu = 0 = g_\epsilon^-([\lambda]_0^1)$ with value $d_\epsilon^+(\lambda, 0) = 0$. When $\lambda \in (0, 1)$ and $\mu \in (g_\epsilon^-(\lambda), 1)$, we have

$$\forall \mu \in (0, 1), \quad d_\epsilon^+(\lambda, \mu) = f_\epsilon^+([\lambda]_0^1, \mu, g_\epsilon^+(\mu)) = -\log\left(1 - \mu(1 - e^{-\epsilon})\right) - \epsilon[\lambda]_0^1\,,$$

by using the properties of $z \to f_\epsilon^+(\lambda, \mu, z)$ on $(g_\epsilon^-(\lambda), 1) = (g_\epsilon^-([\lambda]_0^1), 1)$ by Lemma 24. This function can be extended by continuity to $(\lambda, \mu) = (0, 0) = \lim_{\lambda \to 0^+}(\lambda, g_\epsilon^-([\lambda]_0^1))$ with value $d_\epsilon^+(0, 0) = 0$. In both cases, this function can be extended by continuity to $\mu = 1$ with value $d_\epsilon^+(\lambda, 1) = \epsilon(1 - [\lambda]_0^1)$.

When $\lambda \in (0, 1)$, i.e., $[\lambda]_0^1 = \lambda$, and $\mu \in (\lambda, g_\epsilon^-(\lambda)) \subseteq (0, 1)$ by Lemma 24, we have

$$d_\epsilon^+(\lambda, \mu) = f_\epsilon^+(\lambda, \mu, \lambda) = \text{kl}(\lambda, \mu)\,.$$

This function can be extended by continuity to $\mu = \lambda$ with value $d_\epsilon^+(\lambda, \lambda) = 0$ since $\text{kl}(\lambda, \lambda) = 0$. Using Lemma 24, this function can be extended by continuity to $\mu = g_\epsilon^-(\lambda)$ (i.e., $\lambda = g_\epsilon^+(\mu)$) with value

$$d_\epsilon^+(\lambda, g_\epsilon^-(\lambda)) = \text{kl}\left(\lambda, g_\epsilon^-(\lambda)\right) = \text{kl}\left(g_\epsilon^+(\mu), \mu\right) = -\log\left(1 - \mu(1 - e^{-\epsilon})\right) - \epsilon g_\epsilon^+(\mu)\,.$$

Therefore, we have

$$\forall \lambda \in (0, 1), \ \forall \mu \in [\lambda, g_\epsilon^-(\lambda)], \quad d_\epsilon^+(\lambda, \mu) = \text{kl}(\lambda, \mu)\,.$$

Using that $\lim_{\lambda \to 0^+}[\lambda, g_\epsilon^-(\lambda)] = \{0\}$, this function can be extended by continuity to $\lambda = 0$ with value 0. Using that $\lim_{\lambda \to 1^-}[\lambda, g_\epsilon^-(\lambda)] = \{1\}$, this function can be extended by continuity to $\lambda = 1$ with value $0 = \lim_{(\mu, \lambda) \to 1^-} -\log\left(1 - \mu(1 - e^{-\epsilon})\right) - \epsilon[\lambda]_0^1$.

Putting all the continuity arguments together, we have shown that $(\lambda, \mu) \to d_\epsilon^+(\lambda, \mu)$ is jointly continuous on $\mathbb{R} \times [0, 1]$. Moreover, it is direct to see that, for all $\mu \in [0, 1]$, the function $\lambda \to d_\epsilon^+(\lambda, \mu)$ is constant on $(-\infty, 0]$ and on $[1, +\infty)$. Then,

$$\forall \lambda \in (0, 1), \forall \mu \in [0, 1], \quad d_\epsilon^+(\lambda, \mu) = \begin{cases} 0 & \text{if } \mu \in [0, \lambda] \\ \text{kl}(\lambda, \mu) & \mu \in (\lambda, g_\epsilon^-(\lambda)] \\ -\log\left(1 - \mu(1 - e^{-\epsilon})\right) - \epsilon\lambda & \text{if } \mu \in (g_\epsilon^-(\lambda), 1] \end{cases}\,.$$

Let $\mu \in [0, 1]$ and $\lambda \in (0, \mu)$. Using that $\mu \in (g_\epsilon^-(\lambda), 1]$ if and only if $\lambda \in (0, g_\epsilon^+(\mu))$. For all $\mu \in [0, 1]$, the function $\lambda \to d_\epsilon^+(\lambda, \mu)$ is positive and affine with negative slope $-\epsilon$ on $(0, g_\epsilon^+(\mu))$. Let $\lambda \in (g_\epsilon^+(\mu), \mu)$. Direct computation yields that

$$\frac{\partial d_\epsilon^+}{\partial \lambda}(\lambda, \mu) = \frac{\partial \text{kl}}{\partial \lambda}(\lambda, \mu) = \log\left(\frac{\lambda(1 - \mu)}{(1 - \lambda)\mu}\right) < 0\,,$$
$$\lim_{\lambda \to g_\epsilon^+(\mu)^+} \frac{\partial d_\epsilon^+}{\partial \lambda}(\lambda, \mu) = -\epsilon = \lim_{\lambda \to g_\epsilon^+(\mu)^-} \frac{\partial d_\epsilon^+}{\partial \lambda}(\lambda, \mu)\,,$$
$$\frac{\partial^2 d_\epsilon^+}{\partial \lambda^2}(\lambda, \mu) = \frac{\partial^2 \text{kl}}{\partial \lambda^2}(\lambda, \mu) = \frac{1}{\lambda(1 - \lambda)} > 0\,.$$

For all $\mu \in [0, 1]$, the function $\lambda \to d_\epsilon^+(\lambda, \mu)$ is continuously differentiable, positive, decreasing and convex on $(0, \mu)$. For all $\mu \in [0, 1]$, the function $\lambda \to d_\epsilon^+(\lambda, \mu)$ is twice continuously differentiable, positive and strictly convex on $(g_\epsilon^+(\mu), \mu)$. Combining the above results concludes the part of $\lambda \mapsto d_\epsilon^+(\lambda, \mu)$ on $(0, \mu)$.

Let $\lambda \in (0, 1)$. Let $a > 0$ and $k \in \mathbb{N}$. The $k$-th derivative of $u(x) = a(1 - ax)^{-1}$ on $[0, 1]$ is $u^{(k)}(x) = (k - 1)!a^{k+1}(1 - ax)^{-(k+1)}$. Then,

$$\forall \mu \in (g_\epsilon^-(\lambda), 1], \ \forall k \in \mathbb{N}, \quad \frac{\partial^k d_\epsilon^+}{\partial \mu^k}(\lambda, \mu) = \frac{(1 - e^{-\epsilon})^k(k - 1)!}{(1 - \mu(1 - e^{-\epsilon}))^k} > 0\,,$$

$$\forall \mu \in (\lambda, g_\epsilon^-(\lambda)], \quad \frac{\partial d_\epsilon^+}{\partial \mu}(\lambda, \mu) = \frac{\mu - \lambda}{\mu(1-\mu)} > 0 \,,$$

$$\frac{\partial^2 d_\epsilon^+}{\partial \mu^2}(\lambda, \mu) = \frac{(\mu - \lambda)^2 + \lambda(1-\lambda)}{\mu^2(1-\mu)^2} > 0 \,,$$

$$\frac{\partial^3 d_\epsilon^+}{\partial \mu^3}(\lambda, \mu) > 0 \,.$$

Direct computation yields

$$\lim_{\mu \to g_\epsilon^-(\lambda)} \frac{\mu - \lambda}{\mu(1-\mu)} = (1 - e^{-\epsilon})(1 + \lambda(e^\epsilon - 1)) \,,$$

$$\lim_{\mu \to g_\epsilon^-(\lambda)} \frac{1 - e^{-\epsilon}}{1 - \mu(1 - e^{-\epsilon})} = (1 - e^{-\epsilon})(1 + \lambda(e^\epsilon - 1)) \,,$$

$$\lim_{\mu \to g_\epsilon^-(\lambda)} \left\{ \frac{(\mu - \lambda)^2 + \lambda(1-\lambda)}{\mu^2(1-\mu)^2} - \frac{(1 - e^{-\epsilon})^2}{(1 - \mu(1 - e^{-\epsilon}))^2} \right\} = \frac{\lambda(1-\lambda)}{g_\epsilon^-(\lambda)^2(1 - g_\epsilon^-(\lambda))^2} > 0 \,.$$

For all $\lambda \in (0,1)$, the function $\mu \to d_\epsilon^+(\lambda, \mu)$ is positive, three times differentiable with continuous first derivative and increasing on $(\lambda, 1]$. For all $\lambda \in (0,1)$, the function $\mu \to d_\epsilon^+(\lambda, \mu)$ is strictly convex on $(\lambda, g_\epsilon^-(\lambda)]$ and $(g_\epsilon^-(\lambda), 1]$. The second derivative is discontinuous at $g_\epsilon^-(\lambda)$ with gap $\frac{\partial^2 d_\epsilon^+}{\partial \mu^2}(\lambda, g_\epsilon^-(\lambda)) - \lim_{\mu \to g_\epsilon^-(\lambda)^+} \frac{\partial^2 d_\epsilon^+}{\partial \mu^2}(\lambda, \mu) > 0$. Thanks to the continuity of the first derivative and the sign of the second derivative, the function $\mu \to d_\epsilon^+(\lambda, \mu)$ is strict convexity on $(\lambda, 1]$.

Let $(\mu_1, \mu_2) \in [0,1]^2$ and $(\lambda_1, \lambda_2) \in (0,1)^2$. On the convex set $\mathcal{F}_0 = \{(\lambda, \mu) \in (0,1) \times [0,1] \mid \mu \in [0, \lambda]\}$, the function $d_\epsilon^-$ is null hence jointly convex. Let $((\mu_1, \lambda_1), (\mu_2, \lambda_2)) \in (((0,1) \times [0,1]) \setminus \mathcal{F}_0)^2$. Let $(z_1, z_2) \in [\lambda_1, \mu_1] \times [\lambda_2, \mu_2]$ be the minimizers realizing $d_\epsilon^+(\lambda_1, \mu_1)$ and $d_\epsilon^+(\lambda_2, \mu_2)$. Since it is a convex set, we have $(\alpha\lambda_1 + (1-\alpha)\lambda_2, \alpha\mu_1 + (1-\alpha)\mu_2) \in ((0,1) \times [0,1]) \setminus \mathcal{F}_0$ for all $\alpha \in [0,1]$. Moreover, we have $\alpha z_1 + (1-\alpha)z_2 \in [\alpha\lambda_1 + (1-\alpha)\lambda_2, \alpha\mu_1 + (1-\alpha)\mu_2]$ for all $\alpha \in [0,1]$. Using the definition of $d_\epsilon^+$ as an infimum, we obtain

$$d_\epsilon^+(\alpha\lambda_1 + (1-\alpha)\lambda_2, \alpha\mu_1 + (1-\alpha)\mu_2)$$
$$\leq \mathrm{kl}(\alpha z_1 + (1-\alpha)z_2, \alpha\mu_1 + (1-\alpha)\mu_2) + \epsilon(\alpha z_1 + (1-\alpha)z_2 - (\alpha\lambda_1 + (1-\alpha)\lambda_2))$$
$$\leq \alpha \left( \mathrm{kl}(z_1, \mu_1) + \epsilon(z_1 - \lambda_1) \right) + (1-\alpha) \left( \mathrm{kl}(z_2, \mu_2) + \epsilon(z_2 - \lambda_2) \right)$$
$$= \alpha d_\epsilon^+(\lambda_1, \mu_1) + (1-\alpha)d_\epsilon^+(\lambda_2, \mu_2)$$

where the second inequality comes from the joint convexity of the Kullback-Leibler divergence. Combining both results, we have shown that the function $d_\epsilon^+$ is jointly convex on $(0,1) \times [0,1]$. $\qquad\square$

Lemma 26 gather regularity properties of $d_\epsilon^-$. In particular, it gives a closed-form solution, which is a key property used in our implementation to reduce the computational cost.

**Lemma 26.** *Let $d_\epsilon^-$ as in Eq. (3), and $g_\epsilon^\pm$ as in Eq. (30). For all $\mu \in [0,1]$ and all $\lambda \in \mathbb{R}$, we have*

$$d_\epsilon^-(\lambda, \mu) = \begin{cases} 0 & \text{if } \mu \in [[\lambda]_0^1, 1] \\ -\log(1 + \mu(e^\epsilon - 1)) + \epsilon[\lambda]_0^1 & \text{if } \mu \in [0, g_\epsilon^+([\lambda]_0^1)) \\ \mathrm{kl}(\lambda, \mu) & \text{if } \lambda \in (0,1) \text{ and } \mu \in [g_\epsilon^+([\lambda]_0^1), [\lambda]_0^1) \end{cases} \,.$$

*The function $(\lambda, \mu) \mapsto d_\epsilon^-(\lambda, \mu)$ is jointly continuous on $\mathbb{R} \times [0,1]$. For all $\mu \in [0,1]$, the function $\lambda \mapsto d_\epsilon^-(\lambda, \mu)$ is constant on $(-\infty, 0]$ and on $[1, +\infty)$. Then,*

$$\forall \lambda \in (0,1), \forall \mu \in [0,1], \quad d_\epsilon^-(\lambda, \mu) = \begin{cases} 0 & \text{if } \mu \in [\lambda, 1] \\ \mathrm{kl}(\lambda, \mu) & \text{if } \mu \in [g_\epsilon^+(\lambda), \lambda) \\ -\log(1 + \mu(e^\epsilon - 1)) + \epsilon\lambda & \text{if } \mu \in [0, g_\epsilon^+(\lambda)) \end{cases} \,.$$

*For all $\mu \in [0,1]$, the function $\lambda \mapsto d_\epsilon^-(\lambda, \mu)$ is continuously differentiable, positive, increasing and convex on $(\mu, 1)$; it is affine with positive slope $\epsilon$ on $(g_\epsilon^-(\mu), 1)$ and twice continuously differentiable and strictly convex on $(\mu, g_\epsilon^-(\mu))$.*

*For all $\lambda \in (0, 1)$, the function $\mu \mapsto d_\epsilon^-(\lambda, \mu)$ is positive, three times differentiable with continuous first derivative, decreasing and strictly convex on $[0, \lambda)$; its second derivative is discontinuous at $g_\epsilon^+(\lambda)$ with gap $\lim_{\mu \to g_\epsilon^+(\lambda)^-} \frac{\partial^2 d_\epsilon^-}{\partial \mu^2}(\lambda, \mu) - \frac{\partial^2 d_\epsilon^-}{\partial \mu^2}(\lambda, g_\epsilon^+(\lambda)) < 0$. Moreover, we have*

$$\forall \mu \in [0, \lambda), \quad \frac{\partial d_\epsilon^-}{\partial \mu}(\lambda, \mu) = \begin{cases} -\frac{e^\epsilon - 1}{1 + \mu(e^\epsilon - 1)} & \text{if } \mu \in [0, g_\epsilon^+(\lambda)) \\ -\frac{\lambda - \mu}{\mu(1 - \mu)} & \text{if } \mu \in [g_\epsilon^+(\lambda), \lambda) \end{cases}.$$

*The function $d_\epsilon^-$ is jointly convex on $(0, 1) \times [0, 1]$.*

*Proof.* Using Lemmas 23 and 24, we have

$$d_\epsilon^-(\lambda, \mu) = d_\epsilon^+(1 - \lambda, 1 - \mu) \quad \text{and} \quad g_\epsilon^+(\lambda) = 1 - g_\epsilon^-(1 - \lambda),$$
$$\frac{\partial d_\epsilon^-}{\partial \mu}(\lambda, \mu) = -\frac{\partial d_\epsilon^+}{\partial \mu}(1 - \lambda, 1 - \mu) \quad \text{and} \quad \frac{\partial^2 d_\epsilon^-}{\partial \mu^2}(\lambda, \mu) = \frac{\partial^2 d_\epsilon^+}{\partial \mu^2}(1 - \lambda, 1 - \mu).$$

Moreover, we have $\text{kl}(\lambda, \mu) = \text{kl}(1 - \lambda, 1 - \mu)$ and

$$-\log\left(1 + \mu(e^\epsilon - 1)\right) + \epsilon[\lambda]_0^1 = -\log\left(1 - (1 - \mu)(1 - e^{-\epsilon})\right) - \epsilon[1 - \lambda]_0^1.$$

Combining the above with properties of $d_\epsilon^+$ in Lemma 25 concludes the proof. $\qquad\square$

### G.1.1 Modified Divergence

Let us define

$$\forall x > 0, \quad h(x) := \sqrt{1 + x^2} - 1 + \log\left(\frac{2}{x^2}\left(\sqrt{1 + x^2} - 1\right)\right). \tag{31}$$

For all $(\lambda, \mu, r) \in \mathbb{R} \times (0, 1) \times \mathbb{R}_+^\star$, we define

$$\widetilde{d}_\epsilon^-(\lambda, \mu, r) := \mathbb{1}\left(\mu < [\lambda]_0^1\right) \inf_{z \in (\mu, [\lambda]_0^1)} \left\{\text{kl}(z, \mu) + \frac{1}{r} h(r\epsilon(\lambda - z))\right\},$$

$$\widetilde{d}_\epsilon^+(\lambda, \mu, r) := \mathbb{1}\left(\mu > [\lambda]_0^1\right) \inf_{z \in ([\lambda]_0^1, \mu)} \left\{\text{kl}(z, \mu) + \frac{1}{r} h(r\epsilon(z - \lambda))\right\}. \tag{32}$$

Lemma 27 shows a strong link between $\widetilde{d}_\epsilon^\pm$. This symmetry property can be used to carry regularity properties from $\widetilde{d}_\epsilon^+$ to $\widetilde{d}_\epsilon^-$, and vice versa.

**Lemma 27.** *Let $\widetilde{d}_\epsilon^\pm$ as in Eq. (32). For all $(\lambda, \mu) \in \mathbb{R} \times [0, 1]$, we have*

$$\widetilde{d}_\epsilon^+(1 - \lambda, 1 - \mu, r) = \widetilde{d}_\epsilon^-(\lambda, \mu, r) \quad \text{and} \quad \widetilde{d}_\epsilon^-(1 - \lambda, 1 - \mu, r) = \widetilde{d}_\epsilon^+(\lambda, \mu, r).$$

*Proof.* Using the definitions, the change of variable $\tilde{z} = 1 - z$ and $\text{kl}(1 - \tilde{z}, 1 - \mu) = \text{kl}(\tilde{z}, \mu)$, we obtain

$$\widetilde{d}_\epsilon^+(1 - \lambda, 1 - \mu, r) = \mathbb{1}\left(\mu < [\lambda]_0^1\right) \inf_{z \in [1 - [\lambda]_0^1, 1 - \mu]} \left\{\text{kl}(z, 1 - \mu) + \frac{1}{r} h\left(r\epsilon(\lambda - (1 - z))\right)\right\}$$

$$= \mathbb{1}\left(\mu < [\lambda]_0^1\right) \inf_{\tilde{z} \in [\mu, [\lambda]_0^1]} \left\{\text{kl}(1 - \tilde{z}, 1 - \mu) + \frac{1}{r} h\left(r\epsilon(\lambda - \tilde{z})\right)\right\}$$

$$= \mathbb{1}\left(\mu < [\lambda]_0^1\right) \inf_{\tilde{z} \in [\mu, [\lambda]_0^1]} \left\{\text{kl}(\tilde{z}, \mu) + \frac{1}{r} h\left(r\epsilon(\lambda - \tilde{z})\right)\right\} = \widetilde{d}_\epsilon^-(\lambda, \mu, r).$$

The second equality is a consequence of the first. $\qquad\square$

Lemma 28 gathers regularity properties of the function $h$ defined in Eq. (31).

**Lemma 28.** *Let $h$ as in Eq. (31). Then,*

$$\forall x > 0, \quad h'(x) = \frac{x}{\sqrt{x^2 + 1} + 1} > 0 \quad and \quad h''(x) = \frac{1}{1 + x^2 + \sqrt{1 + x^2}} > 0 .$$

*On $\mathbb{R}_+^\star$, the function $h$ is twice continuously differentiable, increasing and strictly convex. Moreover, it satisfies*

$$h(x) =_{x \to 0} x^2/4 + \mathcal{O}(x^4) \quad and \quad h(x) =_{x \to +\infty} x - \mathcal{O}(\log(x)) .$$

*Proof.* For all $x > 0$, $h_1(x) = x + \log(x)$, $h_2(x) = \sqrt{1 + x^2} - 1$ and $h_3(x) = \sqrt{1 + x^2} - x$. Then

$$h_1'(x) = 1 + \frac{1}{x} \quad , \quad h_2'(x) = \frac{x}{\sqrt{1 + x^2}} \quad and \quad h_3'(x) = \frac{x}{\sqrt{1 + x^2}} - 1 ,$$

Then, we have

$$\forall x > 0, \quad h(x) = h_1(h_2(x)) - 2\log(x) + \log 2 .$$

Therefore, we have

$$h'(x) = h_2'(x)h_1'(h_2(x)) - \frac{2}{x} = \frac{x}{\sqrt{1 + x^2}}\left(1 + \frac{1}{\sqrt{1 + x^2} - 1}\right) - \frac{2}{x}$$

$$= \frac{x}{\sqrt{1 + x^2} - 1} - \frac{2}{x} = \sqrt{1 + \frac{1}{x^2}} - \frac{1}{x} = h_3(1/x) .$$

Note that

$$\sqrt{1 + \frac{1}{x^2}} - \frac{1}{x} = \frac{x}{\sqrt{x^2 + 1} + 1} .$$

Moreover, we have w

$$h''(x) = -\frac{1}{x^2} h_3'(1/x) = -\frac{1}{x^2}\left(\frac{1/x}{\sqrt{1 + (1/x)^2}} - 1\right) = \frac{1}{1 + x^2 + \sqrt{1 + x^2}} .$$

By taking the limit, we have $\lim_{x \to 0^+} h(x) = 0$. Moreover, we see that $\lim_{x \to 0^+} h'(x) = 0$ and $\lim_{x \to 0^+} h''(x) = 1/2$. Therefore, one can conclude that $h(x) =_{x \to 0} x^2/4 + \mathcal{O}(x^4)$ by Taylor expansion. The second result is obtained directly by limit. $\qquad\square$

Lemma 29 provides upper and lower bound on the function $r \mapsto h(rx)/r$ involved in the definition of $\widetilde{d}_\epsilon^\pm$.

**Lemma 29.** *Let $h$ as in Eq. (31). Let $\kappa(r, x) = h(rx)/r - x$ for all $r > 0$ and all $x \in \mathbb{R}_+^\star$. Then, we have*

$$\forall r > 0, \quad \frac{\partial \kappa}{\partial r}(r, x) = \frac{rx h'(rx) - h(rx)}{r^2} = \log\left(\frac{1}{2}(\sqrt{1 + (rx)^2} + 1)\right) > 0 .$$

*On $\mathbb{R}_+^\star$, the function $r \mapsto \kappa(r, x)$ is increasing. Moreover, we have*

$$\forall r > 0, \forall x \in \mathbb{R}_+, \quad 0 \leq r\kappa(r, x) + \log(1 + 2xr) + 1 \leq 1 + \log 4 ,$$

*Proof.* Using Lemma 28 and the definition in Eq. (31), we obtain that

$$\forall x > 0, \quad xh'(x) - h(x) = -\log\left(\frac{2}{x^2}\left(\sqrt{1 + x^2} - 1\right)\right) = \log\left(\frac{1}{2}(\sqrt{1 + x^2} + 1)\right) > 0 ,$$

where we used that $\sqrt{1 + x^2} + 1 > 2$ for the last inequality. Let us define

$$\forall x \in \mathbb{R}_+, \quad g_1(x) = \frac{2(1 + 2x)}{\sqrt{1 + x^2} + 1} .$$

Then, we obtain $g_1(0) = 1$, $\lim_{x \to +\infty} g_1(x) = 4$ and

$$g_1'(x) = 2\frac{2 + 2\sqrt{1 + x^2} - x}{\sqrt{1 + x^2}(\sqrt{1 + x^2} + 1)^2} > 2\frac{2 + x}{\sqrt{1 + x^2}(\sqrt{1 + x^2} + 1)^2} > 0 .$$

Since $g_1$ is strictly increasing on $\mathbb{R}_+^\star$, we obtain $\log g_1(x) \geq \log g_1(0) = 0$ and $\log g_1(x) \leq \log 4$ for all $x \in \mathbb{R}_+$.

By definition, we obtain

$$r\kappa(r,x) + \log(1 + 2xr) + 1 = h(rx) - rx + 1 + \log(1 + 2xr)$$
$$= \sqrt{1 + (rx)^2} - rx + \log\left(\frac{2(1 + 2xr)}{\sqrt{1 + r^2x^2} + 1}\right) .$$

Using that $0 \leq \sqrt{1 + x^2} - x \leq 1$ on $\mathbb{R}_+$, we obtain

$$r\kappa(r,x) + \log(1 + 2xr) + 1 \geq \log\left(\frac{2(1 + 2xr)}{\sqrt{1 + r^2x^2} + 1}\right) = \log(g_1(rx)) \geq 0 ,$$
$$r\kappa(r,x) + \log(1 + 2xr) + 1 \leq 1 + \log(g_1(rx)) \leq 1 + \log 4 .$$

This concludes the proof. $\qquad\qquad\qquad\qquad\qquad\qquad\qquad\qquad\qquad\qquad\qquad\qquad\qquad$ $\square$

Lemma 30 provides lower and upper bounds on the gap between $\widetilde{d}_\epsilon^\pm$ and $d_\epsilon^\pm$.

**Lemma 30.** *Let $d_\epsilon^\pm$ and $\widetilde{d}_\epsilon^\pm$ as in Eq. (3) and (32). For all $(\lambda, \mu, r) \in \mathbb{R} \times (0,1) \times \mathbb{R}_+^\star$ such that $[\lambda]_0^1 < \mu$. Then,*

$$d_\epsilon^+(\lambda, \mu) \leq \widetilde{d}_\epsilon^+(\lambda, \mu, r) + \frac{\log(1 + 2\epsilon r) + 1}{r} .$$

*For all $(\lambda, \mu, r) \in \mathbb{R} \times (0,1) \times \mathbb{R}_+^\star$ such that $[\lambda]_0^1 > \mu$. Then,*

$$d_\epsilon^-(\lambda, \mu) \leq \widetilde{d}_\epsilon^-(\lambda, \mu, r) + \frac{\log(1 + 2\epsilon r) + 1}{r} .$$

*For all $(\lambda, \mu, r) \in [0,1] \times (0,1) \times \mathbb{R}_+^\star$ such that $\lambda < \mu$. Then,*

$$d_\epsilon^+(\lambda, \mu) \geq \widetilde{d}_\epsilon^+(\lambda, \mu, r) - \frac{\log 4}{r} .$$

*For all $(\mu, \lambda, r) \in [0,1] \times \mathbb{R}_+^\star$ such that $\lambda > \mu$. Then,*

$$d_\epsilon^-(\lambda, \mu) \geq \widetilde{d}_\epsilon^-(\lambda, \mu, r) - \frac{\log 4}{r} .$$

*Proof.* Since $\mu \in (0,1)$, we have $[\lambda]_0^1 = \max\{0, \lambda\}$. Therefore, we have $z - \lambda \geq z - [\lambda]_0^1$ and $z - [\lambda]_0^1 \in (0, \mu - [\lambda]_0^1) \subset (0,1)$ for all $z \in ([\lambda]_0^1, \mu)$. Using Lemmas 28 and 29 and $\epsilon > 0$, we obtain, for all $r > 0$ and all $z \in ([\lambda]_0^1, \mu)$,

$$\epsilon(z - [\lambda]_0^1) \leq \frac{1}{r}h(r\epsilon(z - [\lambda]_0^1)) + \frac{\log(1 + 2\epsilon(z - [\lambda]_0^1)r) + 1}{r}$$
$$\leq \frac{1}{r}h(r\epsilon(z - \lambda)) + \frac{\log(1 + 2\epsilon r) + 1}{r} .$$

Therefore, for all $z \in ([\lambda]_0^1, \mu)$, we obtain that

$$\mathrm{kl}(z, \mu) + \epsilon(z - [\lambda]_0^1) \leq \mathrm{kl}(z, \mu) + \frac{1}{r}h(r\epsilon(z - \lambda)) + \frac{\log(1 + 2\epsilon r) + 1}{r} .$$

Taking the infimum over $z \in ([\lambda]_0^1, \mu)$ on both sides of both inequalities and using that

$$d_\epsilon^+(\lambda, \mu) = \inf_{z \in [[\lambda]_0^1, \mu]}\left\{\mathrm{kl}(z, \mu) + \epsilon(z - [\lambda]_0^1)\right\} = \inf_{z \in ([\lambda]_0^1, \mu)}\left\{\mathrm{kl}(z, \mu) + \epsilon(z - [\lambda]_0^1)\right\} ,$$

$$\widetilde{d}_\epsilon^+(\lambda, \mu, r) = \inf_{z \in ([\lambda]_0^1, \mu)}\left\{\mathrm{kl}(z, \mu) + \frac{1}{r}h(r\epsilon(z - \lambda))\right\} ,$$

we obtain

$$d_\epsilon^+(\lambda, \mu) \leq \widetilde{d}_\epsilon^+(\lambda, \mu, r) + \frac{\log(1 + 2\epsilon r) + 1}{r} .$$

This concludes the proof of the first result. Using Lemmas 23 and 27 yields the second result.

Suppose that $\lambda \in [0, 1]$, hence $\lambda = [\lambda]_0^1$. Using Lemmas 28 and 29 and $\epsilon > 0$, we obtain, for all $r > 0$ and all $z \in ([\lambda]_0^1, \mu)$,

$$\frac{1}{r} h(r\epsilon(z - \lambda)) \leq \epsilon(z - \lambda) + \frac{\log 4 - \log(1 + 2\epsilon(z - \lambda)r)}{r} \leq \epsilon(z - [\lambda]_0^1) + \frac{\log 4}{r} \, .$$

Adding $\mathrm{kl}(z, \mu)$ on both sides and taking the infimum over $z \in ([\lambda]_0^1, \mu)$ on both sides of both inequalities yields the proof of third result. Using Lemmas 23 and 27 yields the forth result. $\qquad \square$

Lemma 31 gathers regularity properties on the modified divergences $\widetilde{d}_\epsilon^+$. In particular, it gives a closed-form solution based on an implicit solution of a fixed-point equation. This is a key property used in our implementation to reduce the computational cost.

**Lemma 31.** *Let $\widetilde{d}_\epsilon^+$ as in Eq. (32), and $g_\epsilon^\pm$ as in Eq. (30). For all $\mu \in (0, 1)$, $\lambda \in \mathbb{R}$ and $r > 0$, we have*

$$\widetilde{d}_\epsilon^+(\lambda, \mu, r)$$
$$= \begin{cases} 0 & \text{if } \mu \in (0, [\lambda]_0^1] \\ \mathrm{kl}(x_\epsilon^+(\lambda, \mu, r) + g_\epsilon^+(\mu), \mu) + \frac{1}{r} h(r\epsilon(x_\epsilon^+(\lambda, \mu, r) + g_\epsilon^+(\mu) - \lambda)) & \text{if } \mu \in ([\lambda]_0^1, 1) \end{cases} \, ,$$

*where $x_\epsilon^+(\lambda, \mu, r) \in (\max\{0, \lambda - g_\epsilon^+(\mu)\}, \mu - g_\epsilon^+(\mu))$ is the unique solution for $x \in (\max\{0, \lambda - g_\epsilon^+(\mu)\}, \mu - g_\epsilon^+(\mu))$ of the equation*

$$\log\left(1 + \frac{x}{g_\epsilon^+(\mu)(1 - x - g_\epsilon^+(\mu))}\right) + \epsilon\left(\frac{r\epsilon(x + g_\epsilon^+(\mu) - \lambda)}{\sqrt{(r\epsilon(x + g_\epsilon^+(\mu) - \lambda))^2 + 1} + 1} - 1\right) = 0 \, .$$

*For all $(\mu, r) \in (0, 1) \times \mathbb{R}_+^\star$, the function $\lambda \mapsto \widetilde{d}_\epsilon^+(\lambda, \mu, r)$ is positive, twice continuously differentiable, decreasing and strictly convex on $(-\infty, \mu)$; it satisfies $\lim_{\lambda \to \mu^-} \widetilde{d}_\epsilon^+(\lambda, \mu, r) = 0$ and $\lim_{\lambda \to -\infty} \widetilde{d}_\epsilon^+(\lambda, \mu, r) = +\infty$.*

*For all $(\lambda, r) \in \mathbb{R} \times \mathbb{R}_+^\star$, the function $\mu \mapsto \widetilde{d}_\epsilon^+(\lambda, \mu, r)$ is positive, twice continuously differentiable, increasing and strictly convex on $([\lambda]_0^1, 1)$. Moreover, we have*

$$\forall \mu \in ([\lambda]_0^1, 1), \quad \frac{\partial \widetilde{d}_\epsilon^+}{\partial \mu}(\lambda, \mu, r) = \frac{\mu - g_\epsilon^+(\mu) - x_\epsilon^+(\lambda, \mu, r)}{\mu(1 - \mu)} \, .$$

*For all $(\lambda, \mu) \in \mathbb{R} \times (0, 1)$ such that $\mu \in (0, [\lambda]_0^1]$, the function $r \mapsto \widetilde{d}_\epsilon^+(\lambda, \mu, r)$ is the zero function. For all $(\lambda, \mu) \in \mathbb{R} \times (0, 1)$ such that $\mu \in ([\lambda]_0^1, 1)$, the function $r \mapsto \widetilde{d}_\epsilon^+(\lambda, \mu, r)$ is positive, continuously differentiable and increasing on $\mathbb{R}_+$.*

*Proof.* By definition of the indicator function, we have $\widetilde{d}_\epsilon^+(\lambda, \mu, r) = 0$ if $\mu \in (0, [\lambda]_0^1]$. Let $(\lambda, \mu)$ such that $\mu \notin (0, [\lambda]_0^1]$, i.e., $([\lambda]_0^1, \mu)$ is non-empty. Since $\mu \in (0, 1)$, this implies that $\lambda \in (-\infty, 1)$ necessarily, i.e., $[\lambda]_0^1 = \max\{0, \lambda\}$.

Recall that $\widetilde{d}_\epsilon^+(\lambda, \mu, r) = \mathbb{1}\left(\mu > [\lambda]_0^1\right) \inf_{z \in ([\lambda]_0^1, \mu)} \widetilde{f}_\epsilon^+(\lambda, \mu, r, z)$ where $\widetilde{f}_\epsilon^+(\lambda, \mu, r, z) = \mathrm{kl}(z, \mu) + \frac{1}{r} h(r\epsilon(z - \lambda))$. Using Lemma 28, direct computations yield that, for all $z \in ([\lambda]_0^1, \mu)$,

$$\frac{\partial \widetilde{f}_\epsilon^+}{\partial z}(\lambda, \mu, r, z) = \log\left(\frac{z(1 - \mu)}{(1 - z)\mu}\right) + \epsilon h'(r\epsilon(z - \lambda))$$

$$= \log\left(\frac{z(1 - \mu)}{(1 - z)\mu}\right) + \epsilon \frac{r\epsilon(z - \lambda)}{\sqrt{(r\epsilon(z - \lambda))^2 + 1} + 1} \, ,$$

$$\frac{\partial^2 \widetilde{f}_\epsilon^+}{\partial z^2}(\lambda, \mu, r, z) = \frac{1}{z(1 - z)} + r\epsilon^2 h''(r\epsilon(z - \lambda)) > 0 \, .$$

Therefore, $z \to \widetilde{f}_\epsilon^+(\lambda, \mu, r, z)$ is twice continuously differentiable, positive and strictly convex on $([\lambda]_0^1, \mu)$. Moreover, we have

$$\lim_{z \to \mu} \frac{\partial \widetilde{f}_\epsilon^+}{\partial z}(\lambda, \mu, r, z) = \epsilon h'(r\epsilon(\mu - \lambda)) > 0 \,,$$

$$\frac{\partial \widetilde{f}_\epsilon^+}{\partial z}(\lambda, \mu, r, g_\epsilon^+(\mu)) = -\epsilon \left( 1 - \frac{r\epsilon(z - \lambda)}{\sqrt{(r\epsilon(z - \lambda))^2 + 1} + 1} \right) < 0 \,,$$

When $[\lambda]_0^1 > g_\epsilon^+(\mu)$, $\quad \frac{\partial \widetilde{f}_\epsilon^+}{\partial z}(\lambda, \mu, r, \lambda) = \log \left( \frac{\lambda(1 - \mu)}{(1 - \lambda)\mu} \right) < 0 \,.$

Note that $\max\{[\lambda]_0^1, g_\epsilon^+(\mu)\} = \max\{\lambda, g_\epsilon^+(\mu)\}$ since $\mu \in (0, 1)$. Using that $z \to \frac{\partial \widetilde{f}_\epsilon^+}{\partial z}(\lambda, \mu, r, z)$ is continuously differentiable and increasing on $([\lambda]_0^1, \mu)$, with negative value at $\max\{\lambda, g_\epsilon^+(\mu)\}$ and finite positive limit at $\mu$, $z \mapsto \widetilde{f}_\epsilon^+(\lambda, \mu, r, z)$ admit a unique minimizer on $(\max\{\lambda, g_\epsilon^+(\mu)\}, \mu)$. Let $\widetilde{g}_\epsilon^+(\lambda, \mu, r) \in (\max\{\lambda, g_\epsilon^+(\mu)\}, \mu)$ be defined as this unique minimizer, defined implicitly as solution for $z \in (\max\{\lambda, g_\epsilon^+(\mu)\}, \mu)$ of the equation

$$\log \left( \frac{z(1 - \mu)}{(1 - z)\mu} \right) + \epsilon \frac{r\epsilon(z - \lambda)}{\sqrt{(r\epsilon(z - \lambda))^2 + 1} + 1} = 0 \,.$$

Then, we have $\frac{\partial \widetilde{f}_\epsilon^+}{\partial z}(\lambda, \mu, r, z) = 0$ if and only if $z = \widetilde{g}_\epsilon^+(\lambda, \mu, r)$. Moreover, $z \mapsto \widetilde{f}_\epsilon^+(\lambda, \mu, r, z)$ is decreasing on $([\lambda]_0^1, \widetilde{g}_\epsilon^+(\lambda, \mu, r))$ and increasing on $(\widetilde{g}_\epsilon^+(\lambda, \mu, r), \mu)$.

Let us define $z = g_\epsilon^+(\mu) + x$ where $x \in (\max\{0, \lambda - g_\epsilon^+(\mu)\}, \mu - g_\epsilon^+(\mu))$. Then, we have

$$\frac{\partial \widetilde{f}_\epsilon^+}{\partial z}(\lambda, \mu, r, g_\epsilon^+(\mu) + x)$$

$$= \log \left( 1 + \frac{x}{g_\epsilon^+(\mu)(1 - x - g_\epsilon^+(\mu))} \right) + \epsilon \left( \frac{r\epsilon(x + g_\epsilon^+(\mu) - \lambda)}{\sqrt{(r\epsilon(x + g_\epsilon^+(\mu) - \lambda))^2 + 1} + 1} - 1 \right)$$

Therefore, we have $\widetilde{g}_\epsilon^+(\lambda, \mu, r) = g_\epsilon^+(\mu) + x_\epsilon^+(\lambda, \mu, r)$ where $x_\epsilon^+(\lambda, \mu, r) \in (\max\{0, \lambda - g_\epsilon^+(\mu)\}, \mu - g_\epsilon^+(\mu))$ is the solution for $x \in (\max\{0, \lambda - g_\epsilon^+(\mu)\}, \mu - g_\epsilon^+(\mu))$ of the equation

$$\log \left( 1 + \frac{x}{g_\epsilon^+(\mu)(1 - x - g_\epsilon^+(\mu))} \right) + \epsilon \left( \frac{r\epsilon(x + g_\epsilon^+(\mu) - \lambda)}{\sqrt{(r\epsilon(x + g_\epsilon^+(\mu) - \lambda))^2 + 1} + 1} - 1 \right) = 0 \,.$$

When $\lambda \in (0, 1)$ and $\mu \to \lambda = [\lambda]_0^1$, it is direct to see that $\widetilde{g}_\epsilon^+(\lambda, \mu, r) \to \lambda$. Then, we have

$$\lim_{\mu \to \lambda^+} \widetilde{d}_\epsilon^+(\lambda, \mu, r) = \lim_{\mu \to \lambda^+} \{\mathrm{kl}(\widetilde{g}_\epsilon^+(\lambda, \mu, r), \mu)\} + \frac{1}{r} \lim_{\widetilde{g}_\epsilon^+(\lambda, \mu, r) \to \lambda^+} \{h(r\epsilon(\widetilde{g}_\epsilon^+(\lambda, \mu, r) - \lambda))\} = 0 \,.$$

Direct computation yields that, for $z \in ([\lambda]_0^1, \mu) \subset (0, 1)$,

$$\frac{\partial \widetilde{f}_\epsilon^+}{\partial \mu}(\lambda, \mu, r, z) = \frac{\mu - z}{\mu(1 - \mu)} > 0 \,,$$

$$\frac{\partial^2 \widetilde{f}_\epsilon^+}{\partial \mu^2}(\lambda, \mu, r, z) = \frac{(\mu - z)^2 + z(1 - z)}{\mu^2(1 - \mu)^2} > 0 \,,$$

$$\frac{\partial^2 \widetilde{f}_\epsilon^+}{\partial z^2}(\lambda, \mu, r, z) = \frac{1}{z(1 - z)} + r\epsilon^2 h''(r\epsilon(z - \lambda)) > 0 \,,$$

$$\frac{\partial^2 \widetilde{f}_\epsilon^+}{\partial \mu \partial z}(\lambda, \mu, r, z) = -\frac{1}{\mu(1 - \mu)} < 0 \,.$$

Since $\frac{\partial \widetilde{f}_\epsilon^+}{\partial z}(\lambda, \mu, r, g_\epsilon^+(\lambda, \mu, r)) = 0$, the implicit function theorem yields that

$$\frac{\partial \widetilde{g}_\epsilon^+}{\partial \mu}(\lambda, \mu, r) = -\frac{\frac{\partial^2 \widetilde{f}_\epsilon^+}{\partial \mu \partial z}(\lambda, \mu, r, g_\epsilon^+(\lambda, \mu, r))}{\frac{\partial^2 \widetilde{f}_\epsilon^+}{\partial z^2}(\lambda, \mu, r, g_\epsilon^+(\lambda, \mu, r))} > 0 \,.$$

Moreover, for $\mu \in ([\lambda]_0^1, 1)$,

$$\frac{\partial \widetilde{d}_\epsilon^+}{\partial \mu}(\lambda, \mu, r) = \frac{\partial \widetilde{f}_\epsilon^+}{\partial \mu}(\lambda, \mu, r, \widetilde{g}_\epsilon^+(\lambda, \mu, r)) + \frac{\partial \widetilde{g}_\epsilon^+}{\partial \mu}(\lambda, \mu, r) \frac{\partial \widetilde{f}_\epsilon^+}{\partial z}(\lambda, \mu, r, \widetilde{g}_\epsilon^+(\lambda, \mu, r))$$

$$= \frac{\partial \widetilde{f}_\epsilon^+}{\partial \mu}(\lambda, \mu, r, \widetilde{g}_\epsilon^+(\lambda, \mu, r)) = \frac{\mu - g_\epsilon^+(\mu) - x_\epsilon^+(\lambda, \mu, r)}{\mu(1 - \mu)} > 0 \,,$$

$$\frac{\partial^2 \widetilde{d}_\epsilon^+}{\partial \mu^2}(\lambda, \mu, r) = \frac{\partial^2 \widetilde{f}_\epsilon^+}{\partial \mu^2}(\lambda, \mu, r, \widetilde{g}_\epsilon^+(\lambda, \mu, r)) \frac{\partial \widetilde{g}_\epsilon^+}{\partial \mu}(\lambda, \mu, r) > 0 \,.$$

Therefore, for all $(\lambda, r) \in \mathbb{R} \times \mathbb{R}_+^\star$, the function $\mu \mapsto \widetilde{d}_\epsilon^+(\lambda, \mu, r)$ is positive, twice continuously differentiable, increasing and strictly convex on $([\lambda]_0^1, 1)$.

Let $(\mu, r) \in (0, 1) \times \mathbb{R}_+^\star$. Direct computation yields that, for $z \in ([\lambda]_0^1, \mu) \subset (0, 1)$,

$$\frac{\partial \widetilde{f}_\epsilon^+}{\partial \lambda}(\lambda, \mu, r, z) = -\epsilon h'(r\epsilon(z - \lambda)) < 0 \,,$$

$$\frac{\partial^2 \widetilde{f}_\epsilon^+}{\partial \lambda \partial z}(\lambda, \mu, r, z) = -r\epsilon^2 h''(r\epsilon(z - \lambda)) < 0 \,.$$

Since $\frac{\partial \widetilde{f}_\epsilon^+}{\partial z}(\lambda, \mu, r, g_\epsilon^+(\lambda, \mu, r)) = 0$, the implicit function theorem yields that

$$\frac{\partial \widetilde{g}_\epsilon^+}{\partial \lambda}(\lambda, \mu, r) = -\frac{\frac{\partial^2 f_\epsilon^+}{\partial \lambda \partial z}(\lambda, \mu, r, g_\epsilon^+(\lambda, \mu, r))}{\frac{\partial^2 \widetilde{f}_\epsilon^+}{\partial z^2}(\lambda, \mu, r, g_\epsilon^+(\lambda, \mu, r))} = \frac{r\epsilon^2 h''(r\epsilon(g_\epsilon^+(\lambda, \mu, r) - \lambda))}{\frac{1}{z(1-z)} + r\epsilon^2 h''(r\epsilon(g_\epsilon^+(\lambda, \mu, r) - \lambda))} < 1 \,.$$

Direct computation yields that, for $\lambda \in (-\infty, \mu)$,

$$\frac{\partial \widetilde{d}_\epsilon^+}{\partial \lambda}(\lambda, \mu, r) = \frac{\partial \widetilde{f}_\epsilon^+}{\partial \lambda}(\lambda, \mu, r, \widetilde{g}_\epsilon^+(\lambda, \mu, r)) + \frac{\partial \widetilde{g}_\epsilon^+}{\partial \lambda}(\lambda, \mu, r) \frac{\partial \widetilde{f}_\epsilon^+}{\partial z}(\lambda, \mu, r, \widetilde{g}_\epsilon^+(\lambda, \mu, r))$$

$$= \frac{\partial \widetilde{f}_\epsilon^+}{\partial \lambda}(\lambda, \mu, r, \widetilde{g}_\epsilon^+(\lambda, \mu, r)) = -\epsilon h'(r\epsilon(\widetilde{g}_\epsilon^+(\lambda, \mu, r) - \lambda)) < 0 \,,$$

$$\frac{\partial^2 \widetilde{d}_\epsilon^+}{\partial \lambda^2}(\lambda, \mu, r) = r\epsilon^2 \left(1 - \frac{\partial \widetilde{g}_\epsilon^+}{\partial \lambda}(\lambda, \mu, r)\right) h''(r\epsilon(\widetilde{g}_\epsilon^+(\lambda, \mu, r) - \lambda)) > 0 \,.$$

Therefore, for all $(\mu, r) \in (0, 1) \times \mathbb{R}_+^\star$, the function $\lambda \mapsto \widetilde{d}_\epsilon^+(\lambda, \mu, r)$ is positive, twice continuously differentiable, decreasing and strictly convex on $(-\infty, \mu)$. Similarly as above, it is direct to see that $\lim_{\lambda \to \mu^-} \widetilde{d}_\epsilon^+(\lambda, \mu, r) = 0$ and $\lim_{\lambda \to -\infty} \widetilde{d}_\epsilon^+(\lambda, \mu, r) = +\infty$.

Let $(\lambda, \mu) \in \mathbb{R} \times (0, 1)$. When $\mu \in (0, [\lambda]_0^1]$, we have $\widetilde{d}_\epsilon^+(\lambda, \mu, r) = 0$ for all $r \in [1, +\infty)$, hence $r \mapsto \widetilde{d}_\epsilon^+(\lambda, \mu, r)$ is non-decreasing. Let $\kappa$ as in Lemma 29. Using Lemma 29, we have

$$\forall z > \lambda, \quad \frac{\partial \widetilde{f}_\epsilon^+}{\partial r}(\lambda, \mu, r, z) = \frac{\partial \kappa}{\partial r}(r, \epsilon(z - \lambda)) > 0 \,.$$

When $\mu \in ([\lambda]_0^1, 1)$, we have $\widetilde{g}_\epsilon^+(\lambda, \mu, r) \in (\max\{\lambda, g_\epsilon^+(\mu)\}, \mu)$ and, for all $r > 0$,

$$\frac{\partial \widetilde{d}_\epsilon^+}{\partial r}(\lambda, \mu, r) = \frac{\partial \widetilde{f}_\epsilon^+}{\partial r}(\lambda, \mu, r, \widetilde{g}_\epsilon^+(\lambda, \mu, r)) + \frac{\partial \widetilde{g}_\epsilon^+}{\partial r}(\lambda, \mu, r) \frac{\partial \widetilde{f}_\epsilon^+}{\partial z}(\lambda, \mu, r, \widetilde{g}_\epsilon^+(\lambda, \mu, r))$$

$$= \frac{\partial \widetilde{f}_\epsilon^+}{\partial r}(\lambda, \mu, r, \widetilde{g}_\epsilon^+(\lambda, \mu, r)) > 0 \,,$$

where we used that $\widetilde{g}_\epsilon^+(\lambda, \mu, r) > \lambda$. This concludes the last part of the proof. $\qquad \square$

Lemma 32 gathers regularity properties on the modified divergences $\widetilde{d}_\epsilon^-$. In particular, it gives a closed-form solution based on an implicit solution of a fixed-point equation. This is a key property used in our implementation to reduce the computational cost.

**Lemma 32.** *Let $\widetilde{d}_\epsilon^-$ as in Eq. (32), and $g_\epsilon^\pm$ as in Eq. (30). For all $\mu \in (0,1)$, $\lambda \in \mathbb{R}$ and $r > 0$, we have*

$$\widetilde{d}_\epsilon^-(\lambda, \mu, r)$$
$$= \begin{cases} 0 & \text{if } \mu \in [[\lambda]_0^1, 1) \\ \mathrm{kl}(g_\epsilon^-(\mu) - x_\epsilon^-(\lambda,\mu,r), \mu) + \frac{1}{r}h(r\epsilon(x_\epsilon^-(\lambda,\mu,r) + \lambda - g_\epsilon^-(\mu))) & \text{if } \mu \in (0, [\lambda]_0^1) \end{cases},$$

*where $x_\epsilon^-(\lambda,\mu,r) := x_\epsilon^+(1-\lambda, 1-\mu, r) \in (\max\{g_\epsilon^-(\mu) - \lambda, 0\}, g_\epsilon^-(\mu) - \mu)$ is the solution for $x \in (\max\{g_\epsilon^-(\mu) - \lambda, 0\}, g_\epsilon^-(\mu) - \mu)$ of the equation*

$$\log\left(1 + \frac{x}{(1 - g_\epsilon^-(\mu))(g_\epsilon^-(\mu) - x)}\right) + \epsilon\left(\frac{r\epsilon(x - g_\epsilon^-(\mu) + \lambda)}{\sqrt{(r\epsilon(x - g_\epsilon^-(\mu) + \lambda))^2 + 1} + 1} - 1\right) = 0.$$

*For all $(\mu, r) \in (0,1) \times \mathbb{R}_+^\star$, the function $\lambda \mapsto \widetilde{d}_\epsilon^-(\lambda, \mu, r)$ is positive, twice continuously differentiable, increasing and strictly convex on $(\mu, +\infty)$; it satisfies $\lim_{\lambda \to \mu^+} \widetilde{d}_\epsilon^+(\lambda, \mu, r) = 0$ and $\lim_{\lambda \to +\infty} \widetilde{d}_\epsilon^-(\lambda, \mu, r) = +\infty$.*

*For all $(\lambda, r) \in \mathbb{R} \times \mathbb{R}_+^\star$, the function $\mu \mapsto \widetilde{d}_\epsilon^-(\lambda, \mu, r)$ is positive, twice continuously differentiable, decreasing and strictly convex on $(0, [\lambda]_0^1)$. Moreover, we have*

$$\forall \mu \in (0, [\lambda]_0^1), \quad \frac{\partial \widetilde{d}_\epsilon^-}{\partial \mu}(\lambda, \mu, r) = \frac{\mu - g_\epsilon^-(\mu) + x_\epsilon^-(\lambda, \mu, r)}{\mu(1-\mu)}.$$

*For all $(\lambda, \mu) \in \mathbb{R} \times (0,1)$ such that $\mu \in (0, [\lambda]_0^1]$, the function $r \mapsto \widetilde{d}_\epsilon^-(\lambda, \mu, r)$ is the zero function. For all $(\lambda, \mu) \in \mathbb{R} \times (0,1)$ such that $\mu \in ([\lambda]_0^1, 1)$, the function $r \mapsto \widetilde{d}_\epsilon^-(\lambda, \mu, r)$ is positive, continuously differentiable and increasing on $\mathbb{R}_+$.*

*Proof.* Using Lemmas 27 and 24, we have

$$\widetilde{d}_\epsilon^-(\lambda, \mu, r) = \widetilde{d}_\epsilon^+(1-\lambda, 1-\mu, r) \quad \text{and} \quad g_\epsilon^+(\lambda) = 1 - g_\epsilon^-(1-\lambda),$$
$$\frac{\partial \widetilde{d}_\epsilon^-}{\partial \mu}(\lambda, \mu, r) = -\frac{\partial \widetilde{d}_\epsilon^+}{\partial \mu}(1-\lambda, 1-\mu, r) \quad \text{and} \quad \frac{\partial^2 \widetilde{d}_\epsilon^-}{\partial \mu^2}(\lambda, \mu, r) = \frac{\partial^2 \widetilde{d}_\epsilon^+}{\partial \mu^2}(1-\lambda, 1-\mu, r).$$

Let $x_\epsilon^+(1-\lambda, 1-\mu, r) \in (\max\{0, g_\epsilon^-(\mu) - \lambda\}, g_\epsilon^-(\mu) - \mu)$ be the unique solution for $x \in (\max\{0, g_\epsilon^-(\mu) - \lambda\}, g_\epsilon^-(\mu) - \mu)$ of the equation

$$\log\left(1 + \frac{x}{(1 - g_\epsilon^-(\mu))(g_\epsilon^-(\mu) - x)}\right) + \epsilon\left(\frac{r\epsilon(x - g_\epsilon^-(\mu) + \lambda)}{\sqrt{(r\epsilon(x - g_\epsilon^-(\mu) + \lambda))^2 + 1} + 1} - 1\right) = 0,$$

where we used $g_\epsilon^+(1-\mu) = 1 - g_\epsilon^-(\mu)$ to simplify the formula given in Lemma 31. Therefore, we define $x_\epsilon^-(\lambda, \mu, r) = x_\epsilon^+(1-\lambda, 1-\mu, r)$. Then, we have

$$\mathrm{kl}(g_\epsilon^-(\mu) - x_\epsilon^-(\lambda, \mu, r), \mu) + \frac{1}{r}h(r\epsilon(x_\epsilon^-(\lambda, \mu, r) + \lambda - g_\epsilon^-(\mu))) =$$

$$\mathrm{kl}(x_\epsilon^+(1-\lambda, 1-\mu, r) + g_\epsilon^+(1-\mu), 1-\mu) + \frac{h(r\epsilon(x_\epsilon^+(1-\lambda, 1-\mu, r) + g_\epsilon^+(1-\mu) - 1 + \lambda))}{r}$$

where we used that $\mathrm{kl}(g_\epsilon^-(\mu) - x_\epsilon^-(\lambda, \mu, r), \mu) = \mathrm{kl}(1 - g_\epsilon^-(\mu) + x_\epsilon^-(\lambda, \mu, r), 1 - \mu)$. Combining the above with the properties on $\widetilde{d}_\epsilon^+$ in Lemma 31 concludes the proof. $\square$

Lemma 33 shows that we can invert $\widetilde{d}_\epsilon^\pm$ with respect to their first argument, which is a key property used in Appendix F.

**Lemma 33.** *Let $\widetilde{d}_\epsilon^\pm$ as in Eq. (32). For all $(\mu, r, c) \in (0,1) \times \mathbb{R}_+^\star \times \mathbb{R}_+^\star$, there exists $x > 0$ such that $\widetilde{d}_\epsilon^+(\mu - x, \mu, r) = c$. For all $(\mu, r, c) \in (0,1) \times \mathbb{R}_+^\star \times \mathbb{R}_+^\star$, there exists $x > 0$ such that $\widetilde{d}_\epsilon^-(\mu + x, \mu, r) = c$.*

*Proof.* Let us define $f(x) = \widetilde{d}_\epsilon^+(\mu - x, \mu, r)$ for all $x > 0$. Using Lemma 31, we know that $f$ is continuous and increasing on $\mathbb{R}_+^\star$ and it satisfies $\lim_{x \to 0^+} f(x) = 0$ and $\lim_{x \to +\infty} f(x) = +\infty$. Therefore, there exists a unique $x > 0$ such that $\widetilde{d}_\epsilon^+(\mu - x, \mu, r) = c$. Using Lemma 32, we can conclude similarly for $\widetilde{d}_\epsilon^-$. $\qquad\square$

Lemma 33 shows that $\widetilde{d}_\epsilon^\pm$ is non-decreasing with respect to their first argument, which is a key property used in Appendix F.

**Lemma 34.** *Let $\widetilde{d}_\epsilon^\pm$ as in Eq. (32). For all $(\mu, r) \in (0, 1) \times \mathbb{R}_+^\star$ and all $(\lambda_1, \lambda_2) \in \mathbb{R} \times (-\infty, \mu)$,*

$$\widetilde{d}_\epsilon^+(\lambda_1, \mu, r) \geq \widetilde{d}_\epsilon^+(\lambda_2, \mu, r) \quad \Longrightarrow \quad \lambda_1 \leq \lambda_2$$

*For all $(\mu, r) \in (0, 1) \times \mathbb{R}_+^\star$ and all $(\lambda_1, \lambda_2) \in \mathbb{R} \times (\mu, +\infty)$,*

$$\widetilde{d}_\epsilon^-(\lambda_1, \mu, r) \geq \widetilde{d}_\epsilon^-(\lambda_2, \mu, r) \quad \Longrightarrow \quad \lambda_1 \geq \lambda_2$$

*Proof.* Using Lemma 31, we known that $\lambda \mapsto \widetilde{d}_\epsilon^+(\lambda, \mu, r)$ is decreasing on $(-\infty, \mu)$. Let $(\lambda_1, \lambda_2) \in \mathbb{R} \times (-\infty, \mu)$. Then, we have

$$\lambda_1 > \lambda_2 \quad \Longrightarrow \quad \widetilde{d}_\epsilon^+(\lambda_1, \mu, r) < \widetilde{d}_\epsilon^+(\lambda_2, \mu, r) \,,$$

which is equivalent to the statement of the lemma by contraposition. Using Lemma 32, we can conclude similarly for $\widetilde{d}_\epsilon^-$. $\qquad\square$

### G.2 Transportation Cost

Recall that $W_{\epsilon, a, b}$ is defined in Eq. (50), i.e., for all $(\mu, w) \in \mathbb{R}^K \times \mathbb{R}_+^K$,

$$\forall (a, b) \in [K]^2, \quad W_{\epsilon, a, b}(\mu, w) := \mathbb{1}\left([\mu_a]_0^1 > [\mu_b]_0^1\right) \inf_{u \in [0, 1]} \left\{w_a d_\epsilon^-(\mu_a, u) + w_b d_\epsilon^+(\mu_b, u)\right\} \,,$$

where $d_\epsilon^\pm$ are defined in Eq. (3).

Lemma 35 gathers regularity properties on the transportation costs.

**Lemma 35.** *Let $d_\epsilon^\pm$ as in Eq. (3). For all $(\lambda, \mu) \in (0, 1)^2$ such that $\lambda \geq \mu$ and $w \in \mathbb{R}_+^2$.*

- *The function $u \mapsto w_1 d_\epsilon^-(\lambda, u) + w_2 d_\epsilon^+(\mu, u)$ is strictly convex on $[\mu, \lambda]$ when $\max\{w_1, w_2\} > 0$ and on $[0, 1]$ when $\min\{w_1, w_2\} > 0$. Then,*

$$\inf_{u \in [0, 1]} \left\{w_1 d_\epsilon^-(\lambda, u) + w_2 d_\epsilon^+(\mu, u)\right\} = \inf_{u \in [\mu, \lambda]} \left\{w_1 d_\epsilon^-(\lambda, u) + w_2 d_\epsilon^+(\mu, u)\right\} \,.$$

- *The function $(\lambda, \mu, w) \mapsto \inf_{u \in [0,1]}\{w_1 d_\epsilon^-(\lambda, u) + w_2 d_\epsilon^+(\mu, u)\}$ is continuous on $(0, 1) \times (0, 1) \times \mathbb{R}_+^2$.*

- *If $\max\{w_1, w_2\} > 0$, $u_\star(\lambda, \mu, w) = \arg\min_{u \in [0,1]}\{w_1 d_\epsilon^-(\lambda, u) + w_2 d_\epsilon^+(\mu, u)\}$ is unique and continuous on $(0, 1) \times (0, 1) \times \mathbb{R}_+^2$.*

- *If $\min\{w_1, w_2\} > 0$ and $\lambda > \mu$, $u_\star(\lambda, \mu, w) \in (\mu, \lambda)$ and $\min\{d_\epsilon^-(\lambda, u_\star(\lambda, \mu, w)), d_\epsilon^+(\mu, u_\star(\lambda, \mu, w))\} > 0$.*

*Moreover,*

$$\inf_{u \in [0, 1]} \left\{w_1 d_\epsilon^-(\lambda, u) + w_2 d_\epsilon^+(\mu, u)\right\} = \inf_{(u_1, u_2) \in [0,1]^2 \,:\, u_1 \leq u_2} \left\{w_1 d_\epsilon^-(\lambda, u_1) + w_2 d_\epsilon^+(\mu, u_2)\right\} \,.$$

*Proof.* These results are obtained by leveraging Lemmas 25 and 26 at each step.

For $u \leq \mu$, the function is equal to $w_1 d_\epsilon^-(\lambda, u)$, which is decreasing and strictly convex on $[0, \lambda)$ unless $w_1 = 0$ since $u \leq \mu \leq \lambda$. Therefore, the minimum over that interval is attained at $\mu$. For $u \geq \lambda$, the function is equal to $w_2 d_\epsilon^+(\mu, u)$, which is increasing and strictly convex on $(\mu, 1]$ unless $w_2 = 0$ since $u \geq \lambda \geq \mu$. Therefore, the minimum over that interval is attained at $\lambda$. On the interval $(\mu, \lambda)$, the function is equal to $w_1 d_\epsilon^-(\lambda, u) + w_2 d_\epsilon^+(\mu, u)$, hence it is the sum of two convex functions,

one of which is strictly convex. Furthermore, the function is continuous at $\mu$ and $\lambda$. This concludes the first part of the proof.

As we have just shown, we can restrict the infimum to $[\mu, \lambda]$. We apply Berge's Maximum theorem [Berge, 1997, page 116]. Let

$$
\begin{aligned}
\phi(u, \lambda, \mu, w) &= -w_1 d_\epsilon^-(\lambda, u) - w_2 d_\epsilon^+(\mu, u) \,, \\
\Gamma(\lambda, \mu, w) &= [\mu, \lambda] \,, \\
M(\lambda, \mu, w) &= \max\{\phi(u, \lambda, \mu, w) \mid u \in \Gamma(\lambda, \mu, w)\} \,, \\
\Phi(\lambda, \mu, w) &= \arg\max\{\phi(u, \lambda, \mu, w) \mid u \in \Gamma(\lambda, \mu, w)\} \,.
\end{aligned}
$$

We verify the hypotheses of the theorem:

- $\phi$ is continuous on $[\mu, \lambda] \times (0, 1) \times (0, 1) \times \mathbb{R}_+^2$, by using the properties in Lemmas 25 and 26 since $(\lambda, \mu) \in (0, 1)^2$.

- $\Gamma$ is nonempty, compact-valued and continuous (since constant).

We obtain that $M$ is continuous on $(0, 1) \times (0, 1) \times \mathbb{R}_+^2$ and that $\Phi$ is upper hemicontinuous. This concludes the second part of the proof.

When $\max\{w_1, w_2\} > 0$, we have just shown that $\phi$ is a strictly concave function of $u$. Combining this with the fact that $\Gamma$ is convex, we can argue as in [Sundaram, 1996, Theorem 9.17] to prove that $\Phi$ is a single-valued upper hemicontinuous correspondence, hence a continuous function. This concludes the third part of the proof.

Suppose that $\min\{w_1, w_2\} > 0$ and $\lambda > \mu$. Using Lemmas 25 and 26, the function $u \mapsto w_1 d_\epsilon^-(\lambda, u) + w_2 d_\epsilon^+(\mu, u)$ is continuously differentiable on $(\mu, \lambda)$ with derivative $w_1 \frac{\partial d_\epsilon^-}{\partial u}(\lambda, u) + w_2 \frac{\partial d_\epsilon^+}{\partial u}(\mu, u)$ where

$$
\forall u \in (\mu, 1], \quad \frac{\partial d_\epsilon^+}{\partial u}(\mu, u) = \begin{cases} \frac{1 - e^{-\epsilon}}{1 - u(1 - e^{-\epsilon})} & \text{if } u \in (g_\epsilon^-(\mu), 1] \\ \frac{u - \mu}{u(1 - u)} & \text{if } u \in (\mu, g_\epsilon^-(\mu)] \end{cases} ,
$$

$$
\forall u \in [0, \lambda), \quad \frac{\partial d_\epsilon^-}{\partial u}(\lambda, u) = \begin{cases} -\frac{e^\epsilon - 1}{1 + u(e^\epsilon - 1)} & \text{if } u \in [0, g_\epsilon^+(\lambda)) \\ -\frac{\lambda - u}{u(1 - u)} & \text{if } u \in [g_\epsilon^+(\lambda), \lambda) \end{cases} .
$$

Since $\frac{\partial d_\epsilon^-}{\partial u}(\lambda, u) \to_{u \to \lambda^-} 0$ and $\frac{\partial d_\epsilon^+}{\partial u}(\mu, u) \to_{u \to \mu^+} 0$, we obtain

$$
\lim_{u \to \lambda^-} \left\{ w_1 \frac{\partial d_\epsilon^-}{\partial u}(\lambda, u) + w_2 \frac{\partial d_\epsilon^+}{\partial u}(\mu, u) \right\} = w_2 \frac{\partial d_\epsilon^+}{\partial u}(\mu, \lambda) > 0 \,,
$$

$$
\lim_{u \to \mu^+} \left\{ w_1 \frac{\partial d_\epsilon^-}{\partial u}(\lambda, u) + w_2 \frac{\partial d_\epsilon^+}{\partial u}(\mu, u) \right\} = w_1 \frac{\partial d_\epsilon^-}{\partial u}(\lambda, \mu) < 0 \,.
$$

Therefore, the infimum is attained inside the open interval. Using Lemmas 25 and 26, we can conclude the proof of the first part of the fourth property.

Using the strict convexity of $u_1 \mapsto w_1 d_\epsilon^-(\lambda, u_1)$ and $u_2 \mapsto w_2 d_\epsilon^+(\mu, u_2)$ on $(\mu, \lambda)$, we obtain that

$$
\inf_{u \in (\mu, \lambda)} \{w_1 d_\epsilon^-(\lambda, u) + w_2 d_\epsilon^+(\mu, u)\} = \inf_{(u_1, u_2)\,:\,\mu < u_1 \leq u_2 < \lambda} \{w_1 d_\epsilon^-(\lambda, u_1) + w_2 d_\epsilon^+(\mu, u_2)\} \,.
$$

Re-using the same arguments as above, we obtain that

$$
\inf_{(u_1, u_2)\,:\,\mu < u_1 \leq u_2 < \lambda} \{w_1 d_\epsilon^-(\lambda, u_1) + w_2 d_\epsilon^+(\mu, u_2)\}
$$
$$
= \inf_{(u_1, u_2) \in [0, 1]^2\,:\,u_1 \leq u_2} \{w_1 d_\epsilon^-(\lambda, u_1) + w_2 d_\epsilon^+(\mu, u_2)\} \,.
$$

This concludes the proof of the second part of the fourth property. $\qquad\square$

---

Lemma 36 relates the transportation costs $W_{\epsilon, a^\star, a}$ with the transportation costs used in Eq. (2) to define the characteristic time. Crucially, this shows the equivalence with the definitions in Eq. (35).

**Lemma 36.** *Let $W_{\epsilon,a^\star,a}$ and $\mathrm{d}_\epsilon$ as in Eq. (4) and (1). Let $\mu \in (0,1)^K$ such that $a^\star(\mu) = \{a^\star\}$. Let $Alt(\mu) = \{\lambda \in (0,1)^K \mid a^\star(\lambda) \neq \{a^\star\}\}$. Then,*

$$\forall w \in \triangle_K, \quad \inf_{\lambda \in Alt(\mu)} \sum_{a \in [K]} w_a \mathrm{d}_\epsilon(\mu_a, \lambda_a) = \min_{a \neq a^\star} W_{\epsilon,a^\star,a}(\mu, w) \,.$$

*Proof.* It is direct to see that $Alt(\mu) = \bigcup_{a \neq a^\star} \mathcal{C}_a$ where $\mathcal{C}_a = \{\lambda \in (0,1)^K \mid \lambda_a \geq \lambda_{a^\star}\}$. Then,

$$\forall w \in \triangle_K, \quad \inf_{\lambda \in Alt(\mu)} \sum_{a \in [K]} w_a d_\epsilon(\mu_a, \lambda_a) = \min_{a \neq a^\star} \inf_{\lambda \in \mathcal{C}_a} \sum_{c \in [K]} w_c d_\epsilon(\mu_c, \lambda_c) \,.$$

By non-negativity of $d_\epsilon(\mu_a, \lambda_a)$ for all $a \in [K]$, we obtain

$$\inf_{\lambda \in \mathcal{C}_a} \sum_{c \in [K]} w_c d_\epsilon(\mu_c, \lambda_c) = \inf_{\lambda \in \mathcal{C}_a} \sum_{c \in \{a, a^\star\}} w_c d_\epsilon(\mu_c, \lambda_c)$$

$$= \inf_{(\lambda_a, \lambda_{a^\star}) \in (0,1)^2, \, \lambda_a \geq \lambda_{a^\star}} \sum_{c \in \{a, a^\star\}} w_c d_\epsilon(\mu_c, \lambda_c) \,,$$

where the two equalities are obtained by choosing $\lambda(a) \in (0,1)^K$ such that $\lambda(a)_b = \mu_b$ for all $b \notin \{a, a^\star\}$ with the two other coordinates choosen freely such that $\lambda(a)_a \geq \lambda(a)_{a^\star}$. Using that $\mu_{a^\star} > \mu_a$, we can partition this set as follows

$$\mathcal{C}_{a,a^\star} = \{(\lambda_a, \lambda_{a^\star}) \in (0,1)^2 \mid \lambda_a \geq \lambda_{a^\star}\} = \{(\lambda_a, \lambda_{a^\star}) \in (0, \mu_a)^2 \mid \lambda_a \geq \lambda_{a^\star}\}$$
$$\cup \{(\lambda_a, \lambda_{a^\star}) \in [\mu_a, \mu_{a^\star}] \times (0, \mu_a)\}$$
$$\cup \{(\lambda_a, \lambda_{a^\star}) \in (\mu_{a^\star}, 1)^2 \mid \lambda_a \geq \lambda_{a^\star}\}$$
$$\cup \{(\lambda_a, \lambda_{a^\star}) \in (\mu_{a^\star}, 1) \times [\mu_a, \mu_{a^\star}]\}$$
$$\cup \{(\lambda_a, \lambda_{a^\star}) \in [\mu_a, \mu_{a^\star}]^2 \mid \lambda_a \geq \lambda_{a^\star}\} \,.$$

Using Lemma 22, $\mu_{a^\star} > \mu_a$ and Lemmas 25 and 26, we obtain

$$\inf_{(\lambda_a, \lambda_{a^\star}) \in (0, \mu_a)^2 \mid \lambda_a \geq \lambda_{a^\star}} \{w_{a^\star} d_\epsilon(\mu_{a^\star}, \lambda_{a^\star}) + w_a d_\epsilon(\mu_a, \lambda_a)\}$$
$$= \inf_{(\lambda_a, \lambda_{a^\star}) \in (0, \mu_a)^2 \mid \lambda_a \geq \lambda_{a^\star}} \{w_{a^\star} d_\epsilon^-(\mu_{a^\star}, \lambda_{a^\star}) + w_a d_\epsilon^-(\mu_a, \lambda_a)\} = w_{a^\star} d_\epsilon^-(\mu_{a^\star}, \mu_a) \,,$$

$$\inf_{(\lambda_a, \lambda_{a^\star}) \in [\mu_a, \mu_{a^\star}] \times (0, \mu_a)} \{w_{a^\star} d_\epsilon(\mu_{a^\star}, \lambda_{a^\star}) + w_a d_\epsilon(\mu_a, \lambda_a)\}$$
$$= \inf_{(\lambda_a, \lambda_{a^\star}) \in [\mu_a, \mu_{a^\star}] \times (0, \mu_a)} \{w_{a^\star} d_\epsilon^-(\mu_{a^\star}, \lambda_{a^\star}) + w_a d_\epsilon^+(\mu_a, \lambda_a)\} = w_{a^\star} d_\epsilon^-(\mu_{a^\star}, \mu_a) \,,$$

$$\inf_{(\lambda_a, \lambda_{a^\star}) \in (\mu_{a^\star}, 1)^2 \mid \lambda_a \geq \lambda_{a^\star}} \{w_{a^\star} d_\epsilon(\mu_{a^\star}, \lambda_{a^\star}) + w_a d_\epsilon(\mu_a, \lambda_a)\}$$
$$= \inf_{(\lambda_a, \lambda_{a^\star}) \in (\mu_{a^\star}, 1)^2 \mid \lambda_a \geq \lambda_{a^\star}} \{w_{a^\star} d_\epsilon^+(\mu_{a^\star}, \lambda_{a^\star}) + w_a d_\epsilon^+(\mu_a, \lambda_a)\} = w_a d_\epsilon^+(\mu_a, \mu_{a^\star}) \,,$$

$$\inf_{(\lambda_a, \lambda_{a^\star}) \in (\mu_{a^\star}, 1) \times [\mu_a, \mu_{a^\star}]} \{w_{a^\star} d_\epsilon(\mu_{a^\star}, \lambda_{a^\star}) + w_a d_\epsilon(\mu_a, \lambda_a)\}$$
$$= \inf_{(\lambda_a, \lambda_{a^\star}) \in (\mu_{a^\star}, 1) \times [\mu_a, \mu_{a^\star}]} \{w_{a^\star} d_\epsilon^-(\mu_{a^\star}, \lambda_{a^\star}) + w_a d_\epsilon^+(\mu_a, \lambda_a)\} = w_a d_\epsilon^+(\mu_a, \mu_{a^\star}) \,,$$

$$\inf_{(\lambda_a, \lambda_{a^\star}) \in [\mu_a, \mu_{a^\star}]^2 \mid \lambda_a \geq \lambda_{a^\star}} \{w_{a^\star} d_\epsilon(\mu_{a^\star}, \lambda_{a^\star}) + w_a d_\epsilon(\mu_a, \lambda_a)\}$$
$$= \inf_{(\lambda_a, \lambda_{a^\star}) \in [\mu_a, \mu_{a^\star}]^2 \mid \lambda_a \geq \lambda_{a^\star}} \{w_{a^\star} d_\epsilon^-(\mu_{a^\star}, \lambda_{a^\star}) + w_a d_\epsilon^+(\mu_a, \lambda_a)\} \,.$$

Therefore, we obtain

$$\inf_{(\lambda_a, \lambda_{a^\star}) \in \mathcal{C}_{a,a^\star}} \{w_{a^\star} d_\epsilon(\mu_{a^\star}, \lambda_{a^\star}) + w_a d_\epsilon(\mu_a, \lambda_a)\}$$
$$= \inf_{(\lambda_a, \lambda_{a^\star}) \in [\mu_a, \mu_{a^\star}]^2 \mid \lambda_a \geq \lambda_{a^\star}} \{w_{a^\star} d_\epsilon^-(\mu_{a^\star}, \lambda_{a^\star}) + w_a d_\epsilon^+(\mu_a, \lambda_a)\}$$
$$= \inf_{u \in [\mu_a, \mu_{a^\star}]^2} \{w_{a^\star} d_\epsilon^-(\mu_{a^\star}, u) + w_a d_\epsilon^+(\mu_a, u)\}$$
$$= \inf_{u \in [0,1]} \{w_{a^\star} d_\epsilon^-(\mu_{a^\star}, u) + w_a d_\epsilon^+(\mu_a, u)\} = W_{\epsilon,a^\star,a}(\mu, w) \,,$$

where the second equality is obtained similarly as in Lemma 35 by leveraging the strict convexity of $d_\epsilon^\pm$ in their second argument (see Lemmas 25 and 26). We used Lemma 35 and the definition of $W_{\epsilon,a^\star,a}(\mu, w)$ for the last two equalities. This concludes the proof. $\qquad\square$

Lemma 37 gathers additional properties on the transportation costs.

**Lemma 37.** *Let $d_\epsilon^\pm$ as in Eq. (3).*

- *Let $(\lambda, \mu) \in (0,1)^2$ such that $\lambda > \mu$. When $w_2 > 0$, the function $w_1 \mapsto \min_{u \in [0,1]}\{w_1 d_\epsilon^-(\lambda, u) + w_2 d_\epsilon^+(\mu, u)\}$ is increasing on $\mathbb{R}_+$. When $w_1 > 0$, the function $w_2 \mapsto \min_{u \in [0,1]}\{w_1 d_\epsilon^-(\lambda, u) + w_2 d_\epsilon^+(\mu, u)\}$ is increasing on $\mathbb{R}_+$.*

- *Let $(\lambda, \mu) \in (0,1)^2$ and $\mu \in (0,1)^K$. The function $w \mapsto \min_{u \in [0,1]}\{w_1 d_\epsilon^-(\lambda, u) + w_2 d_\epsilon^+(\mu, u)\}$ is concave on $\mathbb{R}_+^2$. The function $w \mapsto \min_{a \in [K]\setminus\{1\}} \min_{u \in [0,1]}\{w_1 d_\epsilon^-(\mu_1, u) + w_a d_\epsilon^+(\mu_a, u)\}$ is concave on $\mathbb{R}_+^K$.*

*Proof.* Let $w_2 > 0$ and $w_1' > w_1 \geq 0$. Using Lemma 35, since $w_1' > 0$, there exists $u' \in [0,1]$ with $d_\epsilon^-(\lambda, u') > 0$ such that

$$\min_{u \in [0,1]}\{w_1' d_\epsilon^-(\lambda, u) + w_2 d_\epsilon^+(\mu, u)\} = w_1' d_\epsilon^-(\lambda, u') + w_2 d_\epsilon^+(\mu, u')$$

$$> w_1 d_\epsilon^-(\lambda, u') + w_2 d_\epsilon^+(\mu, u')$$

$$\geq \min_{u \in [0,1]}\{w_1 d_\epsilon^-(\lambda, u) + w_2 d_\epsilon^+(\mu, u)\} .$$

Let $w_1 > 0$ and $w_2' > w_2 \geq 0$. Then, we can show similarly by using Lemma 35 that

$$\min_{u \in [0,1]}\{w_1 d_\epsilon^-(\lambda, u) + w_2' d_\epsilon^+(\mu, u)\} > \min_{u \in [0,1]}\{w_1 d_\epsilon^-(\lambda, u) + w_2 d_\epsilon^+(\mu, u)\} .$$

This concludes the first part of the proof. The proof of the second part is direct since those functions are minimum of linear functions, hence concave. $\qquad\square$

Lemma 38 gives a closed-form solution for the transportation costs. This is a key property used in our implementation to reduce the computational cost.

**Lemma 38.** *Let $d_\epsilon^\pm$ and $g_\epsilon^\pm$ as in Eq. (3) and (30). For all $(a,c) \in \mathbb{R}_+^2$ and $b \in \mathbb{R}$, let $r_{1,+}(a,b,c) := \frac{\sqrt{b^2+4ac}-b}{2a}$. For all $(\lambda, \mu) \in (0,1)^2$ and $w \in \mathbb{R}_+^2$ such that $\min\{w_1, w_2\} > 0$ and $\lambda > \mu$.*

$\bullet$ *When (1) $g_\epsilon^-(\mu) \geq \lambda$, or (2) $g_\epsilon^-(\mu) < \lambda$, $g_\epsilon^+(\lambda) \leq g_\epsilon^-(\mu)$ and $\frac{w_2\mu+w_1\lambda}{w_2+w_1} \in [g_\epsilon^+(\lambda), g_\epsilon^-(\mu)]$, we have $u_\star(\lambda, \mu, w) = \frac{w_2\mu+w_1\lambda}{w_2+w_1}$ and*

$$\min_{u \in [0,1]}\{w_1 d_\epsilon^-(\lambda, u) + w_2 d_\epsilon^+(\mu, u)\} = w_1 \mathrm{kl}(\lambda, u_\star(\lambda, \mu, w)) + w_2 \mathrm{kl}(\mu, u_\star(\lambda, \mu, w)) .$$

$\bullet$ *When (3) $g_\epsilon^-(\mu) < \lambda$, $g_\epsilon^+(\lambda) > g_\epsilon^-(\mu)$ and $u_{3,\star}(w) \in [g_\epsilon^-(\mu), g_\epsilon^+(\lambda)]$ where*

$$u_{3,\star}(w) := \frac{w_1(e^\epsilon - 1) - w_2(1 - e^{-\epsilon})}{(w_2 + w_1)(1 - e^{-\epsilon})(e^\epsilon - 1)} ,$$

*we have $u_\star(\lambda, \mu, w) = u_{3,\star}(w)$ and*

$$\min_{u \in [0,1]}\{w_1 d_\epsilon^-(\lambda, u) + w_2 d_\epsilon^+(\mu, u)\}$$
$$= w_1 \left(-\log\left(1 + u_{3,\star}(w)(e^\epsilon - 1)\right) + \epsilon\lambda\right) + w_2 \left(-\log\left(1 - u_{3,\star}(w)(1 - e^{-\epsilon})\right) - \epsilon\mu\right) .$$

$\bullet$ *When (4) $g_\epsilon^-(\mu) < \lambda$, $g_\epsilon^+(\lambda) \leq g_\epsilon^-(\mu)$ and $\frac{w_2\mu+w_1\lambda}{w_2+w_1} \in (\mu, g_\epsilon^+(\lambda))$, or (5) $g_\epsilon^-(\mu) < \lambda$, $g_\epsilon^+(\lambda) > g_\epsilon^-(\mu)$ and $u_{3,\star}(w) < g_\epsilon^-(\mu)$, we have $u_\star(\lambda, \mu, w) = u_{1,\star}(\mu, w)$ and*

$$u_{1,\star}(\mu, w) := r_{1,+}\left((w_2 + w_1)(e^\epsilon - 1), (w_2 - (w_2\mu + w_1)(e^\epsilon - 1)), w_2\mu\right) ,$$
$$\min_{u \in [0,1]}\{w_1 d_\epsilon^-(\lambda, u) + w_2 d_\epsilon^+(\mu, u)\}$$

$$= w_1 \left( -\log \left( 1 + u_{1,\star}(\mu, w)(e^\epsilon - 1) \right) + \epsilon\lambda \right) + w_2 \mathrm{kl}(\mu, u_{1,\star}(\mu, w)) .$$

• *When (6) $g_\epsilon^-(\mu) < \lambda$, $g_\epsilon^+(\lambda) \leq g_\epsilon^-(\mu)$ and $\frac{w_2\mu + w_1\lambda}{w_2 + w_1} \in (g_\epsilon^-(\mu), \lambda)$, or (7) $g_\epsilon^-(\mu) < \lambda$, $g_\epsilon^+(\lambda) > g_\epsilon^-(\mu)$ and $u_{3,\star}(w) > g_\epsilon^+(\lambda)$, we have $u_\star(\lambda, \mu, w) = u_{2,\star}(\lambda, w)$ and*

$$u_{2,\star}(\lambda, w) := 1 - r_{1,+} \left( (w_2 + w_1)(e^\epsilon - 1), (w_1 - (w_1(1 - \lambda) + w_2)(e^\epsilon - 1)), w_1(1 - \lambda) \right) ,$$

$$\min_{u \in [0,1]} \left\{ w_1 d_\epsilon^-(\lambda, u) + w_2 d_\epsilon^+(\mu, u) \right\}$$

$$= w_1 \mathrm{kl}(\lambda, u_{2,\star}(\lambda, w)) + w_2 \left( -\log \left( 1 - u_{2,\star}(\lambda, w)(1 - e^{-\epsilon}) \right) - \epsilon\mu \right) .$$

*Proof.* Suppose that $g_\epsilon^-(\mu) \geq \lambda$. Using Lemma 24, we know that $g_\epsilon^-(\mu) \geq \lambda$ if and only if $\mu \geq g_\epsilon^+(\lambda)$. Therefore, for all $u \in (\mu, \lambda)$, we have

$$w_1 \frac{\partial d_\epsilon^-}{\partial u}(\lambda, u) + w_2 \frac{\partial d_\epsilon^+}{\partial u}(\mu, u) = -w_1 \frac{\lambda - u}{u(1 - u)} + w_2 \frac{u - \mu}{u(1 - u)} = \frac{(w_2 + w_1)u - (w_2\mu + w_1\lambda)}{u(1 - u)} .$$

Therefore, we have

$$u_\star(\lambda, \mu, w) = \frac{w_2\mu + w_1\lambda}{w_2 + w_1} \in (\mu, \lambda) .$$

Suppose that $g_\epsilon^-(\mu) < \lambda$. Using Lemma 24, we know that $g_\epsilon^-(\mu) < \lambda$ if and only if $\mu < g_\epsilon^+(\lambda)$. Using strict convexity of the function on $(\mu, \lambda)$, it is enough to exhibit one local minimum to obtain a global minimum on $(\mu, \lambda)$.

Suppose that $g_\epsilon^+(\lambda) \leq g_\epsilon^-(\mu)$. Similarly as above, we obtain, for all $u \in [g_\epsilon^+(\lambda), g_\epsilon^-(\mu)]$,

$$w_1 \frac{\partial d_\epsilon^-}{\partial u}(\lambda, u) + w_2 \frac{\partial d_\epsilon^+}{\partial u}(\mu, u) = \frac{(w_2 + w_1)u - (w_2\mu + w_1\lambda)}{u(1 - u)} .$$

Suppose that $\frac{w_2\mu + w_1\lambda}{w_2 + w_1} \in [g_\epsilon^+(\lambda), g_\epsilon^-(\mu)]$. Then, we can conclude as above that

$$u_\star(\lambda, \mu, w) = \frac{w_2\mu + w_1\lambda}{w_2 + w_1} \in [g_\epsilon^+(\lambda), g_\epsilon^-(\mu)] ,$$

since it is a local minimum of a strictly convex function.

Suppose that $\frac{w_2\mu + w_1\lambda}{w_2 + w_1} < g_\epsilon^+(\lambda)$. Since the gradient is positive on $[g_\epsilon^+(\lambda), g_\epsilon^-(\mu)]$, we know that the minimum on $(\mu, \lambda)$ is achieved on $(\mu, g_\epsilon^+(\lambda))$, i.e., $u_\star(\lambda, \mu, w) \in (\mu, g_\epsilon^+(\lambda))$. Then, for all $u \in (\mu, g_\epsilon^+(\lambda))$,

$$w_1 \frac{\partial d_\epsilon^-}{\partial u}(\lambda, u) + w_2 \frac{\partial d_\epsilon^+}{\partial u}(\mu, u) = -w_1 \frac{e^\epsilon - 1}{1 + u(e^\epsilon - 1)} + w_2 \frac{u - \mu}{u(1 - u)} .$$

Using Lemma 24, direct computation yields

$$w_1 \frac{\partial d_\epsilon^-}{\partial u}(\lambda, u) + w_2 \frac{\partial d_\epsilon^+}{\partial u}(\mu, u) > 0 \quad \Longleftrightarrow \quad \mu < u \left( 1 + \frac{w_1}{w_2} \left( 1 - \frac{g_\epsilon^-(u)}{u} \right) \right) ,$$

$$\lim_{u \to \mu^+} u \left( 1 + \frac{w_1}{w_2} \left( 1 - \frac{g_\epsilon^-(u)}{u} \right) \right) = \mu \left( 1 + \frac{w_1}{w_2} \left( 1 - \frac{g_\epsilon^-(\mu)}{\mu} \right) \right) < \mu ,$$

$$\lim_{u \to g_\epsilon^+(\lambda)^-} u \left( 1 + \frac{w_1}{w_2} \left( 1 - \frac{g_\epsilon^-(u)}{u} \right) \right) = g_\epsilon^+(\lambda) \left( 1 + \frac{w_1}{w_2} \left( 1 - \frac{\lambda}{g_\epsilon^+(\lambda)} \right) \right) > \mu ,$$

where the second result uses that $u < g_\epsilon^-(u)$ and the last result is obtained by continuity of the differentials (Lemmas 25 and 26) and the positivity on $[g_\epsilon^+(\lambda), g_\epsilon^-(\mu)]$. For all $(a, c) \in \mathbb{R}_+^2$ and $b \in \mathbb{R}$, we define $r_{1,+}(a, b, c) = \frac{\sqrt{b^2 + 4ac} - b}{2a}$. Therefore, we have

$$w_1 \frac{\partial d_\epsilon^-}{\partial u}(\lambda, u) + w_2 \frac{\partial d_\epsilon^+}{\partial u}(\mu, u) = 0$$

$$\Longleftrightarrow \quad (w_2 + w_1)(e^\epsilon - 1)u^2 + (w_2 - (w_2\mu + w_1)(e^\epsilon - 1)) u - w_2\mu = 0$$

$$\Longleftrightarrow \quad u_\star(\lambda, \mu, w) = r_{1,+} \left( (w_2 + w_1)(e^\epsilon - 1), (w_2 - (w_2\mu + w_1)(e^\epsilon - 1)), w_2\mu \right) \in (\mu, g_\epsilon^+(\lambda))$$

where we used that $u_\star(\lambda, \mu, w) \in (\mu, g_\epsilon^+(\lambda))$ is unique for the last equivalence, and that the second root of the second order polynomial equation is negative. Notice that $u_\star(\lambda, \mu, w)$ is independent of $\lambda$.

Suppose that $\frac{w_2\mu + w_1\lambda}{w_2 + w_1} > g_\epsilon^-(\mu)$. Since the gradient is negative on $[g_\epsilon^+(\lambda), g_\epsilon^-(\mu)]$, we know that the minimum on $(\mu, \lambda)$ is achieved on $(g_\epsilon^-(\mu), \lambda)$, i.e., $u_\star(\lambda, \mu, w) \in (g_\epsilon^-(\mu), \lambda)$. Then, for all $u \in (g_\epsilon^-(\mu), \lambda)$,

$$w_1 \frac{\partial d_\epsilon^-}{\partial u}(\lambda, u) + w_2 \frac{\partial d_\epsilon^+}{\partial u}(\mu, u) = -w_1 \frac{\lambda - u}{u(1 - u)} + w_2 \frac{1 - e^{-\epsilon}}{1 - u(1 - e^{-\epsilon})}.$$

Using Lemma 24, direct computation yields

$$w_1 \frac{\partial d_\epsilon^-}{\partial u}(\lambda, u) + w_2 \frac{\partial d_\epsilon^+}{\partial u}(\mu, u) < 0 \quad \Longleftrightarrow \quad \lambda > u\left(1 + \frac{w_2}{w_1}\left(1 - \frac{g_\epsilon^+(u)}{u}\right)\right),$$

$$\lim_{u \to \lambda^-} u\left(1 + \frac{w_2}{w_1}\left(1 - \frac{g_\epsilon^+(u)}{u}\right)\right) = \lambda\left(1 + \frac{w_2}{w_1}\left(1 - \frac{g_\epsilon^+(\lambda)}{\lambda}\right)\right) > \lambda,$$

$$\lim_{u \to g_\epsilon^-(\mu)^+} u\left(1 + \frac{w_2}{w_1}\left(1 - \frac{g_\epsilon^+(u)}{u}\right)\right) = g_\epsilon^-(\mu)\left(1 + \frac{w_2}{w_1}\left(1 - \frac{\mu}{g_\epsilon^-(\mu)}\right)\right) < \lambda,$$

where the second result uses that $u > g_\epsilon^+(u)$ and the last result is obtained by continuity of the differentials (Lemmas 25 and 26) and the negativity on $[g_\epsilon^+(\lambda), g_\epsilon^-(\mu)]$.

Using Lemma 23, we obtain

$$\arg\min_{u \in [0,1]}\{w_1 d_\epsilon^-(\lambda, u) + w_2 d_\epsilon^+(\mu, u)\} = 1 - \arg\min_{u \in [0,1]}\{w_1 d_\epsilon^+(1 - \lambda, u) + w_2 d_\epsilon^-(1 - \mu, u)\}.$$

Using Lemma 24, we obtain

$$g_\epsilon^-(\mu) < \lambda \quad \Longleftrightarrow \quad g_\epsilon^-(1 - \lambda) < 1 - \mu,$$

$$\frac{w_2\mu + w_1\lambda}{w_2 + w_1} \in (g_\epsilon^-(\mu), \lambda) \quad \Longleftrightarrow \quad \frac{w_2(1 - \mu) + w_1(1 - \lambda)}{w_2 + w_1} \in (1 - \lambda, g_\epsilon^+(1 - \mu)).$$

Therefore, we can leverage the above case to obtain $u_\star(\lambda, \mu, w) = u_{2,\star}(\lambda, w)$ where

$$u_{2,\star}(\lambda, w) = 1 - r_{1,+}\left((w_2 + w_1)(e^\epsilon - 1), (w_1 - (w_1(1 - \lambda) + w_2)(e^\epsilon - 1)), w_1(1 - \lambda)\right)$$

Notice that $u_\star(\lambda, \mu, w)$ is independent of $\mu$.

Suppose that $g_\epsilon^-(\mu) < \lambda$ and $g_\epsilon^+(\lambda) > g_\epsilon^-(\mu)$. Similarly as above, we obtain, for all $u \in [g_\epsilon^-(\mu), g_\epsilon^+(\lambda)]$,

$$w_1 \frac{\partial d_\epsilon^-}{\partial u}(\lambda, u) + w_2 \frac{\partial d_\epsilon^+}{\partial u}(\mu, u) = -w_1 \frac{e^\epsilon - 1}{1 + u(e^\epsilon - 1)} + w_2 \frac{1 - e^{-\epsilon}}{1 - u(1 - e^{-\epsilon})}.$$

Therefore, we obtain

$$w_1 \frac{\partial d_\epsilon^-}{\partial u}(\lambda, u) + w_2 \frac{\partial d_\epsilon^+}{\partial u}(\mu, u) > 0$$

$$\Longleftrightarrow \quad w_2(1 - e^{-\epsilon})(1 + u(e^\epsilon - 1)) - w_1(e^\epsilon - 1)(1 - u(1 - e^{-\epsilon})) > 0$$

$$\Longleftrightarrow \quad u > u_{3,\star}(w) := \frac{w_1(e^\epsilon - 1) - w_2(1 - e^{-\epsilon})}{(w_2 + w_1)(1 - e^{-\epsilon})(e^\epsilon - 1)}.$$

Suppose that $u_{3,\star}(w) \in [g_\epsilon^-(\mu), g_\epsilon^+(\lambda)]$. Then, we can conclude as above that

$$u_\star(\lambda, \mu, w) = u_{3,\star}(w) \in [g_\epsilon^-(\mu), g_\epsilon^+(\lambda)],$$

since it is a local minimum of a strictly convex function. Notice that $u_{3,\star}(w)$ is independent of $(\lambda, \mu)$.

Suppose that $u_{3,\star}(w) > g_\epsilon^+(\lambda)$. Since the gradient is negative on $[g_\epsilon^-(\mu), g_\epsilon^+(\lambda)]$, we know that the minimum on $(\mu, \lambda)$ is achieved on $(g_\epsilon^+(\lambda), \lambda)$, i.e., $u_\star(\lambda, \mu, w) \in (g_\epsilon^+(\lambda), \lambda)$. Then, for all $u \in (g_\epsilon^+(\lambda), \lambda)$,

$$w_1 \frac{\partial d_\epsilon^-}{\partial u}(\lambda, u) + w_2 \frac{\partial d_\epsilon^+}{\partial u}(\mu, u) = -w_1 \frac{\lambda - u}{u(1 - u)} + w_2 \frac{1 - e^{-\epsilon}}{1 - u(1 - e^{-\epsilon})}.$$

This recovers the condition solved above. As we know that $u_\star(\lambda, \mu, w) \in (g_\epsilon^+(\lambda), \lambda)$, we obtain $u_\star(\lambda, \mu, w) = u_{2,\star}(\lambda, w)$ where

$$u_{2,\star}(\lambda, w) = 1 - r_{1,+}\left((w_2 + w_1)(e^\epsilon - 1), (w_1 - (w_1(1 - \lambda) + w_2)(e^\epsilon - 1)), w_1(1 - \lambda)\right)$$

Suppose that $u_{3,\star}(w) < g_\epsilon^-(\mu)$. Since the gradient is positive on $[g_\epsilon^-(\mu), g_\epsilon^+(\lambda)]$, we know that the minimum on $(\mu, \lambda)$ is achieved on $(\mu, g_\epsilon^-(\mu))$, i.e., $u_\star(\lambda, \mu, w) \in (\mu, g_\epsilon^-(\mu))$. Then, for all $u \in (\mu, g_\epsilon^-(\mu))$,

$$w_1 \frac{\partial d_\epsilon^-}{\partial u}(\lambda, u) + w_2 \frac{\partial d_\epsilon^+}{\partial u}(\mu, u) = -w_1 \frac{e^\epsilon - 1}{1 + u(e^\epsilon - 1)} + w_2 \frac{u - \mu}{u(1 - u)} .$$

This recovers the condition solved above. As we know that $u_\star(\lambda, \mu, w) \in (\mu, g_\epsilon^-(\mu))$, we obtain $u_\star(\lambda, \mu, w) = u_{1,\star}(\mu, w)$ where

$$u_{1,\star}(\mu, w) = r_{1,+}\left((w_2 + w_1)(e^\epsilon - 1), (w_2 - (w_2\mu + w_1)(e^\epsilon - 1)), w_2\mu\right) .$$

This concludes the proof.

$\square$

### G.2.1 Modified Transportation Cost

Let $\eta > 0$ be the geometric parameter used for the geometric grid update of our private empirical mean estimator. Let us define

$$\forall x \geq 1, \quad r(x) := \frac{x}{1 + \log_{1+\eta} x} , \tag{33}$$

which is increasing if and only if $x > \frac{e}{1+\eta}$. For all $(\mu, w) \in \mathbb{R}^K \times \mathbb{R}_+^K$ and all $(a, b) \in [K]^2$ such that $a \neq b$, we define

$$\widetilde{W}_{\epsilon,a,b}(\mu, w) := \mathbb{1}\left([\mu_a]_0^1 > [\mu_b]_0^1\right) \inf_{u \in (0,1)} \left\{w_a \widetilde{d}_\epsilon^-(\mu_a, u, r(w_a)) + w_b \widetilde{d}_\epsilon^+(\mu_b, u, r(w_b))\right\} , \tag{34}$$

where $\widetilde{d}_\epsilon^\pm$ are defined in Eq. (32).

Lemma 39 gathers regularity properties of the function $r$ defined in Eq. (33).

**Lemma 39.** *Let $r$ as in Eq. (33). Then,*

$$\forall x \geq 1, \quad r'(x) = \frac{\log(x(1 + \eta)/e)}{\log(1 + \eta)(1 + \log_{1+\eta} x)^2} ,$$

$$r''(x) = -\frac{1}{x(\log(1 + \eta))^2} \frac{\log((1 + \eta)xe^{-2})}{(1 + \log_{1+\eta} x)^3} .$$

*On $[1, +\infty)$, the function $r$ is twice continuously differentiable. It is decreasing on $[1, e/(1 + \eta))$ and increasing on $(e/(1 + \eta), +\infty)$; its minium is $r(e/(1 + \eta)) \in (0, 1)$. It is strictly convex on $[1, e^2/(1 + \eta))$ and strictly concave on $(e^2/(1 + \eta), +\infty)$.*

*Proof.* The proof is obtained by direct differentiation and manipulation. We have

$$\forall \eta > 0, \quad r(e/(1 + \eta)) = \frac{e \log(1 + \eta)}{1 + \eta} \in (0, 1) .$$

$\square$

Lemma 40 shows that the modified transportation costs can be rewritten differently, which is a key property used in Appendix F.

**Lemma 40.** *Let $\widetilde{d}_\epsilon^\pm$ as in Eq. (32), and $r$ as in Eq. (33). For all $(\lambda, \mu) \in \mathbb{R}^2$ such that $[\lambda]_0^1 > [\mu]_0^1$ and $(w_1, w_2) \in [1, +\infty)^2$. Then,*

$$\inf_{u \in (0,1)} \{w_1 \widetilde{d}_\epsilon^-(\lambda, u, r(w_1)) + w_2 \widetilde{d}_\epsilon^+(\mu, u, r(w_2))\}$$

$$= \inf_{u \in ([\mu]_0^1, [\lambda]_0^1)} \{w_1 \widetilde{d}_\epsilon^-(\lambda, u, r(w_1)) + w_2 \widetilde{d}_\epsilon^+(\mu, u, r(w_2))\}$$

$$= \inf_{(u_1, u_2) \in (0,1)^2 : u_1 \leq u_2} \{w_1 \widetilde{d}_\epsilon^-(\lambda, u_1, r(w_1)) + w_2 \widetilde{d}_\epsilon^+(\mu, u_2, r(w_2))\} .$$

*Proof.* These results are obtained by leveraging Lemmas 31 and 32.

Note that the condition $[\lambda]_0^1 > [\mu]_0^1$ implies that $\mu \in (-\infty, 1)$ and $\lambda \in (0, +\infty)$, i.e., $[\mu]_0^1 = \max\{0, \mu\}$ and $[\lambda]_0^1 = \min\{1, \lambda\}$.

Suppose that $\mu \leq 0$ and $\lambda \geq 1$. Then, we have $[\mu]_0^1 = 0$ and $[\lambda]_0^1 = 1$. Therefore, the first part of the result holds by definition.

Suppose that $\mu \leq 0$ and $\lambda \in (0, 1)$. Then, we have $[\mu]_0^1 = 0$ and $[\lambda]_0^1 = \lambda$. For $u \in [\lambda, 1)$, the function is equal to $w_2 \widetilde{d}_\epsilon^+(\mu, u, r(w_2))$, which is increasing and strictly convex on $(0, 1)$. Therefore, the minimum over that interval is attained at $\lambda$. For $u \in (0, \lambda)$, the function is equal to $w_1 \widetilde{d}_\epsilon^-(\lambda, u, r(w_1)) + w_2 \widetilde{d}_\epsilon^+(\mu, u, r(w_2))$. Since it is the sum of two strictly convex function, the minimum over that interval is achieved in $(0, \lambda)$. This concludes the proof of the first part of the result for this case.

Suppose that $\mu \in (0, 1)$ and $\lambda \geq 1$. Then, we have $[\mu]_0^1 = \mu$ and $[\lambda]_0^1 = 1$. For $u \in (0, \mu]$, the function is equal to $w_1 \widetilde{d}_\epsilon^-(\lambda, u, r(w_2))$, which is decreasing and strictly convex on $(0, 1)$. Therefore, the minimum over that interval is attained at $\mu$. For $u \in (\mu, 1)$, the function is equal to $w_1 \widetilde{d}_\epsilon^-(\lambda, u, r(w_1)) + w_2 \widetilde{d}_\epsilon^+(\mu, u, r(w_2))$. Since it is the sum of two strictly convex function, the minimum over that interval is achieved in $(\mu, 1)$. This concludes the proof of the first part of the result for this case.

Suppose that $(\mu, \lambda) \in (0, 1)^2$. Then, we have $[\mu]_0^1 = \mu$ and $[\lambda]_0^1 = \lambda$. For $u \in [\lambda, 1)$, the function is equal to $w_2 \widetilde{d}_\epsilon^+(\mu, u, r(w_2))$, which is increasing and strictly convex on $(0, 1)$. Therefore, the minimum over that interval is attained at $\lambda$. For $u \in (0, \mu]$, the function is equal to $w_1 \widetilde{d}_\epsilon^-(\lambda, u, r(w_2))$, which is decreasing and strictly convex on $(0, 1)$. Therefore, the minimum over that interval is attained at $\mu$. For $u \in (\mu, \lambda)$, the function is equal to $w_1 \widetilde{d}_\epsilon^-(\lambda, u, r(w_1)) + w_2 \widetilde{d}_\epsilon^+(\mu, u, r(w_2))$. Since it is the sum of two strictly convex function, the minimum over that interval is achieved in $(\mu, \lambda)$. This concludes the proof of the first part of the result for this case.

In summary, we have shown that

$$\inf_{u \in (0,1)} \{w_1 \widetilde{d}_\epsilon^-(\lambda, u, r(w_1)) + w_2 \widetilde{d}_\epsilon^+(\mu, u, r(w_2))\}$$
$$= \inf_{u \in ([\mu]_0^1, [\lambda]_0^1)} \{w_1 \widetilde{d}_\epsilon^-(\lambda, u, r(w_1)) + w_2 \widetilde{d}_\epsilon^+(\mu, u, r(w_2))\}.$$

Using the strict convexity of $u_1 \mapsto w_1 \widetilde{d}_\epsilon^-(\lambda, u_1, r(w_1))$ and $u_2 \mapsto w_2 \widetilde{d}_\epsilon^+(\mu, u_2, r(w_2))$ on $([\mu]_0^1, [\lambda]_0^1)$, we obtain that

$$\inf_{u \in ([\mu]_0^1, [\lambda]_0^1)} \{w_1 \widetilde{d}_\epsilon^-(\lambda, u, r(w_1)) + w_2 \widetilde{d}_\epsilon^+(\mu, u, r(w_2))\}$$
$$= \inf_{(u_1, u_2) \,:\, [\mu]_0^1 < u_1 \leq u_2 < [\lambda]_0^1} \{w_1 \widetilde{d}_\epsilon^-(\lambda, u_1, r(w_1)) + w_2 \widetilde{d}_\epsilon^+(\mu, u_2, r(w_2))\}.$$

Re-using the same arguments as above, we obtain that

$$\inf_{(u_1, u_2) \,:\, [\mu]_0^1 < u_1 \leq u_2 < [\lambda]_0^1} \{w_1 \widetilde{d}_\epsilon^-(\lambda, u_1, r(w_1)) + w_2 \widetilde{d}_\epsilon^+(\mu, u_2, r(w_2))\} =$$
$$= \inf_{(u_1, u_2) \in (0,1)^2 \,:\, u_1 \leq u_2} \{w_1 \widetilde{d}_\epsilon^-(\lambda, u_1, r(w_1)) + w_2 \widetilde{d}_\epsilon^+(\mu, u_2, r(w_2))\}.$$

This concludes the proof. □

Lemma 41 gives a closed-form solution for the modified transportation costs based on an implicit solution of a fixed-point equation. This is a key property used in our implementation to reduce the computational cost.

**Lemma 41.** *Let $\widetilde{d}_\epsilon^\pm$ as in Eq. (32), $x_\epsilon^\pm$ as in Lemmas 31 and 32, and $r$ as in Eq. (33). For all $(\lambda, \mu) \in \mathbb{R}^2$ such that $[\lambda]_0^1 > [\mu]_0^1$ and $w \in [1, +\infty)^2$. Then,*

$$\inf_{u \in (0,1)} \{w_1 \widetilde{d}_\epsilon^-(\lambda, u, r(w_1)) + w_2 \widetilde{d}_\epsilon^+(\mu, u, r(w_2))\}$$
$$= w_1 \widetilde{d}_\epsilon^-(\lambda, u^\star(\lambda, \mu, w), r(w_1)) + w_2 \widetilde{d}_\epsilon^+(\mu, u^\star(\lambda, \mu, w), r(w_2)),$$

*where $u^\star(\lambda, \mu, w) \in ([\mu]_0^1, [\lambda]_0^1)$ is the unique solution for $u \in ([\mu]_0^1, [\lambda]_0^1)$ of the equation*

$$u(w_1 + w_2) - w_1 g_\epsilon^-(u) - w_2 g_\epsilon^+(u) + w_1 x_\epsilon^-(\lambda, u, r(w_1)) - w_2 x_\epsilon^+(\mu, u, r(w_2)) = 0 \,.$$

*Proof.* Using Lemma 40, we have

$$\inf_{u \in (0,1)} \{w_1 \widetilde{d}_\epsilon^-(\lambda, u, r(w_1)) + w_2 \widetilde{d}_\epsilon^+(\mu, u, r(w_2))\}$$

$$= \inf_{u \in ([\mu]_0^1, [\lambda]_0^1)} \{w_1 \widetilde{d}_\epsilon^-(\lambda, u, r(w_1)) + w_2 \widetilde{d}_\epsilon^+(\mu, u, r(w_2))\} \,.$$

Using Lemmas 32 and 31, we obtain

$$\forall u \in (0, [\lambda]_0^1), \quad \frac{\partial \widetilde{d}_\epsilon^-}{\partial u}(\lambda, u, r(w_1)) = \frac{u - g_\epsilon^-(u) + x_\epsilon^-(\lambda, u, r(w_1))}{u(1-u)} \,,$$

$$\forall u \in ([\mu]_0^1, 1), \quad \frac{\partial \widetilde{d}_\epsilon^+}{\partial u}(\mu, u, r(w_2)) = \frac{u - g_\epsilon^+(u) - x_\epsilon^+(\mu, u, r(w_2))}{u(1-u)} \,.$$

Therefore, for all $u \in ([\mu]_0^1, [\lambda]_0^1)$,

$$w_1 \frac{\partial \widetilde{d}_\epsilon^-}{\partial u}(\lambda, u, r(w_1)) + w_2 \frac{\partial \widetilde{d}_\epsilon^+}{\partial u}(\mu, u, r(w_2))$$

$$= \frac{w_1(u - g_\epsilon^-(u) + x_\epsilon^-(\lambda, u, r(w_1))) + w_2(u - g_\epsilon^+(u) - x_\epsilon^+(\mu, u, r(w_2)))}{u(1-u)}$$

$$= \frac{u(w_1 + w_2) - (w_1 g_\epsilon^-(u) + w_2 g_\epsilon^+(u)) + w_1 x_\epsilon^-(\lambda, u, r(w_1)) - w_2 x_\epsilon^+(\mu, u, r(w_2))}{u(1-u)} \,.$$

For $u \in ([\mu]_0^1, [\lambda]_0^1)$, let us define

$$g_1(u) := u(w_1 + w_2) - w_1 g_\epsilon^-(u) - w_2 g_\epsilon^+(u) + w_1 x_\epsilon^-(\lambda, u, r(w_1)) - w_2 x_\epsilon^+(\mu, u, r(w_2)) \,.$$

Using the proof of Lemmas 32 and 31, we know that

$$\lim_{u \to [\mu]_0^1} \frac{\partial \widetilde{d}_\epsilon^+}{\partial u}(\mu, u, r(w_1)) = 0 \quad \text{and} \quad \lim_{u \to [\lambda]_0^1} \frac{\partial \widetilde{d}_\epsilon^-}{\partial u}(\lambda, u, r(w_1)) = 0 \,,$$

$$\forall u \in (0, [\lambda]_0^1), \quad \frac{\partial \widetilde{d}_\epsilon^-}{\partial u}(\lambda, u, r(w_1)) < 0 \quad \text{and} \quad \forall u \in ([\mu]_0^1, 1), \quad \frac{\partial \widetilde{d}_\epsilon^+}{\partial u}(\mu, u, r(w_1)) > 0 \,.$$

Combined with the strict convexity of $\widetilde{d}_\epsilon^\pm$ in their second argument, the equation $g_1(u) = 0$ admits a unique solution on $([\mu]_0^1, [\lambda]_0^1)$. Since $u(1-u) > 0$, we obtain the implicit equation defining $u^\star(\lambda, \mu, w)$ as above. $\qquad \square$

### G.3 Characteristic Time

Let $\nu$ be a Bernoulli instance with means $\mu \in (0,1)^2$ and unique best arm $a^\star \in [K]$, i.e., $\arg\max_{a \in [K]} \mu_a = \{a^\star\}$. For all $\beta \in (0,1)$, we define

$$T_\epsilon^\star(\nu)^{-1} = \sup_{w \in \triangle_K} \min_{a \neq a^\star} W_{\epsilon, a^\star, b}(\mu, w) \quad \text{and} \quad w_\epsilon^\star(\nu) = \arg\max_{w \in \triangle_K} \min_{a \neq a^\star} W_{\epsilon, a^\star, b}(\mu, w) \,, \tag{35}$$

$$T_{\epsilon,\beta}^\star(\nu)^{-1} = \sup_{w \in \triangle_K, w_{a^\star} = \beta} \min_{a \neq a^\star} W_{\epsilon, a^\star, b}(\mu, w) \quad \text{and} \quad w_{\epsilon,\beta}^\star(\nu) = \arg\max_{w \in \triangle_K, w_{a^\star} = \beta} \min_{a \neq a^\star} W_{\epsilon, a^\star, b}(\mu, w)$$

where $W_{\epsilon, a, b}$ are defined in Eq. (4).

Lemma 42 gathers regularity properties on the characteristic times and their optimal allocations.

**Lemma 42.** *Let $W_{\epsilon, a, b}$ as in Eq. (4). Let $(T_\epsilon^\star, T_{\epsilon,\beta}^\star)$ and $(w_\epsilon^\star, w_{\epsilon,\beta}^\star)$ as in Eq. (35). The function $(\mu, w) \mapsto \min_{a \neq a^\star(\mu)} W_{\epsilon, a^\star(\mu), a}(\mu, w)$ is continuous on $(0,1)^K \times \triangle_K$. The functions $\nu \mapsto T_\epsilon^\star(\nu)^{-1}$ and $\nu \mapsto T_{\epsilon,\beta}^\star(\nu)^{-1}$ are continuous on $\mathcal{F}^K$. The correspondences $\nu \mapsto w_\epsilon^\star(\nu)$ and $\nu \mapsto w_{\epsilon,\beta}^\star(\nu)$ are upper hemicontinuous on $\mathcal{F}^K$ with compact convex values.*

*Proof.* Let $\mathcal{F}_a^K = \{\boldsymbol{\nu} \in \mathcal{F}^K \mid a \in a^\star(\boldsymbol{\nu})\}$. Since $\bigcup_{a \in [K]} \mathcal{F}_a^K = \mathcal{F}^K$, it is enough to show the property for all $\mathcal{F}_a^K$ for $a \in [K]$. Let $a^\star \in [K]$.

First, the function $(w, \boldsymbol{\nu}) \mapsto \min_{a \neq a^\star} \inf_{u \in [0,1]} \{w_{a^\star} d_\epsilon^-(\mu_{a^\star}, u) + w_a d_\epsilon^+(\mu_{a^\star}, u)\}$ is continuous on $\triangle_K \times \mathcal{F}^K$ by Lemma 35 and the fact that a minimum of continuous functions is continuous. It is concave in $w$ by Lemma 37.

The correspondence $(w, \boldsymbol{\nu}) \mapsto \triangle_K$ is nonempty compact-valued and continuous (since constant). By Berge's maximum theorem, we get that $\boldsymbol{\nu} \mapsto T_\epsilon^\star(\boldsymbol{\nu})^{-1}$ is continuous on $\mathcal{F}_{a^\star}^K$ and that $\boldsymbol{\nu} \mapsto w_\epsilon^\star(\boldsymbol{\nu})$ is upper hemicontinuous with compact values. By [Sundaram, 1996, Theorem 9.17], the concavity of the function being maximized implies that $\boldsymbol{\nu} \mapsto w_\epsilon^\star(\boldsymbol{\nu})$ is convex-valued.

The correspondence $(w, \boldsymbol{\nu}) \mapsto \triangle_K \cap \{w_{a^\star} = \beta\}$ is nonempty compact-valued and continuous (since constant). By Berge's maximum theorem, we get that $\boldsymbol{\nu} \mapsto T_{\epsilon,\beta}^\star(\boldsymbol{\nu})^{-1}$ is continuous on $\mathcal{F}_{a^\star}^K$ and that $w_\beta^\star(\boldsymbol{\nu})$ is upper hemicontinuous with compact values. By [Sundaram, 1996, Theorem 9.17], the concavity of the function being maximized implies that $\boldsymbol{\nu} \mapsto w_{\epsilon,\beta}^\star(\boldsymbol{\nu})$ is convex-valued. $\qquad\square$

Lemma 43 provides additional properties on the characteristic times and their optimal allocations. In particular, this results show that the ($\beta$-)optimal allocations is unique, has positive allocation for each arm and that the transportation costs are equal at equilibrium. Those properties are key in the analysis of a sampling rule.

**Lemma 43.** *Let $W_{\epsilon,a,b}$ as in Eq. (4). Let $(T_\epsilon^\star, T_{\epsilon,\beta}^\star)$ and $(w_\epsilon^\star, w_{\epsilon,\beta}^\star)$ as in Eq. (35). Let $\beta \in (0,1)$ and $\boldsymbol{\nu} \in \mathcal{F}^K$ such that $a^\star(\boldsymbol{\nu}) = \{a^\star\}$ is a singleton.*

- $T_\epsilon^\star(\boldsymbol{\nu})^{-1} > 0$ *and* $T_{\epsilon,\beta}^\star(\boldsymbol{\nu})^{-1} > 0$.

- $\min_{a \in [K]} w_a^\star > 0$ *and* $\min_{a \in [K]} w_{\beta,a}^\star > 0$ *for all* $w^\star \in w_\epsilon^\star(\boldsymbol{\nu})$ *and* $w_\beta^\star \in w_{\epsilon,\beta}^\star(\boldsymbol{\nu})$.

- *the ($\beta$-)optimal allocations are unique and the transportation costs are all equals at equilibrium*

$$w_\epsilon^\star(\boldsymbol{\nu}) = \{w_\epsilon^\star\} \quad and \quad \forall a \neq a^\star, \ \inf_{u \in [0,1]} \left\{ w_{\epsilon,a^\star}^\star d_\epsilon^-(\mu_{a^\star}, u) + w_{\epsilon,a}^\star d_\epsilon^+(\mu_a, u) \right\} = T_\epsilon^\star(\boldsymbol{\nu})^{-1} \, ,$$

$$w_{\epsilon,\beta}^\star(\boldsymbol{\nu}) = \{w_{\epsilon,\beta}^\star\} \quad and \quad \forall a \neq a^\star, \ \inf_{u \in [0,1]} \left\{ w_{\epsilon,\beta,a^\star}^\star d_\epsilon^-(\mu_{a^\star}, u) + w_{\epsilon,\beta,a}^\star d_\epsilon^+(\mu_a, u) \right\} = T_{\epsilon,\beta}^\star(\boldsymbol{\nu})^{-1}$$

*Proof.* Using the definition of the supremum with $1_K/K \in \triangle_K$ and Lemma 35, we obtain

$$T_\epsilon^\star(\boldsymbol{\nu})^{-1} = \sup_{w \in \triangle_K} \min_{a \neq a^\star} \inf_{u \in [0,1]} \left\{ w_{a^\star} d_\epsilon^-(\mu_{a^\star}, u) + w_a d_\epsilon^+(\mu_a, u) \right\}$$

$$\geq \frac{1}{K} \min_{a \neq a^\star} \inf_{u \in [0,1]} \left\{ d_\epsilon^-(\mu_{a^\star}, u) + d_\epsilon^+(\mu_a, u) \right\} > 0 \, ,$$

where the last inequality strict uses Lemma 35 and $\mu_a < \mu_{a^\star}$ for all $a \neq a^\star$. Similarly, we can prove that $T_{\epsilon,\beta}^\star(\boldsymbol{\nu})^{-1} > 0$. This concludes the first part of the proof.

We proceed towards contradiction. Suppose that there exists $w^\star \in w_\epsilon^\star(\boldsymbol{\nu})$ and $b$ with $w_b^\star = 0$. Then, we will show $T_\epsilon^\star(\boldsymbol{\nu})^{-1} = 0$, which is a contradiction with the above result. If $b = a^\star$ we have

$$T_\epsilon^\star(\boldsymbol{\nu})^{-1} = \min_{a \neq a^\star} \inf_{u \in [0,1]} w_a^\star d_\epsilon^+(\mu_a, u) \leq \min_{a \neq a^\star} w_a^\star d_\epsilon^+(\mu_a, \mu_a) = 0 \, .$$

If $b \neq a^\star$, we have

$$T_\epsilon^\star(\boldsymbol{\nu})^{-1} = \min_{a \neq a^\star} \inf_{u \in [0,1]} \left\{ w_{a^\star}^\star d_\epsilon^-(\mu_{a^\star}, u) + w_a^\star d_\epsilon^+(\mu_a, u) \right\}$$

$$\leq \inf_{u \in [0,1]} \left\{ w_{a^\star}^\star d_\epsilon^-(\mu_{a^\star}, u) + w_b^\star d_\epsilon^+(\mu_b, u) \right\} = \inf_{u \in [0,1]} w_{a^\star}^\star d_\epsilon^-(\mu_{a^\star}, u) = 0 \, .$$

A similar proof allows to show the result for $w_{\epsilon,\beta}^\star(\boldsymbol{\nu})$ by reasoning on $T_{\epsilon,\beta}^\star(\boldsymbol{\nu})^{-1}$. This concludes the second part of the proof.

For notational simplicity, we assume without loss of generality that $a^\star = 1$ is the best arm. At the optimal allocations, all $w_a$ are positive. Let us define $G_b(x) = \inf_{u \in [0,1]} \{d_\epsilon^-(\mu_1, u) + x d_\epsilon^+(\mu_b, u)\}$ for all $b \neq 1$. Let $w^\star \in w_\epsilon^\star(\boldsymbol{\nu})$. Then, we have

$$T_\epsilon^\star(\boldsymbol{\nu})^{-1} = \max_{w \in \triangle_K, w_1 > 0} w_1 \min_{b \neq 1} G_b\left(\frac{w_b}{w_1}\right) \quad and \quad w^\star \in \arg\max_{w \in \triangle_K} w_1 \min_{b \neq 1} G_b\left(\frac{w_b}{w_1}\right) \, .$$

Introducing $x_b^\star = \frac{w_b^\star}{w_1^\star}$ for all $b \neq 1$, using that $\sum_{b \in [K]} w_b^\star = 1$, one has

$$w_1^\star = \frac{1}{1 + \sum_{c \neq 1} x_c^\star} \quad \text{and} \quad \forall b \neq 1, \ w_b^\star = \frac{x_b^\star}{1 + \sum_{c \neq 1} x_c^\star} \ .$$

If $x^\star$ is unique, then so is $w^\star$. Since it is optimal, $\{x_b^\star\}_{b=2}^K \in \mathbb{R}^{K-1}$ belongs to

$$\underset{\{x_b\}_{b=2}^K \in \mathbb{R}^{K-1}}{\arg\max} \ \frac{\min_{b \neq 1} G_b(x_b)}{1 + \sum_{c=2}^K x_c} \ . \tag{36}$$

Let's show that all the $G_b(x_b^\star)$ have to be equal. Let $\mathcal{O} = \{a \in [K] \setminus \{1\} \mid G_a(x_a^\star) = \min_{b \neq 1} G_b(x_b^\star)\}$ and $\mathcal{A} = [K] \setminus (\{1\} \cup \mathcal{O})$. Assume that $\mathcal{A} \neq \emptyset$. For all $a \in \mathcal{A}$ and $b \in \mathcal{O}$, one has $G_b(x_b^\star) > G_a(x_a^\star)$. Using the continuity of the $G_b$ functions and the fact that they are increasing (Lemma 37), there exists $\epsilon > 0$ such that

$$\forall b \in \mathcal{A}, a \in \mathcal{O}, \quad G_b(x_b^\star - \epsilon/|\mathcal{A}|) > G_a(x_a^\star + \epsilon/|\mathcal{O}|) > G_a(x_a^\star) \ .$$

We introduce $\bar{x}_b = x_b^\star - \epsilon/|\mathcal{A}|$ for all $b \in \mathcal{A}$ and $\bar{x}_a = x_a^\star + \epsilon/|\mathcal{O}|$ for all $a \in \mathcal{O}$, hence $\sum_{b=2}^K \bar{x}_b = \sum_{b=2}^K x_b^\star$. There exists $a \in \mathcal{O}$ such that $\min_{b \neq 1} G_b(\bar{x}_b) = G_a(x_a^\star + \epsilon/|\mathcal{O}|)$, hence

$$\frac{\min_{b \neq 1} G_b(\bar{x}_b)}{1 + \bar{x}_2 + \ldots \bar{x}_K} = \frac{G_a(x_a^\star + \epsilon/|\mathcal{O}|)}{1 + x_2^\star + \cdots + x_K^\star} > \frac{G_a(x_a^\star)}{1 + x_2^\star + \cdots + x_K^\star} = \frac{\min_{b \neq 1} G_b(x_b^\star)}{1 + x_2^\star + \cdots + x_K^\star} \ .$$

This is a contradiction with the fact that $x^\star$ belongs to (36). Therefore, we have $\mathcal{A} = \emptyset$.

We have proved that there is a unique value by $y^\star \in \mathbb{R}_+$, such that for all $b \neq 1$, $G_b(x_b^\star) = y^\star$. Now since $G_b$ is increasing, this defines a unique value for $x_b^\star$, equal to $G_b^{-1}(y^\star)$.

For $y$ in the intersection of the ranges of all $G_b$, let $x_b(y) = G_b^{-1}(y)$. Then, $y^\star$ belongs to

$$\underset{y \in [0, \min_{b \neq 1} \lim_{+\infty} G_b(x))}{\arg\max} \ \frac{y}{1 + \sum_{b \neq 1} x_b(y)} \ . \tag{37}$$

For $\beta \in (0,1)$, the same results (and proof) hold for $w_{\epsilon,\beta}^\star(\boldsymbol{\nu})$ by noting that

$$T_{\epsilon,\beta}^\star(\boldsymbol{\nu})^{-1} = \max_{w \in \triangle_K : w_1 = \beta} \beta \min_{b \neq 1} G_b(w_b/\beta) \ .$$

Let $w_{\epsilon,\beta}^\star \in w_{\epsilon,\beta}^\star(\boldsymbol{\nu})$, since we have equality at the equilibrium, we obtain $\beta G_b\left(w_{\epsilon,\beta,b}^\star/\beta\right) = T_{\epsilon,\beta}^\star(\boldsymbol{\nu})^{-1}$ for all $b \neq 1$. Using the inverse mapping $x_b$, we obtain $w_{\epsilon,\beta,b}^\star = \beta x_b\left(T_{\epsilon,\beta}^\star(\boldsymbol{\nu})^{-1}/\beta\right)$ for all $b \neq 1$. This concludes the third part of the proof. $\qquad \square$

Lemma 44 shows that an asymptotically $1/2$-optimal algorithm has an asymptotic expected sample complexity which is at worse twice the asymptotic expected sample complexity of an asymptotically optimal algorithm. This result motivates the recommendation to the practitioner of using $\beta = 1/2$ when no prior information is available on the true instance $\boldsymbol{\nu}$.

**Lemma 44.** *Let $(T_\epsilon^\star, T_{\epsilon,\beta}^\star, w_\epsilon^\star)$ as in Eq. (35). Let $\beta \in (0,1)$ and $\boldsymbol{\nu} \in \mathcal{F}^K$ such that $a^\star(\boldsymbol{\nu}) = \{a^\star\}$ is a singleton. Then,*

$$T_{\epsilon,1/2}^\star(\boldsymbol{\nu}) \leq 2T_\epsilon^\star(\boldsymbol{\nu}) \quad \text{and} \quad \frac{T_\epsilon^\star(\boldsymbol{\nu})^{-1}}{T_{\epsilon,\beta}^\star(\boldsymbol{\nu})^{-1}} \leq \max\left\{\frac{\beta^\star}{\beta}, \frac{1-\beta^\star}{1-\beta}\right\} \quad \text{with} \quad \beta^\star = w_{\epsilon,a^\star}^\star \ .$$

*Proof.* Define for each non-negative vector $\boldsymbol{\psi} \in \mathbb{R}_+^K$,

$$f(\boldsymbol{\psi}) := \min_{a \neq a^\star} \inf_{u \in [0,1]} \left\{\psi_{a^\star} d_\epsilon^-(\mu_{a^\star}, u) + \psi_a d_\epsilon^+(\mu_a, u)\right\} \ .$$

$T_\epsilon^\star(\boldsymbol{\nu})^{-1}$ is the maximum of $f(\boldsymbol{\psi})$ over probability vectors $\boldsymbol{\psi} \in \triangle_K$. Here, we instead define $f$ for all non-negative vectors, and proceed by varying the total budget of measurement effort available

$\sum_{a \in [K]} \psi_a$. Using Lemma 37, $f$ is non-decreasing in $\psi_a$ for all $a$. $f$ is homogeneous of degree 1. That is $f(c\psi) = cf(\psi)$ for all $c \geq 1$. For each $c_1, c_2 > 0$ define

$$g(c_1, c_2) = \max \left\{ f(\psi) \mid \psi \in \mathbb{R}_+^K, \; \psi_{a^\star(\nu)} = c_1, \; \sum_{a \neq a^\star(\nu)} \psi_a \leq c_2, \right\}.$$

The function $g$ inherits key properties of $f$; it is also non-decreasing and homogeneous of degree 1. We have

$$T_{\epsilon,\beta}^\star(\nu)^{-1} = \max \left\{ f(\psi) \mid \psi \in \mathbb{R}_+^K, \; \psi_{a^\star} = \beta, \; \sum_{a \in [K]} \psi_a = 1 \right\}$$

$$= \max \left\{ f(\psi) \mid \psi \in \mathbb{R}_+^K, \; \psi_{a^\star} = \beta, \; \sum_{a \neq a^\star} \psi_a \leq 1 - \beta \right\} = g(\beta, 1 - \beta),$$

where the second equality uses that $f$ is non-decreasing. Similarly, $T_\epsilon^\star(\nu)^{-1} = g(\beta^\star, 1 - \beta^\star)$ where $\beta^\star = w_{\epsilon,a^\star}^\star$. Setting $r := \max \left\{ \frac{\beta^\star}{\beta}, \frac{1 - \beta^\star}{1 - \beta} \right\}$ implies $r\beta \geq \beta^\star$ and $r(1 - \beta) \geq 1 - \beta^\star$. Therefore

$$r T_{\epsilon,\beta}^\star(\nu)^{-1} = rg(\beta, 1 - \beta) = g(r\beta, r(1 - \beta)) \geq g(\beta^\star, 1 - \beta^\star) = T_\epsilon^\star(\nu)^{-1}.$$

Taking $\beta = \frac{1}{2}$, yields that $T_\epsilon^\star(\nu)^{-1} \leq 2 \max\{\beta^\star, 1 - \beta^\star\} T_{1/2}^\star(\nu)^{-1} \leq 2 T_{1/2}^\star(\nu)^{-1}$. $\qquad \square$

Lemma 45 gives sufficient conditions on the means and allocations in order for the transportation costs to be equals to the non-private transportation costs. Moreover, it gives sufficient conditions on the means in order for this equality to hold irrespective of the considered allocation. Taken together, this result allows to have fine and coarse understanding of the separation between the high privacy regime and the low privacy regime for $\epsilon$-global DP BAI.

**Lemma 45.** *Let $W_{\epsilon,a,b}$ as in Eq. (4). Let $\mu \in (0,1)^K$ such that $a^\star = \arg\max_{a \in [K]} \mu_a$ is unique. Let $w \in (\mathbb{R}_+^\star)^K$. Let $\epsilon > 0$. For all $x \in (0,1)$, we define $f_\epsilon(x) := (1 - x)\left(1 - \frac{1}{1 + x(e^\epsilon - 1)}\right) = (1 - x)g_\epsilon^-(x)(1 - e^{-\epsilon})$. Let us define $\mu_{a^\star,a}^w := \frac{w_{a^\star}\mu_{a^\star} + w_a \mu_a}{w_{a^\star} + w_a}$ for all $a \neq a^\star$. For all $a \neq a^\star$, we have*

$$\mu_{a^\star} - \mu_a \leq \min \left\{ \left(1 + \frac{w_{a^\star}}{w_a}\right) f_\epsilon(1 - \mu_{a^\star}), \left(1 + \frac{w_a}{w_{a^\star}}\right) f_\epsilon(\mu_a) \right\}$$

$$\implies W_{\epsilon,a^\star,a}(\mu, w) = w_{a^\star} \mathrm{kl}(\mu_{a^\star}, \mu_{a^\star,a}^w) + w_a \mathrm{kl}(\mu_a, \mu_{a^\star,a}^w).$$

*Moreover, we have*

$$\max_{a^\star \in [K], \, \mu \in (0,1)^K, \, a^\star(\mu) = \{a^\star\}, \, w \in (\mathbb{R}_+^\star)^K} \min \left\{ \left(1 + \frac{w_{a^\star}}{w_a}\right) f_\epsilon(1 - \mu_{a^\star}), \left(1 + \frac{w_a}{w_{a^\star}}\right) f_\epsilon(\mu_a) \right\} \leq \epsilon/2$$

*and, for all $a \neq a^\star$, we have*

$$\epsilon \geq \log \left( \frac{\mu_{a^\star}(1 - \mu_a)}{\mu_a(1 - \mu_{a^\star})} \right) = \frac{\partial \mathrm{kl}}{\partial x_1}(\mu_{a^\star}, \mu_a) = \frac{\partial \mathrm{kl}}{\partial x_1}(\mu_a, \mu_{a^\star})$$

$$\implies \forall w \in (\mathbb{R}_+^\star)^K, \quad W_{\epsilon,a^\star,a}(\mu, w) = w_{a^\star} \mathrm{kl}(\mu_{a^\star}, \mu_{a^\star,a}^w) + w_a \mathrm{kl}(\mu_a, \mu_{a^\star,a}^w).$$

*Proof.* Let us define $f_\epsilon(x) = (1 - x)\left(1 - \frac{1}{1 + x(e^\epsilon - 1)}\right)$ for all $x \in (0,1)$. Then, we have

$$\frac{\mu_a(1 - \mu_a)(e^\epsilon - 1)}{1 + \mu_a(e^\epsilon - 1)} = (1 - \mu_a)\left(1 - \frac{1}{1 + \mu_a(e^\epsilon - 1)}\right) = f_\epsilon(\mu_a),$$

$$\frac{\mu_{a^\star}(1 - \mu_{a^\star})(e^\epsilon - 1)}{e^\epsilon - \mu_{a^\star}(e^\epsilon - 1)} = \mu_{a^\star}\left(1 - \frac{1}{1 + (1 - \mu_{a^\star})(e^\epsilon - 1)}\right) = f_\epsilon(1 - \mu_{a^\star}).$$

Using Lemma 24, direct manipulation yields that

$$f_\epsilon(1 - \mu_{a^\star}) < \mu_{a^\star} - \mu_a \iff g_\epsilon^+(\mu_{a^\star}) > \mu_a \iff f_\epsilon(\mu_a) < \mu_{a^\star} - \mu_a$$

$$g_\epsilon^-(\mu_a) < \mu_{a^\star,a}^w \quad \Longleftrightarrow \quad f_\epsilon(\mu_a) < \frac{w_{a^\star}}{w_{a^\star} + w_a}(\mu_{a^\star} - \mu_a),$$

$$g_\epsilon^+(\mu_{a^\star}) > \mu_{a^\star,a}^w \quad \Longleftrightarrow \quad f_\epsilon(1 - \mu_{a^\star}) < \frac{w_a}{w_{a^\star} + w_a}(\mu_{a^\star} - \mu_a).$$

Using that $\max\left\{\frac{w_a}{w_{a^\star} + w_a}, \frac{w_{a^\star}}{w_{a^\star} + w_a}\right\} \le 1$, we obtain that

$$\left(g_\epsilon^-(\mu_a) < \mu_{a^\star} \wedge g_\epsilon^-(\mu_a) < \mu_{a^\star,a}^w\right) \quad \Longleftrightarrow \quad \left(1 + \frac{w_a}{w_{a^\star}}\right) f_\epsilon(\mu_a) < \mu_{a^\star} - \mu_a,$$

$$\left(g_\epsilon^-(\mu_a) < \mu_{a^\star} \wedge g_\epsilon^+(\mu_{a^\star}) > \mu_{a^\star,a}^w\right) \quad \Longleftrightarrow \quad \left(1 + \frac{w_{a^\star}}{w_a}\right) f_\epsilon(1 - \mu_{a^\star}) < \mu_{a^\star} - \mu_a,$$

$$\left(g_\epsilon^-(\mu_a) \ge \mu_{a^\star} \vee \left(g_\epsilon^-(\mu_a) < \mu_{a^\star} \wedge \mu_{a^\star,a}^w \in [g_\epsilon^+(\mu_{a^\star}), g_\epsilon^-(\mu_a)]\right)\right)$$

$$\Longleftrightarrow \quad \left(g_\epsilon^-(\mu_a) \ge \mu_{a^\star} \vee \mu_{a^\star,a}^w \in [g_\epsilon^+(\mu_{a^\star}), g_\epsilon^-(\mu_a)]\right)$$

$$\Longleftrightarrow \quad (\min\{f_\epsilon(\mu_a), f_\epsilon(1 - \mu_{a^\star})\} \ge \mu_{a^\star} - \mu_a$$

$$\vee \mu_{a^\star} - \mu_a \le \min\left\{\left(1 + \frac{w_{a^\star}}{w_a}\right) f_\epsilon(1 - \mu_{a^\star}), \left(1 + \frac{w_a}{w_{a^\star}}\right) f_\epsilon(\mu_a)\right\}\right)$$

$$\Longleftrightarrow \quad \mu_{a^\star} - \mu_a \le \max\left\{\min\{f_\epsilon(\mu_a), f_\epsilon(1 - \mu_{a^\star})\},\right.$$

$$\left.\min\left\{\left(1 + \frac{w_{a^\star}}{w_a}\right) f_\epsilon(1 - \mu_{a^\star}), \left(1 + \frac{w_a}{w_{a^\star}}\right) f_\epsilon(\mu_a)\right\}\right\}$$

$$\Longleftrightarrow \quad \mu_{a^\star} - \mu_a \le \min\left\{\left(1 + \frac{w_{a^\star}}{w_a}\right) f_\epsilon(1 - \mu_{a^\star}), \left(1 + \frac{w_a}{w_{a^\star}}\right) f_\epsilon(\mu_a)\right\}.$$

Combining those conditions with Lemma 38 concludes the first part of the proof.

For all $x \in (0, 1)$, we have

$$f_\epsilon'(x) = \frac{(1-x)(e^\epsilon - 1) - x(e^\epsilon - 1)(1 + x(e^\epsilon - 1))}{(1 + x(e^\epsilon - 1))^2} = -(e^\epsilon - 1)\frac{x^2(e^\epsilon - 1) + 2x - 1}{(1 + x(e^\epsilon - 1))^2},$$

$$f_\epsilon'(x) = 0 \quad \Longleftrightarrow \quad x = \frac{e^{\epsilon/2} - 1}{e^\epsilon - 1},$$

$$f_\epsilon''(x) = -2(e^\epsilon - 1)\frac{(1 + x(e^\epsilon - 1))^2 - (e^\epsilon - 1)\left(x^2(e^\epsilon - 1) + 2x - 1\right)}{(1 + x(e^\epsilon - 1))^3}$$

$$= -\frac{2e^\epsilon(e^\epsilon - 1)}{(1 + x(e^\epsilon - 1))^3} \le 0.$$

As $f_\epsilon$ is strictly concave, the maximum is achieved at $\frac{e^{\epsilon/2} - 1}{e^\epsilon - 1}$ with value

$$\max_{x \in (0,1)} f_\epsilon(x) = f_\epsilon\left(\frac{e^{\epsilon/2} - 1}{e^\epsilon - 1}\right) = \frac{e^\epsilon - e^{\epsilon/2}}{e^\epsilon - 1}\left(1 - e^{-\epsilon/2}\right) = \frac{(e^{\epsilon/2} - 1)^2}{e^\epsilon - 1}.$$

Let $\kappa_1(x) = x(e^x - 1) - 4(e^{x/2} - 1)^2$ for all $x > 0$. Then, we have

$$\frac{(e^{\epsilon/2} - 1)^2}{e^\epsilon - 1} \le \epsilon/4 \quad \Longleftrightarrow \quad \kappa_1(\epsilon) \ge 0.$$

Then, we have $\kappa_1(0) = 0$ and

$$\kappa_1'(x) = 4e^{x/2} - 3e^x - 1 + xe^x \quad \text{and} \quad \kappa_1''(x) = e^x\left(2(e^{-x/2} - 1) + x\right).$$

Using that $e^{-x/2} - 1 \ge -x/2$, we obtain $\kappa_1''(x) \ge 0$. Using that $\kappa_1'(0) = 0$, we obtain $\kappa_1'(x) \ge 0$. Using that $\kappa_1(0) = 0$, we obtain $\kappa_1(x) \ge 0$. Therefore, we have shown that

$$\forall \epsilon > 0, \quad \max_{x \in (0,1)} f_\epsilon(x) \le \epsilon/4.$$

Direct manipulation yields that

$$\min\left\{\left(1 + \frac{w_{a^\star}}{w_a}\right) f_\epsilon(1 - \mu_{a^\star}), \left(1 + \frac{w_a}{w_{a^\star}}\right) f_\epsilon(\mu_a)\right\}$$

$$\leq \left(1 + \min\left\{\frac{w_{a^\star}}{w_a}, \frac{w_a}{w_{a^\star}}\right\}\right) \max_{x \in (0,1)} f_\epsilon(x) \leq \epsilon/2\,.$$

Taking the supremum over $w \in (\mathbb{R}_+^\star)^K$, $\mu \in (0,1)^K$ such that $a^\star = a^\star(\mu)$ and over $a^\star \in [K]$ concludes the second part of the proof.

Let $a \neq a^\star$. Direct manipulations yield that

$$\mu_{a^\star} - \mu_a \leq \min\{f_\epsilon(1 - \mu_{a^\star}), f_\epsilon(\mu_a)\}$$

$$\implies \quad \forall w \in (\mathbb{R}_+^\star)^K, \quad \mu_{a^\star} - \mu_a \leq \min\left\{\left(1 + \frac{w_{a^\star}}{w_a}\right) f_\epsilon(1 - \mu_{a^\star}), \left(1 + \frac{w_a}{w_{a^\star}}\right) f_\epsilon(\mu_a)\right\}$$

$$\implies \quad \forall w \in (\mathbb{R}_+^\star)^K, \quad W_{\epsilon,a^\star,a}(\mu, w) = w_{a^\star}\mathrm{kl}(\mu_{a^\star}, \mu_{a^\star,a}^w) + w_a\mathrm{kl}(\mu_a, \mu_{a^\star,a}^w)\,.$$

Recall that $f_\epsilon(x) = (1 - x)\left(1 - \frac{1}{1 + x(e^\epsilon - 1)}\right)$. Then, we have directly that

$$f_\epsilon(x) \geq y \quad \Longleftrightarrow \quad \frac{1 - x - y}{1 - x} \geq \frac{1}{1 + x(e^\epsilon - 1)} \quad \Longleftrightarrow \quad \frac{(y + x)(1 - x)}{x(1 - x - y)} \leq e^\epsilon\,.$$

Plugging this result, we obtain

$$\mu_{a^\star} - \mu_a \leq \min\{f_\epsilon(1 - \mu_{a^\star}), f_\epsilon(\mu_a)\} \quad \Longleftrightarrow \quad e^\epsilon \geq \frac{\mu_{a^\star}(1 - \mu_a)}{\mu_a(1 - \mu_{a^\star})}$$

$$\Longleftrightarrow \quad \epsilon \geq \log\left(\frac{\mu_{a^\star}(1 - \mu_a)}{\mu_a(1 - \mu_{a^\star})}\right)\,.$$

Recall that

$$\frac{\partial \mathrm{kl}}{\partial x_1}(\mu_{a^\star}, \mu_a) = \frac{\partial \mathrm{kl}}{\partial x_1}(\mu_a, \mu_{a^\star}) = \log\left(\frac{\mu_{a^\star}(1 - \mu_a)}{\mu_a(1 - \mu_{a^\star})}\right)\,.$$

This concludes the proof of the last part of the result. $\qquad\square$

Lemma 46 shows that our lower bound is larger (hence better) than the one derived in Azize et al. [2024].

**Lemma 46.** *Let $T_g^\star(\boldsymbol{\nu}, \epsilon)$ as in Theorem 13 in Azize et al. [2024], and $T_\epsilon^\star(\boldsymbol{\nu})$ as in Eq. (35). Then, we have $T_g^\star(\boldsymbol{\nu}, \epsilon) \leq T_\epsilon^\star(\boldsymbol{\nu})$.*

*Proof.* Let $T_g^\star(\boldsymbol{\nu}, \epsilon)$ as in Theorem 13 in Azize et al. [2024]. A sufficient condition to obtain $T_g^\star(\boldsymbol{\nu}, \epsilon) \leq T_\epsilon^\star(\boldsymbol{\nu})$ is to show that, for all $\lambda \in \mathrm{Alt}(\mu)$, we have

$$\sum_{a \in [K]} w_a d_\epsilon(\mu_a, \lambda_a) \leq \min\left\{\sum_{a \in [K]} w_a \mathrm{kl}(\mu_a, \lambda_a), 6\epsilon \sum_{a \in [K]} w_a |\mu_a - \lambda_a|\right\}\,,$$

since we can conclude by taking the infimum over $\lambda \in \mathrm{Alt}(\mu)$ and the supremum over $w \in \triangle_K$ on both sides of the inequalities. By definition of $d_\epsilon$ and evaluation the function at $z = \mu$ and $z = \lambda$ respectively, we obtain

$$d_\epsilon(\lambda, \mu) = \inf_{z \in (0,1)} \{\mathrm{kl}(z, \mu) + \epsilon|\lambda - z|\} \leq \min\{\mathrm{kl}(\lambda, \mu), \epsilon|\lambda - \mu|\}\,.$$

By summing those inequalities over arms $a \in [K]$, we obtain

$$\sum_{a \in [K]} w_a d_\epsilon(\mu_a, \lambda_a) \leq \sum_{a \in [K]} w_a \min\{\mathrm{kl}(\mu_a, \lambda_a), \epsilon|\mu_a - \lambda_a|\}$$

$$\leq \min\left\{\sum_{a \in [K]} w_a \mathrm{kl}(\mu_a, \lambda_a), \epsilon \sum_{a \in [K]} w_a |\mu_a - \lambda_a|\right\}\,.$$

Using that $\sum_{a \in [K]} w_a |\mu_a - \lambda_a| \geq 0$ and $6\epsilon \geq \epsilon$, this concludes the proof. $\qquad\square$

In Garivier and Kaufmann [2016], the authors show how to rewrite the optimization problem underlying the characteristic time and its optimal allocation as a simpler optimization problem. Lemma 47 shows that similar properties holds for $\epsilon$-global DP BAI. In particular, it shows that computing the characteristic time $T_\epsilon^\star(\boldsymbol{\nu})$ and their optimal allocation $w_\epsilon^\star(\boldsymbol{\nu})$ can be done explicitly based on solving nested fixed-point equations. This result is key to implement computationally tractable Track-and-Stop algorithms. Additionally, Lemma 47 gives an explicit lower bound on the characteristic time $T_\epsilon^\star(\boldsymbol{\nu})$.

**Lemma 47.** *Let $d_\epsilon^\pm$ as in Eq. (3), and $(T_\epsilon^\star, w_\epsilon^\star)$ as in Eq. (35). Let $a \neq a^\star$. For $x \in [0, +\infty)$, let*

$$G_a(x) := \inf_{u \in [0,1]} \left\{ d_\epsilon^-(\mu_{a^\star}, u) + x d_\epsilon^+(\mu_a, u) \right\} \text{ and } u_a(x) := \operatorname*{arg\,min}_{u \in [\mu_a, \mu_{a^\star}]} \left\{ d_\epsilon^-(\mu_{a^\star}, u) + x d_\epsilon^+(\mu_a, u) \right\}.$$

- *The function $G_a$ is an increasing and strictly concave one-to-one mapping from $[0, +\infty)$ to $[0, d_\epsilon^-(\mu_{a^\star}, \mu_a))$; it satisfies that $G_a(0) = 0$ and $\lim_{x \to +\infty} G_a(x) = d_\epsilon^-(\mu_{a^\star}, \mu_a)$.*

- *The function $u_a$ is a decreasing one-to-one mapping from $[0, +\infty)$ to $(\mu_a, \mu_{a^\star}]$; it satisfies that $u_a(0) = \mu_{a^\star}$ and $\lim_{x \to +\infty} u_a(x) = \mu_a$.*

- *Let $x_a(y)$ be defined as the unique solution of $G_a(x) = y$ for all $y \in [0, d_\epsilon^-(\mu_{a^\star}, \mu_a))$. The function $x_a$ is an increasing and strictly convex one-to-one mapping from $[0, d_\epsilon^-(\mu_{a^\star}, \mu_a))$ to $[0, +\infty)$; it satisfies that $x_a(0) = 0$ and $\lim_{y \to d_\epsilon^-(\mu_{a^\star}, \mu_a)} x_a(y) = +\infty$.*

*For all $y \in [0, \min_{a \neq a^\star} d_\epsilon^-(\mu_{a^\star}, \mu_a))$, let us define*

$$G(y) := \frac{y}{1 + \sum_{a \neq a^\star} x_a(y)} \quad and \quad F(y) := \sum_{a \neq a^\star} \frac{d_\epsilon^-(\mu_{a^\star}, u_a(x_a(y)))}{d_\epsilon^+(\mu_a, u_a(x_a(y)))} .$$

- *The function $F$ is an increasing one-to-one mapping from $[0, \min_{a \neq a^\star} d_\epsilon^-(\mu_{a^\star}, \mu_a))$ to $[0, +\infty)$; it satisfies that $F(0) = 0$ and $\lim_{y \to \min_{a \neq a^\star} d_\epsilon^-(\mu_{a^\star}, \mu_a)} F(y) = +\infty$.*

- *On $[0, \min_{a \neq a^\star} d_\epsilon^-(\mu_{a^\star}, \mu_a))$, the function $G$ is maximized at the unique $y^\star$ solution in $[0, \min_{a \neq a^\star} d_\epsilon^-(\mu_{a^\star}, \mu_a))$ of the fixed-point equation $F(y) = 1$. Moreover, we have $w_\epsilon^\star(\boldsymbol{\nu})_a = w_\epsilon^\star(\boldsymbol{\nu})_{a^\star} x_a(y^\star)$ for all $a \neq a^\star$,*

$$w_\epsilon^\star(\boldsymbol{\nu})_{a^\star} = \frac{1}{1 + \sum_{a \neq a^\star} x_a(y^\star)} \quad and \quad T_\epsilon^\star(\boldsymbol{\nu})^{-1} = \frac{y^\star}{1 + \sum_{a \neq a^\star} x_a(y^\star)} .$$

- *Moreover, we have*

$$T_\epsilon^\star(\boldsymbol{\nu}) \geq \frac{1}{\min_{a \neq a^\star} d_\epsilon^-(\mu_{a^\star}, \mu_a)} + \sum_{a \neq a^\star} \frac{1}{d_\epsilon^+(\mu_a, \mu_{a^\star})} .$$

*If $\epsilon < \log\left(\frac{\mu_a(1 - \mu_b)}{\mu_b(1 - \mu_a)}\right)$, we have $d_\epsilon^+(\mu_a, \mu_{a^\star}) = -\log\left(1 - \mu_{a^\star}(1 - e^{-\epsilon})\right) - \epsilon\mu_a$ and $d_\epsilon^-(\mu_{a^\star}, \mu_a) = -\log\left(1 + \mu_a(e^\epsilon - 1)\right) + \epsilon\mu_{a^\star}$.*

*Proof.* Using Lemma 37, we know that $G_a$ is concave. Let $u_a(x) \in \operatorname*{arg\,min}_{u \in [0,1]} \left\{ d_\epsilon^-(\mu_{a^\star}, u) + x d_\epsilon^+(\mu_a, u) \right\}$ for all $x \in [0, +\infty)$, whose explicit formula is given in Lemma 45. It is direct to see that $G_a(0) = 0$ and $u_a(0) = \mu_{a^\star}$. Using the optimality condition of $u_a(x)$, we obtain, for all $x \in [0, +\infty)$,

$$G_a'(x) = u_a'(x) \left( \frac{\partial d_\epsilon^-}{\partial u}(\mu_{a^\star}, u_a(x)) + x \frac{\partial d_\epsilon^+}{\partial u}(\mu_a, u_a(x)) \right) + d_\epsilon^+(\mu_a, u_a(x))$$

$$= d_\epsilon^+(\mu_a, u_a(x)) > 0 ,$$

where the last inequality is obtained by Lemma 35 and using that $d_\epsilon^+(\mu_a, u_a(0)) = d_\epsilon^+(\mu_a, \mu_{a^\star}) > 0$. Therefore, $G_a$ is an increasing one-to-one mapping from $[0, +\infty)$ to $[0, \lim_{x \to +\infty} G_a(x))$.

Let $\mu_{a^\star, a}^x = \frac{\mu_{a^\star} + x\mu_a}{1 + x}$ for all $x \in [0, +\infty)$. It is easy to see that $G_a(0) = 0$, $u_a(0) = \mu_{a^\star}$ and $\lim_{x \to +\infty} \mu_{a^\star, a}^x = \mu_a$. Using Lemma 45, we obtain that

$$\lim_{x \to +\infty} \min \left\{ (1 + 1/x) f_\epsilon(1 - \mu_{a^\star}), (1 + x) f_\epsilon(\mu_a) \right\} = f_\epsilon(1 - \mu_{a^\star}) .$$

When $\mu_{a^\star} - \mu_a \leq f_\epsilon(1 - \mu_{a^\star})$, we obtain

$$\lim_{x \to +\infty} G_a(x) = \lim_{x \to +\infty} \left\{ \mathrm{kl}(\mu_{a^\star}, \mu_{a^\star,a}^x) + x\mathrm{kl}(\mu_a, \mu_{a^\star,a}^x) \right\}$$

$$= \mathrm{kl}(\mu_{a^\star}, \mu_a) + \lim_{x \to +\infty} \left\{ x\mathrm{kl}(\mu_a, \mu_{a^\star,a}^x) \right\} = \mathrm{kl}(\mu_{a^\star}, \mu_a) ,$$

where we used that

$$x\mathrm{kl}(\mu_a, \mu_{a^\star,a}^x) = x \left( \mu_a \log \left( 1 - \frac{\mu_{a^\star} - \mu_a}{\mu_a(1 - \mu_a)} \frac{1}{\frac{\mu_{a^\star}}{\mu_a} + x} \right) + \log \left( 1 + \frac{\mu_{a^\star} - \mu_a}{1 - \mu_a} \frac{1}{\frac{1 - \mu_{a^\star}}{1 - \mu_a} + x} \right) \right) ,$$

$$\lim_{x \to +\infty} \left\{ x\mathrm{kl}(\mu_a, \mu_{a^\star,a}^x) \right\} = \frac{(\mu_{a^\star} - \mu_a)^2}{\mu_a(1 - \mu_a)^2} \lim_{x \to +\infty} \left\{ \frac{x}{\left( \frac{\mu_{a^\star}}{\mu_a} + x \right) \left( \frac{1 - \mu_{a^\star}}{1 - \mu_a} + x \right)} \right\} = 0 ,$$

where we used that $\log(1 + x) =_{x \to 0} x + \mathcal{O}(x^2)$. Using Lemma 26 and the proof of Lemma 45, we know that $\mu_{a^\star} - \mu_a \leq f_\epsilon(1 - \mu_{a^\star})$ if and only if $\mu_a \in [g_\epsilon^+(\mu_{a^\star}), \mu_{a^\star})$, hence we have $\mathrm{kl}(\mu_{a^\star}, \mu_a) = d_\epsilon^-(\mu_{a^\star}, \mu_a)$. This concludes the proof in the first case.

When $\mu_{a^\star} - \mu_a > f_\epsilon(1 - \mu_{a^\star})$, we obtain

$$\lim_{x \to +\infty} G_a(x) = \lim_{x \to +\infty} \left\{ -\log\left( 1 + u_{1,\star}(\mu_a, x)(e^\epsilon - 1) \right) + \epsilon\mu_{a^\star} + x\mathrm{kl}(\mu_a, u_{1,\star}(\mu_a, x)) \right\} ,$$

where

$$u_{1,\star}(\mu_a, x) =$$
$$\frac{\sqrt{(x(1 - \mu_a(e^\epsilon - 1)) - (e^\epsilon - 1))^2 + 4(1 + x)(e^\epsilon - 1)x\mu_a} - (x(1 - \mu_a(e^\epsilon - 1)) - (e^\epsilon - 1))}{2(1 + x)(e^\epsilon - 1)} .$$

Direct manipulation yields that $\lim_{x \to +\infty} u_{1,\star}(\mu_a, x) = \mu_a$, hence

$$\lim_{x \to +\infty} G_a(x) = -\log\left( 1 + \mu_a(e^\epsilon - 1) \right) + \epsilon\mu_{a^\star} + \lim_{x \to +\infty} \left\{ x\mathrm{kl}(\mu_a, u_{1,\star}(\mu_a, x)) \right\} .$$

Let us denote $v_{1,\star}(\mu_a, x) = u_{1,\star}(\mu_a, x) - \mu_a \geq 0$, i.e., $\lim_{x \to +\infty} v_{1,\star}(\mu_a, x) = 0$. Direct manipulation yields that

$$v_{1,\star}(\mu_a, x)$$
$$= \frac{1 + \mu_a(e^\epsilon - 1)}{2(e^\epsilon - 1)} \left( 1 - \frac{1}{x + 1} \right)$$
$$\left( \sqrt{1 - \frac{2x(1 - \mu_a(e^\epsilon + 1))(e^\epsilon - 1) - (e^\epsilon - 1)^2}{x^2(1 + \mu_a(e^\epsilon - 1))^2}} - 1 + \frac{(e^\epsilon - 1)(1 - 2\mu_a)}{x(1 + \mu_a(e^\epsilon - 1))} \right)$$
$$= \sqrt{1 - \frac{2x(1 - \mu_a(e^\epsilon + 1))(e^\epsilon - 1) - (e^\epsilon - 1)^2}{x^2(1 + \mu_a(e^\epsilon - 1))^2}} - 1 + \frac{(e^\epsilon - 1)(1 - 2\mu_a)}{x(1 + \mu_a(e^\epsilon - 1))}$$
$$=_{x \to +\infty} \frac{(e^\epsilon - 1)(1 - 2\mu_a)}{x(1 + \mu_a(e^\epsilon - 1))} - \frac{2x(1 - \mu_a(e^\epsilon + 1))(e^\epsilon - 1) - (e^\epsilon - 1)^2}{2x^2(1 + \mu_a(e^\epsilon - 1))^2} + \mathcal{O}(1/x^2)$$
$$=_{x \to +\infty} \frac{2x(e^\epsilon - 1)2(1 - \mu_a)\mu_a(e^\epsilon - 1) + (e^\epsilon - 1)^2}{2x^2(1 + \mu_a(e^\epsilon - 1))^2} + \mathcal{O}(1/x^2) ,$$

hence $v_{1,\star}(\mu_a, x) =_{x \to +\infty} \frac{2(1 - \mu_a)\mu_a(e^\epsilon - 1)^2}{x(1 + \mu_a(e^\epsilon - 1))^2} + \mathcal{O}(1/x^2) .$

where we used that $\sqrt{1 - x} - 1 =_{x \to 0} -x/2 + \mathcal{O}(x^2)$ to obtain the last result. Similarly as before, we derive

$$x\mathrm{kl}(\mu_a, u_{1,\star}(\mu_a, x)) = x \left( \mu_a \log \left( 1 - \frac{1}{\mu_a(1 - \mu_a)} \frac{v_{1,\star}(\mu_a, x)}{1 + v_{1,\star}(\mu_a, x)/\mu_a} \right) \right.$$
$$\left. + \log \left( 1 + \frac{1}{1 - \mu_a} \frac{v_{1,\star}(\mu_a, x)}{1 - v_{1,\star}(\mu_a, x)/(1 - \mu_a)} \right) \right)$$

$$\lim_{x \to +\infty} \left\{ x\mathrm{kl}(\mu_a, \mu_{a^\star, a}^x) \right\} = \frac{1}{\mu_a(1 - \mu_a)^2} \lim_{x \to +\infty} \left\{ \frac{xv_{1,\star}(\mu_a, x)^2}{\left(1 - \frac{v_{1,\star}(\mu_a, x)}{1 - \mu_a}\right)\left(1 + \frac{v_{1,\star}(\mu_a, x)}{\mu_a}\right)} \right\}$$

$$= \frac{\lim_{x \to +\infty} xv_{1,\star}(\mu_a, x)^2}{\mu_a(1 - \mu_a)^2} = 0 \,,$$

where we used that $v_{1,\star}(\mu_a, x) =_{x \to +\infty} \mathcal{O}(1/x)$ to conclude. Therefore, we have shown that $\lim_{x \to +\infty} G_a(x) = -\log\left(1 + \mu_a(e^\epsilon - 1)\right) + \epsilon\mu_{a^\star}$. Using Lemma 26 and the proof of Lemma 45, we know that $\mu_{a^\star} - \mu_a > f_\epsilon(1 - \mu_{a^\star})$ if and only if $\mu_a \in [0, g_\epsilon^+(\mu_{a^\star}))$, hence we have $-\log\left(1 + \mu_a(e^\epsilon - 1)\right) + \epsilon\mu_{a^\star} = d_\epsilon^-(\mu_{a^\star}, \mu_a)$. This concludes the proof in the second case.

Therefore, $G_a$ is a strictly increasing one-to-one mapping from $[0, +\infty)$ to $[0, d_\epsilon^-(\mu_{a^\star}, \mu_a))$. Using the implicit function theorem, we obtain

$$\forall x \in [0, +\infty), \quad u_a'(x) = -\frac{\frac{\partial d_\epsilon^+}{\partial u}(\mu_a, u_a(x))}{\frac{\partial^2 d_\epsilon^-}{\partial u^2}(\mu_{a^\star}, u_a(x)) + x\frac{\partial^2 d_\epsilon^+}{\partial u^2}(\mu_a, u_a(x))} < 0 \,,$$

where the strict inequality is obtained by using properties in Lemmas 25 and 26, since $u_a(x) \in (\mu_a, \mu_{a^\star})$ by Lemmas 35 and 25. Similarly, we obtain

$$\forall x \in [0, +\infty), \quad G_a''(x) = u_a'(x)\frac{\partial d_\epsilon^+}{\partial u}(\mu_a, u_a(x)) > 0 \,,$$

Therefore, we have shown that $G_a$ is strictly concave and that $u_a$ is decreasing.

Let us define $x_a(y)$ as the unique solution of $G_a(x) = y$, which is well-defined based on our above computations. Therefore, we have

$$y = d_\epsilon^-\left(\mu_{a^\star}, u_a(x_a(y))\right) + x_a(y)d_\epsilon^+\left(\mu_a, u_a(x_a(y))\right) \,.$$

Using the derivative of the inverse function, we obtain

$$\forall y \in [0, d_\epsilon^-(\mu_{a^\star}, \mu_a)), \quad x_a'(y) = \frac{1}{G_a'(x_a(y))} = \frac{1}{d_\epsilon^+(\mu_a, u_a(x_a(y)))} > 0 \,,$$

hence $x_a$ is increasing on $[0, d_\epsilon^-(\mu_{a^\star}, \mu_a))$. Moreover, we have

$$\forall y \in [0, d_\epsilon^-(\mu_{a^\star}, \mu_a)), \quad x_a''(y) = -\frac{u_a'(x_a(y))}{d_\epsilon^+(\mu_a, u_a(x_a(y)))^3}\frac{\partial d_\epsilon^+}{\partial u}(\mu_a, u_a(x_a(y))) > 0 \,,$$

hence $x_a$ is strictly convex on $[0, d_\epsilon^-(\mu_{a^\star}, \mu_a))$.

For all $y \in [0, \min_{a \neq a^\star} d_\epsilon^-(\mu_{a^\star}, \mu_a))$, let us define $G(y) = \frac{y}{1 + \sum_{a \neq a^\star} x_a(y)}$ and $F(y) = \sum_{a \neq a^\star} \frac{d_\epsilon^-(\mu_{a^\star}, u_a(x_a(y)))}{d_\epsilon^+(\mu_a, u_a(x_a(y)))}$. Using the above results, direct manipulations yield that, for all $y \in [0, \min_{a \neq a^\star} d_\epsilon^-(\mu_{a^\star}, \mu_a))$,

$$G'(y) = \frac{1 + \sum_{a \neq a^\star} x_a(y) - y\sum_{a \neq a^\star} x_a'(y)}{(1 + \sum_{a \neq a^\star} x_a(y))^2} = \frac{1 + \sum_{a \neq a^\star} x_a(y) - \sum_{a \neq a^\star} \frac{y}{d_\epsilon^+(\mu_a, u_a(x_a(y)))}}{(1 + \sum_{a \neq a^\star} x_a(y))^2}$$

$$= \frac{1 - F(y)}{(1 + \sum_{a \neq a^\star} x_a(y))^2} \,,$$

hence we obtain that $G'(y) = 0$ if and only if $F(y) = 1$. Using that $x_a(0) = 0$, $u_a(0) = \mu_{a^\star}$ and $d_\epsilon^-(\mu_{a^\star}, \mu_{a^\star}) = 0$, we obtain that $F(0) = 0$.

Using that $\lim_{y \to d_\epsilon^-(\mu_{a^\star}, \mu_a)} x_a(y) = +\infty$, $\lim_{x \to +\infty} u_a(x) = \mu_a$, $d_\epsilon^-(\mu_{a^\star}, \mu_a) > 0$ and $d_\epsilon^+(\mu_a, \mu_a) = 0$, we obtain that $\lim_{y \to \min_{a \neq a^\star} d_\epsilon^-(\mu_{a^\star}, \mu_a)} F(y) = +\infty$.

Let $H(y) = \sum_{a \neq a^\star} \frac{1}{d_\epsilon^+(\mu_a, u_a(x_a(y)))}$ for all $y \in [0, \min_{a \neq a^\star} d_\epsilon^-(\mu_{a^\star}, \mu_a))$. Then, we have

$$\sum_{a \neq a^\star} x_a(y) = yH(y) - F(y) \quad, \quad \sum_{a \neq a^\star} x_a'(y) = H(y) \,,$$

$$\frac{\partial d_\epsilon^-}{\partial u}(\mu_{a^\star}, u_a(x)) + x\frac{\partial d_\epsilon^+}{\partial u}(\mu_a, u_a(x)) = 0\,,$$

$$d_\epsilon^-(\mu_{a^\star}, u_a(x_a(y))) + x_a(y)d_\epsilon^+(\mu_a, u_a(x_a(y))) = y\,.$$

Then, for all $y \in [0, \min_{a\neq a^\star} d_\epsilon^-(\mu_{a^\star}, \mu_a))$, we have

$$H'(y) = -\sum_{a\neq a^\star} \frac{u_a'(x_a(y))x_a'(y)}{(d_\epsilon^+(\mu_a, u_a(x_a(y))))^2}\frac{\partial d_\epsilon^+}{\partial u}(\mu_a, u_a(x_a(y)))$$

$$= -\sum_{a\neq a^\star} \frac{u_a'(x_a(y))}{(d_\epsilon^+(\mu_a, u_a(x_a(y))))^3}\frac{\partial d_\epsilon^+}{\partial u}(\mu_a, u_a(x_a(y)))\,,$$

$$F'(y) = \sum_{a\neq a^\star} \frac{u_a'(x_a(y))x_a'(y)}{(d_\epsilon^+(\mu_a, u_a(x_a(y))))^2}$$

$$\left(d_\epsilon^+(\mu_a, u_a(x_a(y)))\frac{\partial d_\epsilon^-}{\partial u}(\mu_{a^\star}, u_a(x_a(y))) - d_\epsilon^-(\mu_{a^\star}, u_a(x_a(y)))\frac{\partial d_\epsilon^+}{\partial u}(\mu_a, u_a(x_a(y)))\right)$$

$$= -\sum_{a\neq a^\star} \frac{u_a'(x_a(y))x_a'(y)}{(d_\epsilon^+(\mu_a, u_a(x_a(y))))^2}\frac{\partial d_\epsilon^+}{\partial u}(\mu_a, u_a(x_a(y)))$$

$$\left(x_a(y)d_\epsilon^+(\mu_a, u_a(x_a(y))) + d_\epsilon^-(\mu_{a^\star}, u_a(x_a(y)))\right)$$

$$= -y\sum_{a\neq a^\star} \frac{u_a'(x_a(y))x_a'(y)}{(d_\epsilon^+(\mu_a, u_a(x_a(y))))^2}\frac{\partial d_\epsilon^+}{\partial u}(\mu_a, u_a(x_a(y))) = yH'(y)\,,$$

Therefore, showing that $H$ is increasing is a sufficient condition to show that $F$ is increasing. Using the above results, we have, for all $a \neq a^\star$ and all $y \in [0, \min_{a\neq a^\star} d_\epsilon^-(\mu_{a^\star}, \mu_a))$, we have

$$\frac{1}{(d_\epsilon^+(\mu_a, u_a(x_a(y))))^3}\frac{\partial d_\epsilon^+}{\partial u}(\mu_a, u_a(x_a(y))) > 0 \quad\text{and}\quad u_a'(x_a(y)) < 0\,.$$

Therefore, $H$ is increasing as a summation of increasing function, hence $F$ is increasing.

Let $y^\star$ such that $F(y^\star) = 1$. Reusing the above manipulation, we obtain

$$G''(y) = -\frac{F'(y)(1 + \sum_{a\neq a^\star} x_a(y)) + 2(1 - F(y))\sum_{a\neq a^\star} x_a'(y)}{(1 + \sum_{a\neq a^\star} x_a(y))^3}$$

$$= -\frac{yH'(y)(1 + yH(y) - F(y)) + 2(1 - F(y))H(y)}{(1 + yH(y) - F(y))^3}\,,$$

$$G''(y^\star) = -\frac{H'(y^\star)}{y^\star H(y^\star)^2} < 0\,,$$

Therefore, $y^\star$ is the unique maximum of $G$. We conclude this part of the proof by using the intermediate results in the proof of Lemma 43.

By strict convexity of $x_a$ and using its properties proven above, we obtain

$$x_a(y) \geq x_a(0) + yx_a'(0) = \frac{y}{d_\epsilon^+(\mu_a, \mu_{a^\star})}\,.$$

Summing those inequalities, we obtain

$$\forall y \in [0, \min_{a\neq a^\star} d_\epsilon^-(\mu_{a^\star}, \mu_a)), \quad G(y) = \frac{y}{1 + \sum_{a\neq a^\star} x_a(y)} \leq \frac{1}{\frac{1}{y} + \sum_{a\neq a^\star} \frac{1}{d_\epsilon^+(\mu_a, \mu_{a^\star})}}\,.$$

Using that $y \mapsto 1/(1/y + \alpha)$ is increasing for $\alpha > 0$, we obtain that

$$T_\epsilon^\star(\boldsymbol{\nu})^{-1} = \max_{y\in[0, \min_{a\neq a^\star} d_\epsilon^-(\mu_{a^\star}, \mu_a))} G(y) \leq \frac{1}{\frac{1}{\min_{a\neq a^\star} d_\epsilon^-(\mu_{a^\star}, \mu_a)} + \sum_{a\neq a^\star} \frac{1}{d_\epsilon^+(\mu_a, \mu_{a^\star})}}\,.$$

This concludes the proof of the second to last result. The last result is obtained by combining Lemmas 26 and 25 and the derivation in the proof of Lemma 45. $\qquad\square$

Lemma 48 is a technical result used in the proof of sufficient exploration of our sampling rule.

**Lemma 48.** *Let $d_\epsilon^\pm$ as in Eq. (3). Let $\mu \in (0,1)^K$. There exists $\alpha > 0$ such that*

$$C_\mu := \min_{(a,b):\mu_a > \mu_b} \quad \inf_{\substack{\lambda_a, \lambda_b: \\ \max_{c \in \{a,b\}} |\mu_c - \lambda_c| \le \alpha}} \quad \inf_{u \in [0,1]} \left\{ d_\epsilon^-(\lambda_a, u) + d_\epsilon^+(\lambda_b, u) \right\} > 0 . \tag{38}$$

*Proof.* Using Lemma 35 for $w_1 = w_2 = 1$, the function $\mu \mapsto \inf_{u \in [0,1]} \left\{ d_\epsilon^-(\mu_a, u) + d_\epsilon^+(\mu_b, u) \right\}$ is continuous on $\mathcal{F}^K$. Since it has strictly positive values when $\mu_a > \mu_b$ (Lemma 35), there exists $\alpha$ such that

$$\inf_{\substack{\lambda_a, \lambda_b: \\ \max_{c \in \{a,b\}} |\mu_c - \lambda_c| \le \alpha}} \quad \inf_{u \in [0,1]} \left\{ d_\epsilon^-(\lambda_a, u) + d_\epsilon^+(\lambda_b, u) \right\} > 0 .$$

Further lower bounding by a finite number of strictly positive constants yields the result. $\square$

Lemma 48 is a technical result used in the proof of convergence towards the optimal allocation of our sampling rule.

**Lemma 49.** *Let $d_\epsilon^\pm$ as in Eq. (3). Let $(\phi_1, \phi_2) \in (0,1)^2$. Let $\mathcal{I}_a := \left\{ \mu \in (0,1)^K \mid a \in a^\star(\mu) \right\}$ for all $a \in [K]$. For all $a^\star \in [K]$, all $\mu \in \mathcal{I}_{a^\star}$, all $(a,b) \in ([K] \setminus \{a^\star\})^2$ such that $a \ne b$, and all $\beta \in [0,1]$, define*

$$G_{a,b}(\mu, \beta) := \inf_{u \in [0,1]} \left\{ \beta d_\epsilon^-(\mu_{a^\star}, u) + \phi_1 d_\epsilon^+(\mu_a, u) \right\} - \inf_{u \in (0,1)} \left\{ \beta d_\epsilon^-(\mu_{a^\star}, u) + \phi_2 d_\epsilon^+(\mu_b, u) \right\} .$$

*The function $(\mu, \beta) \mapsto G_{a,b}(\mu, \beta)$ is continuous on $(0,1)^K \times [0,1]$. For all $\xi > 0$, the function $(\mu, \beta) \mapsto \inf_{\tilde\beta: |\beta - \tilde\beta| \le \xi} G_{a,b}(\mu, \beta)$ is continuous on $(0,1)^K$.*

*Proof.* Since $\bigcup_{a \in [K]} \mathcal{I}_a^K = (0,1)^K$, it is enough to show the property for all $a \in [K]$. Let $a^\star \in [K]$, $\mu \in \mathcal{I}_{a^\star}$, $(a,b) \in ([K] \setminus \{a^\star\})^2$ such that $a \ne b$. As done in Lemma 42 by using Lemma 35, we obtain that the function $(\mu, \beta) \mapsto G_{a,b}(\mu, \beta)$ is continuous on $\mathcal{I}_{a^\star} \times [0,1]$ for all $a^\star \in [K]$, hence on $(0,1)^K \times [0,1]$. Let $\Phi : \mu \mapsto \left\{ \tilde\beta : |\beta - \tilde\beta| \le \xi \right\}$, it is a continuous (constant), compact valued and non-empty correspondence. Using the above continuity, Berge's theorem yields that $\mu \mapsto \inf_{\tilde\beta: |\beta - \tilde\beta| \le \xi} G_{a,b}(\mu, \tilde\beta)$ is continuous on $(0,1)^K$. $\square$

# H  Asymptotic Upper Bound on the Expected Sample complexity

Let $\boldsymbol{\nu}$ be a Bernoulli instance with means $\mu \in (0,1)^2$ and unique best arm $a^\star \in [K]$, i.e., $\arg\max_{a \in [K]} \mu_a = \{a^\star\}$. Let $\beta \in (0,1)$. Let $w_{\epsilon,\beta}^\star(\boldsymbol{\nu}) = \{w_{\epsilon,\beta}^\star\}$ be the unique $\beta$-optimal allocation defined in Eq. (35), which satisfies $\min_{a \in [K]} w_{\epsilon,\beta,a}^\star > 0$ by Lemma 43. At equilibrium, we have equality of the transportation costs by Lemma 43, namely

$$\forall a \ne a^\star, \quad W_{\epsilon,a^\star,a}(\mu, w_{\epsilon,\beta}^\star) = T_{\epsilon,\beta}^\star(\boldsymbol{\nu})^{-1} , \tag{39}$$

where $W_{\epsilon,a,b}$ is defined in Eq. (4) and $T_{\epsilon,\beta}^\star$ is defined in Eq. (35).

Let $\gamma > 0$. Let $\omega \in \triangle_K$ be any allocation over arms such that $\min_a \omega_a > 0$. We denote by $T_\gamma(\omega)$ the *convergence time* towards $\omega$, which is a random variable quantifying the number of samples required for the global empirical allocations $N_n/(n-1)$ to be $\gamma$-close to $\omega$ for any subsequent time, namely

$$T_\gamma(\omega) := \inf \left\{ T \ge 1 \mid \forall n \ge T, \left\| \frac{N_n}{n-1} - \omega \right\|_\infty \le \gamma \right\} . \tag{40}$$

The proof of Theorem 7 follows the same analysis as the unified analysis of Top Two algorithms, see, e.g., Jourdan et al. [2022]. Appendix H is organised as follows. After recalling some technical results (Appendix H.1), we prove sufficient exploration of our sampling rule (Appendix H.2). Second, we prove that convergence time towards the $\beta$-optimal allocation of our sampling rule (Appendix H.3) has finite expectation. Finally, we conclude the proof of Theorems 7 (Appendix H.4).

### H.1 Technical Results from the Literature

Lemma 50 relates the global counts $(N_{n,a})_{a \in [K]}$ and the local counts $(\tilde{N}_{n,a})_{a \in [K]}$.

**Lemma 50.** *Let $\eta > 0$ be the geometric parameter used for the geometric grid update of our private empirical mean estimator. For all $(a, k) \in [K] \times \mathbb{N}$ s.t. $\mathbb{E}_{\boldsymbol{\nu}\pi}[T_k(a)] < +\infty$, $N_{T_k(a),a} = \tilde{N}_{k,a} = \lceil (1+\eta)^{k-1} \rceil$. For all $a \in [K]$ and all $n \in \mathbb{N}$, $N_{n,a} \geq \tilde{N}_{n,a} \geq N_{n,a}/(1+\eta)$.*

*Proof.* Let $a \in [K]$. After initialisation, we have $k = 1$, $T_1(a) = K + 1$ and $N_{T_1(a),a} = 1$. Using the definition of the phase switch, it is direct to see that $N_{T_2(a),a} = \tilde{N}_{2,a} = \lceil 1 + \eta \rceil$ when $\mathbb{E}_{\boldsymbol{\nu}\pi}[T_2(a)] < +\infty$. Similarly, we obtain $N_{T_k(a),a} = \tilde{N}_{k,a} = \lceil (1+\eta)^{k-1} \rceil$ when $\mathbb{E}_{\boldsymbol{\nu}\pi}[T_k(a)] < +\infty$. The last result is a direct consequence of the definition of the per-arm geometric update grid. $\square$

Lemma 51 controls the deviation $N_{n,a}^a - \beta L_{n,a}$ enforced by the tracking procedure.

**Lemma 51** (Lemma 2.2 in Jourdan and Degenne [2024]). *For all $n > K$ and all $a \in [K]$, $-1/2 \leq N_{n,a}^a - \beta L_{n,a} \leq 1$.*

Lemma 52 gathers properties on the $\overline{W}_{-1}$ function used in the stopping threshold.

**Lemma 52** (Jourdan et al. [2023]). *Let $\overline{W}_{-1}(x) := -W_{-1}(-e^{-x})$ for all $x \geq 1$, where $W_{-1}$ is the negative branch of the Lambert W function. The function $\overline{W}_{-1}$ is increasing on $(1, +\infty)$ and strictly concave on $(1, +\infty)$. In particular, $\overline{W}'_{-1}(x) = \left(1 - \frac{1}{\overline{W}_{-1}(x)}\right)^{-1}$ for all $x > 1$. Then, for all $y \geq 1$ and $x \geq 1$,*

$$\overline{W}_{-1}(y) \leq x \quad \Longleftrightarrow \quad y \leq x - \log(x).$$

*Moreover, for all $x > 1$,*

$$x + \log(x) \leq \overline{W}_{-1}(x) \leq x + \log(x) + \min\left\{\frac{1}{2}, \frac{1}{\sqrt{x}}\right\}.$$

Lemma 53 gives an upper bound on a time define implicit as a function of $\overline{W}_{-1}$, namely it is an inversion result.

**Lemma 53** (Lemma 32 in Azize et al. [2024]). *Let $\overline{W}_{-1}$ defined in Lemma 52. Let $A > 0$, $B > 0$ such that $B/A + \log A > 1$ and $C(A, B) = \sup\{x \mid x < A\log x + B\}$. Then, $C(A, B) < h_1(A, B)$ with $h_1(z, y) = z\overline{W}_{-1}(y/z + \log z)$.*

Lemma 54 shows that upon correction the supremum of sub-exponential random variables is also a sub-exponential random variable.

**Lemma 54** (Lemma 72 in Jourdan et al. [2022]). *Suppose that $(X_n)_{n \geq 1}$ are sub-exponential random variables with constants $(C_n)$, such that $c := \inf_n C_n > 0$. Then $\sup_n(X_n/\log(e + n))$ is sub-exponential.*

Lemma 55 gives a coarse convergence rate of the private empirical estimators of the means towards their true means.

**Lemma 55.** *There exist sub-exponential random variable $W_\mu$ such that almost surely, for all $a \in [K]$ and all $n$ such that $\tilde{N}_{n,a} \geq 1$,*

$$\tilde{N}_{n,a}|\tilde{\mu}_{n,a} - \mu_a| \leq W_\mu \log(e + \tilde{N}_{n,a}).$$

*In particular, any random variable which is polynomial in $W_\mu$ has a finite expectation.*

*Proof.* Let us define

$$W_\mu = \max_{a \in [K]} \sup_{n \in \mathbb{N}} \frac{\tilde{N}_{n,a}|\tilde{\mu}_{n,a} - \mu_a|}{\log(e + \tilde{N}_{n,a})}.$$

Let $a \in [K]$. Let us define the geometric grid $N_k = \lceil (1+\eta)^{k-1} \rceil$ for all $k \in \mathbb{N}$, on which we effectively need to control the concentration. The maximum of a finite number of sub-exponential

random variables is sub-exponential. Therefore, using the geometric update grid, it suffices to show that
$$\sup_{k \in \mathbb{N}} \frac{N_k |(Z_{N_k} + S_k)/N_k - \mu_a|}{\log(e + N_k)}$$
is sub-exponential, where $Z_{N_k}$ is the cumulative sum of $N_k$ i.i.d. observations from $\mathrm{Ber}(\mu_a)$ and $S_k$ is the cumulative sum of $k$ i.i.d. observations from $\mathrm{Lap}(1/\epsilon)$.

Using that $Z_{N_k} - N_k \mu_a$ is sub-Gaussian and $S_k$ is sub-exponential, for a fixed $k$, $|Z_{N_k} - N_k \mu_a + S_k|$ is sub-exponential. Applying Lemma 54, we obtain that
$$\sup_{k \in \mathbb{N}} \frac{N_k |(Z_{N_k} + S_k)/N_k - \mu_a|}{\log(e + N_k)}$$
is sub-exponential. We finally obtain that the maximum over the finitely many arms has the same property. □

## H.2 Sufficient Exploration

The first step of in the generic analysis of Top Two algorithms [Jourdan et al., 2022] consists in showing sufficient exploration. The main idea is that, if there are still undersampled arms, either the leader or the challenger will be among them. Therefore, after a long enough time, no arm can still be undersampled. We emphasise that there are multiple ways to select the leader/challenger pair in order to ensure sufficient exploration. Therefore, other choices of leader/challenger pair would yield similar results.

Given an arbitrary phase $p \in \mathbb{N}$, we define the sampled enough set, i.e., the arms having reached phase $p$, and the arm with highest mean in this set (when not empty) as
$$S_n^p = \{a \in [K] \mid N_{n,a} \geq (1+\eta)^{p-1}\} \quad \text{and} \quad a_n^\star = \arg\max_{a \in S_n^p} \mu_a . \tag{41}$$

Since $\min_{a \neq b} |\mu_a - \mu_b| > 0$, $a_n^\star$ is unique. Let $p \in \mathbb{N}$ such that $(p-1)/4 \in \mathbb{N}$. We define the highly and the mildly under-sampled sets as
$$U_n^p := \{a \in [K] \mid N_{n,a} < (1+\eta)^{(p-1)/2}\} \quad \text{and} \quad V_n^p := \{a \in [K] \mid N_{n,a} < (1+\eta)^{3(p-1)/4}\} . \tag{42}$$
Those arms have not reached phase $(p-1)/2$ and phase $3(p-1)/4$, respectively.

Lemma 56 shows that, when the leader is sampled enough, it is the arm with highest true mean among the sampled enough arms.

**Lemma 56.** *Let $S_n^p$ and $a_n^\star$ as in (41). There exists $p_0$ with $\mathbb{E}_{\nu\pi}[\exp(\alpha p_0)] < +\infty$ for all $\alpha > 0$ such that if $p \geq p_0$, for all $n$ such that $S_n^p \neq \emptyset$, we have*

- *For all $a \in S_n^p$, we have $\tilde{\mu}_{n,a} \in (0,1)$ and $a_n^\star = \arg\max_{a \in S_n^p} \tilde{\mu}_{n,a}$.*

- *If $B_n \in S_n^p$, then $B_n = a_n^\star$.*

*Proof.* Let $p_0$ to be specified later. Let $p \geq p_0$. Let $n \in \mathbb{N}$ such that $S_n^p \neq \emptyset$, where $S_n^p$ and $a_n^\star$ as in Equation (41). Since $N_{n,a} \geq (1+\eta)^{p-1}$ for all $a \in S_n^p$, we have $\tilde{N}_{n,a} \geq (1+\eta)^{p-1}$. Using Lemma 55 and $x \to \log(e+x)/x$ is decreasing, we obtain that
$$\tilde{\mu}_{n,a_n^\star} \geq \mu_{a_n^\star} - W_\mu \frac{\log(e + (1+\eta)^{p-1})}{(1+\eta)^{p-1}} ,$$
$$\forall a \in S_n^p \setminus \{a_n^\star\}, \quad \tilde{\mu}_{n,a} \leq \mu_a + W_\mu \frac{\log(e + (1+\eta)^{p-1})}{(1+\eta)^{p-1}} .$$

Let $\overline{\Delta}_{\min} = \min_{a \neq b} |\mu_a - \mu_b|$ and $\Delta_0 = \min_{a \in [K]} \min\{\mu_a, 1 - \mu_a\} > 0$. By assumption on the considered instances, we know that $\overline{\Delta}_{\min} > 0$. Let $p_1 = \lceil \log_{1+\eta}(X_1 - e) \rceil + 1$ with
$$X_1 = \sup \left\{ x > 1 \mid x \leq 4(\min\{\overline{\Delta}_{\min}, \Delta_0\})^{-1} W_\mu \log x + e \right\}$$
$$\leq h_1(4(\min\{\overline{\Delta}_{\min}, \Delta_0\})^{-1} W_\mu, e) ,$$

where we used Lemma 53, and $h_1$ defined therein. Then, for all $p \in \mathbb{N}$ such that $p \geq p_1 + 1$ and all $n \in \mathbb{N}$ such that $S_n^p \neq \emptyset$, we have

$$\forall a \in S_n^p, \quad \mu_a - \min\{\overline{\Delta}_{\min}, \Delta_0\}/4 \leq \tilde{\mu}_{n,a} \leq \mu_a + \min\{\overline{\Delta}_{\min}, \Delta_0\}/4 \, .$$

Therefore, we have $\tilde{\mu}_{n,a} \in (0,1)$ for all $a \in S_n^p$. Since $\tilde{\mu}_{n,a_n^\star} \geq \mu_{a_n^\star} - \min\{\overline{\Delta}_{\min}, \Delta_0\}/4$ and $\tilde{\mu}_{n,a} \leq \mu_a + \min\{\overline{\Delta}_{\min}, \Delta_0\}/4$ for all $a \in S_n^p \setminus \{a^\star\}$, we obtain $a_n^\star = \arg\max_{a \in S_n^p} \tilde{\mu}_{n,a}$ since $\arg\max_{a \in S_n^p} \tilde{\mu}_{n,a}$ is unique. The leader is defined as $B_n = \arg\max_{a \in [K]} [\tilde{\mu}_{n,a}]_0^1$. If $B_n \in S_n^p$, we obtain

$$B_n = \arg\max_{a \in S_n^p} [\tilde{\mu}_{n,a}]_0^1 = \arg\max_{a \in S_n^p} \tilde{\mu}_{n,a} = a_n^\star \, .$$

For all $\alpha \in \mathbb{R}_+$, we have $\exp(\alpha p_1) \leq e^{3\alpha}(X_1 - e)^{\alpha/\log 2}$, hence $\mathbb{E}_{\nu\pi}[\exp(\alpha p_1)] < +\infty$ by using Lemma 55 and $h_1(x,e) \sim_{x \to +\infty} x \log x$ to obtain that $\exp(\alpha p_1)$ is at most polynomial in $W_\mu$. Taking $p_0 = p_1$ concludes the proof. $\qquad\square$

Lemma 57 shows that the transportation costs between the sampled enough arms with largest true means and the other sampled enough arms are increasing fast enough.

**Lemma 57.** *Let $S_n^p$ as in Eq. (41). There exists $p_1$ with $\mathbb{E}_{\nu\pi}[\exp(\alpha p_1)] < +\infty$ for all $\alpha > 0$ such that if $p \geq p_1$, for all $n$ such that $S_n^p \neq \emptyset$, for all $(a,b) \in (S_n^p)^2$ such that $\mu_a > \mu_b$, we have*

$$W_{\epsilon,a,b}(\tilde{\mu}_n, N_n) \geq (1+\eta)^{p-1} C_\mu \, ,$$

*where $C_\mu > 0$ is a problem dependent constant.*

*Proof.* Let $p_0$ as in Lemma 56. Let $p \geq p_0$. Let $n \in \mathbb{N}$ such that $S_n^p \neq \emptyset$, where $S_n^p$ as in Eq. (41). Since $N_{n,a} \geq (1+\eta)^{p-1}$ for all $a \in S_n^p$, we have $\tilde{N}_{n,a} \geq (1+\eta)^{p-1}$ by using Lemma 50. Let $(a,b) \in (S_n^p)^2$ such that $\mu_a > \mu_b$. Using Lemma 48, there exists $\alpha_\mu > 0$ such that

$$C_\mu = \min_{\substack{(a,b):\mu_a > \mu_b}} \inf_{\substack{\lambda_a, \lambda_b: \\ \max_{c \in \{a,b\}} |\mu_c - \lambda_c| \leq \alpha_\mu}} \inf_{u \in [0,1]} \left\{ d_\epsilon^-(\lambda_a, u) + d_\epsilon^+(\lambda_b, u) \right\} > 0 \, .$$

Let $\eta > 0$ s.t. $\eta < \frac{1}{4} \min\{\overline{\Delta}_{\min}, \Delta_0, \alpha_\mu\}$ where $\overline{\Delta}_{\min} = \min_{a \neq b} |\mu_a - \mu_b|$ and $\Delta_0 = \min_{a \in [K]} \min\{\mu_a, 1 - \mu_a\}$. Similarly as in the proof of Lemma 56, we can construct $p_2$ with $\mathbb{E}_{\nu\pi}[\exp(\alpha p_2)] < +\infty$ for all $\alpha > 0$ such that if $p \geq p_2$, for all $n$ such that $S_n^p \neq \emptyset$, we have $|\tilde{\mu}_{n,a} - \mu_a| \leq \eta$ for all $a \in S_n^p$. Therefore, we have $\tilde{\mu}_{n,a} = [\tilde{\mu}_{n,a}]_0^1$ and $[\tilde{\mu}_{n,b}]_0^1 = \tilde{\mu}_{n,b}$. Moreover, we have $\tilde{\mu}_{n,a} \geq \mu_a - \eta > \mu_b + \eta \geq \tilde{\mu}_{n,b}$. Then, we obtain

$$\begin{aligned}
W_{\epsilon,a,b}(\tilde{\mu}_n, N_n) &= \inf_{u \in [0,1]} \left\{ N_{n,a} d_\epsilon^-(\tilde{\mu}_{n,a}, u) + N_{n,b} d_\epsilon^+(\tilde{\mu}_{n,b}, u) \right\} \\
&\geq (1+\eta)^{p-1} \inf_{u \in [0,1]} \left\{ d_\epsilon^-(\tilde{\mu}_{n,a}, u) + d_\epsilon^+(\tilde{\mu}_{n,b}, u) \right\} \\
&\geq (1+\eta)^{p-1} \inf_{\substack{\lambda_a, \lambda_b: \\ \max_{c \in \{a,b\}} |\mu_c - \lambda_c| \leq \alpha_\mu}} \inf_{u \in [0,1]} \left\{ d_\epsilon^-(\lambda_a, u) + d_\epsilon^+(\lambda_b, u) \right\} \geq (1+\eta)^{p-1} C_\mu \, .
\end{aligned}$$

This concludes the proof. $\qquad\square$

Lemma 58 shows that the transportation costs between sampled enough arms and undersampled arms are not increasing too fast.

**Lemma 58.** *Let $S_n^p$ be as in Eq. (41). There exists $p_2$ with $\mathbb{E}_{\nu\pi}[\exp(\alpha p_2)] < +\infty$ for all $\alpha > 0$ such that if $p \geq p_2$, for all $n$ such that $S_n^p \neq \emptyset$, For all $p \geq p_2$ and all $n$ such that $S_n^p \neq \emptyset$, for all $a \in S_n^p$ and $b \notin S_n^p$,*

$$W_{\epsilon,a,b}(\tilde{\mu}_n, N_n) \leq (1+\eta)^{p-1} D_\mu \, ,$$

*where $D_\mu \in (0, +\infty)$ is a problem dependent constant.*

*Proof.* Let $n \in \mathbb{N}$ such that $S_n^p \neq \emptyset$, where $S_n^p$ as in Eq. (41). Since $N_{n,a} \geq (1+\eta)^{p-1}$ for all $a \in S_n^p$, we have $\tilde{N}_{n,a} \geq (1+\eta)^{p-1}$ by using Lemma 50. Likewise, $N_{n,b} < (1+\eta)^{p-1}$ for all

$b \notin S_n^p$, we have $\tilde{N}_{n,b} < (1+\eta)^{p-1}$. Let $a \in S_n^p$ and $b \notin S_n^p$. Since the result is direct when $[\tilde{\mu}_{n,a}]_0^1 \leq [\tilde{\mu}_{n,b}]_0^1$, we assume $[\tilde{\mu}_{n,a}]_0^1 > [\tilde{\mu}_{n,b}]_0^1$ in the following.

Let $\eta > 0$ s.t. $\eta < \frac{1}{4}\min\{\overline{\Delta}_{\min}, \Delta_0\}$ where $\overline{\Delta}_{\min} = \min_{a \neq b}|\mu_a - \mu_b|$ and $\Delta_0 = \min_{a \in [K]}\min\{\mu_a, 1 - \mu_a\} > 0$. Similarly as in the proof of Lemma 56, we can construct $p_2$ with $\mathbb{E}_{\boldsymbol{\nu}_\pi}[\exp(\alpha p_2)] < +\infty$ for all $\alpha > 0$ such that if $p \geq p_2$, for all $n$ such that $S_n^p \neq \emptyset$, we have $|\tilde{\mu}_{n,a} - \mu_a| \leq \eta$ for all $a \in S_n^p$. Let $g_\epsilon^+(x) = \frac{x}{x(1-e^\epsilon)+e^\epsilon}$ as in Lemma 24. Using Lemma 24, for all $a \in S_n^p$, we have

$$1 > \mu_a + \min\{\overline{\Delta}_{\min}, \Delta_0\}/4 \geq \tilde{\mu}_{n,a} > g_\epsilon^+(\tilde{\mu}_{n,a}) \geq g_\epsilon^+(\mu_a - \min\{\overline{\Delta}_{\min}, \Delta_0\}/4) > 0 \,.$$

Taking $u = \tilde{\mu}_{n,a} \in [0,1]$ and using that $d_\epsilon^-(\tilde{\mu}_{n,a}, \tilde{\mu}_{n,a}) = 0$, we obtain

$$W_{\epsilon,a,b}(\tilde{\mu}_n, N_n) = \inf_{u \in [0,1]} \left\{N_{n,a}d_\epsilon^-(\tilde{\mu}_{n,a}, u) + N_{n,b}d_\epsilon^+(\tilde{\mu}_{n,b}, u)\right\}$$

$$\leq N_{n,b}d_\epsilon^+(\tilde{\mu}_{n,b}, \tilde{\mu}_{n,a}) \leq (1+\eta)^{p-1}d_\epsilon^+(\tilde{\mu}_{n,b}, \tilde{\mu}_{n,a}) \,,$$

where the last term is positive since $\tilde{\mu}_{n,a} > [\tilde{\mu}_{n,b}]_0^1$ and $\tilde{\mu}_{n,a} \in (0,1)$ by Lemma 25.

When $\tilde{\mu}_{n,b} \leq 0$, Lemma 25 yields that

$$d_\epsilon^+(\tilde{\mu}_{n,b}, \tilde{\mu}_{n,a}) = -\log\left(1 - \tilde{\mu}_{n,a}(1 - e^{-\epsilon})\right) \leq \epsilon \,,$$

where we used that $x \to -\log\left(1 - x(1 - e^{-\epsilon})\right)$ is increasing on $(0,1)$. When $\tilde{\mu}_{n,b} \in (0, g_\epsilon^+(\tilde{\mu}_{n,a}))$, Lemma 25 yields that

$$d_\epsilon^+(\tilde{\mu}_{n,b}, \tilde{\mu}_{n,a}) = -\log\left(1 - \tilde{\mu}_{n,a}(1 - e^{-\epsilon})\right) - \epsilon\tilde{\mu}_{n,b} \leq \epsilon \,.$$

When $\tilde{\mu}_{n,b} \in [g_\epsilon^+(\tilde{\mu}_{n,a}), \tilde{\mu}_{n,a})$, Lemma 25 yields that

$$d_\epsilon^+(\tilde{\mu}_{n,b}, \tilde{\mu}_{n,a}) = \mathrm{kl}(\tilde{\mu}_{n,b}, \tilde{\mu}_{n,a}) \leq -\log\min\{\tilde{\mu}_{n,a}, 1 - \tilde{\mu}_{n,a}\}$$

$$\leq -\log\min\{\mu_a - \min\{\overline{\Delta}_{\min}, \Delta_0\}/4, 1 - \mu_a - \min\{\overline{\Delta}_{\min}, \Delta_0\}/4\} \,,$$

where we used the classical result that $\mathrm{kl}(q,p) \leq -\log\min\{p, 1-p\}$. Let us define

$$D_\mu = \epsilon + \max_{a \in [K]}\left\{-\log\min\{\mu_a - \min\{\overline{\Delta}_{\min}, \Delta_0\}/4, 1 - \mu_a - \min\{\overline{\Delta}_{\min}, \Delta_0\}/4\}\right\} \,.$$

Then, we have shown that $d_\epsilon^+(\tilde{\mu}_{n,b}, \tilde{\mu}_{n,a}) \leq D_\mu$ where $D_\mu \in (0, +\infty)$. This yields the result. □

Lemma 59 shows that the challenger is mildly undersampled if the leader is not mildly undersampled.

**Lemma 59.** *Let $V_n^p$ be as in Equation* (42). *There exists $p_3$ with $\mathbb{E}_{\boldsymbol{\nu}_\pi}[\exp(\alpha p_3)] < +\infty$ for all $\alpha > 0$ such that if $p \geq p_3$, for all $n$ such that $U_n^p \neq \emptyset$, $B_n \notin V_n^p$ implies $C_n \in V_n^p$.*

*Proof.* Let $p_3$ to be specified later. Let $p \geq p_3$. Let $n \in \mathbb{N}$ such that $U_n^p \neq \emptyset$ and $V_n^p \neq [K]$, where $U_n^p \subseteq V_n^p$ are defined in Eq. (42). Since the statement holds when $B_n \in V_n^p$, we suppose that $B_n \notin V_n^p$ in the following.

Let $p_0$ as in Lemma 56, $p_1$ and $C_\mu$ as in Lemma 57, and $p_2$ and $D_\mu$ as in Lemma 58. Let $p_4 = \max\{2p_2 - 1, \frac{4}{3}\max\{p_0, p_1\} - 1/3\}$, which satisfied that $\mathbb{E}_{\boldsymbol{\nu}_\pi}[\exp(\alpha p_4)] < +\infty$ for all $\alpha > 0$ by using Lemmas 56, 57 and 58. Then, for all $p \geq p_4 = \max\{2p_2 - 1, \frac{4}{3}\max\{p_0, p_1\} - 1/3\}$ and all $n$ such that $B_n \notin V_n^p$, we have $\tilde{\mu}_{n,a} \in (0,1)$ for all $a \notin V_n^p$, $B_n = b_n^\star := \arg\max_{a \notin V_n^p}\mu_a$, $B_n \notin U_n^p$ and

$$\forall b \notin \{b_n^\star\} \cup V_n^p, \quad W_{\epsilon,b_n^\star,b}(\tilde{\mu}_n, N_n) + \log N_{n,b} \geq (1+\eta)^{3(p-1)/4}C_\mu + \frac{3(p-1)}{4}\log(1+\eta) \,,$$

$$\forall b \in U_n^p, \quad W_{\epsilon,b_n^\star,b}(\tilde{\mu}_n, N_n) + \log N_{n,b} \leq (1+\eta)^{(p-1)/2}D_\mu + \frac{p-1}{2}\log(1+\eta) \,,$$

where we used Lemmas 56, 57 and 58. Direct manipulations yield that

$$(1+\eta)^{3(p-1)/4}C_\mu + \frac{3(p-1)}{4}\log(1+\eta) \geq (1+\eta)^{(p-1)/2}D_\mu + \frac{p-1}{2}\log(1+\eta)$$

$$\Longleftarrow \quad p \geq p_5 = 4\lceil \log_{1+\eta}(D_\mu/C_\mu)\rceil + 1 \,,$$

where $\mathbb{E}_{\boldsymbol{\nu}_\pi}[\exp(\alpha p_5)] < +\infty$ for all $\alpha > 0$ since it is a deterministic constant. Let $p_3 = \max\{p_4, p_5\}$ which satisfies $\mathbb{E}_{\boldsymbol{\nu}_\pi}[\exp(\alpha p_3)] < +\infty$ for all $\alpha > 0$. Then, we have shown that for all $p \geq p_3$, for all $n$ such that $B_n \notin V_n^p$, we have $B_n = b_n^\star$ and

$$\min_{b \notin \{b_n^\star\} \cup V_n^p}\{W_{\epsilon, b_n^\star, b}(\tilde{\mu}_n, N_n) + \log N_{n,b}\} > \max_{b \in U_n^p}\{W_{\epsilon, b_n^\star, b}(\tilde{\mu}_n, N_n) + \log N_{n,b}\} \,.$$

By definition of the TC challenger, i.e., $C_n \in \arg\min_{b \neq B_n}\{W_{\epsilon, B_n, b}(\tilde{\mu}_n, N_n) + \log N_{n,b}\}$, we obtain that $C_n \in V_n^p$. Otherwise, there would be a contradiction since we assumed $U_n^p \neq \emptyset$. This concludes the proof. $\qquad \square$

Lemma 60 shows that all the arms are sufficient explored for large enough $n$.

**Lemma 60.** *There exists $N_0$ with $\mathbb{E}_{\boldsymbol{\nu}_\pi}[N_0] < +\infty$ such that, for all $n \geq N_0$ and all $a \in [K]$, $N_{n,a} \geq \sqrt{n/K}$.*

*Proof.* Let $p_0$ and $p_3$ as in Lemmas 56 and 59. Combining Lemmas 56 and 59 yields that, for all $p \geq p_4 = \max\{p_3, 4p_0/3 - 1/3\}$ and all $n$ such that $U_n^p \neq \emptyset$, we have $B_n \in V_n^p$ or $C_n \in V_n^p$. We have $\mathbb{E}_{\boldsymbol{\nu}_\pi}[(1+\eta)^{p_2}] < +\infty$. We have $(1+\eta)^{p-1} \geq K(1+\eta)^{3(p-1)/4}$ for all $p \geq p_5 = 4\lceil \log_{1+\eta} K\rceil + 1$. Let $p \geq \max\{p_5, p_4\}$. For notational simplicity, we conduct the proof as if that $k(1+\eta)^{p-1} \in \mathbb{N}$ for all $k \in [K]$. It is direct to adapt the proof by using the operator $\lceil \cdot \rceil$.

Suppose towards contradiction that $U_{K(1+\eta)^{p-1}}^p$ is not empty. Then, for any $1 \leq t \leq K(1+\eta)^{p-1}$, $U_t^p$ and $V_t^p$ are non empty as well. Using the pigeonhole principle, there exists some $a \in [K]$ such that $N_{(1+\eta)^{p-1},a} \geq (1+\eta)^{3(p-1)/4}$. Thus, we have $\left|V_{(1+\eta)^{p-1}}^p\right| \leq K - 1$. Our goal is to show that $\left|V_{2(1+\eta)^{p-1}}^p\right| \leq K - 2$. A sufficient condition is that one arm in $V_{(1+\eta)^{p-1}}^p$ is pulled at least $(1+\eta)^{3(p-1)/4}$ times between $(1+\eta)^{p-1}$ and $2(1+\eta)^{p-1} - 1$.

**Case 1.** Suppose there exists $a \in V_{(1+\eta)^{p-1}}^p$ such that $L_{2(1+\eta)^{p-1},a} - L_{(1+\eta)^{p-1},a} \geq \beta^{-1}\left((1+\eta)^{3(p-1)/4} + 3/2\right)$. Using Lemma 51, we obtain

$$N_{2(1+\eta)^{p-1},a}^a - N_{(1+\eta)^{p-1},a}^a \geq \beta(L_{2(1+\eta)^{p-1},a} - L_{(1+\eta)^{p-1},a}) - 3/2 \geq (1+\eta)^{3(p-1)/4} \,,$$

hence $a$ is sampled $(1+\eta)^{3(p-1)/4}$ times between $(1+\eta)^{p-1}$ and $2(1+\eta)^{p-1} - 1$.

**Case 2.** Suppose that for all $a \in V_{(1+\eta)^{p-1}}^p$, we have $L_{2(1+\eta)^{p-1},a} - L_{(1+\eta)^{p-1},a} < \beta^{-1}\left((1+\eta)^{3(p-1)/4} + 3/2\right)$. Then,

$$\sum_{a \notin V_{(1+\eta)^{p-1}}^p}(L_{2(1+\eta)^{p-1},a} - L_{(1+\eta)^{p-1},a}) \geq (1+\eta)^{p-1} - K\beta^{-1}\left((1+\eta)^{3(p-1)/4} + 3/2\right) \,.$$

Using Lemma 51, we obtain

$$\left|\sum_{a \notin V_{(1+\eta)^{p-1}}^p}(N_{2(1+\eta)^{p-1},a}^a - N_{(1+\eta)^{p-1},a}^a) - \beta \sum_{a \notin V_{(1+\eta)^{p-1}}^p}(L_{2(1+\eta)^{p-1},a} - L_{(1+\eta)^{p-1},a})\right| \leq 3(K-1)/2 \,.$$

Combining all the above, we obtain

$$\sum_{a \notin V_{(1+\eta)^{p-1}}^p}(L_{2(1+\eta)^{p-1},a} - L_{(1+\eta)^{p-1},a}) - \sum_{a \notin V_{(1+\eta)^{p-1}}^p}(N_{2(1+\eta)^{p-1},a}^a - N_{(1+\eta)^{p-1},a}^a)$$

$$\geq (1-\beta)\sum_{a \notin V_{(1+\eta)^{p-1}}^p}(L_{2(1+\eta)^{p-1},a} - L_{(1+\eta)^{p-1},a}) - 3(K-1)/2$$

$$\geq (1-\beta)\left((1+\eta)^{p-1} - K\beta^{-1}\left((1+\eta)^{3(p-1)/4} + 3/2\right)\right) - 3(K-1)/2 \geq K(1+\eta)^{3(p-1)/4}$$

where the last inequality is obtained for $p \geq p_6 + 1$ with

$$p_6 = \sup \left\{ p \in \mathbb{N} \mid (1-\beta)\left((1+\eta)^{p-1} - K\beta^{-1}\left((1+\eta)^{3(p-1)/4} + 3/2\right)\right) - \frac{3}{2}(K-1) \right.$$
$$\left. < K(1+\eta)^{3(p-1)/4} \right\}.$$

The left hand side summation is exactly the number of times where an arm $a \notin V^p_{(1+\eta)^{p-1}}$ was leader but wasn't sampled, hence we have shown that

$$\sum_{t=(1+\eta)^{p-1}}^{2(1+\eta)^{p-1}-1} \mathbb{1}\left(B_t \notin V^p_{(1+\eta)^{p-1}}, \, a_t = C_t\right) \geq K(1+\eta)^{3(p-1)/4}.$$

For any $(1+\eta)^{p-1} \leq t \leq 2(1+\eta)^{p-1} - 1$, $U^p_t$ is non-empty, hence we have $B_t \notin V^p_{(1+\eta)^{p-1}}$ (hence $B_t \notin V^p_t$) implies $C_t \in V^p_t \subseteq V^p_{(1+\eta)^{p-1}}$. Therefore, we have shown that

$$\sum_{t=(1+\eta)^{p-1}}^{2(1+\eta)^{p-1}-1} \mathbb{1}\left(a_t \in V^p_{(1+\eta)^{p-1}}\right) \geq \sum_{t=(1+\eta)^{p-1}}^{2(1+\eta)^{p-1}-1} \mathbb{1}\left(B_t \notin V^p_{(1+\eta)^{p-1}}, \, a_t = C_t\right) \geq K(1+\eta)^{3(p-1)/4}$$

Therefore, there is at least one arm in $V^p_{(1+\eta)^{p-1}}$ that is sampled $(1+\eta)^{3(p-1)/4}$ times between $(1+\eta)^{p-1}$ and $2(1+\eta)^{p-1} - 1$.

In summary, we have shown $\left|V^p_{2(1+\eta)^{p-1}}\right| \leq K - 2$ for all $p \geq p_7 = \max\{p_6, p_4, p_5\}$. By induction, for any $1 \leq k \leq K$, we have $\left|V^p_{k(1+\eta)^{p-1}}\right| \leq K - k$, and finally $U^p_{K(1+\eta)^{p-1}} = \emptyset$ for all $p \geq p_7$. Defining $N_0 = K(1+\eta)^{p_7-1}$, we have $\mathbb{E}_{\boldsymbol{\nu}\pi}[N_0] < +\infty$ by using Lemmas 56 and 59 for $p_4 = \max\{p_3, 4p_0/3 - 1/3\}$ and $p_6$ and $p_5$ are deterministic. For all $n \geq N_0$, we let $(1+\eta)^{p-1} = \frac{n}{K}$. Then, by applying the above, we have $U^p_{K(1+\eta)^{p-1}} = U^{\log_{1+\eta}(n/K)+1}_n$ is empty, which shows that $N_{n,a} \geq \sqrt{n/K}$ for all $a \in [K]$. $\qquad\square$

### H.3 Convergence Towards $\beta$-Optimal Allocation

The second step of in the generic analysis of Top Two algorithms Jourdan et al. [2022] is to show the convergence of the empirical proportions towards the $\beta$-optimal allocation. First, we show that the leader coincides with the best arm. Hence, the tracking procedure will ensure that the empirical proportion of time we sample it is exactly $\beta$. Second, we show that a sub-optimal arm whose empirical proportion overshoots its $\beta$-optimal allocation will not be sampled next as challenger. Therefore, this "overshoots implies not sampled" mechanism will ensure the convergence towards the $\beta$-optimal allocation. We emphasise that there are multiple ways to select the leader/challenger pair in order to ensure convergence towards the $\beta$-optimal allocation. Therefore, other choices of leader/challenger pair would yield similar results. Note that our results heavily rely on having obtained sufficient exploration first.

Lemma 61 shows the leader and the candidate answer are equal to the best arm for large enough $n$.

**Lemma 61.** *Let $N_0$ be as in Lemma 60. There exists $N_1 \geq N_0$ with $\mathbb{E}_{\boldsymbol{\nu}\pi}[N_1] < +\infty$ such that, for all $n \geq N_1$, we have $\tilde{\mu}_n \in (0,1)^K$ and $\tilde{a}_n = B_n = a^\star$.*

*Proof.* Let $\Delta_{\min} = \min_{a \neq a^\star}(\mu_{a^\star} - \mu_a)$ and $\Delta_0 = \min_{a \in [K]} \min\{\mu_a, 1 - \mu_a\} > 0$. Using Lemma 55, we obtain, for all $n \geq N_0$,

$$\tilde{\mu}_{n,a^\star} \geq \mu_{a^\star} - W_\mu \frac{\log(e + \sqrt{n/K}/(1+\eta))}{\sqrt{n/K}/(1+\eta)}$$

$$\forall a \neq a^\star, \quad \tilde{\mu}_{n,a} \leq \mu_a + W_\mu \frac{\log(e + \sqrt{n/K}/(1+\eta))}{\sqrt{n/K}/(1+\eta)},$$

where we used that $x \to \log(e+x)/x$ is decreasing and $\tilde{N}_{n,a} \geq N_{n,a}/(1+\eta) \geq \sqrt{n/K}/(1+\eta)$. Let $N_1 = \max\{N_0, \lceil K(1+\eta)^2 X_1^2 \rceil\}$ where

$$X_1 = \sup\left\{x > 1 \mid x \leq 4(\Delta_{\min}, \Delta_0)^{-1} W_\mu \log x + e\right\} \leq h_1(4(\Delta_{\min}, \Delta_0)^{-1} W_\mu, e),$$

where we used Lemma 53, and $h_1$ defined therein. Using Lemmas 55 and 60, we obtain $\mathbb{E}_{\boldsymbol{\nu}\pi}[N_1] < +\infty$. Then, we have $0 < \mu_a - \Delta_0/4 \leq \tilde{\mu}_{n,a} \leq \mu_a + \Delta_0/4 < 1$ for all $a \in [K]$. Moreover, for all $n \geq N_1$, we have $\tilde{\mu}_{n,a^\star} \geq \mu_{a^\star} - \Delta_{\min}/4$ and $\tilde{\mu}_{n,a} \leq \mu_a + \Delta_{\min}/4$ for all $a \neq a^\star$, hence

$$a^\star = \arg\max_{a \in [K]} \tilde{\mu}_{n,a} = \arg\max_{a \in [K]} [\tilde{\mu}_{n,a}]_0^1 = \tilde{a}_n = B_n.$$

This concludes the proof. $\qquad\qquad\square$

Lemma 62 shows that that the pulling proportion of the best arm converges towards $\beta$. It is a direct consequence of Lemma 61 by using the same proof as Lemma 39 in Azize et al. [2024], hence we omit the proof.

**Lemma 62** (Lemma 39 in Azize et al. [2024]). *Let $\gamma > 0$, and $N_1$ be as in Lemma 61. There exists a deterministic constant $C_0 \geq 1$ such that, for all $n \geq C_0 N_1$, we have $\left|\frac{N_{n,a^\star}}{n-1} - \beta\right| \leq \gamma$.*

Lemma 63 shows that if a sub-optimal arm overshoots its $\beta$-optimal allocation then it cannot be selected as challenger for large enough $n$.

**Lemma 63.** *Let $\gamma \in (0, \gamma_\mu)$ where $\gamma_\mu$ is a problem dependent constant. Let $N_1$ and $C_0$ be as in Lemma 61 and 62. There exists $N_2 \geq C_0 N_1$ with $\mathbb{E}_{\boldsymbol{\nu}\pi}[N_2] < +\infty$ such that, for all $n \geq N_2$,*

$$\exists a \neq a^\star, \quad \frac{N_{n,a}}{n-1} \geq \gamma + \omega^\star_{\epsilon,\beta,a} \quad \implies \quad C_n \neq a.$$

*Proof.* Let $\eta > 0$ and $\gamma > 0$ be small enough, which we will specify below. Let $\tilde{\gamma} \in (0, \gamma)$. Let $N_1$ as in Lemma 61 and $C_0$ as in Lemma 62 for $\tilde{\gamma}$. Let $n \geq C_0 N_1$. Therefore, we have $\tilde{\mu}_n \in (0,1)^K$ and $\tilde{a}_n = B_n = a^\star$ and $\left|\frac{N_{n,a^\star}}{n-1} - \beta\right| \leq \tilde{\gamma}$. Using the same proof as in Lemma 61, there exists $N_3$ with $\mathbb{E}_{\boldsymbol{\nu}\pi}[N_3] < +\infty$ such that, for all $n \geq N_3$, we have $\|\tilde{\mu}_n - \mu\|_\infty \leq \eta$. Let $n \geq \max\{C_0 N_1, N_3\}$.

Let $a \neq a^\star$ such that $\frac{N_{n,a}}{n-1} \geq \omega^\star_{\epsilon,\beta,a} + \gamma$. Suppose towards contradiction that $\frac{N_{n,b}}{n-1} > \omega^\star_{\epsilon,\beta,b}$ for all $b \notin \{a^\star, a\}$. Then, for all $n \geq C_0 N_1$, we have

$$1 - \beta + \tilde{\gamma} \geq 1 - \frac{N_{n,a^\star}}{n-1} = \sum_{b \neq a^\star} \frac{N_{n,b}}{n-1} > \gamma + \sum_{b \neq a^\star} \omega^\star_{\epsilon,\beta,b} = 1 - \beta + \gamma,$$

which yields a contradiction since $\tilde{\gamma} < \gamma$. Therefore, for all $n \geq C_0 N_1$, we have

$$\exists a \neq a^\star, \quad \frac{N_{n,a}}{n-1} \geq \omega^\star_{\epsilon,\beta,a} + \gamma \quad \implies \quad \exists b \notin \{a^\star, a\}, \quad \frac{N_{n,b}}{n-1} \leq \omega^\star_{\epsilon,\beta,b}.$$

Let $b \notin \{a^\star, a\}$ such that $\frac{N_{n,b}}{n-1} \leq \omega^\star_{\epsilon,\beta,b}$. By definition of the TC challenger, we obtain

$$C_n \neq a \impliedby W_{\epsilon,a^\star,a}(\tilde{\mu}_n, N_n) + \log N_{n,a} > W_{\epsilon,a^\star,b}(\tilde{\mu}_n, N_n) + \log N_{n,b}$$

$$\impliedby \frac{1}{n-1}\left(W_{\epsilon,a^\star,a}(\tilde{\mu}_n, N_n) - W_{\epsilon,a^\star,b}(\tilde{\mu}_n, N_n)\right) > \frac{1}{n-1} \log \frac{\omega^\star_{\epsilon,\beta,b}}{\omega^\star_{\epsilon,\beta,a} + \gamma}$$

$$\impliedby \frac{1}{n-1}\left(W_{\epsilon,a^\star,a}(\tilde{\mu}_n, N_n) - W_{\epsilon,a^\star,b}(\tilde{\mu}_n, N_n)\right) > \frac{1}{n-1} \max_{a \neq b}\left|\log \frac{\omega^\star_{\epsilon,\beta,b}}{\omega^\star_{\epsilon,\beta,a}}\right|,$$

where we used the positivity of the $\beta$-optimal allocation (Lemma 43) to ensure that $\max_{a \neq b}\left|\log \frac{\omega^\star_{\epsilon,\beta,b}}{\omega^\star_{\epsilon,\beta,a}}\right| \in (0, +\infty)$. Using that $\tilde{\mu}_{n,a^\star} > \max\{\tilde{\mu}_{n,a}, \tilde{\mu}_{n,b}\}$, we obtain

$$\frac{1}{n-1}\left(W_{\epsilon,a^\star,a}(\tilde{\mu}_n, N_n) - W_{\epsilon,a^\star,b}(\tilde{\mu}_n, N_n)\right)$$

$$\geq \inf_{u \in [0,1]}\left\{\frac{N_{n,a^\star}}{n-1} d_\epsilon^-(\tilde{\mu}_{n,a^\star}, u) + (\omega^\star_{\epsilon,\beta,a} + \gamma) d_\epsilon^+(\tilde{\mu}_{n,a}, u)\right\}$$

$$- \inf_{u \in [0,1]} \left\{ \frac{N_{n,a^\star}}{n-1} d_\epsilon^-(\tilde{\mu}_{n,a^\star}, u) + \omega_{\epsilon,\beta,b}^\star d_\epsilon^+(\tilde{\mu}_{n,b}, u) \right\}$$

$$\geq \inf_{\tilde{\beta}:|\tilde{\beta}-\beta| \leq \tilde{\gamma}} G_{a,b}(\tilde{\mu}_n, \tilde{\beta}) \geq \inf_{\lambda:\|\lambda-\mu\|_\infty \leq \eta} \inf_{\tilde{\beta}:|\tilde{\beta}-\beta| \leq \tilde{\gamma}} G_{a,b}(\lambda, \tilde{\beta}) ,$$

where, for all $(a,b) \in ([K] \setminus \{a^\star\})^2$ such that $a \neq b$,

$$G_{a,b}(\lambda, \tilde{\beta}) = \inf_{u \in [0,1]} \left\{ \tilde{\beta} d_\epsilon^-(\lambda_{a^\star}, u) + (\omega_{\epsilon,\beta,a}^\star + \gamma) d_\epsilon^+(\lambda_a, u) \right\}$$

$$- \inf_{u \in [0,1]} \left\{ \tilde{\beta} d_\epsilon^-(\lambda_{a^\star}, u) + \omega_{\epsilon,\beta,b}^\star d_\epsilon^+(\lambda_b, u) \right\} .$$

Using the equality at equilibrium from (39) (see Lemma 43) and the fact that the transportation costs are increasing in their allocation argument (see Lemma 37), we obtain $G_{a,b}(\mu, \beta) > 0$ for all $(a,b) \in ([K] \setminus \{a^\star\})^2$ such that $a \neq b$, since

$$\inf_{u \in [0,1]} \left\{ \beta d_\epsilon^-(\mu_{a^\star}, u) + (\omega_{\epsilon,\beta,a}^\star + \gamma) d_\epsilon^+(\mu_a, u) \right\} > W_{\epsilon,a^\star,a}(\mu, w_{\epsilon,\beta}^\star) = W_{\epsilon,a^\star,b}(\mu, w_{\epsilon,\beta}^\star) .$$

By Lemma 49, the functions $(\lambda, \tilde{\beta}) \to G_{a,b}(\lambda, \tilde{\beta})$ and $\lambda \to \inf_{\tilde{\beta}:|\tilde{\beta}-\beta| \leq \tilde{\gamma}} G_{a,b}(\lambda, \tilde{\beta})$ are continuous. Therefore, there exists $\eta_\mu$ and $\gamma_\mu$ small enough such that

$$\inf_{\lambda:\|\lambda-\mu\|_\infty \leq \eta} \inf_{\tilde{\beta}:|\tilde{\beta}-\beta| \leq \tilde{\gamma}} G_{a,b}(\lambda, \tilde{\beta}) \geq G_{a,b}(\mu, \beta)/2 \geq \frac{1}{2} \min_{a \neq b, a \neq a^\star, b \neq a^\star} G_{a,b}(\mu, \beta) > 0 ,$$

where the last strict inequality uses that the minimum of a finite number of positive constants is also positive. Considering such $(\eta_\mu, \gamma_\mu)$ at the beginning of the proof and taking $N_2 = \max\{C_0 N_1, N_3, \kappa_\mu\}$ where

$$\kappa_\mu = 2 + \frac{2 \max_{a \neq b} \left| \log \frac{\omega_{\epsilon,\beta,b}^\star}{\omega_{\epsilon,\beta,a}^\star} \right|}{\min_{a \neq b, a \neq a^\star, b \neq a^\star} G_{a,b}(\mu, \beta)} < +\infty ,$$

As it satisfies $\mathbb{E}_{\boldsymbol{\nu}\pi}[N_2] < +\infty$, this concludes the proof. $\qquad\square$

Lemma 64 shows that the convergence time towards the $\beta$-optimal allocation has finite expectation. It is a direct consequence of Lemmas 61, 62 and 63 by using the same proof as Lemma 41 in Azize et al. [2024], hence we omit the proof.

**Lemma 64** (Lemma 41 in Azize et al. [2024]). *Let $\gamma \in (0, \gamma_\mu)$ where $\gamma_\mu$ is a problem dependent constant, and $T_\gamma(w)$ as in Eq. (40). Then, we have $\mathbb{E}_{\boldsymbol{\nu}\pi}[T_\gamma(\omega_{\epsilon,\beta}^\star)] < +\infty$.*

### H.4 Asymptotic Upper Bound

The final step of the generic analysis of Top Two algorithms [Jourdan et al., 2022] is to invert the GLR stopping rule in Eq. (7) by leveraging the convergence of the empirical proportions towards the $\beta$-optimal allocation. Provided this convergence is shown, the asymptotic upper bound on the expected sample complexity only depends on the dependence in $\log(1/\delta)$ of the threshold that ensures $\delta$-correctness. Compared to the non-private GLR stopping rule, the GLR stopping rule in Eq. (7) pay an extra cost to ensure privacy.

**Lemma 65.** *Let $\epsilon > 0$, $\eta > 0$ and $(\delta, \beta) \in (0,1)^2$. Let $T_{\epsilon,\beta}^\star(\boldsymbol{\nu})$ as in Eq. (35) and $\omega_{\epsilon,\beta}^\star$ be its associated $\beta$-optimal allocation. Assume that there exists $\gamma_\mu > 0$ such that $\mathbb{E}_{\boldsymbol{\nu}\pi}[T_\gamma(\omega_{\epsilon,\beta}^\star)] < +\infty$ for all $\gamma \in (0, \gamma_\mu)$, where $T_\gamma(w)$ is defined in Eq. (40). Combining such a sampling rule, using the $GPE_\eta(\epsilon)$ update, with the GLR stopping rule as in Eq. (7) and the stopping threshold $c$ as in Eq. (8) yields an $\epsilon$-global DP and $\delta$-correct algorithm which satisfies that, for all $\boldsymbol{\nu}$ with mean $\mu$ such that $|a^\star(\mu)| = 1$,*

$$\limsup_{\delta \to 0} \frac{\mathbb{E}_{\boldsymbol{\nu}\pi}[\tau_{\epsilon,\delta}]}{\log(1/\delta)} \leq 2(1+\eta) T_{\epsilon,\beta}^\star(\boldsymbol{\nu}) .$$

*Proof.* Lemma 5 yields the $\epsilon$-global DP. Theorem 6 yields the $\delta$-correctness.

Let $\zeta > 0$ and $a^\star$ be the unique best arm. Using the equality at equilibrium from (39) (see Lemma 43) and the continuity of $(\mu, w) \mapsto \min_{a \neq a^\star(\mu)} W_{\epsilon, a^\star(\mu), a}(\mu, w)$ (see Lemma 42), there exists $\gamma_\zeta > 0$ such that $\left\| \frac{N_n}{n-1} - \omega^\star_{\epsilon, \beta} \right\|_\infty \leq \gamma_\zeta$ and $\|\tilde{\mu}_n - \mu\|_\infty \leq \gamma_\zeta$ implies that

$$\forall a \neq a^\star, \quad W_{\epsilon, a^\star, a}(\tilde{\mu}_n, N_n/(n-1)) \geq \frac{(1-\zeta)}{T^\star_{\epsilon, \beta}(\boldsymbol{\nu})} .$$

We choose such a $\gamma_\zeta$. Let $\gamma_\mu > 0$ be such that for $\mathbb{E}_{\boldsymbol{\nu}\pi}[T_\gamma(\omega^\star_{\epsilon, \beta})] < +\infty$ for all $\gamma \in (0, \gamma_\mu)$, where $T_\gamma(\omega)$ is defined in Eq. (40). Let $\gamma \in (0, \min\{\gamma_\mu, \gamma_\zeta, \min_{a \in [K]} \omega^\star_{\epsilon, \beta, a}/4, \Delta_{\min}/4, \Delta_0/4\})$ where $\Delta_{\min} = \min_{a \neq a^\star}(\mu_{a^\star} - \mu_a)$ and $\Delta_0 = \min_{a \in [K]} \min\{\mu_a, 1 - \mu_a\}$. For all $n \geq T_\gamma(\omega^\star_{\epsilon, \beta})$, we have

$$\tilde{N}_{n,a} \geq N_{n,a}/(1+\eta) \geq (n-1)(\omega^\star_{\epsilon, \beta, a} - \gamma)/(1+\eta) \geq (n-1)\frac{3}{4(1+\eta)} \min_{a \in [K]} \omega^\star_{\epsilon, \beta, a} > 0 ,$$

where the last inequality used the positivity of the $\beta$-optimal allocation (Lemma 43). Since arms are sampled linearly, it is direct to construct $N_3 \geq T_\gamma(\omega^\star_{\epsilon, \beta})$ with $\mathbb{E}_{\boldsymbol{\nu}\pi}[N_3] < +\infty$ such that $\|\tilde{\mu}_n - \mu\|_\infty \leq \gamma$ and $\left\| \frac{N_n}{n-1} - \omega^\star_{\epsilon, \beta} \right\|_\infty \leq \gamma$ (as well as $\min_{a \in [K]} N_{n,a} > e$ trivially).

Recall that $c(n, \epsilon, \delta) = c_1(n, \delta) + c_2(n, \epsilon)$ where $n \mapsto c_1(n, \delta)$ and $n \mapsto c_1(n, \delta)$ are increasing (see Lemmas 52 and 39). Since $\tilde{N}_{n,a} \leq N_{n,a} \leq n$, we obtain

$$\sum_{b \in \{a^\star, a\}} c(\tilde{N}_n, \epsilon, \delta) \leq 2(c_1(n, \delta) + c_2(n, \epsilon)) .$$

Using Lemma 37 and $\tilde{N}_{n,a} \geq N_{n,a}/(1+\eta)$ for all $a \in [K]$ (Lemma 50), we obtain

$$W_{\epsilon, a^\star, a}(\tilde{\mu}_n, \tilde{N}_n) \geq \frac{n-1}{1+\eta} W_{\epsilon, a^\star, a}\left(\tilde{\mu}_n, \frac{N_n}{n-1}\right) .$$

Let $\kappa \in (0, 1)$ and $T > N_3/\kappa$. For all $n \in [\kappa T, T]$, we have $\tilde{a}_n = a^\star$ and, for all $a \neq a^\star$,

$$\tau_{\epsilon, \delta} > n$$
$$\implies \exists a \neq a^\star, \quad W_{\epsilon, a^\star, a}(\tilde{\mu}_n, \tilde{N}_n) \leq \sum_{b \in \{a^\star, a\}} c(\tilde{N}_n, \epsilon, \delta)$$
$$\implies \exists a \neq a^\star, \quad \frac{n-1}{1+\eta} W_{\epsilon, a^\star, a}\left(\tilde{\mu}_n, \frac{N_n}{n-1}\right) \leq 2(c_1(n, \delta) + c_2(n, \epsilon))$$
$$\implies \exists a \neq a^\star, \quad \frac{n-1}{1+\eta} \frac{(1-\zeta)}{T^\star_{\epsilon, \beta}(\boldsymbol{\nu})} \leq 2c_1(T, \delta) + 2c_2(T, \epsilon) ,$$

where we used that $n \mapsto c_1(n, \delta)$ and $n \mapsto c_2(n, \epsilon)$ are increasing and $n \leq T$. Therefore, we obtain

$$\min\{\tau_{\epsilon, \delta}, T\} \leq \kappa T + \sum_{n=\kappa T}^{T} \mathbb{1}(\tau_\delta > n)$$
$$\leq \kappa T + \sum_{n=\kappa T}^{T} \mathbb{1}\left( \frac{n-1}{1+\eta} \frac{(1-\zeta)}{T^\star_{\epsilon, \beta}(\boldsymbol{\nu})} \leq 2c_1(T, \delta) + 2c_2(T, \epsilon) \right)$$
$$\leq \kappa T + 1 + \frac{2(1+\eta)T^\star_{\epsilon, \beta}(\boldsymbol{\nu})}{1-\zeta}(c_1(T, \delta) + c_2(T, \epsilon)) .$$

Let $T_\zeta(\delta)$ defined as

$$T_\zeta(\delta) := \inf\left\{ T \geq 1 \mid \frac{1}{1-\kappa}\left( 1 + \frac{2(1+\eta)T^\star_{\epsilon, \beta}(\boldsymbol{\nu})}{1-\zeta}(c_1(T, \delta) + c_2(T, \epsilon)) \right) \leq T \right\} .$$

Using Lemma 52, we know that $\overline{W}_{-1}(x) =_{x \to \infty} x + \log x$, hence we have $\limsup_{\delta \to 0} c_1(T, \delta)/\log(1/\delta) \leq 1$. Since $\lim_{\delta \to 0} c_2(T, \epsilon)/\log(1/\delta) = 0$, we obtain

$\limsup_{\delta \to 0} \frac{T_\zeta(\delta)}{\log(1/\delta)} \le \frac{2(1+\eta)T^\star_{\epsilon,\beta}(\nu)}{(1-\zeta)(1-\kappa)}$. For every $T \ge \max\{T_\zeta(\delta), N_3/\kappa\}$, we have $\tau_{\epsilon,\delta} \le T$, hence $\mathbb{E}_{\nu\pi}[\tau_{\epsilon,\delta}] \le T_\zeta(\delta) + \mathbb{E}_{\nu\pi}[N_3]/\kappa < +\infty$. Therefore, for all $\zeta, \kappa > 0$, we obtain

$$\limsup_{\delta \to 0} \frac{\mathbb{E}_{\nu\pi}[\tau_{\epsilon,\delta}]}{\log(1/\delta)} \le \limsup_{\delta \to 0} \frac{T_\zeta(\delta)}{\log(1/\delta)} \le \frac{2(1+\eta)T^\star_{\epsilon,\beta}(\nu)}{(1-\zeta)(1-\kappa)} .$$

Letting $\zeta$ and $\kappa$ go to zero concludes the proof. $\qquad\square$

**Proof of Theorem 7**   The proof is obtained by combining Theorem 6 and Lemmas 5, 60, 64 and 65.

# I   Variants of Algorithms

In Appendix I, we propose several variants of the algorithmic components used in our algorithm. The objective is to give freedom of choice for the practitioners interested in solving $\epsilon$-global DP BAI. Given the rich literature on BAI, it is unreasonable to provide details for the $\epsilon$-global DP version of all the existing BAI algorithms. Therefore, we settle for a few instances that has received increased scrutiny in the BAI literature.

First, we adapt the Track-and-Stop sampling rule [Garivier and Kaufmann, 2016] to solve $\epsilon$-global DP BAI (Appendix I.1). This leverages the computational tractable procedure to compute the optimal allocation $w^\star_\epsilon$ derived in Lemma 47. Second, we explore some alternative choices of components of the Top Two sampling rule for $\epsilon$-global DP BAI (Appendix I.2). This includes adaptive choice of target for the leader, hence aiming at achieving $T^\star_\epsilon(\nu)$ instead of $T^\star_{\epsilon,\beta}(\nu)$. Third, we adapt the LUCB sampling rule [Kalyanakrishnan et al., 2012] for $\epsilon$-global DP BAI (Appendix I.3).

## I.1   Track-and-Stop Sampling Rule

The Track-and-Stop (TaS) sampling rule was introduced in the seminal paper [Garivier and Kaufmann, 2016]. At each time $n$, it solves the optimization problem defining the characteristic time for the current empirical estimator $\tilde{\mu}_n$. When $\tilde{\mu}_n \in (0,1)^K$, we define $\widetilde{w}_n = w^\star_\epsilon(\tilde{\nu}_n)$ where $\tilde{\nu}_n$ is the Bernoulli instance with means $\tilde{\mu}_n$. When $\tilde{\mu}_n \notin (0,1)^K$, $[\tilde{\mu}_n]^1_0$ corresponds to a degenerate Bernoulli instance, hence we define $\widetilde{w}_n = 1_K/K$. Since $\tilde{\mu}_n$ is updated on a per-arm geometric grid governed by $\eta$, the optimal allocation $\widetilde{w}_n$ is updated on the same per-arm geometric grid. Therefore, choosing a larger $\eta$ yields lower computational cost of TaS at the cost of larger expected sampled complexity, i.e., asymptotic multiplicative factor $1 + \eta$ due to the update grid.

Given the vector $\widetilde{w}_n \in \triangle_K$, the next arm $a_n$ to sample is obtained by using C-Tracking [Garivier and Kaufmann, 2016] with forced exploration in order to ensure that sufficient exploration holds. This is done here by projecting on $\triangle^\epsilon_K = \{w \in [\epsilon,1]^K \mid \sum_{a \in [K]} w_a = 1\}$ for a well chosen $\epsilon \in (0, 1/K)$. Let $\widetilde{w}^{\epsilon_n}_n$ be the $\ell_\infty$ projection of $\widetilde{w}_n$ on $\triangle^{\epsilon_n}_K$ with $\epsilon_n = (K^2 + n)^{-1/2}/2$. While we consider a projection that changes at each time $n$ (due to $\epsilon_n$), $\widetilde{w}^{\epsilon_n}_n$ could also be updated on a per-arm geometric grid, i.e., when $\widetilde{w}_n$ is updated itself. For all $n \ge K + 1$, the TaS sampling rule defines

$$a_n \in \arg\max_{a \in [K]} \left\{ \sum_{t \in [n]} \widetilde{w}^{\epsilon_t}_{t,a} - N_{n,a} \right\} . \tag{43}$$

In summary, our proposed Track-and-Stop algorithm is defined as in DP-TT with the sole modification that Lines 13-14 are replaced by the sampling rule defined in Eq. (43).

**Optimal Allocation Oracle**   In Lemma 47, we show that $w^\star_\epsilon(\nu)$ can be computed explicitly based on the unique fixed-point solution $F_\mu(y) = 1$ for $y \in [0, \min_{a \ne a^\star(\mu)} d^-_\epsilon(\mu_{a^\star(\mu)}, \mu_a))$, where $F_\mu$ is an increasing one-to-one mapping from $[0, \min_{a \ne a^\star(\mu)} d^-_\epsilon(\mu_{a^\star(\mu)}, \mu_a))$ to $[0, +\infty)$ defined as

$$F_\mu(y) = \sum_{a \ne a^\star(\mu)} \frac{d^-_\epsilon(\mu_{a^\star(\mu)}, u_a(x_a(y)))}{d^+_\epsilon(\mu_a, u_a(x_a(y)))} . \tag{44}$$

The definitions of $u_a$ and $x_a$ is defered to Lemma 47, $u_a$ is decreasing and $x_a$ is increasing and strictly convex.

**Asymptotic Expected Sample Complexity**  Combining the TaS sampling rule $a_n$ as in Eq. (43) with the $\mathrm{GPE}_\eta(\epsilon)$ update and the GLR stopping rule as in Eq. (7) for the stopping threshold as in Eq. (8) yields a $\delta$-correct and $\epsilon$-global DP algorithm (see Lemma 5 and Theorem 6). Moreover, we conjecture that its satisfies that, for all $\boldsymbol{\nu} \in \mathcal{F}^K$ with unique best arm,

$$\limsup_{\delta \to 0} \frac{\mathbb{E}_{\boldsymbol{\nu}\pi}[\tau_{\epsilon,\delta}]}{\log(1/\delta)} \le 2(1+\eta)T_\epsilon^\star(\boldsymbol{\nu}) \,.$$

The multiplicative factor $1 + \eta$ comes from the per-arm geometric update grid, and the factor 2 comes from the asymptotic scaling in $2\log(1/\delta)$ of the stopping threshold. Using Theorem 7 for $\beta = 1/2$ and $T_{\epsilon,1/2}^\star(\boldsymbol{\nu}) \le 2T_\epsilon^\star(\boldsymbol{\nu})$ (Lemma 44), proving this conjecture would only yield an asymptotic improvement by a factor of at most 2. However, this would come at the price of a significantly higher computational cost.

**Proof Sketch of Conjecture**  While the detailed proof of this conjecture is beyond the scope of this work, an astute reader could notice that all the necessary steps were proven to derive Theorem 7 for DP-TT. At a high level, it is intuitive that the asymptotic analysis of Track-and-Stop is simpler than the one of DP-TT.

First, the forced exploration is enforced algorithmically, hence an equivalent of Lemma 60 can be shown for the Track-and-Stop sampling rule. In contrast, the proof of sufficient exploration for DP-TT is more challenging and involves a subbtle reasoning towards contradiction, see Appendix H.2 for more details.

Second, the convergence towards the optimal allocation is also enforced algorithmically. Thanks to the forced exploration and due to the continuity of $\boldsymbol{\nu} \mapsto w_\epsilon^\star(\boldsymbol{\nu})$ (Lemma 42) and the convergence $\tilde{\mu}_n \to_{n \to +\infty} \mu$, the empirical optimal allocation $\widetilde{w}_n$ converges towards the true optimal allocation $w_\epsilon^\star(\boldsymbol{\nu})$. Therefore, an equivalent of Lemma 64 can be shown for the Track-and-Stop sampling rule. In contrast, the proof of convergence towards $\beta$-optimal allocation for DP-TT is more challenging and leverage subbtle regularity properties of the $\beta$ characteristic time and its optimal allocation, e.g., the equality at equilibrium of all the transportations costs in Eq. (39), see Appendix H.3 for more details.

Third, the invertion of the GLR stopping rule can be done similarly as for DP-TT. The sole modification lies in using our derived regularity properties for $w_\epsilon^\star(\boldsymbol{\nu})$ instead of $w_{\epsilon,\beta}^\star(\boldsymbol{\nu})$, e.g., the equality at equilibrium of all the transportations costs in Lemma 43. Therefore, an equivalent of Lemma 65 can be shown for the Track-and-Stop sampling rule with $2(1+\eta)T_\epsilon^\star(\boldsymbol{\nu})$ instead of $2(1+\eta)T_{\epsilon,\beta}^\star(\boldsymbol{\nu})$, see Appendix H.4 for more details.

## I.2  Top Two Sampling Rule

As detailed in Chapter 2.2 in Jourdan [2024], a Top Two sampling rule is defined by four choices: a leader arm $B_n \in [K]$, a challenger arm $C_n \in [K] \setminus \{B_n\}$, a target $\beta_n(B_n, C_n) \in [0,1]$ and a mechanism to reach the target, i.e., $a_n \in \{B_n, C_n\}$ by using $\beta_n(B_n, C_n)$. For instance, the sampling rule in DP-TT uses the EB leader, the TCI challenger, a fixed target $\beta \in (0,1)$ and $K$ independent $\beta$-tracking procedures (one per leader). We propose adaptive choice of target (Appendix I.2.1), as well as leader fostering implicit exploration (Appendix I.2.2).

### I.2.1  Adaptive Target

When the target is fixed to $\beta$ beforehand, the Top Two sampling rule can achieve $T_{\epsilon,\beta}^\star(\boldsymbol{\nu})$ at best. We propose adaptive choices of the target inspired by the recent literature on asymptotically optimal Top Two algorithms [You et al., 2023; Bandyopadhyay et al., 2024].

**BOLD Target**  Given the EB-TCI leader/challenger pair $(B_n, C_n)$ defined in DP-TT, we adapt the BOLD target from Bandyopadhyay et al. [2024]. Let us define

$$u_{\epsilon,B_n,a}(\tilde{\mu}_n, N_n) = \underset{u \in [0,1]}{\arg\min} \left\{ N_{n,B_n} d_\epsilon^-(\tilde{\mu}_{n,B_n}, u) + N_{n,a} d_\epsilon^+(\tilde{\mu}_{n,b}, u) \right\} \,, \tag{45}$$

whose closed-form solution is given in Lemma 45. Then, the deterministic BOLD target defines the next arm to pull as

$$a_n = B_n \quad \text{if} \quad \sum_{a \neq B_n} \frac{d_\epsilon^-(\tilde{\mu}_{n,B_n}, u_{\epsilon,B_n,a}(\tilde{\mu}_n, N_n))}{d_\epsilon^+(\tilde{\mu}_{n,a}, u_{\epsilon,B_n,a}(\tilde{\mu}_n, N_n))} > 1 \quad \text{and} \quad a_n = C_n \quad \text{otherwise.} \quad (46)$$

In summary, the sole modification in DP-TT is Line 14 that is replaced by the sampling rule defined in Eq. (46).

For any single-parameter exponential family of distributions, Bandyopadhyay et al. [2024] shows that the BOLD target allows to reach asymptotic optimality. Forced exploration is added by Bandyopadhyay et al. [2024] to ensure that sufficient exploration holds. Showing that the BOLD target can achieve asymptotic optimality without forced exploration, i.e., meaning that it ensures sufficient exploration on its own, is an open problem.

**IDS Target**   Given the EB-TCI leader/challenger pair $(B_n, C_n)$ defined in DP-TT, we adapt the IDS target from You et al. [2023]. Namely, the randomized IDS target defines the next arm to pull from as

$$a_n = \begin{cases} B_n & \text{with proba } \beta_n(B_n, C_n) \\ C_n & \text{otherwise} \end{cases} \quad \text{where } \beta_n(B_n, C_n) = \frac{N_{n,B_n} d_\epsilon^-(\tilde{\mu}_{n,B_n}, u_{\epsilon,B_n,C_n}(\tilde{\mu}_n, N_n))}{W_{\epsilon,B_n,C_n}(\tilde{\mu}_n, N_n)},$$
$$(47)$$

where $u_{\epsilon,B_n,C_n}(\tilde{\mu}_n, N_n)$ is defined in Eq. (45). In summary, the sole modification in DP-TT is Line 14 that is replaced by the sampling rule defined in Eq. (47).

While we could use $K(K-1)$ tracking procedures to select $a_n \in \{B_n, C_n\}$, we use randomization above for the sake of simplicity. For Gaussian distributions with known variance, You et al. [2023] shows that the IDS target allows to reach asymptotic optimality. Showing that the IDS target can achieve optimality for other classes of distributions is an open problem.

**Asymptotic Expected Sample Complexity**   Sampling $a_n$ as in Eq. (46) or (47) for the EB-TCI leader/challenger pair $(B_n, C_n)$ defined in DP-TT based on the $\text{GPE}_\eta(\epsilon)$ update and the GLR stopping rule as in Eq. (7) for the stopping threshold as in Eq. (8) yields a $\delta$-correct and $\epsilon$-global DP algorithm (see Lemma 5 and Theorem 6).

While we conjecture that their asymptotic expected sample complexities $\limsup_{\delta \to 0} \frac{\mathbb{E}_{\nu \pi}[\tau_{\epsilon,\delta}]}{\log(1/\delta)}$ are both upper bounded by $2(1 + \eta)T_\epsilon^\star(\nu)$, we emphasize that our analysis doesn't provide the necessary steps for this result to hold. This is an interesting research direction left for future work.

### I.2.2   Implicit Exploring Leaders and TC Chalenger

The empirical best (EB) leader is a greedy choice of leader that doesn't foster implicit exploration. Without additional exploration mechanism, it can suffer from large empirical stopping time despite being enough for an asymptotic analysis, see [Jourdan et al., 2022]. This motivated the choice of the TCI challenger for DP-TT, since it fosters additional implicit exploration by penalizing over sampled challengers with the $\log N_{n,a}$ term. We propose other choices of leaders that foster implicit exploration, and define the TC challenger that removes this penalization.

The UCB leader is defined as

$$B_n^{\text{UCB}} \in \arg\max_{a \in [K]} U_{n,a} \quad \text{where} \quad U_{n,a} = \max \left\{ u \in [0,1] \mid N_{n,a} d_\epsilon^+([\tilde{\mu}_{n,a}]_0^1, u) \leq \log(n) \right\}. \quad (48)$$

By adding a bonus to the empirical mean, we are optimistic since we consider that the means are better than suggested by our observations.

The IMED leader builds on the IMED algorithm [Honda and Takemura, 2015] is defined as

$$B_n^{\text{IMED}} \in \arg\min_{a \in [K]} \left\{ N_{n,a} d_\epsilon^+([\tilde{\mu}_{n,a}]_0^1, \tilde{\mu}_n^\star) + \log N_{n,a} \right\} \quad \text{where} \quad \tilde{\mu}_n^\star = \max_{a \in [K]} [\tilde{\mu}_{n,a}]_0^1. \quad (49)$$

The TC challenger is defined as

$$C_n^{\text{TC}} \in \arg\min_{a \neq B_n} W_{\epsilon,B_n,b}(\tilde{\mu}_n, N_n), \quad (50)$$

where $W_{\epsilon,a,b}$ is defined as in Eq. (4).

In summary, the sole modification in DP-TT is Line 13 which can be replaced by choosing the leader as in Eq. (48) or Eq. (49), or choosing the challenger as in Eq. (50).

**Asymptotic Expected Sample Complexity**   Choosing the leader as in Eq. (48) or Eq. (49) or the challenger as in Eq. (50) based on the $\beta$-tracking as in DP-TT, the $\mathrm{GPE}_\eta(\epsilon)$ update and the GLR stopping rule as in Eq. (7) for the stopping threshold as in Eq. (8) yields a $\delta$-correct and $\epsilon$-global DP algorithm (see Lemma 5 and Theorem 6). Moreover, we conjecture that its satisfies that, for all $\boldsymbol{\nu} \in \mathcal{F}^K$ with distinct means,

$$\limsup_{\delta \to 0} \frac{\mathbb{E}_{\boldsymbol{\nu}\pi}[\tau_{\epsilon,\delta}]}{\log(1/\delta)} \leq 2(1+\eta)T^\star_{\epsilon,\beta}(\boldsymbol{\nu}) \,.$$

While the detailed proof of this conjecture is beyond the scope of this work, an astute reader could notice that all the necessary steps were proven to derive Theorem 7 for DP-TT. When using the TC challenger as in Eq. (50), the proofs of Lemmas 59 and 63 can be readily adapted. When using the UCB leader as in Eq. (48) or the IMED leader as in Eq. (49), the proofs of Lemmas 56 and 61 could also be adapted.

### I.3   LUCB Sampling Rule

While the Top Two terminology was introduced in Russo [2016], the first sampling rule having a Top Two structure is the greedy sampling strategy in LUCB1 introduced by Kalyanakrishnan et al. [2012]. At each time $n$, it selects the EB leader $B_n^{\mathrm{EB}} = \tilde{a}_n$ and the UCB challenger defined as

$$C_n^{\mathrm{UCB}} \in \arg\max_{a \neq B_n^{\mathrm{EB}}} U_{n,a} \quad \text{where} \quad U_{n,a} \quad \text{as in Eq. (48)} \,. \tag{51}$$

Then, it samples both $B_n^{\mathrm{EB}}$ and $C_n^{\mathrm{UCB}}$. Instead of using the GLR stopping rule as in Eq.(7), LUCB1 stops when the LCB (lower confidence bound) of the leader exceeds the UCB of the challenger, i.e.,

$$\tau_{\epsilon,\delta}^{\mathrm{LUCB1}} = \inf\left\{n \mid \widetilde{L}_{n,B_n^{\mathrm{EB}}} > U_{n,C_n^{\mathrm{UCB}}}\right\} \,, \tag{52}$$

where

$$\widetilde{L}_{n,a} = \max\left\{u \in [0,1] \mid N_{n,a}d_\epsilon^-([\tilde{\mu}_{n,a}]_0^1, u) \leq \log(n)\right\} \,. \tag{53}$$

In summary, the modifications in DP-TT are: (1) the sampling rule in Lines 13-15 is replaced by sampling both $B_n^{\mathrm{EB}}$ and $C_n^{\mathrm{UCB}}$, and (2) the stopping rule in Line 10 is replaced by Eq. (52). While studying this algorithm is beyond the scope of this work, we emphasize that LUCB is known to not reach asymptotic ($\beta$-)optimality.

## J   Implementation Details and Supplementary Experiments

Appendix J is organized as follows. First, we provide additional detail on the implementation details for our algorithm (Appendix J.1). Second, we provide supplementary experiments to illustrate the good performance of our algorithm (Appendix J.2).

### J.1   Implementation Details

We present additional experiments comparing the algorithms in different bandit instances with Bernoulli distributions, as defined by Sajed and Sheffet [2019], namely

$\mu_1 = (0.95, 0.9, 0.9, 0.9, 0.5)$,   $\mu_2 = (0.75, 0.7, 0.7, 0.7, 0.7)$,
$\mu_3 = (0.1, 0.3, 0.5, 0.7, 0.9)$,   $\mu_4 = (0.75, 0.625, 0.5, 0.375, 0.25)\}$,
$\mu_5 = (0.75, 0.53125, 0.375, 0.28125, 0.25)$,   $\mu_6 = (0.75, 0.71875, 0.625, 0.46875, 0.25)\}$.

For each Bernoulli instance, we implement the algorithms with

$$\epsilon \in \{0.001, 0.005, 0.01, 0.05, 0.1, 0.2, 0.3, 0.4, 0.5, 0.6, 0.7, 0.8, 0.9, 1, 10, 100, 125\} \,.$$

The risk level is set at $\delta = 0.01$. We verify empirically that the algorithms are $\delta$-correct by running each algorithm 1000 times.

We implement all the algorithms in Python (version 3.8) and on an 8 core 64-bits Intel i5@1.6 GHz CPU.

**Choice of Hyperparameters**    The default choice $\beta = 1/2$ is motivated by the worst-case inequality $T^\star_{\epsilon,1/2}(\nu) \leq 2T^\star_\epsilon(\nu)$ proven in Lemma 44. It implies that an asymptotically $1/2$-optimal algorithm has an expected sample complexity that is at worst twice as high as an asymptotically optimal algorithm. Moreover, this choice ensures that we sample the leader arm (i.e., the empirical estimate for $a^\star$) half of the time, and spend the remaining samples to explore all the other suboptimal arms (i.e., the challengers) until we are confident enough about $\tilde{a}_n$ being $a^\star$.

The default choice $\eta = 1$ is motivated by the trade-off existing between theoretical guarantees and empirical performance. Based on Theorem 7, smaller $\eta$ yields "better" theoretical asymptotic guarantees, i.e., smaller upper bound on the expected sample complexity of DP-TT. However, the asymptotic regime of $\delta \to 0$ fails to capture the empirical trade-off for moderate values of $\delta$. Choosing $\delta$ arbitrarily close to 0 will result in a large stopping threshold as $k_\eta$ scales as $1/\log(1 + \eta)$, see Eq. (8) in Theorem 6. Smaller $\eta$ also implies that a larger cumulative Laplacian noise is added to the cumulative Bernoulli signal. The limit $\eta = 0$ coincides with adding one Laplace noise per Bernoulli observation. The concentration results in Appendix F.6 (Lemmas 18 and 19) show that the rate is governed by $\tilde{d}^\pm_\epsilon(\mu \pm x, \mu, 1)$, which is not equivalent to $d^\pm_\epsilon(\mu \pm x, \mu)$ asymptotically.

**Remark 2.** *To implement the thresholds of* AdaP-TT, AdaP-TT$^\star$ *and DP-TT, we use empirical thresholds that we get by approximating the theoretical thresholds. The expressions of the empirical thresholds used can be found in the code in the supplementary material.*

## J.2    Supplementary Experiments

Figure 2 confirms our experimental findings from Section 6. DP-TT outperforms all the other $\delta$-correct and $\epsilon$-global DP BAI algorithms, for different values of $\epsilon$ and in all the instances tested. The empirical performance of DP-TT demonstrates two regimes. A high-privacy regime, where the stopping time depends on the privacy budget $\epsilon$, and a low privacy regime, where the performance of DP-TT is independent of $\epsilon$, and requires twice the number of samples used by the non-private EB-TCI-$\beta$.

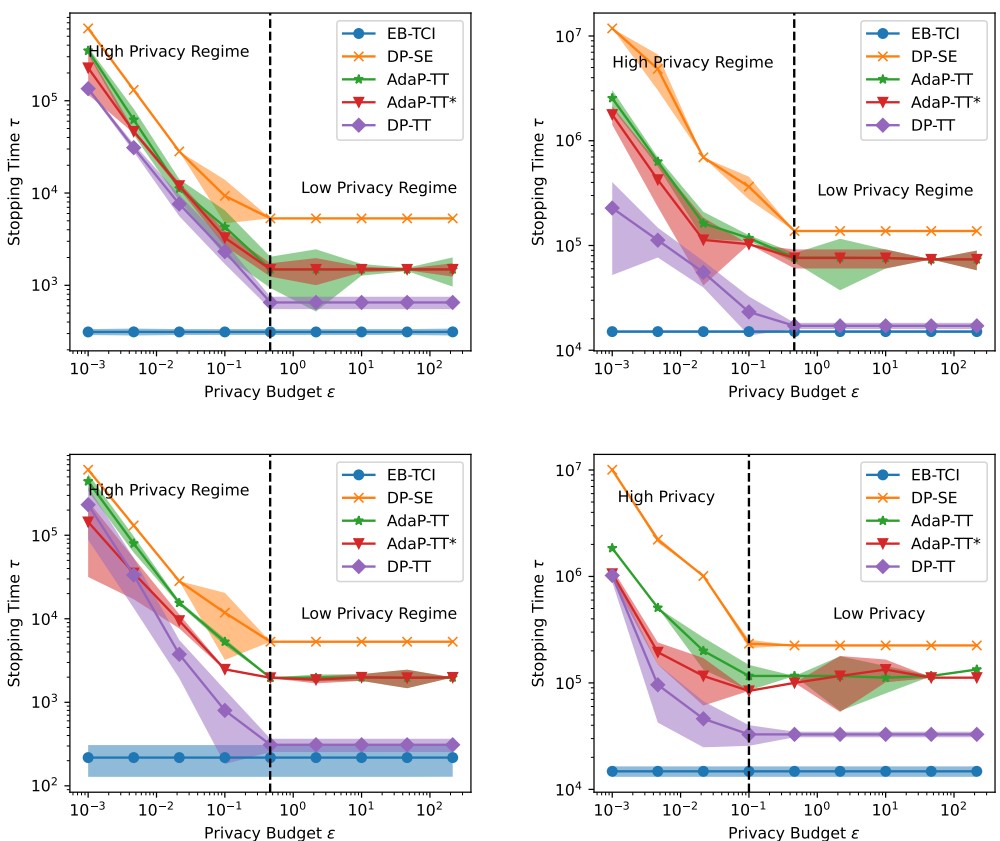

Figure 2: Empirical stopping time $\tau_{\epsilon,\delta}$ (mean $\pm 2$ std. over 1000 runs) for $\delta = 10^{-2}$ with respect to the privacy budget $\epsilon$ for $\epsilon$-global DP on Bernoulli instances $\mu_3$, $\mu_4$, $\mu_5$ and $\mu_6$ (top left to bottom right). The shaded vertical line separates the two privacy regimes.

