# OpenReview forum: "Optimal Best Arm Identification under Differential Privacy"
_NeurIPS.cc/2025/Conference — NeurIPS 2025 poster_

### Official Review · Reviewer_YnCC · 2025-06-11

**Clarity:** 3
**Significance:** 3
**Originality:** 3
**Rating:** 5
**Confidence:** 3

**Summary:**

The authors study the problem of Best Arm Identification (BAI) with fixed confidence $\delta$ under global $\epsilon$-DP, assuming Bernoulli bandits. In this setting, they improve upon a previously established lower bound and propose an asymptotically optimal algorithm that matches the new lower bound, leaving only a small multiplicative factor gap between the bounds. The proposed algorithm (DP Top-Two Sampling with new sampling rule) achieves better sample complexity than previously established algorithms, requiring only about twice as many samples as the non-private BAI in a low privacy regime.

**Questions:**

I have a few questions regarding numerical experiments:

1) How would Figure 1 change for different values of $\delta$?

2) Could you quantify the boundary between the low- and high-privacy regimes shown in Figure 1?

**Ethical Concerns:**

["NO or VERY MINOR ethics concerns only"]

**Final Justification:**

The paper is technically solid, well written, and builds upon a series of previous works. The Related Work section in the appendix provides a comprehensive overview of the literature on private bandits.

The paper proves a new lower bound for the BAI problem. Even though the tradeoff between KL and TV distances has already been introduced for regret minimization, the BAI setting required a novel proof technique. The authors propose a new sampling rule for the DP Top-Two algorithm, achieving provably asymptotically optimal sample complexity and demonstrating better numerical results than alternative methods.


The limitations of the paper remain: it focuses on best-arm identification with a fixed confidence level, $\epsilon$-DP, only Bernoulli bandits, and provides only asymptotic bounds. However, I find the contributions of the paper to outweigh these limitations.

My concerns regarding the paper’s motivation and presentation have been addressed by the authors and are promised to be further resolved.

Therefore, I continue to support the acceptance of the paper.

**Limitations:**

yes

**Paper Formatting Concerns:**

I do not have any concerns

**Quality:**

4

**Strengths And Weaknesses:**

The paper is technically solid, well written, and builds upon a series of previous works. The Related Work section in the appendix provides a comprehensive overview of the literature on private bandits.

The paper proves a new lower bound for the BAI problem. Even though the tradeoff between KL and TV distances has already been introduced for regret minimization, the BAI setting required a novel proof technique. The authors propose a new sampling rule for the DP Top-Two algorithm, achieving provably asymptotically optimal sample complexity and demonstrating better numerical results than alternative methods.

Even though this might be of limited interest to the broader DP community, I believe it will be a valuable contribution to DP bandit research. Therefore, I support the acceptance of the paper.

**Weaknesses:**

1) The setting is quite limited, as it focuses on BAI with a fixed confidence level, \epsilon-DP, and only Bernoulli bandits.

2) The problem is only solved asymptotically; a gap still remains, and it does not shrink in the limit to the non-private case.

3) I do not find the paper’s motivation for private BAI convincing. While I do not doubt the importance of the problem, I believe it could be presented better, possibly with more examples.

4) The presentation would benefit from polishing. For instance, the abbreviation "GLR" is used in the contributions section but is only defined in full in Section 4.

---

> ### Author Rebuttal · Authors · 2025-07-30
>
> We thank Reviewer YnCC for the time spent and the support given towards acceptance.
> We address the reviewer's questions below.
>
> **W1 - Limited Setting.** Here, we provide some comments on going beyond our setting.
>
> (a) *Beyond Bernoulli distributions.* See the answer to W for Reviewer inBP.
>
> (b) *Beyond $\epsilon$-DP.* See answer to Q3 for Reviewer VJtw.
>
> (c) *Beyond fixed-confidence.*
> The extension of our results to the fixed-budget and the anytime settings is an ongoing research direction.
> See paragraph ``Performance Metrics'' in Appendix C.1 for detailed references.
> Based on the guarantees achieved by Top Two algorithms in the non-private BAI setting [52], similar results might hold for DP-TT, or some variant of it.
> However, this requires a non-asymptotic understanding of DP-TT.
> As described below, this raises new technical challenges.
>
> (d) *Beyond BAI*
> The technical arguments used to show Theorem 2 allow obtaining a similar lower bound for (i) one-parameter exponential families, e.g., Gaussian, (ii) pure exploration problems with a unique correct answer, e.g., top-$m$ best arm identification, and (iii) structured instances without arm correlation, e.g., unimodal bandits.
> This is straightforwardly done by adapting the definition of Alt$(\nu)$, $a^\star(\nu)$, and $\mathcal F$ in Equation 1.
> For settings that do not fall into the above three categories, the characteristic time requires more subtle modifications; yet our intermediate technical results can still be used.
> Based on the non-private fixed-confidence literature, we conjecture the changes of characteristic times described below.
>
> - For bounded distribution (non-parametric) or Gaussian with unknown variance (two-parameters exponential family), the KL should be replaced by an infimum over KL [2,51]. The plurality of distributions having the same mean implies the existence of distribution that is the most confusing given a specific constraint on the mean, which is captured by this ``inf KL'' term.
>
> - For $\gamma$-Best Arm Identification (multiple correct answers), an outer maximization over all the correct answers should be added [Degenne and Koolen, 2019, Pure exploration with multiple correct answers]. The plurality of correct answers implies the existence of a correct answer that is the easiest to verify, which is captured by this ``max'' over correct answers.
>
> - For linear bandits (correlated means), the $\inf$ in the definition of $d_{\epsilon}$ per-arm should be replaced by a joint infimum over all the arms and moved outside the sum over arms [Degenne et al., 2020, Gamification of Pure Exploration for Linear Bandits].
>
> Since our intermediate technical results jointly and tightly capture the interplay between the $\delta$-correct and the $\epsilon$-global DP constraints, it paves the way for numerous studies of private pure exploration settings.
>
> **W2 - Asymptotic analysis, with gap remaining.**
>
> (a) *Beyond asymptotic guarantees.*
> Theorem 2 is already a non-asymptotic lower bound, that holds for any values of $\epsilon$ and $\delta)$.
> Theorem 5 shows that DP-TT is $\delta$-correct and $\epsilon$-global DP for any $(\epsilon,\delta)$.
> While Theorem 6 is an asymptotic upper bound for $\delta \to 0$, it holds for any $\epsilon$.
> Since our experiments reveal the good performance of DP-TT for the moderate value of $\delta=0.01$, deriving a non-asymptotic upper bound on its expected sample complexity is an interesting direction for future work.
> Based on the non-asymptotic guarantees achieved by Top Two algorithms in the non-private BAI setting [52], DP-TT might also enjoy similar properties.
>
> (b) *Constant factor between the upper and lower bounds.*
> Appendix C.2 contains a discussion on how to further reduce the multiplicative problem-independent constant gap between the upper and the lower bound.
> Combined with the appropriate analysis, an adaptive choice of $\beta$ such as IDS [86] or BOLD [14] would replace $T^\star_{\epsilon,\beta}(\nu)$ by $T^\star_{\epsilon}(\nu)$, which is the optimal scaling asymptotically.
> Based on finer concentration results, the stopping threshold could scale asymptotically as $\log(1/\delta)$ instead of $2\log(1/\delta)$.
> Provided these improvements are proven, the multiplicative factor would be reduced to $1+\eta$, where $\eta$ is the parameter that controls the geometrically increasing phases. In theory, one can set $\eta$ arbitrarily close to $0$, thus completely shaving the multiplicative gap in the asymptotic results. However, this may come at the cost of the non-asymptotic performance of DP-TT with this choice of $\eta$, see the answer to Reviewer iuVL.
>
> **W3 - Motivation for private BAI.** In the updated version, we will add an extended motivation section with multiple examples of privacy constraints in bandits. In addition to clinical trials, two other interesting applications are: hyperparameter tuning when learning under privacy constraints, and ad recommendations through click feedback.
>
> **W4 - Improved presentation.**
> See answer to Reviewer VJtw.
>
> **Q1 - Varying $\delta$ in Figure 1.**
> While it is not allowed to include figures in the rebuttal, we will add the equivalent of Figure 1 for $\delta \in \{0.1, 0.001\}$ in the revised version.
> Based on our upper and lower bounds, the $\delta$-dependent dominating term of the expected sample complexity of DP-TT scales as $\Theta(T^\star_{\epsilon}(\nu)\log(1/\delta))$.
> This is tight when $\delta \to 0$, up to a small instance-independent constant.
> For moderate values of $\delta$, we conjecture that $\delta$-independent additive terms in the upper and lower bounds exist and that they cannot be ignored.
> As those terms already exist for non-private BAI, additional ones are expected for private BAI.
> For a study of non-private Top Two algorithms in the non-asymptotic regime, we defer to [49].
> In Figure 6 in Appendix G.2.1 of [49] for EB-TCI on the “equal means” instance with $(K,\mu_{a^\star},\Delta) = (35,0,0.5)$, we see a ratio of roughly $2$ between the $\delta \in \{0.1, 0.001\}$, while it should be $3$ when using a cross product solely based on $T^\star(\nu)\log(1/\delta)$.
>
> **Q2 - Low and high-privacy regimes in Figure 1.**
> In Figure 1, the vertical lines are at (a) $\epsilon = 0.45$ and (b) $\epsilon = 0.45$.
> In the `Allocation Dependent Low Privacy Regime' paragraph (Lines 184-191), we provide both an allocation-dependent and allocation-independent theoretical values of the change-of-regime $\epsilon$.
> We compare with the allocation-independent condition from Line 191, i.e., $\max_{a \ne a^\star} \epsilon_{a^\star,a}$ where $\epsilon_{a,b} = \log \left( \frac{\mu_{a^\star}(1 - \mu_{a})}{\mu_{a}(1 - \mu_{a^\star})} \right)$.
> For $\mu_{2}$, we have $\epsilon_{1,a} = 0.25$ for all $a \ne 1$.
> This separation boundary roughly coincides with the one stipulated by Line 191.
> For $\mu_{1}$, we have $\epsilon_{1,5} = 2.94$ and $\epsilon_{1,a} = 0.75$ for all $a \in \{2,3,4\}$.
> The condition of Line 191 is stronger than the empirical separation when there are arms with significantly lower mean (arm $5$).
> This highlights the need for a ``tighter'' allocation-dependent condition, such as Eq.(6).
> Intuitively, an arm with a significantly lower mean will also be sampled significantly less, hence there is a compounding effect.

---

> > ### Comment · Reviewer_YnCC · 2025-08-01
> >
> > Thank you for your answers. I will continue supporting the paper's acceptance!

---

### Official Review · Reviewer_VJtw · 2025-07-02

**Clarity:** 3
**Significance:** 3
**Originality:** 3
**Rating:** 5
**Confidence:** 2

**Summary:**

This paper studies fixed-confidence best-arm identification under global differential privacy for Bernoulli bandits. It develops a new information-theoretic lower bound on sample complexity, refining previous KL/TV-based transportation cost characterizations, and proposes an algorithm that achieves this lower bound to within a small constant factor.

**Questions:**

1. Theorem 6 relies on a unique best arm. In practice, many arms could have nearly identical means. How does DP-TT behave in these near-tie scenarios, and would the stopping rule or estimator become unstable?
2. Experiment section describes the experiments on Bernoulli means but does not explore systematically how the sample complexity changes for a wider range of K, δ, or ε. Could the authors expand on how to design experiments to stress-test DP-TT, and share best practices for tuning these hyperparameters?
3. The conclusion mentions a possible extension to other DP frameworks, but no technical discussion is provided. Could the authors outline what would need to change in DP-TT or the lower bound if moving to (ε,δ)-DP or Rényi DP?

**Ethical Concerns:**

["NO or VERY MINOR ethics concerns only"]

**Limitations:**

Yes.

**Paper Formatting Concerns:**

No issues.

**Quality:**

3

**Strengths And Weaknesses:**

**Strengths:**

- **Strong theoretical contributions**: Developing novel concentration inequalities for Laplace distributions and mixed Bernoulli-Laplace sums, the `d_epsilon` divergence is elegant and leads to provably tighter lower bounds.
- **Near-optimal performance**: The elimination of "observation forgetting" is clever, while it introduces more Laplace noise, the logarithmic growth is well-controlled and preserves more signal. The geometric batching approach is elegant.
- **Empirical support**: Good experimental results showing DP-TT outperforms existing methods across different privacy regimes.

**Weaknesses:**

- **Limited scope**: Focused only on Bernoulli rewards; no extension to other sub-Gaussian families.
- **Presentation density**: Some sections (e.g., signed `d_epsilon` definitions) are hard to parse; clearer intuition would help.
- **Practical considerations**: Little discussion on how to set ε or interpret privacy-utility tradeoffs in real applications.

---

> ### Author Rebuttal · Authors · 2025-07-30
>
> We thank Reviewer VJtw for the time spent and the encouraging feedback.
> We address the reviewer's questions below.
>
> **W1 - Beyond Bernoulli distributions.**
> See answer to Reviewer inBP.
>
> **W2 - Presentation.**
> In the revised version, we will use the extra page to polish the presentation, add additional intuition, polish the grammar/punctuation, unpack the definitions of the quantities defined in Theorems 2 and 3 (e.g., $T^\star_{\epsilon}(\nu)$, $g_{\epsilon}^{\pm}$, $d_{\epsilon}$, $d_{\epsilon}^{\pm}$, $W_{\epsilon,a,b}$, $\Delta_{\epsilon,a}$, etc) outside the theorem statements, clarify the notation $\mathcal F^K$ by introducing a notation $\mathcal M$ for the set of mean instances with unique best arm, add the meaning of the abbreviation ``GLR'' before its first use.
>
> **W3 - Privacy-utility tradeoffs and choice of $\epsilon$.**
> Our matching upper and lower bounds suggest the existence of two privacy regimes: (i) a low privacy regime (high values of $\epsilon$) where the hardness of the problem reduces to the non-private KL-based characteristic time ($T^\star_\mathrm{KL}$) and privacy can be achieved for `free' (up to some multiplicative constants). And (ii) a high privacy regime (low values of $\epsilon$) where privacy has an additional cost characterised by $T^\star_\epsilon$, and at the limit when $\epsilon \rightarrow 0$, reduces to a TV-based characteristic time. We also provide exact characterisations of the value of $\epsilon$ at which the change of regimes happens. Please refer to the ``Allocation Dependent Low Privacy Regime'' paragraph (Lines 184-191) where we discuss in depth both an allocation-dependent and a weaker allocation-independent value of $\epsilon$ of change of regimes. In light of this discussion, the best choice of $\epsilon$ that provides the highest privacy protection while maintaining a great sample complexity is exactly the change-of-regime $\epsilon$. We will add a discussion about this at the end of Line 191.
>
> **Q1 - Behavior with near-optimal arms.**
> The assumption in Theorem 6 that the means are distinct is used to prove sufficient exploration; it can be removed by using forced exploration or a fine-grained analysis [49, 52].
> The unique best arm assumption is a standard assumption in the fixed-confidence BAI setting.
> Even in the non-private setting, this assumption is necessary to obtain a $\delta$-correct algorithm with finite expected stopping time, as the characteristic time becomes infinite otherwise.
> In the private setting, this phenomenon persists as can be seen by the lower bound in Eq.(5), i.e., $T_{\epsilon}^{\star}(\nu) \to +\infty$ when $\Delta_{\epsilon,a^\star} \to 0$.
> In near-tie scenarios, DP-TT will perform as badly as any other $\delta$-correct and $\epsilon$-global DP algorithms due to the fundamental lower bound.
> Empirically, this will be reflected by empirical transportation costs that are too small for the stopping condition to be met.
> In applications where the near-tie scenario is common, practitioners consider $\gamma$-Best Arm Identification, i.e., identify an arm that is $\gamma$-close to the best arm.
> For the non-private fixed-confidence $\gamma$-BAI setting, we refer to [52] for references and guarantees satisfied by Top Two algorithms.
> To tackle $\gamma$-BAI, DP-TT should be modified by using the appropriate transportation costs in both the stopping rule and the definition of the challenger, i.e., $W_{\epsilon,\gamma,a,b}$ instead of $W_{\epsilon,a,b}$.
> Studying this setting with multiple correct answers is left for future work.
>
> **Q2 - Additional experiments.**
> In Figures 1 and 2, we study the impact of $\epsilon$ with $17$ distinct values (see Appendix J.1), ranging from $0.001$ to $125$.
> Based on our upper and lower bounds, the $\delta$-dependent dominating term of the expected sample complexity of DP-TT scales as $\Theta(T^\star_{\epsilon}(\nu)\log(1/\delta))$.
> On the “equal means” instance (one optimal arm, and other arms with the same strictly lower mean), Eq.(5) suggests that $T^\star_{\epsilon}(\nu)$ scales linearly with $K$.
> Empirically, this linear scaling has been observed for Top Two algorithms in the non-private regime (e.g., Figure 1 in [49] for EB-TCI); hence, we conjecture that it would hold for DP-TT.
> See answer to Reviewer YnCC for the impact of varying $\delta$.
> See answer to Reviewer iuVL for the choice of hyperparameters.
> As figures are prohibited in the rebuttal, we will include supporting evidence in the revised version.
>
> **Q3 - Beyond $\epsilon$-DP.**
>
> (a) *Lower bound.* Our lower bound technique can be seen as a generalisation of the packing argument, and thus depends on the group privacy property. Specifically, in Appendix D, just before Equation 11, we used the group privacy of $\epsilon$-DP to bound $\text{KL}(M_{D}, M_{D'}) \leq \epsilon \text{dham}(D, D')$.
>
> - For $\rho$-zCDP, there is a similar group privacy property that can directly be plugged here, stating that $\text{KL}(M_{D}, M_{D'}) \leq \rho \text{dham}(D, D')^2$. This means that for $\rho$-zCDP, $\epsilon \times \mathrm{dham}$ is replaced by $\rho  \times \mathrm{dham}^2$
> in the fundamental optimal transport inequality of Equation 11. We leave solving tightly this new optimal transport for future work, and add a detailed discussion about this in an extended Future extensions paragraph in the appendix.
>
> - For $(\epsilon, \delta)$-DP, the group privacy property is not tight. This is a classic issue in $(\epsilon, \delta)$-DP lower bounds, where other techniques are used (e.g., fingerprinting). Adapting these techniques for bandits is an interesting open problem (even for bandits with regret minimisation).
>
> (b) *Privacy analysis.* It is straightforward to make DP-TT achieve either $\rho$-zCDP or $(\epsilon, \delta)$, by replacing the Laplace mechanism with the Gaussian mechanism.
>
> (c) *Concentration inequality.* To design a correct stopping rule with the Gaussian mechanism, it is possible to use the same fine-grained tail bounds of the sum of two independent random variables (Lemma 9) by only replacing the concentration of Laplace random variables with the concentration of Gaussian random variables.
>
> (d) *Algorithm design.* In the Top Two family of algorithms, an important design choice is the transportation cost, used for stopping and sampling the challenger. DP-TT uses a $d_\epsilon$ based transport, inspired by the lower bound of Theorem 1. Thus, to go beyond $\epsilon$-DP for algorithm design, it is important to derive a tight lower bound that suggests the use of a new transportation cost.
>
> To sum up, there are multiple technical challenges to have a complete solution with matching upper and lower bounds, up to a constant.

---

> > ### Comment · Reviewer_VJtw · 2025-08-07
> >
> > Thank you for your response!

---

### Official Review · Reviewer_inBP · 2025-07-03

**Clarity:** 2
**Significance:** 3
**Originality:** 3
**Rating:** 5
**Confidence:** 2

**Summary:**

The authors consider a bandit problem with $K$ arms generating a Bernoulli distributed random variable when picked. The goal is to find the arm with highest corresponding mean $\mu \in (0,1)^K$, whilst adhering to an $\epsilon$-differential privacy guarantee, in the sense that the arm identification algorithm's output should not reveal any information about the underlying sample.

Formally, an algorithm consists of a stopping time $\tau$, an arm recommendation $\tilde{a}$ and a sequence of arm recommendations up until $\tau$; $(a_1,\dots,a_\tau)$. Given two sequences of Hamming distance 1-apart arm draws $R,R'$,
$$P( (T,\tilde{a},(a_1,\dots,a_T) \in A | R) \leq e^\epsilon P( (T,\tilde{a},(a_1,\dots,a_T) \in A | R').$$

Within this setup, the authors:
* Provide a lower bound on the expected stopping time of a approximately corrected and $\epsilon$-DP algorithm.
* They provide a $\epsilon$-DP stopping rule that matches the lower bound up to a constant factor.

**Questions:**

Do the authors think that the results hold under $(\epsilon,\delta)$-differential privacy as well? I.e. can the lower bound be shown to hold for the more general differential privacy definition?

**Ethical Concerns:**

["NO or VERY MINOR ethics concerns only"]

**Limitations:**

The main limitation of the paper is its setting; which only covers Bernoulli observations.

**Quality:**

3

**Strengths And Weaknesses:**

As a main strength: I believe the authors provide a novel and technically sound contribution to bandit problems under privacy, improving over existing bounds in the literature.

The main weakness of the paper I see is its setting: considering only Bernoulli samples is quite limiting. I would like to see more discussion on why this limited setting is studied and which parts do no hold if one has for example a Gaussian BAI problem.

Minor comments:
* The grammar and punctuation could be improved in certain places.
* I would suggest (1) and (2) not to be given *inside* of Theorem 2 but before it.
* Similarly, Theorem 3, a lot is defined inside of the theorem statement, unnecessarily so? I would suggest the authors use the additional space to properly introduce and unpack the necessary quantities for Theorem 3 (and 2).
* I would avoid the name "global DP" personally; it seems that the setting is close to what is commonly called central DP in the i.i.d. setting.
* The choice of the $\delta$ notation is somewhat unfortunate given the differential privacy definition in full generality involving $\epsilon$ and $\delta$.
* The notation $\cF^K$ is poorly explained and seems superfluous; as it is a set that is characterized by $\mu$.

---

> ### Author Rebuttal · Authors · 2025-07-30
>
> We thank Reviewer inBP for the time spent reviewing and the positive feedback.
> We address the reviewer's questions below.
>
> **W - Beyond Bernoulli distributions.**
> We would like to stress the following points about our results and analysis.
>
> (a) *Lower bound.*
> The non-asymptotic lower bound on the expected sample complexity (Theorem 2) holds for any class of distributions $\mathcal F$, and thus is already true beyond Bernoullis (e.g. exponential families, sub-Gaussians, etc). However, by going beyond Bernoullis, $d_\epsilon$ might not admit a closed-form solution, e.g., the TV between Gaussian distributions has no simple formula. We conjecture that most regularity properties used to derive Theorem 3, and needed for the upper bound proofs, also hold for other classes of distributions. From a theoretical perspective, this might render the proof of the sufficient regularity properties more challenging.
> From a practical perspective, this increases the computational cost of the stopping rule and the challenger arm, as they require computing an empirical transportation cost based on the $d_\epsilon$ divergence. Finally, since the objective function inside the $d_\epsilon$ is continuous and convex over a compact interval, using off-the-shelf convex optimisation solvers to compute $d_\epsilon$ is still straightforward.
>
> (b) *Privacy analysis.* Our privacy analysis (Lemma 4) holds for any distribution with support in $[0,1]$, and thus is already true beyond Bernoullis. It is straightforward to extend this to any distribution with a support in $[a, b]$, by multiplying the noise terms by the range $(b-a)$. Also, all prior works in bandits with DP are either for Bernoulli or bounded distributions, for the simple reason that this assumption makes estimating the empirical mean privately using the Laplace mechanism straightforward, as the sensitivity is controlled. This helps focus on the more interesting tradeoffs between DP, exploration and exploitation, without any additional technical overheads that may be introduced by estimating privately the mean of unbounded distributions. We believe that considering $(\epsilon,\delta)$-DP and unbounded Gaussian rewards may be an interesting direction for future work.
>
> (c) *Concentration inequality.*
> The concentration results (Theorem 5) used to derive a stopping threshold ensuring $\delta$-correctness are highly specific to Bernoulli distributions.
> However, the proof builds on fine-grained tail bounds of the sum of two independent random variables, i.e., we bound those probabilities by the maximal product between their respective survival functions (Lemma 9).
> Therefore, the technical tools used for this intermediate result can be also used to study other one-parameter exponential families, e.g., Gaussian observation, with other noise mechanisms, e.g., Gaussian noise.
>
> (d) *Expected sample complexity upper bound.*
> The proof of Theorem 6 (Appendix H) builds on the unified analysis of the Top Two algorithm [50], coping with many classes of distributions, e.g., one-parameter exponential families and non-parametric bounded distributions.
> It relies heavily on the derived regularity properties (Appendix G) for the signed divergences, transportation costs, characteristic times, and optimal allocations.
> Based on the non-private BAI literature, we conjecture that most regularity properties used to derive Theorem 6 also hold for other classes of distributions.
> Due to the lack of explicit formulae, the proofs are challenging.
> As mentioned above, the DP-TT algorithm becomes computationally costly due to the intertwined optimisation procedure when computing the transportation costs numerically.
>
> (e) *Bernoulli setting.*
> Bernoulli bandits are a fundamental setting found in many applications (clinical trials, ad recommendations through clicks, etc).
> Our intermediate results can improve upon existing state-of-the-art in other settings, e.g., bandits with regret minimisation, sequential hypothesis testing, and DP under continual observations.
>
> **Q - Beyond $\epsilon$-DP.**
> See answer to Reviewer VJtw.
>
> **Minor comments.** We thank the reviewer for the precise recommendations and suggestions. In the revised version, we will use the extra page to polish the presentation, add additional intuition, polish the grammar/punctuation, unpack the definitions of the quantities defined in Theorems 2 and 3 (e.g., $T^\star_{\epsilon}(\nu)$, $g_{\epsilon}^{\pm}$, $d_{\epsilon}$, $d_{\epsilon}^{\pm}$, $W_{\epsilon,a,b}$, $\Delta_{\epsilon,a}$, etc) outside the theorem statements, clarify the notation $\mathcal F^K$ by introducing a notation $\mathcal M$ for the set of mean instances with unique best arm, add the meaning of the abbreviation ``GLR'' before its first use.
>
> - *Notation $\delta$.* To the best of our knowledge, the notation $\delta$ is used in almost all the papers that study pure exploration problems for stochastic multi-armed bandits in the fixed-confidence setting.
> While it is also standard to use this notation for $(\epsilon,\delta)$-DP, we chose to be consistent with the BAI literature when studying $(\epsilon,0)$-DP.
> See answer to Reviewer VJtw for comments on the presentation.
>
> - *$\epsilon$-global DP.*
> We chose the name "global DP" as it is the name used in prior works for bandits under central DP, in both regret and BAI settings.
> It is indeed exactly the central DP model with a trusted central decision maker.
> We add a comment about this in the Background section.

---

### Official Review · Reviewer_iuVL · 2025-07-05

**Clarity:** 3
**Significance:** 3
**Originality:** 3
**Rating:** 4
**Confidence:** 1

**Summary:**

The paper describes a fixed-confidence Best Arm Identification (BAI) setup under epsilon-global Differential Privacy (DP) for Bernoulli bandits. The paper presents a lower bound and a corresponding algorithm, DP-TT, which closes the gap between the two to within a small constant factor.

**Questions:**

First, I have to note that I do not work in this field, I tried my best to read around and understand the work.

* For the constant factor in the upper bound, is there a way to improve the allocation strategy to help the sample complexity further? Can it similarly be problem independent?
* Current derivation appears quite dependent on Bernoulli distributions, is there a way to generalise the setup to cover other types of bandits? What would be the main blocker in such extension?
* It appears that not forgetting samples here leads to improved sample complexity. As the same time, it feels that this statefullness can also be a problem for perhaps slightly different threat models since data keeps on being accumulated. I wonder what other trade-offs appear here e.g. memory?
* How does one pick (n, b) in practice? What would sample counts actually look like in practical scenarios?

**Ethical Concerns:**

["NO or VERY MINOR ethics concerns only"]

**Final Justification:**

Its a strong paper and the authors response made sense to me.

**Limitations:**

Yes

**Quality:**

3

**Strengths And Weaknesses:**

Strengths:
* Adoption of DP framwork to a new domain with signifincally better guarantees

Weaknesses:
* Still a constant factor in the bound
* Currently focusing only on Bernoulli

---

> ### Author Rebuttal · Authors · 2025-07-30
>
> We thank Reviewer iuVL for the time spent reviewing and the effort made despite not working in this field.
> We address the reviewer's questions below.
>
> **W1/Q1 - Constant factor between the upper and lower bounds.**
> See answer to Reviewer YnCC.
>
> **W2/Q2 - Beyond Bernoulli distributions.**
> See answer to Reviewer inBP.
>
> **Q3 - Implications of not forgetting.**
>
> (a) *Memory.* DP-TT has the same memory complexity as the forgetting algorithms. The reason is that DP-TT only needs to store one *accumulated* noisy sum of rewards *across phases*, while the forgetting ones store the noisy sum of rewards of *only* the last phase.
>
> (b) *$\epsilon$-global DP.* For privacy considerations, under our threat model where the adversary only observes the output of the algorithm (sequence of actions, final recommendation and stopping time), both forgetting and non-forgetting algorithms provide the same DP guarantees thanks to post-processing.
>
> (c) *Pan privacy.* One can indeed imagine other threat models where forgetting the rewards provides better privacy guarantees. One possible example of this is the pan-private threat model [32], where the adversary can also *intrude* into the internal states of the algorithm during its execution. In this threat model, if an adversary observes the internal states of the execution of DP-TT at some phase $\ell > 1$, they can see the full sum of rewards and maybe infer the membership of, say, the first user (up to tradeoffs guaranteed by the $\epsilon$-DP constraint, since the sum is noisy). However, for the forgetting algorithms, if the adversary observes the internal states of the execution of forgetting algorithms at some phase $\ell > 1$, the adversary can infer *nothing* about the reward of the first patient, since its reward has been deleted completely.
>
> **Q4.1 - Choice of hyperparameter $(\eta,\beta)$.**
> Without prior information on the mean vector $\mu$, we recommend using the default choice of hyperparameters as described in Algorithm 1.
>
> (a) *Leader target $\beta$.*
> The default choice $\beta = 1/2$ is motivated by the worst-case inequality $T^\star_{\epsilon, 1/2}(\nu) \le 2 T^\star_{\epsilon}(\nu)$ proven in Lemma 43.
> It implies that an asymptotically $1/2$-optimal algorithm has an expected sample complexity that is at worst twice as high as an asymptotically optimal algorithm.
> Moreover, this choice ensures that we sample the leader arm (i.e., the empirical estimate for $a^\star$) half of the time, and spend the remaining samples to explore all the other suboptimal arms (i.e., the challengers) until we are confident enough about $\tilde a_n$ being $a^\star$.
> With prior knowledge on the range of the optimal allocation for the best arm, one should choose $\beta$ accordingly to be closer to asymptotic optimality.
> See Appendix C.2 for references on an adaptive choice of $\beta$ to achieve $T^\star_{\epsilon}(\nu)$ instead of $T^\star_{\epsilon, \beta}(\nu)$.
>
> (b) *Geometric grid $\eta$.*
> The default choice $\eta = 1$ is motivated by the trade-off existing between theoretical guarantees and empirical performance.
> Based on Theorem 6, smaller $\eta$ yields ``better'' theoretical asymptotic guarantees, i.e., smaller upper bound on the expected sample complexity of DP-TT.
> However, the asymptotic regime of $\delta \to 0$ fails to capture the empirical trade-off for moderate values of $\delta$.
> Choosing $\delta$ arbitrarily close to $0$ will result in a large stopping threshold as $k_{\eta}$ scales as $1/\log(1+\eta)$, see Eq.(8) in Theorem 5.
> Smaller $\eta$ also implies that a larger cumulative Laplacian noise is added to the cumulative Bernoulli signal.
> The limit $\eta = 0$ coincides with adding one Laplace noise per Bernoulli observation.
> The concentration results in Appendix F.6 (Lemmas 17 and 18) show that the rate is governed by $\tilde d_{\epsilon}^{\pm}(\mu \pm  x, \mu, 1)$, which is not equivalent to $d_{\epsilon}^{\pm}(\mu \pm  x, \mu)$ asymptotically.
>
> **Q4.2 - Value of sample counts in practical scenarios.** In practice, the sample counts depend on the hardness of the bandit instance (characterised by $T^\star_\epsilon$, the characteristic time of Eq.1 based on $d_\epsilon$), the choice of the privacy budget $\epsilon$ and the $\delta$-correctness parameter. For easy instances (big gaps), $\delta$ in the order of $0.01$, and $\epsilon$ in the order of  $0.1$, the sample count can be in the order of the thousands.

---

> > ### Comment · Reviewer_iuVL · 2025-08-05
> >
> > Thank you for your response!

---

### Decision · Program_Chairs · 2025-09-17

**Decision:**

Accept (poster)

**Comment:**

The authors consider the Best Arm Identification (BAI) problem, where the goal is to find the arm with highest mean, under the constraint of $\epsilon$-differential privacy guarantee. Prior work showed tight upper and lower bounds for this problem for the small $\epsilon$ regime. This work closes the gap for the large $\epsilon$ regime. This is a classic problem with solid theoretical contributions that will likely be of interest to the community. All reviewers support the paper.